# Monte Carlo Based Techniques for Quantum Magnets with Long-Range Interactions

**DOI:** 10.3390/e26050401

**Published:** 2024-05-01

**Authors:** Patrick Adelhardt, Jan A. Koziol, Anja Langheld, Kai P. Schmidt

**Affiliations:** Department of Physics, Friedrich-Alexander Universität Erlangen-Nürnberg (FAU), 91058 Erlangen, Germany; patrick.adelhardt@fau.de (P.A.); jan.koziol@fau.de (J.A.K.); anja.langheld@fau.de (A.L.)

**Keywords:** quantum spin systems, long-range interactions, Ising interactions, XY interactions, Heisenberg interactions, Monte Carlo, series expansion, perturbative continuous unitary transformation, stochastic series expansion, quantum phase transitions, critical exponents, quantum simulation

## Abstract

Long-range interactions are relevant for a large variety of quantum systems in quantum optics and condensed matter physics. In particular, the control of quantum–optical platforms promises to gain deep insights into quantum-critical properties induced by the long-range nature of interactions. From a theoretical perspective, long-range interactions are notoriously complicated to treat. Here, we give an overview of recent advancements to investigate quantum magnets with long-range interactions focusing on two techniques based on Monte Carlo integration. First, the method of perturbative continuous unitary transformations where classical Monte Carlo integration is applied within the embedding scheme of white graphs. This linked-cluster expansion allows extracting high-order series expansions of energies and observables in the thermodynamic limit. Second, stochastic series expansion quantum Monte Carlo integration enables calculations on large finite systems. Finite-size scaling can then be used to determine the physical properties of the infinite system. In recent years, both techniques have been applied successfully to one- and two-dimensional quantum magnets involving long-range Ising, XY, and Heisenberg interactions on various bipartite and non-bipartite lattices. Here, we summarise the obtained quantum-critical properties including critical exponents for all these systems in a coherent way. Further, we review how long-range interactions are used to study quantum phase transitions above the upper critical dimension and the scaling techniques to extract these quantum critical properties from the numerical calculations.

## 1. Introduction

Since the advent of the theoretical description of classical and quantum phase transitions (QPTs), long-range interactions between degrees of freedom challenged the established concepts and propelled the development of new ideas in the field [1,2,3,4,5]. It is remarkable that, only a few years after the introduction of the renormalisation group (RG) theory by K.G. Wilson in 1971 as a tool to study phase transitions and as an explanation for universality classes [6,7,8,9,10,11], it was used to investigate ordering phase transitions with long-range interactions. These studies found that the criticality depends on the decay strength of the interaction [1,2,3]. It then took two decades to develop numerical Monte Carlo (MC) tools capable of simulating basic magnetic long-range models with thermal phase transitions following the behaviour predicted by the RG theory [12,13]. The results of these simulations sparked a renewed interest in finite-size scaling above the upper critical dimension [12,14,15,16,17,18,19] since “hyperscaling is violated” [13] for long-range interactions that decay slowly enough. In this regime, the treatment of dangerous irrelevant variables (DIVs) in the scaling forms is required to extract critical exponents from finite systems.

Meanwhile, a similar historic development took place regarding the study of QPTs under the influence of long-range interactions. By virtue of pioneering RG studies [20,21], the numerical investigation of long-range interacting magnetic systems has been triggered [22,23,24,25,26,27,28,29,30,31,32,33,34,35,36,37,38]. In particular, Monte Carlo-based techniques became a popular tool to gain quantitative insight into these long-range interacting quantum magnets [22,25,26,29,30,31,32,33,34,35,36,37,38,39,40]. On the one hand, this includes high-order series expansion techniques, where classical Monte Carlo integration is applied for the graph embedding scheme, allowing extracting energies and observables in the thermodynamic limit [25,29]. On the other hand, there is stochastic series expansion quantum Monte Carlo [39], which enables calculations on large finite systems. To determine the physical properties of the infinite system, finite-size scaling is performed with the results of these computations. Inspired by the recent developments for classical phase transitions [15,16,17,18,19,41], a theory for finite-size scaling above the upper critical dimension for QPTs was introduced [32,34].

When investigating algebraically decaying long-range interactions $∼r^{-(d+\sigma )}$ with the distance *r* and the dimension *d* of the system, there are two distinct regimes: one for σ≤0 (strong long-range interaction) and another one for σ>0 (weak long-range interaction) [5,42,43,44,45]. In the case of strong long-range interactions, common definitions of internal energy and entropy in the thermodynamic limit are not applicable and standard thermodynamics breaks down [5,42,43,44,45]. We will not focus on this regime in this review. For details specific to strong long-range interactions, we refer to other review articles such as Refs. [5,42,43,44,45]. For the sake of this work, we restrict the discussion to weak long-range interaction or competing antiferromagnetic strong long-range interactions, for which an extensive ground-state energy can be defined without rescaling of the coupling constant [5].

The interest in quantum magnets with long-range interactions is further fuelled by the relevance of these models in state-of-the-art quantum–optical platforms [5,46,47,48,49,50,51,52,53,54,55,56,57,58,59,60,61,62,63,64,65,66,67,68,69,70,71,72,73,74,75,76,77,78,79,80,81,82,83,84,85,86,87]. To realise long-range interacting quantum lattice models with a tunable algebraic decay exponent, one can use trapped ions, which are coupled off-resonantly to motional degrees of freedom [5,81,82,83,84,85,88]. Another possibility is to couple trapped neutral atoms to photonic modes of a cavity [5,86,87]. Alternatively, one can realise long-range interactions decaying with a fixed algebraic decay exponent of six or three using Rydberg atom quantum simulators [46,47,48,49,50,51,52,53,54,55] or ultracold dipolar quantum atomic or molecular gases in optical lattices [56,57,58,59,60,61,62,63,64,65,66,67,68,69,70,71,72,73]. Note that, in many of the above-listed cases, it is possible to map the long-range interacting atomic degrees of freedom onto quantum spin models [5,52,89]. Therefore, they can be exploited as analogue quantum simulators for long-range interacting quantum magnets, and the relevance of the theoretical concepts transcends the boundary between the fields.

From the perspective of condensed matter physics, there are multiple materials with relevant long-range interactions [90,91,92,93,94,95,96,97,98,99,100,101,102,103,104,105,106]. The compound LiHoF_4_ in an external field realises an Ising magnet in a transverse magnetic field [102,103,104,105]. A recent experiment with the two-dimensional Heisenberg ferromagnet Fe_3_GeTe_2_ demonstrates that phase transitions and continuous symmetry breaking can be implemented by circumventing the Hohenberg–Mermin–Wagner theorem with long-range interactions [106]. This material is in the recently discovered material class of 2D magnetic van der Waals systems [107,108]. Further, dipolar interactions play a crucial role in the spin ice state in the frustrated magnetic pyrochlore materials Ho_2_Ti_2_O_7_ and Dy_2_Ti_2_O_7_ [90,91,92,93,94,95,96,97,98,99,100,101].

In this review, we are interested in physical systems described by quantum spin models, where the magnetic degrees of freedom are located on the sites of a lattice. We concentrate on the following three paradigmatic types of magnetic interactions between lattice sites: first, Ising interactions, where the magnetic interaction is oriented only in the direction of one quantisation axis; second, XY interactions with a U(1)-symmetric magnetic interaction invariant under planar rotations; and third, Heisenberg interactions with a SU(2)-symmetric magnetic interaction invariant under rotations in 3D spin space. In the microscopic models of interest, a competition between magnetic ordering and trivial product states, external fields, or quasi-long-range order leads to QPTs.

In this context, the primary research pursuit revolves around how the properties of the QPT depend on the long-range interaction. The upper critical dimension of a QPT in magnetic models with non-competing algebraically decaying long-range interactions is known to depend on the decay exponent of the interaction for a small enough exponent, and decreases as the decay exponent decreases [20,21]. If the dimension of a system is equal to or exceeds the upper critical dimension, the QPT displays mean-field critical behaviour. At the same time, standard finite-size scaling, as well as standard hyperscaling relations are no longer applicable. Therefore, these systems are primary workhorse models to study finite-size scaling above the upper critical dimension. In this case, the numerical simulation of these systems is crucial in order to gauge novel theoretical developments. Further, QPTs in systems with competing long-range interactions do not tend to depend on the long-range nature of the interaction [23,24,25,26,29,30,32]. In several cases, long-range interactions then lead to the emergence of ground states and QPTs, which are not present in the corresponding short-range interacting models [27,30,54,55,109,110,111,112].

In this review, we are mainly interested in the description and discussion of two Monte Carlo-based numerical techniques, which were successfully used to study the low-energy physics of long-range interacting quantum magnets, in particular with respect to the quantitative investigation of QPTs [22,25,29,30,31,32,34,35,36,37,38,40]. The success of Monte Carlo techniques in this field is due to the occurrence of high-dimensional sums and integrals that commonly arise in the formulation of many-particle statistics. In contrast to many deterministic integration techniques, for which the standard error scales exponentially with the dimension of the underlying integral, the standard error of an integral calculated with Monte Carlo integration does not scale with the dimension of the underlying integral. We further chose to review this topic due to our personal involvement with the application and development of these methods [25,29,30,31,32,34,35]. On the one hand, we explain in detail how classical Monte Carlo integration can enhance the capabilities of linked-cluster expansions (LCEs) with the pCUT+MC approach (a combination of the perturbative unitary transform approach (pCUT) and MC embedding). On the other hand, we describe how stochastic series expansion (SSE) quantum Monte Carlo (QMC) integration is used to directly sample the thermodynamic properties of suitable long-range quantum magnets on finite systems.

This review is structured as follows. In Section 2, we review the basic concept of a QPT in a condensed way, focusing on the details relevant for this review. We define the quantum-critical exponents and the relations between them in Section 2.1. Here, we also have the first encounter with the generalised hyperscaling relation, which is also valid above the upper critical dimension where conventional hyperscaling breaks down. As the SSE QMC method discussed in this review is a finite-system simulation, we discuss the conventional finite-size scaling below the upper critical dimension in Section 2.2 and the peculiarities of finite-size scaling above the upper critical dimension in Section 2.3. In Section 3, we summarise the basic concepts of Markov chain Monte Carlo integration: Monte Carlo sampling, Markovian random walks, stationary distributions, the detailed balance condition, and the Metropolis–Hastings algorithm. We continue by introducing the series-expansion Monte Carlo embedding method pCUT+MC in Section 4. We start with the basic concepts of a graph expansion in Section 4.1 and introduce the perturbative method of our choice, the perturbative continuous unitary transformation method, in Section 4.2. We introduce the theoretical concepts for setting up a linked-cluster expansion as a full graph decomposition in Section 4.3 and, subsequently, discuss how to practically calculate perturbative contributions in Section 4.4 and Section 4.5. We prepare the discussion of the white graph decomposition in Section 4.6 with an interlude on the relevant graph theory in Section 4.6.1 and Section 4.6.2 and the important concept of white graphs in Section 4.6.3. Further, in Section 4.7, we discuss the embedding problem for the white graph contributions. Starting from the nearest-neighbour embedding problem in Section 4.7.1, we generalise it to the long-range case in Section 4.7.2 and then introduce a classical Monte Carlo algorithm to calculate the resulting high-dimensional sums in Section 4.7.3. This is followed by some technical aspects on series extrapolations in Section 4.8 and a summary of the entire workflow in Section 4.9. In the next section, the topic changes towards the review of the SSE QMC method, which is an approach to simulate thermodynamic properties of suitable quantum many-body systems on finite systems at a finite temperature. First, we discuss the general concepts of the method in Section 5. We review the algorithm to simulate arbitrary transverse-field Ising models introduced by A. Sandvik [39] in Section 5.1. We then review an algorithm used to simulate non-frustrated Heisenberg models in Section 5.2. After the introduction to the algorithms, we summarise techniques on how to measure common observables in the SSE QMC scheme in Section 5.3. Since the SSE QMC method is a finite-temperature method, we discuss how to rigorously use this scheme to perform simulations at effective zero temperature in Section 5.4. We conclude this section with a brief summary of path integral Monte Carlo techniques used for systems with long-range interactions (see Section 5.5). To maintain the balance between algorithmic aspects and their physical relevance, we summarise several theoretical and numerical results for quantum phase transitions in basics long-range interacting quantum spin models, for which the discussed Monte Carlo-based techniques provided significant results. First, we discuss long-range interacting transverse-field Ising models in Section 6. For ferromagnetic interactions, this model displays three regimes of universality: a long-range mean-field regime for slowly decaying long-range interactions, an intermediate long-range non-trivial regime, and a regime of short-range universality for strong decaying long-range interactions. We discuss the theoretical origins of this behaviour in Section 6.1.1 and numerical results for quantum critical exponents in Section 6.1.2. Since this model is a prime example to study scaling above the upper critical dimension in the long-range mean-field regime, we emphasise these aspects in Section 6.1.3. Further, we discuss the antiferromagnetic long-range transverse-field Ising model on bipartite lattices in Section 6.2 and on non-bipartite lattices in Section 6.3. The next obvious step is to change the symmetry of the magnetic interactions. Therefore, we turn to long-range interacting XY models in Section 7 and Heisenberg models in Section 8. We discuss the long-range interacting transverse-field XY chain in Section 7 starting with the U(1)-symmetric isotropic case in Section 7.1, followed by the anisotropic case for ferromagnetic (see Section 7.2) and antiferromagnetic (see Section 7.3) interactions, which display similar behaviour to the long-range transverse-field Ising model on the chain discussed in Section 6. We conclude the discussion of the results with unfrustrated long-range Heisenberg models in Section 8. We focus on the staggered antiferromagnetic long-range Heisenberg square lattice bilayer model in Section 8.1 followed by long-range Heisenberg ladders in Section 8.2 and the long-range Heisenberg chain in Section 8.3. We conclude in Section 9 with a brief summary and with some comments on the next possible steps in the field.

## 2. Quantum Phase Transitions

This review is part of the Special Issue with the topic “Violations of Hyperscaling in Phase Transitions and Critical Phenomena”. In this work, we summarise investigations of low-dimensional quantum magnets with long-range interactions targeting, in particular, quantum phase transitions (QPTs) above the upper critical dimension, where the naive hyperscaling relation is no longer applicable. In this section, we recapitulate the relevant aspects of QPTs needed to discuss the results of the Monte Carlo-based numerical approaches. First, we give a general introduction to QPTs. After that, we discuss in detail the definition of critical exponents and the relations among them in Section 2.1, as well as the scaling below (see Section 2.2) and above (see Section 2.3) the upper critical dimension.

Any non-analytic point of the ground-state energy of an infinite quantum system as a function of a tuning parameter λ is identified with a QPT [113]. This tuning parameter can, for instance, be a magnetic field or pressure, but not the temperature. Quantum phase transitions are a concept of zero temperature as there are no thermal fluctuations and all excited states are suppressed infinitely strong such that the system remains in its ground state. There are two scenarios for how a non-analytic point in the ground-state energy can emerge [113]: First is an actual (sharp) level crossing between the ground-state energy and another energy level. Second, the non-analytic point can be considered as a limiting case of an avoided level crossing. Historically, phase transitions are classified by the lowest order derivative of the free energy that is discontinuous [113,114]. Therefore, a first-order phase transition is discontinuous in the order parameter (first derivative) and a second-order phase transition is discontinuous in the response functions (second derivative). Since, in second-order phase transitions, the order parameter is still continuous across the phase transition, we use the term “continuous phase transition” as an equivalent for “second-order phase transition”.

In this review, we are interested in second-order QPTs, which fall into the second scenario. At a second-order QPT, the relevant elementary excitations condense into a novel ground state, while the characteristic length and time scales diverge. Apart from topological QPTs involving a long-range entangled topological phase, such continuous transitions are described by the concept of spontaneous symmetry breaking. On one side of the QPT, the ground state obeys a symmetry of the Hamiltonian, while on the other side, this symmetry is broken in the ground state and a ground-state degeneracy arises.

Following the idea of the quantum-to-classical mapping [113,115], *d*-dimensional quantum systems can be mapped in the vicinity of a second-order QPT to models of statistical mechanics with a classical (thermal) second-order phase transition in d+1 dimensions. In many cases, the models obtained from a quantum-to-classical mapping are rather artificial [113]. However, such mappings often allow categorising QPTs in terms of universality classes and associated critical exponents by the non-analytic behaviour of the classical counterparts [10,11,113,116,117]. The mapping further illustrates that the renormalisation group (RG) theory is also applicable to describe QPTs [10,11,113,116,117].

In the RG theory, each QPT belongs to a non-trivial fixed point of the RG transformation [10,11], whereas a trivial fixed point would, for instance, be a fully ordered state with maximal correlation or a fully disordered state with no correlation at all. Critical exponents are connected to the RG flow in the immediate vicinity of these non-trivial fixed points [10,11,113]. The concept of universality classes arises from the fact that different microscopic Hamiltonians can have a quantum critical point that is attracted by the same non-trivial fixed point under successive application of the RG transformation [10,11]. Due to this, the QPTs in these models have the same critical exponents.

Another remarkable result of the RG theory is the scaling of observables in the vicinity of phase transitions. Historically, the theory of scaling at phase transitions was heuristically introduced before the RG approach [118,119,120,121,122,123,124]. The latter provided the theoretical foundation for the scaling hypothesis [6,7]. The main statement of the scaling theory is that the non-analytic contributions to the free energy and correlation functions are mathematically described by generalised homogeneous functions (GHFs) [124]. A function with *n* variables $f(x_{1},x_{2},\cdots ,x_{n})$ is called a GHF, if there exist $a_{1},a_{2},\cdots ,a_{n}\in R$ with at least one being non-zero and $a_{f}\in R$ such that, for $b\in R_{+}$,
(1)$$\begin{array}{c} f\left(b^{a_{1}}x_{1},b^{a_{2}}x_{2},\cdots ,b^{a_{n}}x_{n}\right)=b^{a_{f}}f\left(x_{1},x_{2},\cdots ,x_{n}\right)\;. \end{array}$$ The exponents $a_{1},a_{2},\cdots ,a_{n}$ are the scaling powers of the variables, and $a_{f}$ is the scaling power of the function *f* itself. An in-depth summary of the mathematical properties of GHFs can be found in Appendix B. The most important properties of GHFs are that their derivatives, Legendre transforms, and Fourier transforms are also GHFs. As we will outline in Section 2.2, the theory of finite-size scaling is formulated in terms of GHFs and relates the non-analytic behaviour at QPTs in the thermodynamic limit with the scaling of observables for different finite system sizes. In this, the variables $x_{i}$ are related to physical parameters like the temperature *T*, control parameter λ, symmetry-breaking field *H*, and also irrelevant, more abstract parameters that parameterise the microscopic details of the model like the lattice spacing. Later in this section, we will define irrelevant variables in the context of the RG and GHFs.

Another aspect relevant for this work is that quantum fluctuations are the driving factor with QPTs [113]. In general, fluctuations are more important in low dimensions [117]. The universality class of QPTs for a certain symmetry breaking depends on the dimensionality of the system.

An important aspect regarding this review is the so-called upper critical dimension $d_{\mathrm{uc}}$. The upper critical dimension is defined as a dimensional boundary such that, for systems with dimension $d\geq d_{\mathrm{uc}}$, the critical exponents are those obtained from mean-field considerations. The upper critical dimension is of particular importance for QPTs in systems with non-competing long-range interactions. For sufficiently small decay exponents of an algebraically decaying long-range interaction $1/r^{d+\sigma }$, the upper critical dimension starts to decrease as a function of the decay exponent σ [20,32,34]. In the limiting case of a completely flat decay (σ=−d) of the long-range interaction resulting in an all-to-all coupling, the model is intrinsically of the mean-field type and mean-field considerations become exact. For a certain value of the decay exponent, the upper critical dimension becomes equal to the fixed spatial dimension, and for decay exponents below this value, the dimension of the system is above the upper critical dimension of the transition [20,32,34]. This makes long-range interacting systems an ideal test bed for studying phase transitions above the upper critical dimension in low-dimensional systems. In particular, long-range interactions can make the upper critical dimension accessible in real-world experiments as the upper critical dimension of short-range models is usually not below three.

Although phase transitions above the upper critical dimension display mean-field criticality, they are still a matter worth studying, since naive scaling theory describing the behaviour of finite systems close to a phase transition (see Section 2.2) is no longer applicable [16,19,125]. Moreover, the naive versions of some relations between critical exponents, as discussed in Section 2.1, do not hold any longer [15,16]. The reason for this issue are the dangerous irrelevant variables (DIVs) in the RG framework [126,127,128]. During the application of the RG transformation, the original Hamiltonian is successively mapped to other Hamiltonians, which can have infinitely many couplings. All these couplings, in principle, enter the GHFs. In practice, all but a finite number of these couplings are set to zero since their scaling powers are negative, which means they flow to zero under renormalisation. These couplings are, therefore, called irrelevant. This approach of setting irrelevant couplings to zero can be used to derive the finite-size scaling behaviour as described in Section 2.2. However, above the upper critical dimension, this approach breaks down because it is only possible to set irrelevant variables to zero if the GHF does not have a singularity in this limit [126]. Above the upper critical dimension, such singularities in irrelevant parameters exist, which makes them DIVs [127]. We explain the effect of DIVs on scaling in Section 2.3.

### 2.1. Critical Exponents in the Thermodynamic Limit

As outlined above, a second-order QPT comes with a singularity in the free energy density. In fact, also, other observables experience singular behaviour at the critical point in the form of power-law singularities. For instance, the order parameter *m* as a function of the control parameter λ behaves as
(2)$$m\left(\lambda \to \lambda_{c}^{-}\right)∼{\left|\lambda -\lambda_{c}\right|}^{\beta }$$
in the ordered phase. Without loss of generality, the system is taken to be in the ordered phase for $\lambda <\lambda_{c}$ and the notation $\lambda \to \lambda_{c}^{-}$ means that λ is approaching $\lambda_{c}$ from below, i.e., it is approaching in the ordered phase. In the disordered phase $\lambda >\lambda_{c}$, the order parameter by definition vanishes such that $m(\lambda \to \lambda_{c}^{+})=0$. The observables with their respective power-law singular behaviour, which is characterised by the critical exponents α,β,γ,δ,η,ν, and *z*, are summarised in Table 1 together with how they are commonly defined in terms of the free energy density *f*, the symmetry-breaking field *H*, which couples to the order parameter, and the reduced control parameter $r=\left(\lambda -\lambda_{c}\right)/\lambda_{c}$.

One usually defines reduced parameters like *r* that vanish at the critical point not only to shorten the notation, but also to express the power-law singularities independent of the microscopic details of the specific model one is looking at. While the value of $\lambda_{c}$ depends on these details, the power-law singularities are empirically known to not depend on the microscopic details, but only on more general properties like the dimensionality, the symmetry that is being broken, and, with particular emphasis due to the focus of this review, on the range of the interaction. It is, therefore, common to classify continuous phase transitions in terms of universality classes. These universality classes share the same set of critical exponents. In terms of the RG, this behaviour is understood as distinct critical points of microscopically different models flowing to the same renormalisation group fixed point, which determines the criticality of the system [6,7,10]. Prominent examples for universality classes of magnets are the 2D and 3D Ising ($Z_{2}$ symmetry), 3D XY (O(2) symmetry), and 3D Heisenberg (O(3) symmetry) universality classes [113]. It is important to mention that the dimension in the classifications is referring to classical and not quantum systems, and they should not be confused with each other. In fact, the universality class of a short-range interacting non-frustrated quantum Ising model of dimension *d* lies in the universality of the classical d+1-dimensional Ising model.

There are only a few dimensions for which a separate universality class is defined for the different models. For lower dimensions, the fluctuations are too strong in order for a spontaneous symmetry breaking to occur. In the case of the classical Ising model, there is no phase transition for 1D, while for the classical XY and Heisenberg models with continuous symmetries, there is not even a phase transitions for 2D due to the Hohenberg–Mermin–Wagner (HWM) theorem [130,131]. This dimensional boundary is referred to as lower critical dimension $d_{\mathrm{lc}}$. The lower critical dimension is the highest dimension for which no transition occurs, i.e., $d_{\mathrm{lc}}=1$ for the Ising model and $d_{\mathrm{lc}}=2$ for the XY and Heisenberg model. For higher dimensions d≥4, the critical exponents of the mentioned models do not depend on the dimensionality any longer, and they take on the mean-field critical exponents in all dimensions. The underlying reason is that, with increasing dimensions, the local fluctuations become smaller due to the higher connectivity of the system [132]. This has been also exploited in diagrammatic and series expansions in 1/d [133,134,135]. This dimensional boundary, at which the criticality becomes the mean-field one, is called upper critical dimension $d_{\mathrm{uc}}$. Usually, the upper critical dimension is too large to realise a system above its upper critical dimension in the real world. However, long-rang interactions can increase the connectivity of a system in a similar sense as the dimensionality. A sufficiently long-range interaction can, therefore, lower the upper critical dimension to a value that is accessible in experiments.

Finally, it is worth mentioning that the critical exponents are not independent of each other, but obey certain relations [129], namely
(3)$$\begin{array}{cc} 2-\alpha & =(d+z)\nu \;, \end{array}$$
(4)$$\begin{array}{cc} 2-\alpha & =2\beta +\gamma \;, \end{array}$$
(5)$$\begin{array}{cc} \gamma & =\beta (\delta -1)\;, \end{array}$$
(6)$$\begin{array}{cc} \gamma & =(2-\eta )\nu \;. \end{array}$$ The first relation in Equation (Equation 3) is the so-called hyperscaling relation, whose classical analogue (without *z*) was introduced by Widom [10,136]. The Essam–Fisher relation in Equation (Equation 4) [137,138] is reminiscent of a similar inequality, which was proven rigorously by Rushbrooke using thermodynamic stability arguments. Equation (Equation 5) is called the Widom relation. The last relation in Equation (Equation 6) is the Fisher scaling relation, which can be derived using the fluctuation–dissipation theorem [10,129,138]. Those relations were originally obtained from scaling assumptions of observables close to the critical point, which were only later derived rigorously when the RG formalism introduced to critical phenomena [10,129]. Due to these relations, it is sufficient to calculate three, albeit not arbitrary, exponents to obtain the full set of critical exponents.

The hyperscaling relation Equation (Equation 3) is the only relation containing the dimension of the system and is, therefore, often said to break down above the upper critical dimension, where one expects the same mean-field critical exponents independent of the dimension *d* [10]. It, therefore, deserves special focus in this review since the long-range models discussed will be above the upper critical dimension in certain parameter regimes. Personally, we would not agree that the hyperscaling relation breaks down above the upper critical dimension, but we would rather call Equation (Equation 3) a special case of a more general hyperscaling relation: (7)$$2-\alpha =\left(\frac{d}{ϙ}+z\right)\nu ,$$
with the pseudo-critical exponent ϙ (“koppa”) [34]: (8)$$ϙ=\left\{\begin{array}{cc} 1 & \;\mathrm{for}\;d\leq d_{\mathrm{uc}}\\ \frac{d}{d_{\mathrm{uc}}} & \;\mathrm{for}\;d>d_{\mathrm{uc}}\;. \end{array}\right.$$ Below the upper critical dimension, the general hyperscaling relation, therefore, relaxes to Equation (Equation 3). Above the upper critical dimension, the relation becomes
(9)$$2-\alpha =\left(d_{\mathrm{uc}}+z\right)\nu ,$$
which is independent of the dimension of the system. For the derivation of this generalised version of the hyperscaling relation for QPTs, see Section 2.3 or Ref. [34]. The derivation of the classical counterpart can be found in Ref. [15] and is reviewed in Ref. [41].

### 2.2. Finite-Size Scaling below the Upper Critical Dimension

Even though the singular behaviour of observables at the critical point is only present in the thermodynamic limit, it is possible to study the criticality of an infinite system by investigating their finite counterparts. In finite systems, the power-law singularities of the infinite system are rounded and shifted with respect to the critical point, e.g., the susceptibility with its characteristic divergence at the critical point $\lambda_{c}$ is deformed to a broadened peak of finite height. The peak’s position $r_{L}=\left(\lambda_{L}-\lambda_{c}\right)/\lambda_{c}$ is shifted with respect to the critical point r=0. A possible definition of a pseudo-critical point of a finite system is the peak position $\lambda_{L}$. As the characteristic length scale of fluctuations ξ diverges at the critical point, the finite system approaching the critical point will at some point begin to “feel” its finite extent and the observables start to deviate from the ones in the thermodynamic limit. As ξ diverges with the exponent ν like $\xi ∼{\left|r\right|}^{-\nu }$ at the critical point, the extent of rounding in a finite system is related to the value of ν. Similarly, the peak magnitude of finite-size observables at the pseudo-critical point will depend on how strong the singularity in the thermodynamic limit is, which means it depends on the respective critical exponents α, β, γ, and δ. The shifting, rounding, and varying peak magnitude are shown for the susceptibility of the long-range transverse-field Ising model in Figure 1. This dependence of observables in finite systems on the criticality of the infinite system is the basis of finite-size scaling.

In a more mathematical sense, the relation between critical exponents and finite-size observables has its origin in the renormalisation group (RG) flow close to the corresponding RG fixed point that determines the criticality [139]. Close to this fixed point, one can linearise the RG flow so that the free energy density and the characteristic length ξ become generalised homogeneous functions (GHFs) in their parameters [124,128,129,140,141]. For a thorough discussion of the mathematical properties of GHFs, we refer to Ref. [124] and Appendix B. This means that the free energy density *f* and characteristic length scale ξ as functions of the couplings r,H,T,u and the inverse system length $L^{-1}$ obey the relations:(10)$$\begin{array}{cc} f(r,H,T,L^{-1},u) & =b^{-(d+z)}f\left(b^{y_{r}}r,b^{y_{H}}H,b^{z}T,bL^{-1},b^{y_{u}}u\right) \end{array}$$(11)$$\begin{array}{cc} \xi (r,H,T,L^{-1},u) & =b\xi (b^{y_{r}}r,b^{y_{H}}H,b^{z}T,bL^{-1},b^{y_{u}}u) \end{array}$$
with the respective scaling dimensions $y_{r},y_{H},z>0$, $y_{L}=1$, and $y_{u}<0$ governing the linearised RG flow with spatial rescaling factor b>1 around the RG fixed point, at which all couplings vanish by definition. All of those couplings are relevant except for *u*, which denotes the leading irrelevant coupling [10,113]. Relevant couplings are related to real-world parameters that can be used to tune our system away from the critical point like the temperature *T*, a symmetry-breaking field *H*, or simply the control parameter *r*. The irrelevant couplings *u* do not per se vanish at the critical point like the relevant ones do. However, they flow to zero under the RG transformation and are commonly set to zero in the scaling laws: (12)$$\begin{array}{cc} f(r,H,T,L^{-1}) & =b^{-(d+z)}f\left(b^{y_{r}}r,b^{y_{H}}H,b^{z}T,bL^{-1}\right) \end{array}$$(13)$$\begin{array}{cc} \xi (r,H,T,L^{-1}) & =b\xi (b^{y_{r}}r,b^{y_{H}}H,b^{z}T,bL^{-1}) \end{array}$$
by assuming analyticity in these parameters. The generalised homogeneity of thermodynamic observables can be derived from the one of the free energy density *f*. For example, the generalised homogeneity of the magnetisation:(14)$$\begin{array}{cc} m(r,H,T,L^{-1}) & =b^{-\left(d+z\right)+y_{H}}m\left(b^{y_{r}}r,b^{y_{H}}H,b^{z}T,bL^{-1}\right) \end{array}$$
can be derived by taking the derivative of *f* with respect to the symmetry-breaking field *H*.

By investigating the singularity of $\xi \left(r\right)=\xi \left(r,H=0,T=0,L^{-1}=0\right)$ in *r* via Equation (Equation 13), one can show that the scaling power $y_{r}$ of the control parameter *r* is related to the critical exponent ν by $y_{r}=1/\nu$ [113]. For this, one fixes the value $b^{y_{r}}r$ of the first argument to ±1 in the right-hand side of Equation (Equation 13) by setting $b={\left|r\right|}^{-1/y_{r}}$ such that
(15)$$\begin{array}{cc} \xi (r) & ={\left|r\right|}^{-1/y_{r}}\xi \left(\pm 1\right)∼{\left|r\right|}^{-\nu }\;. \end{array}$$ Analogously, further relations between the scaling powers and other critical exponents can be derived by looking at the singular behaviour of the respective observables in the corresponding parameters. Overall, one further obtains
(16)$$\alpha =-\frac{d+z-2y_{r}}{y_{r}}\;,\;\beta =\frac{d+z-y_{H}}{y_{r}}\;,\;\delta =\frac{y_{H}}{d+z-y_{H}}\;,\;\gamma =-\frac{d+z-2y_{H}}{y_{r}}\;.$$ From these equations, one can already tell that the critical exponents are not independent of each other. In fact, the scaling relations 2−α=(d+z)ν, 2−α=2β+γ and γ=β(δ−1) (see Equations (Equation 3)–(Equation 5)) can be derived from Equation (Equation 16) and $y_{r}^{-1}=\nu$. By expressing the RG scaling powers $y_{x}$ in terms of critical exponents, the homogeneity law for an observable O with a bulk divergence $O\left(r,0,0,0\right)∼{\left|r\right|}^{\omega }$ is given by
(17)$$\begin{array}{cc} O(r,H,T,L^{-1}) & =b^{-\omega y_{r}}O\left(b^{y_{r}}r,b^{y_{H}}H,b^{z}T,bL^{-1}\right) \end{array}$$
(18)$$\begin{array}{cc} \; & =b^{-\omega /\nu }O\left(b^{1/\nu }r,b^{(\beta +\gamma )/\nu }H,b^{z}T,bL^{-1}\right)\;. \end{array}$$

In order to investigate the dependence on the linear system size *L*, the last argument in the homogeneity law is fixed to $bL^{-1}=1$ by inserting b=L. This readily gives the finite-size scaling form
(19)$$\begin{array}{cc} O_{L}\left(r,H,T\right) & =L^{-\omega /\nu }\Psi \left(L^{1/\nu }r,L^{(\beta +\gamma )/\nu }H,L^{z}T\right) \end{array}$$
with Ψ being the universal scaling function of the observables O. The scaling function Ψ itself does not depend on *L* any longer, but in order to compare different linear system sizes, one has to rescale its arguments. To extract the critical exponents from finite systems, the observable $O_{L}\left(r,H,T\right)$ is measured for different system sizes *L* and parameters (r,H,T) close to the critical point (r,H,T)=(0,0,0). The *L*-dependence according to Equation (Equation 19) is then fit with the critical exponents ω, ν, β+γ, and *z*, as well as the critical point $\lambda_{c}$, which is hidden in the definition of *r*, as free parameters. It is advisable to fix two of the three parameters r,H,T to their critical values in order to minimise the amount of free parameters in the fit. For example, with H=T=0 and only r≠0, one can extract the two critical exponents ω and ν alongside $\lambda_{c}$. For further details on a fitting procedure, we refer to Ref. [32]. If one knows the critical point, one can also set (r,H,T)=(0,0,0) and look at the *L*-dependent scaling $O_{L}∼L^{-\omega /\nu }$ directly at the critical point to extract the exponent ratio ω/ν. There are many more possible approaches to extract critical exponents from the FSS law in Equation (Equation 19) [142,143,144]. For relatively small system sizes, it might be required to take corrections to scaling into account [143,144].

### 2.3. Finite-Size Scaling above the Upper Critical Dimension

In the derivation of finite-size scaling below the upper critical dimension, it was assumed that the free energy density is an analytic function in the leading irrelevant coupling *u* and, therefore, one can set it to zero. However, this is not the case above the upper critical dimension any longer, and the free energy density *f* is singular at u=0. Due to this singular behaviour *u* is referred to as a dangerous irrelevant variable (DIV).

As a consequence, one has to take the scaling of *u* close to the RG fixed point into account. This is achieved by absorbing the scaling of *f* in *u* for small *u* into the scaling of the other variables [128]:(20)$$f\left(r,H,T,L^{-1},u\right)=u^{p_{\left(d+z\right)}}\bar{f}\left(u^{p_{r}}r,u^{p_{H}}H,u^{p_{T}}T,u^{p_{L}}L^{-1}\right)\;,$$
up to a global power $p_{\left(d+z\right)}$ of *u*. This leads to a modification of the scaling powers in the homogeneity law for the free energy density [128]: (21)$$\begin{array}{cc} f(r,H,T,L^{-1}) & =b^{-{\left(d+z\right)}^{*}}f\left(b^{y_{r}^{*}}r,b^{y_{H}^{*}}H,b^{z^{*}}T,b^{y_{L}^{*}}L^{-1}\right) \end{array}$$(22)$$\begin{array}{cc} \; & =L^{-{\left(d+z\right)}^{*}/y_{L}^{*}}F\left(L^{y_{r}^{*}/y_{L}^{*}}r,L^{y_{H}^{*}/y_{L}^{*}}H,L^{z^{*}/y_{L}^{*}}T\right) \end{array}$$
with the modified scaling powers [34,128]:(23)$${\left(d+z\right)}^{*}=\left(d+z\right)-p_{\left(d+z\right)}y_{u}\;,\;\begin{array}{cccc} y_{r}^{*} & =y_{r}+p_{r}y_{u}\;,\; & y_{H}^{*} & =y_{H}+p_{H}y_{u}\;,\\ z^{*} & =z+p_{z}y_{u}\;,\; & y_{L}^{*} & =1+p_{L}y_{u}\;. \end{array}$$ In the classical case [128], $y_{L}^{*}$ was commonly set to 1 by choice. This is justified because the scaling powers of a GHF are only determined up to a common non-zero factor [124]. However, for the quantum case [34], this was kept general as it has no impact on the FSS.

As the predictions from the Gaussian field theory and mean field differed for the critical exponents α, β, and δ, but not for the “correlation” critical exponents ν, *z*, η, and γ [145], the correlation sector was thought not to be affected by DIVs at first [128,142,145]. Later, the Q-FSS, another approach to FSS above the upper critical dimension, pioneered by Ralph Kenna and his colleagues, was developed for classical [15,19,125], as well as for quantum systems [34], which explicitly allowed the correlation sector to also be affected by the DIV. In analogy to the free energy density, the homogeneity law of the characteristic length scale is then also modified to
(24)$$\begin{array}{cc} \xi (r,H,T,L^{-1}) & =b^{-y_{\xi }^{*}}\xi \left(b^{y_{r}^{*}}r,b^{y_{H}^{*}}H,b^{z^{*}}T,b^{y_{L}^{*}}L^{-1}\right) \end{array}$$
(25)$$\begin{array}{cc} \; & =L^{ϙ}\Xi \left(L^{y_{r}^{*}/y_{L}^{*}}r,L^{y_{H}^{*}/y_{L}^{*}}H,L^{z^{*}/y_{L}^{*}}T\right) \end{array}$$
with $y_{\xi }^{*}=-1-p_{\xi }y_{u}=-y_{r}^{*}/y_{r}$ in order to reproduce the correct bulk singularity $\xi ∼{\left|r\right|}^{-\nu }$. A new pseudo-critical exponent ϙ (“koppa”): (26)$$ϙ=-\frac{y_{\xi }^{*}}{y_{L}^{*}}=\frac{y_{r}^{*}}{y_{r}y_{L}^{*}}=\nu \frac{y_{r}^{*}}{y_{L}^{*}}$$
is introduced. This exponent describes the scaling of the characteristic length scale with the linear system size. This non-linear scaling of ξ with *L* is one of the key differences to the previous treatments above the upper critical dimension in Ref. [128].

Analogous to the case below the upper critical dimension, the modified scaling powers $y_{x}^{*}$ can be related to the critical exponents: (27)$$\begin{array}{cc} \alpha & =-\frac{{\left(d+z\right)}^{*}-2y_{r}^{*}}{y_{r}^{*}}\;, \end{array}$$(28)$$\begin{array}{cc} \beta & =\frac{{\left(d+z\right)}^{*}-y_{H}^{*}}{y_{r}^{*}}\;, \end{array}$$(29)$$\begin{array}{cc} \delta & =\frac{y_{H}^{*}}{{\left(d+z\right)}^{*}-y_{H}^{*}}\;, \end{array}$$(30)$$\begin{array}{cc} \gamma & =-\frac{{\left(d+z\right)}^{*}-2y_{H}^{*}}{y_{r}^{*}}\;. \end{array}$$ By using the mean-field critical exponents for the O(n) quantum rotor model, one obtains restrictions for the ratios of modified scaling powers:(31)$$\begin{array}{ccc} y_{r}^{*} & =\frac{{\left(d+z\right)}^{*}}{2}, & y_{H}^{*}=\frac{3{\left(d+z\right)}^{*}}{4} \end{array}$$

Furthermore, one can link the bulk scaling powers $y_{r}^{*},y_{H}^{*}$, and ${\left(d+z\right)}^{*}$ to the scaling power $y_{L}^{*}$ of the inverse linear system size [34]:(32)$$\begin{array}{c} {\left(d+z\right)}^{*}=y_{L}^{*}d+\frac{y_{r}^{*}}{y_{r}}z\;, \end{array}$$
by looking at the scaling of the susceptibility in a finite system [34,128]. This relation is crucial for deriving an FSS form above the upper critical dimension as the modified scaling power $y_{L}^{*}$, or rather, its relation to the other scaling powers determines the scaling with the linear system size *L*. For details on the derivation, we refer to Ref. [34]. We want to stress again that the scaling powers of GHFs are only determined up to a common non-zero factor [124]. Therefore, it is evident that one can only determine the ratios of the modified scaling powers, but not their absolute value. The absolute values are subject to choice. Different choices were discussed in Ref. [34], but these choices rather correspond to taking on different perspectives and have no impact on the FSS nor the physics.

From Equation (Equation 32) together with Equations (Equation 26) and (Equation 27), a generalised hyperscaling relation: (33)$$2-\alpha =\left(\frac{d}{ϙ}+z\right)\nu ,$$
can be derived. This also determines the pseudo-critical exponent: (34)$$ϙ=\frac{d}{d_{\mathrm{uc}}}\;\mathrm{for}\;d>d_{\mathrm{uc}}\;.$$

Finally, we can express the modified scaling powers in the FSS law for an observable O with power-law singularity $O\left(r,0,0,0\right)∼{\left|r\right|}^{\omega }$: (35)$$\begin{array}{cc} O(r,H,T,L^{-1}) & =b^{-\omega y_{r}^{*}}O\left(b^{y_{r}^{*}}r,b^{\left(\beta +\gamma \right)y_{r}^{*}}H,b^{z^{*}}T,b^{y_{L}^{*}}L^{-1}\right) \end{array}$$(36)$$\begin{array}{cc} \; & =L^{-\omega ϙ/\nu }\Psi \left(L^{ϙ/\nu }r,L^{(\beta +\gamma )ϙ/\nu }H,L^{z^{*}/y_{L}^{*}}T\right)\;. \end{array}$$ For ϙ=1 below the upper critical dimension, Equation (Equation 36) relaxes to the standard FSS law Equation (Equation 19). The scaling in temperature has not yet been studied for finite quantum systems above the upper critical dimension. However, in Ref. [34], it was conjectured that $z^{*}=y_{r}^{*}/y_{r}z$ based on Equation (Equation 32), which is also in agreement with *z* being the dynamical critical exponent that determines the space–time anisotropy $\xi_{\tau }∼\xi^{z}$, as we will shortly see. This means that the finite-size gap scales as $\Delta_{L}∼L^{-z^{*}/y_{L}^{*}}∼L^{-ϙz}$ with the system size [34]. Of particular interest is the scaling of the characteristic length scale above the upper critical dimension, for which the modified scaling law Equation (Equation 36) also holds with ω=−ν. Hence, the characteristic length scale in dependence of the control parameter *r* scales like
(37)$$\xi_{L}\left(r\right)=L^{ϙ}\Xi \left(L^{ϙ/\nu }r\right)\;$$
with the scaling function Ξ. Directly at the critical point r=0, this leads to $\xi_{L}∼L^{ϙ}$. Comparing this with the scaling of the inverse finite-size gap $\Delta_{L}^{-1}∼\xi_{L,\tau }∼\xi_{L}^{z}$ verifies that *z* still determines the space–time anisotropy. Prior to the Q-FSS [15,17], the characteristic length scale was thought to be bound by the linear system size *L* [128]. However, this was shown not to be the case by measurements of the characteristic length scale for the classical five-dimensional Ising model [14] and for long-range transverse-field Ising models [34].

For the latter, the data collapse of the correlation length according to Equation (Equation 37) is shown in Figure 2 as an example.

## 3. Monte Carlo Integration

In this section, we provide a brief introduction to Monte Carlo integration (MCI). We focus on the aspects of Markov chain MCI as the basis to formulate the white graph Monte Carlo embedding scheme of the pCUT+MC method in Section 4 and the stochastic series expansion (SSE) quantum Monte Carlo (QMC) algorithm in Section 5 in a self-contained fashion. MCI is the foundation for countless numerical applications, which require the integration over high-dimensional integration spaces. As this review has a focus on “Monte Carlo-based techniques for quantum magnets with long-range interactions”, we forward readers with a deeper interest in the fundamental aspects of MCI and Markov chains to Refs. [146,147,148].

MCI summarises numerical techniques to find estimators for integrals of functions f:C→R over an integration space C using random numbers. The underlying idea behind MCI is to estimate the integral, or the sum in the case of discrete variables, of the function *f* over the configuration space by an expectation value:(38)$$\begin{array}{c} I=\int_{C}d\omega \;f\left(\omega \right)=\int_{C}d\omega \;\frac{P(\omega )}{P(\omega )}f\left(\omega \right)=\int_{C}d\omega \;P\left(\omega \right)\tilde{f}\left(\omega \right)=\lim_{S\to \infty }\frac{1}{S}\sum_{i=1}^{i=S}\tilde{f}\left(\omega_{i}\right) \end{array}$$
with $\omega_{i}\in C$ sampled according to a probability density function $P:C\to R_{\geq 0}$ (PDF) and the function $\tilde{f}\left(\omega \right)=f\left(\omega \right)/P\left(\omega \right)$ reweighted by the PDF. A famous direct application of this idea is the calculation of the number “pi”, which is discussed in great detail in Ref. [148].

In this review, MCI is used for the embedding of white graphs on a lattice to evaluate high orders of a perturbative series expansion or to calculate thermodynamic observables using the SSE framework. In both cases, non-normalised relative weights π(ω) within a configuration space C arise, which are used for the sampling of the PDF *P*:(39)$$\begin{array}{c} P\left(\omega \right)=\frac{\pi (\omega )}{\int_{C}d\omega \;\pi \left(\omega \right)}\;, \end{array}$$
being oftentimes not directly accessible. In the context of statistical physics, π(ω) is often chosen to be the relative Boltzmann weight $e^{-\beta E(\omega )}$ of each configuration ω. While this relative Boltzmann weight is accessible as long as E(ω) is known, the full partition function to normalise the weights is in general not.

In order to efficiently sample the configuration space C according to the relative weights, the methods in this review use a Markovian random walk to generate $\left\{\omega_{1},\cdots ,\omega_{m}\right\}$. Let $\omega_{n}$ be the random state of a random walk at a discrete step *n*. The state $\omega_{n+1}$ at the next step is randomly determined according to the conditional probabilities $T(\omega \to \omega^{\prime })$ (transition probabilities). These transition probabilities are normalised by
(40)$$\begin{array}{c} \int_{C}d\omega^{\prime }\;T\left(\omega \to \omega^{\prime }\right)=1\;. \end{array}$$ Markovian random walks obey the Markov property, which means the random walk is memory-free and the transition probability for multiple steps factorises into a product over all time steps:(41)$$\begin{array}{c} T\left(\omega^{\left(0\right)}\to \omega^{\left(1\right)}\to \cdots \to \omega^{\left(m-1\right)}\to \omega^{\left(m\right)}\right)=\prod_{i=0}^{m-1}T\left(\omega^{\left(i\right)}\to \omega^{\left(i+1\right)}\right) \end{array}$$
with $\omega^{\left(0\right)}$ the start configuration. We require the Markovian random walk to fulfil the following conditions: First, the random walk should have a certain PDF P(ω) defined by the weights π(ω) in Equation (Equation 39) as a stationary distribution. By definition, P(ω) is a stationary distribution of the Markov chain if it satisfies the global balance condition:(42)$$\begin{array}{c} \int_{C}d\omega \;P\left(\omega \right)T\left(\omega \to \omega^{\prime }\right)=P\left(\omega^{\prime }\right)\;. \end{array}$$ Second, we require the random walk to be irreducible, which means that the transition graph must be connected and every configuration ω∈C can be reached from any configuration $\omega^{\prime }\in C$ in a finite number of steps. This property is necessary for the uniqueness of the stationary distribution [147]. Lastly, we require the random walk to be aperiodic (see Ref. [147] for a rigorous definition). Together with the irreducibility condition, this ensures convergence to the stationary distribution [147].

There are several possibilities to design a Markov chain with a desired stationary distribution [146,147,148,149,150,151]. Commonly, the Markov chain is constructed to be reversible. This means that it satisfies the detailed balance condition:(43)$$\begin{array}{c} P\left(\omega \right)T\left(\omega \to \omega^{\prime }\right)=P\left(\omega^{\prime }\right)T\left(\omega^{\prime }\to \omega \right)\;, \end{array}$$
which is a stronger condition for the stationarity of *P* than the global balance condition in Equation (Equation 42). One popular choice for the transition probabilities $T(\omega \to \omega^{\prime })$ that satisfies the detailed balance condition is given by the Metropolis–Hastings algorithm. Most applications of MCI reviewed in this work are based on the Metropolis–Hastings algorithm [149,150]. In this approach, the transition probabilities $T(\omega \to \omega^{\prime })$ are decomposed into propositions $\tilde{T}\left(\omega \to \omega^{\prime }\right)$ and acceptance probabilities $p_{\mathrm{acc}}\left(\omega \to \omega^{\prime }\right)$ as follows:(44)$$\begin{array}{c} T\left(\omega \to \omega^{\prime }\right)=\tilde{T}\left(\omega \to \omega^{\prime }\right)p_{\mathrm{acc}}\left(\omega \to \omega^{\prime }\right)\;. \end{array}$$ The probabilities to propose a move $\tilde{T}\left(\omega \to \omega^{\prime }\right)$ can be any random walk satisfying the irreducibility and aperiodicity condition. By inserting the decomposition of the transition probabilities Equation (Equation 44) into the detailed balance condition Equation (Equation 43), one obtains for the acceptance probabilities:(45)$$\begin{array}{cc} \frac{p_{\mathrm{acc}}\left(\omega \to \omega^{\prime }\right)}{p_{\mathrm{acc}}\left(\omega^{\prime }\to \omega \right)} & =\frac{P\left(\omega^{\prime }\right)\tilde{T}\left(\omega^{\prime }\to \omega \right)}{P\left(\omega \right)\tilde{T}\left(\omega \to \omega^{\prime }\right)}=\frac{\pi \left(\omega^{\prime }\right)\tilde{T}\left(\omega^{\prime }\to \omega \right)}{\pi \left(\omega \right)\tilde{T}\left(\omega \to \omega^{\prime }\right)}\;, \end{array}$$
where, in the last step, the idea that the unknown normalisation factors (see Equation (Equation 39)) of the PDF cancel was used. The condition in Equation (Equation 45) is fulfilled by the Metropolis–Hastings acceptance probabilities [150]:(46)$$\begin{array}{c} p_{\mathrm{acc}}\left(\omega \to \omega^{\prime }\right)=\min\left(1,\frac{\pi \left(\omega^{\prime }\right)\tilde{T}\left(\omega^{\prime }\to \omega \right)}{\pi \left(\omega \right)\tilde{T}\left(\omega \to \omega^{\prime }\right)}\right)\;. \end{array}$$ For the special case, for which the proposition probabilities are symmetric $\tilde{T}\left(\omega \to \omega^{\prime }\right)=\tilde{T}\left(\omega^{\prime }\to \omega \right)$, Equation (Equation 46) reduces to the Metropolis acceptance probabilities:(47)$$\begin{array}{c} p_{\mathrm{acc}}\left(\omega \to \omega^{\prime }\right)=\min\left(1,\frac{\pi (\omega^{\prime })}{\pi (\omega )}\right)\;. \end{array}$$

As an example, we regard a classical thermodynamic system with Boltzmann weights $e^{-\beta E(\omega )}$ given by the energies of configurations and the inverse temperature to give an intuitive interpretation of the Metropolis acceptance probabilities in Equation (Equation 47). The proposition to move from a configuration ω to a configuration $\omega^{\prime }$ with a smaller energy $E\left(\omega^{\prime }\right)<E\left(\omega \right)$ is always accepted independent of the temperature. On the other hand, the proposition to move to a configuration $\omega^{\prime }$ with a larger energy than ω is only accepted with a probability depending on the ratio of the Boltzmann weights. If the temperature is higher, it is more likely to move to states with a larger energy. This reflects the physics of the system in the algorithm, focusing on the low-energy states at low temperatures and going to the maximum entropy state at large temperatures.

## 4. Series-Expansion Monte Carlo Embedding

In this section, we provide a self-contained and comprehensive overview of linked-cluster expansions for long-range interacting systems [25,29,30,31,34,35] using white graphs [152] in combination with Monte Carlo integration for the graph embedding. First, we introduce linked-cluster expansions (LCEs) and discuss perturbative continuous unitary transformations (pCUTs) [153,154] as a suitable high-order series expansion method. We then establish an adequate formalism for setting up LCEs and discuss the calculation of suitable physical quantities in practice. With the help of white graphs, we can employ LCEs for models with long-range interactions and use the resulting contributions in a Monte Carlo algorithm to deal with the embedding problem posed by long-range interactions. This approach we dub pCUT + MC.

### 4.1. Motivation and Basic Concepts

The goal of all our efforts is to calculate physical observables in the thermodynamic limit, i.e., for an infinite lattice L, using high-order series expansions. The starting point of every perturbative problem is
(48)$$H=H_{0}+\lambda V,$$
where the Hamiltonian describing the full physical problem H can be split up into an unperturbed part $H_{0}$, which is readily diagonalisable, and a perturbation V associated with a perturbation parameter λ, which is small compared to the energy scales of $H_{0}$. We aim to obtain a power series up to a maximal reachable order $o_{\max}$ as an approximation of a desired physical quantity:(49)$$f\left(\lambda \right)\approx p_{0}+p_{1}\lambda +p_{2}\lambda^{2}+\cdots p_{o_{\max}}\lambda^{o_{\max}}\;,$$
where the coefficients $p_{i}$ are to be determined by the series expansion. We want to use the information contained in the power series to infer the properties of the approximated function f(λ) [155]. The cost of determining the coefficients is associated with an exponential growth in complexity with increasing order [155]. Hence, calculations are performed with the help of a computer programme. Obviously, the computer cannot deal with an infinitely large lattice. Instead, we must look at finite cut-outs consisting of a finite set of lattice sites that are connected by bonds (or links) symbolising the interactions of the Hamiltonian on the lattice. We term these cut-outs *clusters*. If two clusters *A* and *B* do not share a common site or conterminously do not have a link that connects any site of A and B with each other (A∩B=∅), then the cluster C=A∪B is called a *disconnected cluster*. Otherwise, if no such partition into disconnected clusters *A* and *B* exists (A∩B≠∅), the cluster *C* is called *connected*. We can define quantum operators M (e.g., a physical observable) on these clusters just as on the infinite lattice.

There are essentially two ways of performing high-order series expansions. The first one is the naive approach of taking a single finite cluster C⊂L [153,154,156,157,158] and designing it such that the contribution of M(C) coincides with the contributions on the infinite lattice M(L)=M(C) up to the considered order in the perturbation parameter. The cluster needs to be chosen large enough such that the perturbative calculations up to the considered order are not affected by the boundaries of the cluster. Another way of performing calculations is to construct the operator contribution on a cluster—coinciding with the infinite lattice contributions up to a given order—by decomposing it into all possible contributions on smaller clusters [155,159,160,161,162,163,164,165,166,167,168,169,170]. Now, the contributions on many, but smaller clusters must be determined and added up to obtain the contribution on the infinite lattice:(50)$$M\left(L\right)=M\left(C\right)=\sum_{C^{\prime }\subset C}M\left(C^{\prime }\right)\;.$$ In contrast to the previous approach, we willingly accept boundary effects for the many subclusters. Such a cluster decomposition is known to be computationally more efficient because it suffices to calculate the contributions on the Hilbert space of the smaller clusters reducing the overhead of non-contributing processes, and it also suffices to perform the calculations only on a few clusters as many give identical contributions due to symmetries. This can significantly reduce the overhead.

However, there are subtleties about the validity of performing calculations on finite clusters, e.g., when setting up a linked-cluster expansion (linked-cluster means only connected clusters contribute), the operator M must satisfy a certain property, namely cluster additivity. The quantity M is called *cluster additive* if and only if the contribution on disconnected clusters A∪B solely comes from the contributions on its constituent connected clusters *A* and *B*. This means we can simply add the contributions of *A* and *B* from the smaller connected clusters to obtain the one for the disconnected cluster, i.e.,
(51)M(A∪B)=M(A)+M(B),
as illustrated schematically in Figure 3.

We can also understand cluster additivity in the language of perturbation theory where acting on bonds in every order forms a cluster of perturbatively active bonds. If such a cluster is connected, we call these processes *linked*. So, cluster additivity simultaneously means that only linked processes will contribute. Cluster additivity is at the heart of the *linked-cluster theorem*, which states that only linked processes will contribute to the overall contribution in the thermodynamic limit. To set up a linked-cluster expansion, we want to exploit cluster additivity and the linked-cluster theorem so that we can “simply” add up the contributions from individual connected clusters to obtain the desired power series in the thermodynamic limit.

An example of a cluster additive quantity is the ground-state energy $E_{0}$. Imagine we want to calculate the ground-state energy of a non-degenerate ground-state subspace, then cluster additivity is naturally fulfilled:(52)$$E_{0}\left(A\cup B\right)=E_{0}\left(A\right)+E_{0}\left(B\right),$$
and we can calculate the ground-state energy on A∪B from its individual parts *A* and *B*. However, cluster additivity is not satisfied in general. We can construct a counterexample by considering the first excited state with energy $E_{1}$. For example, consider the first excitation above the ferromagnetic ground state of the transverse-field Ising model in the low-field limit that is a single spin flip dressed with quantum fluctuations induced by the transverse field [113]. We usually refer to such excitations as *quasiparticles* (qp). Here, we cannot add the contributions on clusters *A* and *B* to obtain the excitation energy on cluster A∪B:(53)$$E_{1}\left(A\cup B\right)\neq E_{1}\left(A\right)+E_{1}\left(B\right)\;.$$ How to set up a linked-cluster expansion for intensive properties is not obvious, and it seemed out of reach after the introduction of linked-cluster expansions in the 1980s [159,160,161]. Only several years later, it was noticed by Gelfand [162] that additivity can be restored for excited states when properly subtracting the ground-state energy. This approach was later generalised to multiparticle excitations [154,156,164,165] and observables [154,163].

In the following, we first introduce a perturbation theory method that maps the original problem in Equation (Equation 48) to an effective one. We will show that the derived effective Hamiltonian and observables satisfy cluster additivity. In the subsequent section, we make use of the property and show how we can set up a linked-cluster expansion for energies of excited states and observables by properly subtracting contributions from lower energy states.

### 4.2. Perturbation Method: Perturbative Continuous Unitary Transformations

The first step towards setting up a linked-cluster expansion is to find a perturbation method that satisfies cluster additivity, which is generically not given and a non-trivial task [171]. Here, we use perturbative continuous unitary transformations (pCUTs) [153,154] that transform the original Hamiltonian perturbatively order by order into a quasiparticle-conserving Hamiltonian, reducing the original many-body problem to an effective few-body problem. We start discussing how to solve the flow equation to obtain the pCUT method and show afterwards how the Hamiltonian decomposes into additive parts that can be used for a linked-cluster expansion.

We strive to solve the usual problem of perturbation theory of Equation (Equation 48). The unperturbed part $H_{0}$ can be easily diagonalised exactly with a spectrum that has to be equidistant and bounded from below. Additionally, the perturbation V must be a sum of operators $T_{n}$:(54)$$V=\sum_{n=-N}^{N}T_{n}\;,$$
containing all processes changing the energy by *n* quanta and—if properly rescaled—corresponding to the same number of quasiparticles *n*. The goal of the pCUT method is to find an optimal basis in which the many-body problem of the original Hamiltonian reduces to an effective few-body problem. For that, we introduce a unitary transformation depending on a continuous flow parameter *ℓ* and define
(55)$$H\left(\ell \right)=U^{†}\left(\ell \right)HU\left(\ell \right)\;.$$ In the limiting case ℓ=0, we require H(0)=H to recover the original Hamiltonian, and for ℓ=∞, we require $\lim_{\ell \to \infty }H\left(\ell \right)=H_{\mathrm{eff}}$ so that the unitary transformation maps the original to the desired effective Hamiltonian. We can rewrite the unitary transformation as
(56)$$U\left(\ell \right)=T_{\ell }\exp\left(-\int_{0}^{\ell }\eta \left(\ell^{\prime }\right)d\ell^{\prime }\right)\;,$$
where η is the anti-hermitian generator generating the unitary transformation and $T_{\ell }$ the ordering operator for the flow parameter. Taking the derivatives of Equation (Equation 55) and Equation (Equation 56) in *ℓ*, we eventually arrive at the flow equation:(57)$$\frac{dH(\ell )}{d\ell }=\left[\eta (\ell ),H(\ell )\right]\;.$$ Flow equations have been studied for quite some time in mathematics and physics with a variety of applications [172,173,174,175,176,177,178,179,180]. It was Knetter and Uhrig [153] who proposed a perturbative ansatz for the generator of continuous unitary transformations along the lines of Mielke [180], introducing the quasiparticle generator (also known as the “MKU generator”) for the pCUT method:(58)$$\eta^{\mathrm{qp}}{\left(\ell \right)}_{i,j}:=\mathrm{sgn}\left(H_{0\;i,i}-H_{0\;j,j}\right)H_{i,j}\left(\ell \right)\;,$$
where the indices i,j refer to blocks of the Hamiltonian labelling the quasiparticle number. Diagonal blocks $H_{i,i}$ contain all processes conserving the number of quasiparticles *i*, while off-diagonal blocks $H_{i,j}$ contain all processes changing the quasiparticle number from *i* to *j*. The reasoning behind the ansatz can be explained by looking at $\mathrm{sgn}(H_{0\;i,i}-H_{0\;j,j})$, where processes i→j in $H_{i,j}$ are assigned the opposite sign of the inverse processes j→i, and therefore, the idea is to “rotate away” off-diagonal blocks by the unitary transformation during the flow of *ℓ*, while processes that do not change the quasiparticle number are not transformed away due to sgn(0)=0, but get renormalised during the flow. Consequently, in the limit ℓ→∞, we obtain an effective Hamiltonian $H_{\mathrm{eff}}$ that is block diagonal in *n*. This idea is depicted in Figure 4.

Next, we make a perturbative ansatz for the Hamiltonian during the flow:(59)$$H\left(\ell \right)=H_{0}+\sum_{\begin{array}{c} k=1\\ \sum_{j}n_{j}=k \end{array}}^{\infty }\lambda_{1}^{n_{1}}\cdots \lambda_{N_{\lambda }}^{n_{N_{\lambda }}}\sum_{\dim(m)=k}F\left(\ell ;m\right)T\left(m\right)\;,$$
with the notation
(60)$$\begin{array}{cc} m & =(m_{1},m_{2},m_{3},\cdots ,m_{k})\;, \end{array}$$
(61)$$\begin{array}{cc} m_{i} & \in \{0,\pm 1,\pm 2,\cdots ,\pm N\}\;, \end{array}$$
(62)$$\begin{array}{cc} \dim(m) & =k\;, \end{array}$$
(63)$$\begin{array}{cc} T(m) & =T_{m_{1}}T_{m_{2}}T_{m_{3}}\cdots T_{m_{k}}\;, \end{array}$$
with F(ℓ;m) being undetermined real functions. We introduce $N_{\lambda }$ distinct expansion parameters instead of just a single λ to keep the notation as general as possible because in Section 4.6.3, about white graphs, we will need multiple expansion parameters to encode additional information. Inserting Equations (Equation 58) and (Equation 59) into the flow equation (Equation 57), we can solve the equation perturbatively order-by-order as we obtain a recursive set of differential equations for F(ℓ;m).

To recover the original Hamiltonian H, we have to demand the correct initial conditions F(0;m)=1 for |m|=1 and F(0;m)=0 for |m|>1. We can solve the differential equations (cf. Ref. [153]) exactly for ℓ→∞, yielding
(64)$$H_{\mathrm{eff}}=H_{0}+\sum_{\begin{array}{c} k=1\\ \sum_{j}n_{j}=k \end{array}}^{\infty }\lambda_{1}^{n_{1}}\cdots \lambda_{N_{\lambda }}^{n_{N_{\lambda }}}\sum_{\begin{array}{c} \dim(m)=k,\\ M(m)=0 \end{array}}C\left(m\right)T\left(m\right),$$
with F(∞;m)=C(m)∈Q being exact rational coefficients and the restriction $M\left(m\right)=\sum_{i=1}^{k}m_{i}=0$ making the products T(m) quasiparticle-conserving [153]. Hence, the commutator of the effective Hamiltonian with the unperturbed diagonal part of the original Hamiltonian vanishes ($[H_{\mathrm{eff}},H_{0}]=0$). Note that, so far, the effective Hamiltonian (Equation 64) is model independent. It only depends on the overall structure of Equation (Equation 54). The generic form of the Hamiltonian comes at the cost of an additional normal ordering usually by applying the Hamiltonian to a cluster. Of course, it could also be performed explicitly by using the hard-core bosonic commutation relations, but the former approach can be handled much easier by a computer programme. Yet, we achieved our goal of obtaining a block-diagonal Hamiltonian:(65)$$H_{\mathrm{eff}}=\underset{n=0}{⨁}H_{\mathrm{eff}\;n}\;,$$
where $H_{\mathrm{eff}\;n}$ is the effective irreducible Hamiltonian of *n* quasiparticle processes (see also Section 4.3). Let us emphasise again that this block-diagonal structure allows us to solve the *n* quasiparticle blocks individually, which significantly reduces the complexity of the original many-body problem to an effective one that is block-diagonal in the quasiparticle number *n*.

If we want to calculate an effective observable, we can make an ansatz along the same lines [154]. We insert the perturbative ansatz:(66)$$O\left(\ell \right)=\sum_{\begin{array}{c} k=1\\ \sum_{j}n_{j}=k \end{array}}^{\infty }\lambda_{1}^{n_{1}}\cdots \lambda_{N_{\lambda }}^{n_{N_{\lambda }}}\sum_{i=1}^{k+1}\sum_{\dim(m)=k}G\left(\ell ;m;i\right)O\left(m;i\right)$$
with undetermined functions G(ℓ;m;i). The operator product is defined as
(67)$$O\left(m;i\right)=T_{m_{1}}\cdots T_{m_{i-1}}OT_{m_{i+1}}\cdots T_{m_{k}}\;.$$ Inserting exactly the same generator (Equation 58) and the ansatz for the observable in Equation (Equation 66) instead of the Hamiltonian into the flow Equation (Equation 57), we arrive at
(68)$$O_{\mathrm{eff}}=\sum_{\begin{array}{c} k=1\\ \sum_{j}n_{j}=k \end{array}}^{\infty }\lambda_{1}^{n_{1}}\cdots \lambda_{N_{\lambda }}^{n_{N_{\lambda }}}\sum_{i=1}^{k+1}\sum_{\dim(m)=k}\tilde{C}\left(m;i\right)O\left(m;i\right)$$
with $\tilde{C}\left(m;i\right)=G\left(\infty ;m;i\right)\in Q$ by solving the resulting set of differential equations for ℓ→∞ [154]. Note that the last sum does not contain a restriction M(m)=0, and therefore—in contrast to the effective Hamiltonian—effective observables are not (necessarily) quasiparticle-conserving.

We have just derived the effective form of the Hamiltonian and observables in the pCUT method that have a very generic form depending only on the structure of the perturbation of Equation (Equation 54). As already stated, the model dependence of our approach comes into play when performing a linked-cluster expansion by applying the effective Hamiltonian or observable to finite clusters. But how do we know if the effective quantities are cluster additive? We follow the argumentation of Refs. [181,182] by looking at the original Hamiltonian (Equation 48) that trivially satisfies cluster additivity as long as all bonds represent a non-vanishing term in V between sites (“bond equals interaction”). Thus, the Hamiltonian on a disconnected cluster A∪B:(69)$${H|}_{A\cup B}=H{{|}_{A}+H|}_{B}$$
is cluster additive because ${H|}_{A}$ and ${H|}_{B}$ are non-interacting. Here, we denote the restriction of the Hamiltonian H to a cluster *C* as ${H|}_{C}$. We can further insert this property into the flow equation:(70)$$\begin{array}{cc} \frac{{dH\left(\ell \right)|}_{A\cup B}}{d\ell } & =\left[{\eta \left(\ell \right)|}_{A\cup B}{,H\left(\ell \right)|}_{A\cup B}\right]\\ \frac{{dH\left(\ell \right)|}_{A}}{d\ell }+\frac{{dH\left(\ell \right)|}_{B}}{d\ell } & =\left[{\eta \left(\ell \right)|}_{A}{+\eta \left(\ell \right)|}_{B},H\left(\ell \right){{|}_{A}+H\left(\ell \right)|}_{B}\right]\\ \; & =\left[{\eta \left(\ell \right)|}_{A}{,H\left(\ell \right)|}_{A}\right]+\left[{\eta \left(\ell \right)|}_{B}{,H\left(\ell \right)|}_{B}\right]\;. \end{array}$$ Here, we used the property of H(0) that it commutes on disconnected clusters and the fact that the Hamiltonian is continuously transformed during the flow starting from ℓ=0. Therefore, the derivative and the commutator can be split up, acting on each cluster individually and preserving cluster additivity during the flow. Consequently, the effective Hamiltonian:(71)$$H_{\mathrm{eff}}{|}_{A\cup B}=H_{\mathrm{eff}}{{|}_{A}+H_{\mathrm{eff}}|}_{B}$$
in the limit ℓ→∞ is cluster-additive as well. The same proof holds for effective observables. Another more physical argument is that the effective pCUT Hamiltonian can be written as a sum of nested commutators of *T*-operators [152,167]. For instance, considering the perturbation $V=T_{-1}+T_{0}+T_{+1}$, the effective Hamiltonian looks like
(72)$$H_{\mathrm{eff}}=H_{0}+\lambda T_{0}+\lambda^{2}\left[T_{+1},T_{-1}\right]+\frac{\lambda^{3}}{2}\left(\left[\left[T_{+1},T_{0}\right],T_{-1}\right]+\left[T_{+1},\left[T_{0},T_{-1}\right]\right]\right)+\cdots \;.$$ Splitting up $T_{n}=\sum_{l}\tau_{n,l}$ into local operators acting on bonds *l*, the nested commutators vanish for processes that are not linked. Hence, the linked-cluster theorem is fulfilled, and the effective Hamiltonian is cluster-additive. To emphasise the linked-cluster property, the generic effective Hamiltonian is often written as
(73)$$H_{\mathrm{eff}}=H_{0}+\sum_{\begin{array}{c} k=1\\ \sum_{j}n_{j}=k \end{array}}^{\infty }\lambda_{1}^{n_{1}}\cdots \lambda_{N_{\lambda }}^{n_{N_{\lambda }}}\sum_{\begin{array}{c} \dim(m)=k\\ M(m)=0 \end{array}}\sum_{\begin{array}{c} C\\ |E_{C}|\leq k \end{array}}C\left(m\right)\;\sum_{\begin{array}{c} l_{1},\cdots ,l_{k}\\ {⋃}_{i=1}^{k}l_{i}=C \end{array}}\tau_{m_{1},l_{1}}\cdots \tau_{m_{k},l_{k}}\;,$$
where the sum over *C* runs over all possible connected clusters with maximal *k* bonds ($k\geq |E_{C}|$) [152]. The notation $E_{C}$ will be clarified in Section 4.6.2, where graphs are formally introduced. In this context, it is simply the set of bonds of a connected cluster *C* and $|E_{C}|$ the number of bonds in this set. The condition ${⋃}_{i=1}^{k}l_{i}=C$ arising from the linked-cluster theorem ensures that the cluster consisting of active links and sites during a process must match with the bonds and sites of the connected cluster *C*. For observables, the generalised condition ${⋃}_{i=1}^{k}l_{i}\cup x=C$ holds, where the index *x* can either refer to a site (local observable) or a link (non-local observable), and we have
(74)$$\begin{array}{cc} O_{\mathrm{eff}}=\sum_{\begin{array}{c} k=1\\ \sum_{j}n_{j}=k \end{array}}^{\infty }\lambda_{1}^{n_{1}}\cdots \lambda_{N_{\lambda }}^{n_{N_{\lambda }}}\sum_{i=1}^{k+1}\sum_{\dim(m)=k} & \sum_{\begin{array}{c} C\\ |E_{C}|\leq k \end{array}}\tilde{C}\left(m;i\right)\\ \; & \times \;\sum_{\begin{array}{c} l_{1},\cdots ,l_{k}\\ {⋃}_{i=1}^{k}l_{i}\cup x=C \end{array}}\;\tau_{m_{1},l_{1}}\cdots \tau_{m_{i-1},l_{i-1}}O_{x}\tau_{m_{i},l_{i}}\cdots \tau_{m_{k},l_{k}}\;. \end{array}$$ Although we showed that the effective Hamiltonian and observables are cluster-additive and, therefore, fulfil the linked-cluster theorem, to set up a linked-cluster expansion, there are important subtleties remaining when we restrict the effective Hamiltonian and observables to the quasiparticle basis, which we need to address before we can discuss how to perform the calculations in practice.

### 4.3. Unravelling Cluster Additivity

In this subsection, we need to clarify how we can use the cluster-additive property of the effective pCUT Hamiltonian and observables to set up a linked-cluster expansion not only for the ground-state energy, but also for the energies of excited states. Many aspects of this section are based on the original work of Ref. [154], in which a general formalism was developed for how to derive suitable quantities for the calculation of multiparticle excitations and observables. We further develop this formalism by inferring the concept of cluster additivity for the quasiparticle basis, introducing the notion of particle additivity. The term “additivity” in this context was recently introduced by Ref. [171].

We start by recalling that the effective Hamiltonian is block-diagonal, and we can write the Hamiltonian operator as a sum of irreducible operators of *n* quasiparticle processes:(75)$$H_{\mathrm{eff}}=\sum_{n=0}H_{\mathrm{eff},n}\;.$$ We can express the *n* quasiparticle processes in second quantisation in terms of local (hard-core) bosonic operators $b_{i}^{†}$ creating and bi† annihilating a quasiparticle at site *i*. When considering quantum magnets like we do in this review, a hard-core repulsion comes into play allowing only a single quasiparticle at a given site [154]. For instance, in the ferromagnetic ground-state of the 2D Ising model, an elementary excitation is given by a single spin flip, which can be interpreted as a quasiparticle excitation [113]. Obviously, at most one excitation on the same site is allowed. Different particle flavours τ can also be accounted for by incorporating an additional index τ of the operator $b_{i,\tau }^{\left(†\right)}$. To keep the notation simple, we will drop this additional index in the following. The irreducible operators in the second quantisation and normal-ordered form then read
(76)Heff0=ϵ01,Heff1=∑i∑jti;jbj†bi†,Heff2=∑i1,i2∑j1,j2ti1,i2;j1,j2bj2†bj1†bi2†bi1†,⋮Heffn=∑i1,⋯,in∑j1,⋯,jnti1⋯,in;j1,⋯,jnbjn†⋯bj1†bin†⋯bi1†. Written in normal order, the meaning of these processes is directly clear when acting on states in the quasiparticle basis. The prefactors $t_{i_{1}\cdots ,i_{n};j_{1},\cdots ,j_{n}}$ are to be determined by applying the effective pCUT Hamiltonian (Equation 64) to an appropriately designed cluster *C*. Let us consider the quasiparticle basis on a connected cluster *C*:(77)$$\left\{{|0\rangle }_{C}\;,{|1\rangle }_{C}\;,{|2\rangle }_{C}\;,\cdots ,{|n\rangle }_{C}\right\}=\left\{{|0\rangle }_{C}\;,{|1;\;i\rangle }_{C}\;,{|2;\;i_{1},i_{2}\rangle }_{C}\;,\cdots ,{|n;\;i_{1},i_{2},\cdots ,i_{n}\rangle }_{C}\right\}\;,$$
where the number *n* specifies the number of particle excitations and the indices $i_{j}$ denote the positions of the *n* (local) excitations. The effective pCUT Hamiltonian is quasiparticle conserving, so let us restrict it to *N* particle states, like when evaluating its matrix elements in this basis. If we evaluate by acting on a state with fewer particles than particles involved in the process, then the irreducible operator annihilates more particles than there are and the contribution is zero, that is $H_{n}{|}^{N}=0$ for N<n. When we determine the action of $H_{\mathrm{eff}\;n}{|}^{n}$ for N=n, this allows us to determine all prefactors $t_{i_{1}\cdots ,i_{n};j_{1},\cdots ,j_{n}}$ defining the action of $H_{\mathrm{eff}\;n}$ on the entire unrestricted Hilbert space. The second quantisation presents a natural generalisation of the Hamiltonian restricted to a finite number of particles to an arbitrary number of particles [154]. We can construct $H_{n}{|}^{N}$ for N>n from $H_{n}{|}^{n}$ since the latter completely defines the action $H_{\mathrm{eff}\;n}$ on the entire Hilbert space.

Although everything seems fine so far to set up a linked-cluster expansions, let us tell you that it is not. We finished the motivation in Section 4.1 with the statement that we cannot simply add up contributions for energies of excited states (cf. Equation (Equation 53)). The reason is that, although we showed that $H_{\mathrm{eff}}$ is cluster additive, the irreducible operators restricted to the *N* particle basis $H_{\mathrm{eff}\;n}{|}^{N}$ are in fact not. To grasp a better understanding of the abstract concept of cluster additivity and why setting up a linked cluster expansion for higher particle channels usually fails, let us consider the following basis on a disconnected cluster A∪B:(78)$$\begin{array}{cc} \left\{{|0\rangle }_{A\cup B}\;,{|1\rangle }_{A\cup B}\;,{|2\rangle }_{A\cup B}\;,\cdots \right\}=\{ & {|0\rangle }_{A}⊗{|0\rangle }_{B}\;,{|0\rangle }_{A}⊗{|1\rangle }_{B}\;,{|1\rangle }_{A}⊗{|0\rangle }_{B}\;,\\ \; & {|0\rangle }_{A}⊗{|2\rangle }_{B}\;,{|1\rangle }_{A}⊗{|1\rangle }_{B}\;,{|2\rangle }_{A}⊗{|0\rangle }_{B}\;,\cdots \}\;, \end{array}$$
where ${|n\rangle }_{C}$ represents all possible *n*-particle states living on a cluster *C*. While there is only one way to decompose the zero-particle states ${|0\rangle }_{A\cup B}$ on the disconnected cluster A∪B, one-particle states ${|1\rangle }_{A\cup B}$ decompose into two sets of states with a particle on cluster *A* (${|1\rangle }_{A}⊗{|0\rangle }_{B}$) and a particle on *B* (${|0\rangle }_{A}⊗{|1\rangle }_{B}$). For two-particle excitations ${|2\rangle }_{A\cup B}$, there are three possibilities to distribute the particles. In general, *N*-particle states have the form ${|k\rangle }_{A}⊗{|N-k\rangle }_{B}$ with k∈{0,1,⋯,N}, and there are N+1 possibilities to decompose the states.

When restricting Equation (Equation 51) to these *N*-quasiparticle states, a cluster-additive Hamiltonian must decompose as
(79)$$H_{\mathrm{eff}}^{\mathrm{cl}.\;\mathrm{add}.}{|}_{A\cup B}^{N}=\overset{N}{\underset{k=0}{⨁}}\left(H_{\mathrm{eff}}{{|}_{A}^{k}+H_{\mathrm{eff}}|}_{B}^{N-k}\right)\;,$$
where we introduce the notation ${|}_{C}^{n}$ restricting the Hamiltonian to all *n*-quasiparticle states on a cluster *C*. The direct sum in Equation (Equation 79) is introduced to emphasise that, for a cluster additive Hamiltonian, there must not be any particle processes between the two disconnected clusters *A* and *B*. The Hilbert space on the disconnected cluster A∪B can be seen as the natural extension of the Hilbert spaces on cluster *A* and *B*, and we can define the operators on the clusters *A* and *B* in terms of the Hilbert space on A∪B as a tensor product:(80)$$\begin{array}{cc} H_{\mathrm{eff}}{|}_{A}^{k} & :=H_{\mathrm{eff}}{{|}^{k}⊗1|}^{N-k}\;,\\ H_{\mathrm{eff}}{|}_{B}^{k} & {:=1|}^{N-k}⊗H_{\mathrm{eff}}{|}^{k}\;, \end{array}$$
where operators on the left of ⊗ are defined on the Hilbert space of *A* and the operators to the right on the Hilbert space of *B*. The issue with Equation (Equation 79) is that, when we restrict the particle basis to *N* on the disconnected cluster A∪B, there are contributions from lower particles channels coming from the N+1 possibilities to distribute the *N* particles on the two clusters. For example, if we look at the one-particle space:(81)$$H_{\mathrm{eff}}^{\mathrm{cl}.\;\mathrm{add}.}{|}_{A\cup B}^{1}=\left(H_{\mathrm{eff}}{{|}_{A}^{1}+H_{\mathrm{eff}}|}_{B}^{0}\right)⊕\left(H_{\mathrm{eff}}{{|}_{A}^{0}+H_{\mathrm{eff}}|}_{B}^{1}\right),$$
we see that, in addition to the one-particle contributions, we obtain additional zero-particle contributions. The left part of the direct sum stems from acting on ${|1\rangle }_{A}⊗{|0\rangle }_{B}$ and the right side from acting on ${|0\rangle }_{A}⊗{|1\rangle }_{B}$. There are always two possibilities to distribute the one-particle excitation on the two clusters where the other cluster is always unoccupied, which gives an additional zero-particle contribution. This fact is schematically illustrated in Figure 5a.

This is not the desired behaviour for a linked-cluster expansion. We would like the general notion of Equation (Equation 51) to directly translate to the particle-restricted basis, which is illustrated in Figure 5b. In other words, our goal is to find a notion of (cluster) additivity in the restricted particle basis such that we can simply add up the *N*-particle contributions of individual clusters without caring about lower particle channels. We define *particle additivity* as
(82)$$H_{\mathrm{eff}}^{\mathrm{add}.}{|}_{A\cup B}^{N}:=H_{\mathrm{eff}}{{|}_{A}^{N}⊕H_{\mathrm{eff}}|}_{B^{N}},$$
where we demand the other contributions from Equation (Equation 79) to vanish. The crucial thing to notice is that irreducible operators (Equation 76) in the particle basis:(83)$$H_{\mathrm{eff}\;N}{{|}_{A\cup B}^{N}\equiv H_{\mathrm{eff}}^{\mathrm{add}}|}_{A\cup B}^{N}$$
are, in fact, particle-additive [154,171]. In the following, we will show that this is indeed the case. First, we remember that, for the ground-state energy, we can trivially add up the contributions. Starting from the definition for cluster additivity (Equation 79), we have
(84)$$\begin{array}{cc} H_{\mathrm{eff}}^{\mathrm{cl}.\;\mathrm{add}}{|}_{A\cup B}^{0} & =H_{\mathrm{eff}}{{|}_{A}^{0}+H_{\mathrm{eff}}|}_{B}^{0}\\ \; & \equiv H_{\mathrm{eff}\;0}{|}_{A}^{0}+H_{\mathrm{eff}\;0}{{|}_{B}^{0}\equiv H_{\mathrm{eff}}^{\mathrm{add}}|}_{A\cup B}^{0}\;. \end{array}$$ Second, from restricting the decomposition of Equation (Equation 75) to the N=1 particle channel, we can express the irreducible one-particle operator as
(85)$$H_{\mathrm{eff}\;1}{|}_{A\cup B}^{1}=H_{\mathrm{eff}}^{\mathrm{cl}.\;\mathrm{add}.}{{|}_{A\cup B}^{1}-H_{\mathrm{eff}\;0}|}_{A\cup B}^{1}\;.$$ We recall that, by calculating $H_{\mathrm{eff}\;0}{|}^{0}$, we can automatically derive $H_{\mathrm{eff}\;0}$ on the entire Hilbert space, which subsequently defines $H_{\mathrm{eff}\;0}{|}^{1}$. Therefore, by inserting the definition for cluster additivity (Equation 79), we obtain
(86)$$\begin{array}{cc} H_{\mathrm{eff}}^{\mathrm{cl}.\;\mathrm{add}.}{{|}_{A\cup B}^{1}-H_{\mathrm{eff}\;0}|}_{A\cup B}^{1}= & \left[\left(H_{\mathrm{eff}}{{|}_{A}^{1}+H_{\mathrm{eff}}|}_{B}^{0}\right)⊕\left(H_{\mathrm{eff}}{{|}_{A}^{0}+H_{\mathrm{eff}}|}_{B}^{1}\right)\right]\\ \; & -\left[\left(H_{\mathrm{eff}\;0}{{|}_{A}^{1}+H_{\mathrm{eff}\;0}|}_{B}^{0}\right)⊕\left(H_{\mathrm{eff}\;0}{{|}_{A}^{0}+H_{\mathrm{eff}\;0}|}_{B}^{1}\right)\right]\\ = & \left(H_{\mathrm{eff}}{{|}_{A}^{1}-H_{\mathrm{eff}\;0}|}_{A}^{1}\right)⊕\left(H_{\mathrm{eff}}{{|}_{B}^{1}-H_{\mathrm{eff}\;0}|}_{B}^{1}\right)\\ = & H_{\mathrm{eff}\;1}{{|}_{A}^{1}⊕H_{\mathrm{eff}\;1}|}_{B}^{1}\\ \equiv & H_{\mathrm{eff}}^{\mathrm{add}.}{|}_{A\cup B}^{1}\;, \end{array}$$
where we used the definition of Equation (Equation 82) in the last line. Hence, we have proven Equation (Equation 83) for N<2. The above proof can be readily extended to N≥2.

We achieved our goal of finding a notion of cluster additivity in the particle basis, which we termed particle additivity. We can determine the desired particle-additive quantities by using the subtraction scheme:(87)$$H_{\mathrm{eff}}^{\mathrm{add}.}{|}^{N}\equiv H_{\mathrm{eff}\;N}{|}^{N}=H_{\mathrm{eff}}{{|}^{N}-\sum_{n=0}^{N-1}H_{\mathrm{eff}\;n}|}^{N},$$
which comes from Equation (Equation 75) by restricting it to an *N*-particle basis. This is an inductive scheme starting from N=0 calculating the irreducible additive quantity $H_{\mathrm{eff}\;0}{|}^{0}$. This result can be used to calculate the subsequent irreducible additive quantity $H_{\mathrm{eff}\;1}{|}^{1}$ for N=1. Then, for N=2, we use the results from N=0 and N=1 to calculate $H_{\mathrm{eff}\;2}{|}^{2}$, and so on. Again, it is important that $H_{\mathrm{eff}\;n}{|}^{n}$ completely defines the operator $H_{\mathrm{eff}\;n}$ and, therefore, any $H_{\mathrm{eff}\;n}{|}^{m}$.

When considering effective observables, the particle number is no longer conserved and more types of processes are allowed. We need to generalise Equation (Equation 75) for effective observables by introducing an additional sum over *d* that is the change in the quasiparticle number. An effective observable, thus, decomposes as
(88)$$O_{\mathrm{eff}}=\sum_{n=0}\sum_{d\geq -n}O_{\mathrm{eff}\;n,d}\;,$$
where $O_{\mathrm{eff}\;n,d}$ are irreducible contributions [154]. When writing them in second quantisation, we have
(89)Oeffn,d=∑i1,⋯,in∑j1,⋯,jn+dt˜i1⋯,in;j1,⋯,jn+dbjn+d†⋯bj1†bin†⋯bi1†. We can directly see that the *d* quasiparticles are created because there are *d* additional creation operators. When *d* is negative, *d* quasiparticles are annihilated. We can infer a notion for cluster additivity and particle additivity along the same lines:(90)$$\begin{array}{cc} O_{\mathrm{eff}}^{\mathrm{cl}.\;\mathrm{add}.}{|}_{A\cup B}^{N\to N+d} & =\overset{N}{\underset{k=0}{⨁}}\left(O_{\mathrm{eff}}{{|}_{A}^{k\to k+d}+O_{\mathrm{eff}}|}_{B}^{N-k\to N-k+d}\right)\;, \end{array}$$(91)$$\begin{array}{cc} O_{\mathrm{eff}}^{\mathrm{add}.}{|}_{A\cup B}^{N\to N+d} & =O_{\mathrm{eff}}{{|}_{A}^{N\to N+d}⊕O_{\mathrm{eff}}|}_{B}^{N\to N+d}\;. \end{array}$$ To determine the particle additive parts, we can use an analogue subtraction scheme as described in Equation (Equation 87), which can be denoted as
(92)$$O_{\mathrm{eff}\;N,d}{|}^{N\to N+d}=O_{\mathrm{eff}}{{|}^{N\to N+d}-\sum_{n=0}^{N-1}O_{\mathrm{eff}\;n,d}|}^{N\to N+d}\;.$$ If we want to calculate $O_{\mathrm{eff}\;N,d}{|}^{N\to N+d}$, then we have to inductively apply Equation (Equation 92).

There are several things to be noted at this point. First, not all perturbation theory methods satisfy cluster additivity (Equation 79), and in this case, we cannot write operators as a direct sum any longer. There will be quasiparticle processes between one and the other cluster changing the number of particles on each cluster [155,162]. This is sketched in Figure 6 by the presence of an additional term.

When falsely performing a linked-cluster expansion, it can be noticed immediately that the approach breaks down. A symptom of non-cluster additivity is the presence of contributions of lower orders than expected from the number of edges of the graph [155]. When calculating reduced contributions in a linked-cluster expansion, we subtract only contributions of connected subgraphs, which leaves non-zero contributions of disconnected clusters when the perturbation theory method is not cluster-additive. However, there are notable exceptions when a linked-cluster expansion for energies of excited states is still correct even though the perturbation theory method is not cluster-additive [163,164,165,183,184,185,186]. This is only possible when the considered excitation does not couple with a lower particle channel, i.e., lower lying states are described by a distinct set of quantum numbers lying in another symmetry sector [155,165,187]. For instance, consider the elementary quasiparticle excitation in a high-field expansion for the transverse-field Ising model (TFIM), then the structure of the perturbation is $T_{-2}+T_{0}+T_{2}$, and therefore, the first excited state does not couple directly with the ground state (there is no $T_{\pm 1}$, which is due to symmetry). If one wants to draw a comparison, we can think of this as being similar to the case when calculating the excitations in the density matrix renormalisation group. It is no problem to target an excited state if it is in a different symmetry sector than the ground state, but if it is in the same symmetry sector described by the same set of quantum numbers, then the Hamiltonian needs to be modified to project out the ground state [188].

Recently, a minimal transformation to an effective Hamiltonian was discovered that preserves cluster additivity. This method, called “projective cluster-additive transformation” [171], can be used analogously and is even more efficient for the calculation of high-order perturbative series. In this review, however, we stick to the well-established pCUT method.

### 4.4. Calculating Cluster Contributions

At this point, we may ask how to evaluate physical quantities on finite clusters in practice. To evaluate these quantities, we must evaluate them in the quasiparticle basis. In general, when setting up a cluster expansion, may it be non-linked or linked, it is important to subtract the contributions from all possible subclusters to prevent over-counting. Mathematically, for a quantity M ($H_{\mathrm{eff}}$ or $O_{\mathrm{eff}}$), this can be written as
(93)$$\bar{M}{|}_{C}^{\cdots }=M{|}_{C}^{\cdots }-\sum_{C^{\prime }\subset C}\bar{M}{|}_{C^{\prime }}^{\cdots }\;,$$
where the sum runs over all real subclusters $C^{\prime }$ in cluster *C*, and we call the resulting quantity $\bar{M}{|}_{C}^{\cdots }$
*reduced*. Starting from the smallest possible cluster (e.g., a single bond between two sites), this formula can be inductively applied to determine the reduced quantity on increasingly big clusters. An essential observation to make is that reduced operators vanish on disconnected clusters by construction if the operator M is additive since we subtract all contributions from individual subclusters. As the linked-cluster theorem applies, we can set up a cluster expansion:(94)$$\bar{M}\left(L\right){|}_{C}^{\cdots }=\sum_{C\subset L}\bar{M}\left(C\right){|}_{C}^{\cdots }$$
of connected clusters, but we need to consider reduced quantities $\bar{M}$ to prevent over-counting. For a light notation, we will drop the bar in the sections below as we will only consider reduced contributions on graphs anyway.

Now, we are ready to look at the problem from a more practical point of view. From the previous subsection, we know how cluster additivity translates into the particle basis and how to construct particle-additive parts, namely the irreducible quasiparticle contributions. We decompose the effective Hamiltonian into its irreducible contributions by explicitly calculating: (95)$$\begin{array}{cc} \; & H_{\mathrm{eff}\;0}{{|}^{0}=H_{\mathrm{eff}}|}^{0}\; \end{array}$$(96)$$\begin{array}{cc} \; & H_{\mathrm{eff}\;1}{|}^{1}=H_{\mathrm{eff}}{{|}^{1}-H_{\mathrm{eff}\;0}|}^{1}\;, \end{array}$$(97)$$\begin{array}{cc} \; & H_{\mathrm{eff}\;2}{|}^{2}=H_{\mathrm{eff}}{|}^{2}-H_{\mathrm{eff}\;1}{{|}^{2}-H_{\mathrm{eff}\;0}|}^{2}\; \end{array}$$(98)$$\begin{array}{cc} \; & \;\vdots \end{array}$$(99)$$\begin{array}{cc} \; & H_{\mathrm{eff}\;N}{|}^{N}=H_{\mathrm{eff}}{{|}^{N}-\sum_{n=0}^{N-1}H_{\mathrm{eff}\;n}|}^{N}\;. \end{array}$$ Again, consider the effective Hamiltonian in the second quantisation made up of hard-core bosonic operators $b_{i}^{\left(†\right)}$ annihilating (creating) quasiparticles and the quasiparticle counting operator ni=bi†bi† occurring in the unperturbed Hamiltonian $H_{0}$. We also consider a connected cluster *C*, and we denote *n* quasiparticle states on this cluster as ${|n;i_{1},\cdots ,i_{n}\rangle }_{C}$ with the quasiparticles on the sites $i_{1}$ to $i_{n}$. Note that, for multiple quasiparticle flavours or multiple sites within a lattice unit cell, this notation can be generalised to ${|n;i_{1},\cdots ,i_{n},\tau_{1},\cdots ,\tau_{n}\rangle }_{C}$ by introducing additional indices $\tau_{i}$. To lighten the notation, in the following, we stick to the former case. Let us consider the three lowest particle channels:n=0We can directly calculate the ground-state energy $E_{0}\left(C\right)$ on a cluster *C* as it is already additive:
(100)$$E_{0}\left(C\right)=\langle 0|H_{\mathrm{eff}}{|0\rangle }_{C},$$
as can be seen from Equation (Equation 95).n=1To calculate the irreducible amplitudes $t_{i;j}^{\left(1\right)}\left(C\right)$ associated with the hopping process bj†bi† in $H_{\mathrm{eff}\;1}$, we need to subtract the zero-particle channel, as can be seen from Equation (Equation 96). However, we only need to subtract the ground-state energy if the hopping process is local, bi†bi†, since the ground-state energy only contributes to diagonal processes. Thus, we calculate
(101)$$\begin{array}{c} \begin{array}{cccc} t_{i;j}^{\left(1\right)}\left(C\right) & =\langle 1;j|H_{\mathrm{eff}}{|1;i\rangle }_{C} & \; & \mathrm{if}\;i\neq j\;,\\ t_{i;i}^{\left(1\right)}\left(C\right) & =\langle 1;i|H_{\mathrm{eff}}{|1;i\rangle }_{C}-E_{0}\left(C\right) & \; & \mathrm{else}\;. \end{array} \end{array}$$n=2In the two-particle case, we have to distinguish between three processes: pair hoppings (ti,j;k,l(2)(C)bl†bk†bj†bi† with four distinct indices), correlated hoppings (ti,j;i,k(2)(C)bk†bj†ni), and density–density interactions ($t_{i,j;i,j}^{\left(2\right)}\left(C\right)\;n_{j}n_{i}$). The free quasiparticle hopping is already irreducible, and nothing has to be done, but for the correlated hopping contribution, we have to subtract the free one-particle hopping. In the case of the two-particle density–density interactions, we need to subtract the local one-particle hoppings, as well as the ground-state energy, as this process is diagonal (cf. Equation (Equation 97)). Therefore, we calculate
(102)$$\begin{array}{c} \begin{array}{cccc} t_{i,j;k,l}^{\left(2\right)}\left(C\right) & =\langle 2;k,l|H_{\mathrm{eff}}{|2;i,j\rangle }_{C} & \; & \mathrm{if}\;i\neq j\neq k\neq l\;,\\ \;t_{i,j;i,k}^{\left(2\right)}\left(C\right) & =\langle 2;i,k|H_{\mathrm{eff}}{|2;i,j\rangle }_{C}-t_{j;k}^{\left(1\right)}\left(C\right) & \; & \mathrm{if}\;i\neq j\neq k\;,\\ t_{i,j;i,j}^{\left(2\right)}\left(C\right) & =\langle 2;i,j|H_{\mathrm{eff}}{|2;i,j\rangle }_{C}-t_{i;i}^{\left(1\right)}\left(C\right)-t_{j;j}^{\left(1\right)}\left(C\right)-E_{0}\left(C\right) & \; & \mathrm{if}\;i\neq j\;. \end{array} \end{array}$$ An analogous procedure can be applied for effective observables. Here, we need to determine the irreducible contributions for a fixed *d*. The subtraction scheme is given by
(103)$$\begin{array}{cc} \; & O_{\mathrm{eff}\;0,d}{{|}^{0\to d}=O_{\mathrm{eff}}|}^{0\to d} \end{array}$$
(104)$$\begin{array}{cc} \; & O_{\mathrm{eff}\;1,d}{|}^{1\to 1+d}=O_{\mathrm{eff}}{{|}^{1\to 1+d}-O_{\mathrm{eff}\;0,d}|}^{1\to 1+d} \end{array}$$
(105)$$\begin{array}{cc} \; & O_{\mathrm{eff}\;2,d}{|}^{2\to 2+d}=O_{\mathrm{eff}}{|}^{2\to 2+d}-O_{\mathrm{eff}\;1,d}{{|}^{2\to 2+d}-O_{\mathrm{eff}\;0,d}|}^{2\to 2+d} \end{array}$$
(106)$$\begin{array}{cc} \; & \;\vdots \end{array}$$
(107)$$\begin{array}{cc} \; & O_{\mathrm{eff}\;N,d}{|}^{N\to N+d}=O_{\mathrm{eff}}{{|}^{N\to N+d}-\sum_{n=0}^{N-1}O_{\mathrm{eff}\;n,d}|}^{N\to N+d}\;. \end{array}$$ For d=0, we recover exactly the same subtraction procedure as before. It is straightforward to generalise this procedure for d≠0. Let us specifically consider the example we will encounter in the next section, when calculating correlations for the spectral weight. The effective observable is applied to the unperturbed ground state |0〉. Hence, there are only contributions out of the ground state (N=0), and the effective observables decomposes into
(108)$$O_{\mathrm{eff}}=O_{\mathrm{eff}\;0,0}+O_{\mathrm{eff}\;0,1}+O_{\mathrm{eff}\;0,2}+\cdots \;.$$ Since only N=0 processes contribute, nothing needs to be subtracted and the effective observable $O_{\mathrm{eff}}{{|}^{0\to d}=O_{\mathrm{eff}\;0,d}|}^{0\to d}$ is irreducible and already particle-additive.

Although, for these types of calculations, the lowest orders can be analytically determined by hand, the calculations usually become cumbersome quickly and must be evaluated using a computer programme to push to higher perturbative orders (in most cases, a maximal order ranging from 8 to 20 is achievable). Such a programme reads the information of the cluster (site and bond information), the “bra” and “ket” states, the coefficient lists C(m) or $\tilde{C}\left(m;i\right)$ from Equations (Equation 64) and (Equation 68) up to the desired order, the structure of the Hamiltonian H, the local τ-operators from the perturbation V, and if necessary, the observable. The input states, as well as the τ-operators should be efficiently implemented in the eigenbasis of $H_{0}$ bitwise encoding the information as known, for instance, from exact diagonalisation. If possible, the calculation should be performed with rational coefficients for the exact representation of the perturbative series up to a desired order. The routine of the programme is then to iterate through the operator sequences from the coefficient list C or $\tilde{C}$ and to consecutively apply the τ-operators by systematically iterating over all bonds of the cluster and calculating the action of the operator, saving the intermediate states for the action of the next operator in the sequence. As intermediate states are superpositions of basis states, they are saved in associative containers:(109)$$|\psi \rangle =\sum_{j}c_{j}|j\rangle \;,$$
where |j〉 is the bit representation of a basis state. The key of the associative container is the basis state |j〉 and the associated value, the prefactor $c_{j}$. The bitwise representation of the basis states |j〉, as well as the τ-operators allow for a fast access and modification during the calculation.

So far, we have introduced the pCUT method for calculating the perturbative contributions on clusters. We demonstrated that the resulting effective quasiparticle-conserving Hamiltonian is cluster-additive and showed how to extract particle-additive irreducible contributions. We introduced a subtraction scheme for the effective Hamiltonian and observables that can be easily applied to calculate additive quantities to set up a linked-cluster expansion. Finally, we briefly discussed an efficient implementation of the pCUT method.

### 4.5. Energy Spectrum and Observables

Having established the basic theoretical framework for the pCUT method, we want to give a short overview of the physical quantities that are most frequently calculated with this approach. To this end, we assume that we consider a single suitably designed cluster for the calculations instead of setting up an LCE as a full graph decomposition. We will see how we can calculate the desired quantities without thinking about the abstract concepts necessary for linked-cluster expansions, and with the insights from this section, it will be easier to recognise what we are aiming at. Here, we first consider the energy spectrum of the Hamiltonian. We derive both the control-parameter susceptibility as the second derivative of the ground-state energy, as well as the elementary excitation gap from the effective one-quasiparticle Hamiltonian. Second, we consider observables. In the pCUT approach, often, spectral weights are calculated, which are of great importance for inelastic neutron scattering experiments.

#### 4.5.1. Ground-State Energy and Elementary Excitation Gap

Following the above-described recursive scheme, we start with the zero-quasiparticle channel assuming a non-degenerate unperturbed ground state, which is the situation in all applications discussed in this review. The ground-state energy can be directly calculated from the cluster as in Equation (Equation 100). We consider a suitably designed cluster that is large enough to accommodate all fluctuations of a given maximal order $o_{\max}$ and has periodic boundary conditions to correctly account for translational invariance. We calculate $E_{0}=\langle 0|H_{\mathrm{eff}}|0\rangle$ and obtain a high-order perturbative series:(110)$$E_{0}=\sum_{o=0}^{o_{\max}}p_{o}\lambda^{o}$$
in the expansion parameter λ, which is valid in the thermodynamic limit up to the given maximal order $o_{\max}$. We can extract the ground-state energy per site by dividing $E_{0}$ by the number of sites of the cluster $\epsilon_{0}=E_{0}/N$. By taking the second derivative, we obtain the control-parameter susceptibility: (111)$$\chi =-\frac{d^{2}\epsilon_{0}}{d\lambda^{2}}\;.$$ We are usually interested in the quantum-critical properties of the model and, therefore, analyse the behaviour about the quantum-critical point $\lambda_{c}$. The control-parameter susceptibility shows the diverging power-law behaviour:(112)$$\chi \propto |\lambda -\lambda_{c}{|}^{-\alpha }$$
with the associated critical exponent α, as we know from Table 1.

Turning to the one-quasiparticle channel, we calculate the hopping amplitudes following Equation (Equation 101). Here, we use open boundary conditions, again with a cluster large enough to accommodate all fluctuations contributing to the hopping process. Note that, in our notation, we denote $t_{i;j}$ for hopping amplitudes on a graph or cluster level and the character *a* for processes in the thermodynamic limit. As for the ground-state energy, we can directly infer the contribution in the thermodynamic limit if the contributing fluctuations do not feel finite-size effects, and thus, we use a(j−i) in the following. Further, we can generalise our notation to multiple particle types or larger unit cells containing multiple sites, as mentioned earlier, by introducing additional indices ξ, τ. We denote a hopping from unit cell i to j by δ=j−i and within the unit cells from ξ to τ. We calculate $a_{\xi ,\tau }\left(\delta \right)=\langle 1;j,\tau |H_{\mathrm{eff}}|1;j-\delta ,\xi \rangle$ by fixing j, due to translational symmetry. The effective one-quasiparticle Hamiltonian in the second quantisation then is
(113)Heff1qp:=Heff|1=Heff0|1+Heff1|1=ϵ0N+∑j,δ,ξ,τaξ,τ(δ)bj,τ†bj−δ,ξ†. Applying the Fourier transform for a discrete lattice:(114)bj,τ†=1N∑kbj,τ†expikjbj,τ†=1N∑kbj,τ†exp−ikj,
we can diagonalise the resulting Hamiltonian in momentum space:(115)FHeff1qp=ϵ0N+∑kbk†Ω(k)bk†=∑kΩm,n(k)bk,m†bk,n†=∑kων(k)βk,ν†βk,ν†,
introducing the operators $\beta_{k,\nu }^{\left(†\right)}\left(k\right)$ that diagonalise the matrix $\Omega_{m,n}$. The eigenenergies $\omega_{\nu }\left(k\right)$ are the associated bands of the one-quasiparticle dispersion. In case of a trivial unit cell or a single particle flavour, the dispersion matrix becomes a scalar, and we can directly express the single-banded dispersion as
(116)$$\omega \left(k\right)=a\left(0\right)+2\sum_{\delta }a\left(\delta \right)\cos\left(k\delta \right)\;,$$
where the sum over δ is restricted to symmetry representatives and we assumed real hopping amplitudes a(δ). We determine the hopping amplitudes a(δ) as a perturbative series by performing the calculations on the properly designed cluster. Note that, even for a single cluster, we need to perform a subtraction scheme because, in order to obtain the irreducible contribution a(0) explicitly, we need to subtract the ground-state energy from $\langle i|H_{\mathrm{eff}}|i\rangle$. When we determine the elementary excitation gap at the minimum of the lowest band:(117)$$\Delta =\min_{k,\nu }\omega_{\nu }\left(k\right)=\sum_{o=0}^{o_{\max}}p_{o}\lambda^{o}\;,$$
we can as well extract the gap directly as such a series. The gap closing shows a power-law behaviour:(118)$$\Delta \propto |\lambda -\lambda_{c}{|}^{z\nu }$$
about the critical point with the critical exponent zν.

#### 4.5.2. Spectral Properties

Neutron scattering is a powerful method resolving spatial and dynamical structures in condensed matter physics since thermal neutrons have a de Broglie wavelength of similar length scale as interatomic distances and their energy is of the same order of magnitude as typical excitations [189]. By measuring the change in momentum and kinetic energy in inelastic neutron scattering experiments determining the dynamic response, not only static properties like magnetic order can be resolved, but also dynamic properties like spin–spin correlations [190]. The dynamic response $S_{\alpha \beta }\left(k,\omega \right)$ can be determined as it is proportional to the cross-section:(119)$$\frac{d^{2}\sigma }{d\Omega d\omega }\propto \sum_{\alpha ,\beta }S_{\alpha ,\beta }\left(k,\omega \right)$$
of inelastic neutron scattering [189,190]. We follow the derivations in Refs. [155,191,192] and start with the definition of the *dynamic response*:(120)$$S_{\alpha ,\beta }\left(k,\omega \right)=\frac{1}{2\pi N}\sum_{i,j}\int_{-\infty }^{\infty }dt\exp\left\{i\left[\omega t-k\left(j-i\right)\right]\right\}{\langle S_{j}^{\alpha }\left(t\right)S_{i}^{\beta }\left(0\right)\rangle }_{T}\;,$$
which is the space and time Fourier transform of the spin correlation function ${\langle S^{\alpha }\left(t\right)S^{\beta }\left(0\right)\rangle }_{T}$ with α,β∈{x,y,z,+,−} and ${\langle \cdot \rangle }_{T}$ referring to the thermal expectation value. In the limit of vanishing temperature T=0, the expectation value simplifies to $\langle \cdot \rangle =\langle \psi_{0}|\cdot |\psi_{0}\rangle$ with $|\psi_{0}\rangle$ being the ground state. Then, we call $S_{\alpha ,\beta }\left(k,\omega \right)$ the *dynamic structure factor*. We introduce a complete set of energy eigenstates $\left\{|\psi_{\Lambda }\rangle \right\}$, where Λ denotes a set of quantum numbers, for instance n,k, where *n* is the number of quasiparticles and k the lattice momentum. Writing the dynamic structure factor in terms of these energy eigenstates, this yields the dynamic structure factor in the spectral form as a sum:(121)$$S_{\alpha ,\beta }\left(k,\omega \right)=\sum_{\Lambda }S_{\alpha ,\beta }^{\Lambda }\left(k,\omega \right)\;,$$
where $S_{\alpha ,\beta }^{\Lambda }\left(k,\omega \right)$ is called the *exclusive structure factor* or *spectral weight* associated with the quantum numbers Λ. We insert $1=\sum_{\Lambda }|\psi_{\Lambda }\rangle \langle \psi_{\Lambda }|$ into the correlation function and switch to the Heisenberg picture, where $S_{j}^{\alpha }\left(t\right)=e^{iHt}S_{j}^{\alpha }\left(0\right)e^{-iHt}$. The spectral weight then reads
(122)$$\begin{array}{cc} S_{\alpha ,\beta }^{\Lambda }\left(k,\omega \right)=\frac{1}{2\pi N}\sum_{i,j}\int_{-\infty }^{\infty }dt & \exp\left[i\left(\omega -E_{\Lambda }+E_{0}\right)t\right]\exp\left[ik\left(j-i\right)\right]\\ \; & \times \langle \psi_{0}|S_{j}^{\alpha }\left(0\right)|\psi_{\Lambda }\rangle \langle \psi_{\Lambda }|S_{i}^{\beta }\left(0\right)|\psi_{0}\rangle \;, \end{array}$$
which, after integrating over time *t*, yields
(123)$$S_{\alpha ,\beta }^{\Lambda }\left(k,\omega \right)=\delta \left(\omega -E_{\Lambda }+E_{0}\right)\frac{1}{N}\sum_{i,j}\langle \psi_{0}|S_{j}^{\alpha }\left(0\right)|\psi_{\Lambda }\rangle \langle \psi_{\Lambda }|S_{i}^{\beta }\left(0\right)|\psi_{0}\rangle \exp\left[ik\left(j-i\right)\right]\;.$$ If we consider the case ${\left(S_{i}^{\alpha }\right)}^{†}=S_{i}^{\beta }$ or some observable with ${\left(O_{i}\right)}^{†}=O_{i}$, which could be a linear combination of spin operators, then the above expression further simplifies to
(124)$$S^{\Lambda }\left(k,\omega \right)=\delta \left(\omega -E_{\Lambda }+E_{0}\right)\frac{1}{N}{\left|\sum_{i}\langle \psi_{\Lambda }|O_{i}|\psi_{0}\rangle e^{-iki}\right|}^{2}\;.$$ By the above equations, we can identify the spectral weights $S^{\Lambda }\left(k\right)$ as
(125)$$S^{\Lambda }\left(k\right)=\frac{1}{N}{\left|\sum_{i}\langle \psi_{\Lambda }|O_{i}|\psi_{0}\rangle e^{-iki}\right|}^{2}\;.$$ These quantities are usually visualised as a heat map where the dispersion ω(k) is plotted against the momentum k and the value $S^{\Lambda }\left(k,\omega \right)$, that is the intensity of the scattering signal associated with $|\psi_{\Lambda }\rangle$, is colour-coded.

In the pCUT approach, we want to reformulate the observable in terms of an effective one $O_{\mathrm{eff}}$. Here, we want to restrict ourselves to the one-quasiparticle spectral weight. In Ref. [191], you can also find a formulation for the two-quasiparticle case. For the 1qp spectral weight, Λ are the quantum numbers defining one-quasiparticle states, and we denote $S^{\Lambda }\left(k\right)\equiv S_{\tau }^{1\mathrm{qp}}\left(k\right)$ in the following. Since the pCUT method is a perturbative approach, we want to reformulate the problem in the language of $H_{0}$ states:(126)$$\begin{array}{cc} |\psi_{0}\rangle & =U|0\rangle \;, \end{array}$$(127)$$\begin{array}{cc} |\psi_{1\mathrm{qp}}\rangle & =U|1;k,\tau \rangle \;, \end{array}$$
where we introduce the momentum states |1;k,τ〉 with additional index τ denoting a quantum number like a flavour of the excitation or denoting a site in a unit cell. The momentum states are defined via the Fourier transform:(128)$$|1;k,\tau \rangle =\frac{1}{\sqrt{N}}\sum_{j}\exp\left(ikj\right)|1;j,\tau \rangle \;.$$ Inserting these identities, we obtain
(129)$$\begin{array}{cc} S_{\tau }^{1\mathrm{qp}}\left(k\right) & =\frac{1}{N}{\left|\sum_{i}\langle \psi_{1\mathrm{qp}}|O_{i}|\psi_{0}\rangle \exp\left(-iki\right)\right|}^{2}=\frac{1}{N}{\left|\sum_{i}\langle 1;k,\tau |U^{†}O_{i}U|0\rangle \exp\left(-iki\right)\right|}^{2}\\ \; & =\frac{1}{N^{2}}{\left|\sum_{i,j}\langle 1;j,\tau |O_{\mathrm{eff},i}|0\rangle \exp\left[ik\left(j-i\right)\right]\right|}^{2}\;, \end{array}$$
where we defined $O_{\mathrm{eff},i}=U^{†}O_{i}U$. For a problem with translational invariance, we can fix the site i of the observable and introduce δ=j−i, which yields
(130)$$S_{\tau }^{1\mathrm{qp}}\left(k\right)=\frac{1}{N^{2}}{\left|\sum_{i,\delta }\langle 1;i+\delta ,\tau |O_{\mathrm{eff},i}|0\rangle \exp\left(ik\delta \right)\right|}^{2}={\left|\sum_{\delta }{\tilde{a}}_{\tau }\left(\delta \right)\exp\left(ik\delta \right)\right|}^{2}\;,$$
where we used ${\tilde{a}}_{\tau }\left(\delta \right)=\langle 1;i+\delta ,\tau |O_{\mathrm{eff},i}|0\rangle$. Note that the form of the irreducible contribution of the effective observable is $O_{\mathrm{eff}\;0,1}=\sum_{\delta }{\tilde{a}}_{\tau }\left(\delta \right)b_{\delta ,\tau }^{†}$, exactly as denoted above in Equation (Equation 108). As long as the processes $\tilde{a}\left(\delta \right)$ and $\tilde{a}\left(-\delta \right)$ are equivalent by means of the model symmetries, we can use $\tilde{a}\left(\delta \right)=\tilde{a}\left(-\delta \right)$ and further simplify $S_{\tau }^{1\mathrm{qp}}\left(k\right)$ to
(131)$$S_{\tau }^{1\mathrm{qp}}\left(k\right)={\left|{\tilde{a}}_{\tau }\left(0\right)+2\sum_{\delta }{\tilde{a}}_{\tau }\left(\delta \right)\cos\left(k\delta \right)\right|}^{2}\equiv {\left|s_{\tau }\left(k\right)\right|}^{2}\;.$$ So, in the pCUT method, we determine ${\tilde{a}}_{\tau }\left(\delta \right)$ as a perturbative series with which we can determine $S_{\tau }^{1\mathrm{qp}}\left(k\right)$. Note that this observable is already an irreducible contribution according to Equation (Equation 92), and no subtraction scheme needs to be performed. Because we consider the contribution $O_{\mathrm{eff}}{|}^{0\to 1}$ of the observable, the initial particle number is n=0, and nothing needs to be subtracted. To extract the quantum-critical behaviour, we need to evaluate the expression at the critical momentum $k_{c}$ as
(132)$$S_{\tau }^{1\mathrm{qp}}\left(k_{c}\right)=\sum_{o=0}^{o_{\max}}p_{o}\lambda^{o},$$
yielding a perturbative series for this quantity as well. It shows a diverging critical behaviour that goes as
(133)$$S_{\tau }^{1\mathrm{qp}}\left(k_{c}\right)\propto {\left|\lambda -\lambda_{c}\right|}^{-(2-z-\eta )\nu }$$
with the associated critical exponent (2−z−η)ν. After determining the three quantities χ, Δ, and $S_{\tau }^{1\mathrm{qp}}\left(k_{c}\right)$ described above, we can use the extrapolation techniques described in Section 4.8 to extract estimates for the critical point $\lambda_{c}$ and the associated critical exponents. However, we continue with the description of a linked-cluster expansion as a full graph decomposition for long-range interacting systems. The next step on the way is to formally introduce graphs, discuss their generation, and the associated concept of white graphs.

### 4.6. White Graph Decomposition

In this section, we first give a brief introduction to graph theory, which forms the basis for understanding how linked-cluster expansions as a graph decomposition work. Then, we discuss how to generate all topologically distinct graphs up to a certain number of edges corresponding to the maximal number of active edges at a given perturbative order. We conclude this section by explaining the concept of white graphs where edge colours are ignored as a topological attribute in the classification of graphs. White graphs are essential for tackling long-range interactions, and therefore, every graph decomposition in this review is, in fact, a white graph decomposition.

#### 4.6.1. Graph Theory

So far, we already defined clusters as a cut-out of the infinite lattice with a finite set of sites and a set of bonds connecting those sites. More generally, only considering the topology of clusters without restricting them to the geometry of lattices, we can define a *graph* as a tuple $G=(V_{G},E_{G})$ consisting of a (finite) set of vertices $V_{G}$ and a (finite) set of edges $E_{G}$. An edge $e\in E_{G}$ consists of a pair of vertices {μ,ν}, and these vertices $\mu ,\nu \in V_{G}$ are called *adjacent*. The *degree* of a vertex is the number of edges connecting it to other vertices of the graph. In the following, we only consider *undirected*, *simple*, and *connected* graphs, which means there are neither directed edges, multiple edges between two vertices, nor loops (no edge that is joining a vertex to itself), and there always exists a path of edges connecting any two vertices of a graph [155,193,194]. As an example, we depict a graph $G=(V_{G},E_{G})$ with $V_{G}=\left\{0,1,2,3,4,5\right\}$ and $E_{G}=\left\{\left\{0,1\right\},\left\{0,2\right\},\left\{0,4\right\},\left\{0,5\right\},\left\{1,3\right\},\left\{2,3\right\}\right\}$ in Figure 7.

A *subgraph* $G^{\prime }$ of a graph G (we write $G^{\prime }\subset G$) is defined as a subset of $V_{G}^{\prime }\subset V_{G}$ and $E_{G}^{\prime }\subset E_{G}$ [193,194,195]. We call $G^{\prime }$ a proper subgraph if $V_{G}^{\prime }$ and $E_{G}^{\prime }$ are proper subsets of $V_{G}$ and $E_{G}$.

To set up a full graph decomposition, it is essential to define how to distinguish different graphs. If there exists an isomorphism between two graphs, we call them *topologically equivalent*; otherwise, they are *topologically distinct*. A *graph isomorphism* $\mathrm{Iso}(G_{1},G_{2})$ is a bijective map $\varphi_{\mathrm{Iso}}$ between the vertex sets of two graphs, such that
(134)$$\begin{array}{cc} \; & \varphi_{\mathrm{Iso}}:\;V_{G_{1}}\to V_{G_{2}}\\ \; & \left\{\mu ,\nu \right\}\in E_{G_{1}}\Leftrightarrow \left\{\varphi_{\mathrm{Iso}}\left(\mu \right),\varphi_{\mathrm{Iso}}\left(\nu \right)\right\}\in E_{G_{2}}\;. \end{array}$$ So, an isomorphism preserves adjacency and non-adjacency, i.e., μ and ν are adjacent if and only if the vertices φ(μ) and φ(ν) are adjacent for any vertices $\mu ,\nu \in V_{G}$ [193,194,196]. A special case are *graph automorphisms*
Auto(G), which are maps of a graph on itself, so we have
(135)$$\begin{array}{cc} \; & \varphi_{\mathrm{Auto}}:\;V_{G}\to V_{G}\\ \; & \left\{\mu ,\nu \right\}\in E_{G}\Leftrightarrow \left\{\varphi_{\mathrm{Auto}}\left(\mu \right),\varphi_{\mathrm{Auto}}\left(\nu \right)\right\}\in E_{G}\;. \end{array}$$ In other words, a graph automorphism is a permutation of the vertex set preserving adjacency [194,196]. The number of graph automorphisms |Auto(G)| of a given graph gives the number of its symmetries, and we call it the *symmetry number*
$s_{G}=\left|\mathrm{Auto}\left(G\right)\right|$ of a graph [155,195]. Examples of a graph isomorphism and automorphism are depicted in Figure 7. Further, if $G_{1}\subset G_{2}$, the mapping (Equation 134) is injective instead of bijective such that
(136)$$\begin{array}{cc} \; & \varphi_{\mathrm{Mono}}:\;V_{G_{1}}\to V_{G_{2}}\\ \; & \left\{\mu ,\nu \right\}\in E_{G_{1}}\Rightarrow \left\{\varphi_{\mathrm{Mono}}\left(\mu \right),\varphi_{\mathrm{Mono}}\left(\nu \right)\right\}\in E_{G_{2}}, \end{array}$$
and we call it a *subgraph isomorphism* or *monomorphism* $\mathrm{Mono}(G_{1},G_{2})$ [195,197,198].

An example of a monomorphism is depicted in Figure 8a. Monomorphisms will later become of utmost importance as they give the number of embeddings of the subgraph onto a graph, which is the infinite lattice.

To account for hopping processes during the embedding of graphs, we need to assign additional attributes to graphs like colouring their vertices [195]. Then, a coloured graph $G_{c}$ is a tuple $\left(V_{G},E_{G},A_{V}\right)$, where $A_{V}$ is a set of pairs {μ,a} with μ∈V and *a* the colour attribute. In Figure 8b, an example is depicted with $A_{V}=\left\{\left\{0,\mathrm{green}\right\},\left\{2,\mathrm{green}\right\}\right\}$. We can extend the above definitions for isomorphisms and automorphisms and say that they must preserve the vertex colour, i.e., only vertices of the same colour can be mapped onto each other. Of course, this reduces the cardinality $\left|\mathrm{Auto}(G_{c})\right|$ and, therefore, the symmetry number $s_{G_{c}}$ associated with the coloured graph. We can later exploit the colour information for matching hopping vertices of the graph with the actual hopping of the quasiparticle on the lattice. Note that, also, colouring edges of graphs with attributes $A_{E}$ is useful to distinguish different types of interactions of a Hamiltonian on the graph level and very similar properties as for coloured vertices hold. As stated above, two graphs are topologically distinct if there does not exist a graph isomorphism between the two graphs. Thus, for coloured graphs, the colour attribute serves as another topological attribute. The importance of this will become apparent later in Section 4.6.3. For a more elaborate overview of graph theory in the context of linked-cluster expansions, we recommend Ref. [195].

#### 4.6.2. Graph Generation

For the graph generation of (undirected) simple, connected graphs, we need to define an ordering between all graph isomorphs such that it is possible to pick a unique representative of all graph isomorphs that is called *canonical representative* [199]. One challenge lays in efficiently generating graphs as the number of connected graphs grows exponentially with the number of edges and the idea behind every algorithm must be to restrict the number of possible graph isomorph candidates when generating new edge sets by permutation. A well known algorithm is McKay’s algorithm [199,200], which exploits the degree information of the vertices, sets up a search tree for vertices with the same degree, and uses the ordering to check if the canonical representative of the current graph already exists. We recommend using open-source libraries for the graph generation and calculation of additional symmetries. For instance, there is “*nauty*” [201] and “*networkX*” [202] or the “*Boost Graph Library*” [203].

There are various conventions for how to write the graph information to a file. One could simply save the site and edge set as lists or save its *adjacency matrix*, where the rows and columns refer to the sites and a non-zero entry in the corresponding matrix element marks the existence of an edge between the sites [155,193,194]. Here, we suggest to simply use a *bond list*. Each entry in the list contains the edge information of the graph, denoted as $n_{e},\mu ,\nu$, where $e\in E_{G}$ with $\mu ,\nu \in V_{G}$ adjacent to *e* and $n_{e}$ is just a number associated with the edge *e*. The number $n_{e}$ can be interpreted as a specific expansion parameter corresponding to this bond. In the simplest case, there is just one expansion parameter and therefore a single number $n_{e}$ for all edges. Assigning multiple expansion parameters becomes especially important in the next Section 4.6.3. Usually, the symmetry number $s_{G}$ of a graph is calculated on the fly when generating a set of graphs and should be saved as well. In our workflow, we save the generated graphs into bond lists and create a list containing all connected graphs (e.g., the file names) along with their symmetry number $s_{G}$. These types of lists suffice for the calculation of the ground-state energy. When calculating 1qp irreducible processes for the dispersion or for the spectral weight observable, we can think of these processes breaking the graph symmetry. Therefore, after the graph generation, we consider all possible processes on a graph and assign colour attributes to the start and end vertices. Due to the symmetry of the processes, we assign the same colour for hoppings and distinct colours for processes of the spectral weight (When calculating the hopping processes of the effective 1qp Hamiltonian from vertex μ to ν, it has the symmetry $t_{\mu ;\nu }=t_{\nu ;\mu }$ with the inverse process giving the same contributions as long as the prefactors are real due to the Hermiticity of the Hamiltonian. Thus, we can use the same colours for start and end vertex. The 1qp process for the spectral weight ${\tilde{t}}_{\mu ;\nu }$ is distinct because a quasiparticle is created at vertex μ and then subsequently hops to vertex ν. Therefore, ${\tilde{t}}_{\mu ;\nu }$ and ${\tilde{t}}_{\nu ;\mu }$ are, in general, not equivalent processes ${\tilde{t}}_{\mu ;\nu }\neq {\tilde{t}}_{\nu ;\mu }$, and we have to use distinct colours for the start and end vertices. But, as mentioned in Section 4.5.2, the processes can become equivalent by means of general model symmetries, for instance due to underlying graph symmetries). We calculate the symmetry number $s_{G_{c}}$ associated with the coloured graph. In the end, we create a list, where each entry contains the graph, its symmetry number $s_{G}$, a representative process (start and end vertex), and the associated symmetry number of the coloured graph $s_{G_{c}}$, counting the number of processes that give the same contributions as the representative process due to symmetries.

After generating the graphs and the lists, we can employ perturbation theory calculations on the graph level, viewing graphs as abstract clusters with vertices as lattice sites connected by bonds. A programme as described above reads in the graph and process information and repeatedly performs the pCUT calculations on every graph. The resulting graph contributions must be added up in such a way that the information of the lattice geometry is restored by weighting the graph contributions with embedding factors. That is, how many ways a graph can be fit onto a lattice apart from translational invariance (cf. Ref. [155]). For the conventional embedding of contributions from models with just nearest-neighbour interactions, the number of graphs can be reduced as the lattice geometry puts a restriction on the graphs. For example, graphs containing cycles of odd length cannot be embedded on the square lattice. In contrast, for long-range interactions, no such restriction exists, and therefore, the lattice can be seen as a single fully connected graph where every site interacts with each other. Hence, we have to generate every simple connected graph up to a given number of edges as all graphs contribute. Before we turn to the embedding problem in more detail, we will first deal with the challenges long-range interactions pose and address the problem by using white graphs.

#### 4.6.3. White Graphs for Long-Range Interactions

In many cases in perturbation theory, there may be more types of expansion parameters, e.g., due to the geometry of the lattice, as can be seen in the *n*-leg Heisenberg ladder [152]: (137)$$H=H_{0}+\lambda_{⊥}V+\lambda_{\left|\right|}V$$
with an expansion parameter $\lambda_{⊥}$ associated with the rungs and another one $\lambda_{\left|\right|}$ associated with the legs. The interaction V is given by XY interactions, and the series expansion is performed about the Ising limit $H_{0}$. Setting up a graph decomposition for this model, the canonical approach would be to associate each expansion parameter with a distinct edge colour, blue for $\lambda_{⊥}$ and purple for $\lambda_{\left|\right|}$.

In Figure 9a (left), all graphs with two edges and two colours (one for $\lambda_{⊥}$ and one for $\lambda_{\left|\right|}$) are depicted on the left. It is necessary to incorporate the edge colour information as the graphs can only be embedded correctly on the infinite lattice when the edge colour matches the bond type of the lattice. Another common type of perturbation problem is where the perturbation splits into different types of interactions, although associated with the same perturbation parameter. For instance, the problem can look like
(138)$$\begin{array}{cc} H & =H_{0}+\lambda V\\ \; & =H_{0}+\lambda \left(V_{x}+V_{x}+V_{z}\right)\;, \end{array}$$
which is essentially the form of the Hamiltonian when performing a high-field series expansion [204] for Kitaev’s honeycomb model [205], where each site has one *x*-, *y*-, and *z*-type Ising interaction bond to one of its neighbours on the honeycomb lattice, respectively. Here, we associate the three different interaction types with three types of edge colours, blue, green, and purple for *x*-, *y*-, and *z*-bonds. See Figure 9b (left) for an illustration of all three graphs with two edges and three colours (there are only three possibilities because colours must alternate due to the constraints posed by Kitaev’s honeycomb model). In both cases, the edge colour is an additional topological attribute of the graph, leading to exponentially more graphs with the number of colours, which becomes relevant when pushing to high orders in perturbation.

In the case of long-range interactions, such an approach becomes unfeasible. A Hamiltonian with long-range interactions is of the form
(139)$$\begin{array}{cc} H & =H_{0}+\lambda \sum_{\delta }\frac{1}{{\left|\delta \right|}^{d+\sigma }}V\\ \; & =H_{0}+\sum_{\delta }\lambda_{\delta }V, \end{array}$$
where δ is the distance between interacting sites, *d* the dimension of the lattice, and σ the long-range decay exponent and $\lambda_{\delta }=\lambda {\left|\delta \right|}^{-(d+\sigma )}$. Applying the conventional approach from above, we would associate each of the infinitely many perturbations $V_{\delta }$ with its own edge colour. The only obvious way to resolve this problem would be to truncate the sum over δ only considering very small distances. Instead, the use of *white graphs* [152] can be a solution to problems of this kind [25,29]. The idea is to change our view of how to tackle these Hamiltonians. We ignore the edge colours of the graph—significantly reducing the number of graphs for a given order—and instead, encode the colour information in the expansion parameters on the graph level in a more abstract way. This is performed by associating each edge *e* of the graph G with a different “abstract” perturbation parameter $\lambda_{e}$ such that we can track how often τ-operators acted on each edge *e*, yielding a multivariable polynomial:(140)$$P_{G}\left(\left\{\lambda_{e}\right\}\right)=\sum_{m}v_{m}\left(G\right)M_{G,m}=\sum_{m}v_{m}\left(G\right)\prod_{e\in E_{G}}\lambda_{e}^{n_{e,m}}\;,$$
with the sum over all monomial contributions. The individual monomial contributions consist of a prefactor $v_{m}\left(G\right)$, and its monomial dependency $M_{G,m}$. $M_{G,m}$ comprises a product of expansion parameters $\lambda_{e}$ associated with edge *e* and their respective integer powers $n_{e,m}\geq 1$ tracking how often each bond was active during the calculation. Let us emphasise that we simply wrote the white graph contribution for $\epsilon_{0}$, $t_{\mu ;\nu }$, or ${\tilde{t}}_{\mu ;\nu }$ explicitly as a polynomial $P_{G}\left(\left\{\lambda_{e}\right\}\right)$. It is only later during the embedding of these abstract generalised contributions when the proper link colour is reintroduced by substituting the expansion parameters by the actual expansion parameters for each realisation on the actual lattice. This is the origin of the name “white graphs” because, during the calculation on the graph, the colour of the links is unspecified (but encoded in the multivariable polynomial), and only during the embedding, the colour of the edges is reintroduced. In Figure 9, we illustrate the white graph concept in the middle. On the right, we depict how to recover the polynomial of the conventional graph contribution from the abstract white graph contribution for the models in Equations (Equation 137) and (Equation 138) by correctly substituting the abstract expansion parameters. The main difference between the model (Equation 138) compared to (Equation 137) is that, for the different interaction types, multiple parameters are associated with each edge. To account for three types of interaction flavours, we have to consider three expansion parameters $\lambda_{e,f}$ with f∈{x,y,z} per edge *e*.

For long-range interactions, we can straightforwardly apply the exact same white graph scheme. We substitute the abstract expansion parameter with the correct algebraically decaying interaction strength depending on the distance between the interacting sites on the lattice [25,29]. For this, we have to use the substitution:(141)$$\lambda_{e}\mapsto \frac{1}{{\left|\delta \right|}^{\left(d+\sigma \right)}}$$
with $\delta =i_{\nu }-i_{\mu }$ and $e=\left\{\mu ,\nu \right\}\in E_{G}$. It is also possible to incorporate multiple interaction types, but the substitution for the different interaction types must then be performed before the embedding, as was done in Refs. [31,35,206].

So far, we explained how to resolve the problem of infinitely many perturbation parameters by introducing white graphs. We managed to reduce the number of expansion parameters from infinity to the number of edges, i.e., the order of the perturbation. Yet, the polynomial in Equation (Equation 140) still grows exponentially with the number of expansion parameters. Following the description in Ref. [152], we can further mitigate this issue by an efficient representation of white graph contributions. The abstract concept is to track the relevant information in a quantity M(n) as a generalised monomial with the property $M\left(n_{1}+n_{2}\right)=M\left(n_{1}\right)M\left(n_{2}\right)$ and use $n_{i}$ as an abstract parameter that encodes the tracked information. In Equation (Equation 140), the monomial quantity M(n) is just $M_{G,m}$, tracking how often each edge was active during the calculation by associating each edge with its own abstract expansion parameter. We can generalise the expression of Equation (Equation 109) for states comprising additional information:(142)$$|\psi \rangle =\sum_{i,j}c_{i,j}M\left(n_{i}\right)|j\rangle =\sum_{j}|j\rangle \left(\sum_{i}c_{i,j}M\left(n_{i}\right)\right)\;.$$ Instead of a simple superposition of states |j〉 with associated prefactor $c_{j}$ as in Equation (Equation 109), we have an additional superposition over all monomials $M(n_{i})$ comprising the information of all the distinct processes encoded in $M(n_{i})$ leading to the state |j〉. We can make use of this factorisation property by using nested containers. The key of the outer container is the basis state |j〉, and the value contains an inner container with the monomial $M(n_{i})$ as the key and the prefactor $c_{i,j}$ as the value. Thus, the action of a τ-operator on a state |j〉 can be calculated independent of the action on the monomial $M(n_{i})$. For a flat container, we would have to calculate the action on the same state |j〉 multiple times. To further improve the efficiency of the calculation, we can directly encode into the programme which edge of a graph was active in a bitwise representation. Using the information of the number of edges of a graph and tracking the applied perturbative order during the calculation, we can neglect subcluster contributions on the fly and reduce the computational overhead even further. Therefore, we can directly calculate the reduced contribution on the graph without the need for explicitly subtracting subcluster contributions.

Wrapping things up, we explained how the use of white graphs can be applied to models with long-range interactions resolving the issue of infinitely many graphs or expansion parameters at any finite perturbative order. Instead of treating each perturbation of the long-range interaction as another topological attribute in the classification of graphs, we associate an abstract expansion parameter with each edge of a graph, and only during the embedding on the lattice, we substitute these expansion parameters with the actual bond information of long-range interacting sites. An efficient representation of white graphs can further help to reduce the computational overhead.

### 4.7. Monte Carlo Embedding of White Graphs

We discussed ways to set up a linked-cluster expansion, either by designing a single appropriately large cluster hosting all relevant fluctuations up to a given perturbative order or by setting up a linked-cluster expansion as a full graph decomposition, where the latter is the more efficient way to perform high-order series expansions. Of course, a quantity M must be cluster-additive in the first place. To decompose M(L) on the lattice L into many smaller subcluster contributions, we can add up the contributions:(143)$$M\left(L\right)=\sum_{C\subset L}M\left(C\right)\;,$$
where M(C) are reduced contributions on a cluster *C* to prevent overcounting. It is not necessary to calculate the contribution of every possible cluster since many clusters have the same contribution. It suffices to calculate the contribution of a representative cluster, only containing the relevant topological information, that is a graph defined by its vertex set $V_{G}$ and its edge set $E_{G}$. We can see a graph G as representing an equivalence class [G], whose elements are all clusters *C* realising all possible embeddings of a graph on the lattice [195]. Figuratively, all elements of the equivalence class are related by translational and rotational symmetry or are different geometric realisations on the lattice. We can now split the sum over all clusters into one sum over all possible graphs on the lattice and another one over all elements in the equivalence class:(144)$$M\left(L\right)=\sum_{C\subset L}M\left(C\right)=\sum_{G\subset L}\sum_{C\in [G]}M\left(C\right)\;,$$
where it suffices to calculate M(C) once for all C∈[G] [195]. We are left counting the number of elements *C* in the equivalence class such that we can write
(145)$$M\left(L\right)=\sum_{G\subset L}W\left(G,L\right)M\left(G\right)\;,$$
where the embedding factor W(G,L) is simply a number counting the number of embeddings [195]. The important point in our line of thought is that we can calculate the quantity M(G) only once on the graph level and multiply the contribution with a weight that is the embedding factor W and sum up the resulting contribution for all graphs to obtain the desired quantity M(L) in the thermodynamic limit. We are essentially left with determining the embedding factors after calculating the graph contributions M(G). Depending on the topology of the graph, the number of possible embeddings is different and, therefore, also the embedding factor. When calculating one quasiparticle processes for the 1qp dispersion or the 1qp spectral weight, it is important that we account for the graph vertices’ colour attributes for the definition of the equivalence class. If conventional graph contributions are considered, we directly obtain the correct contribution M(G). If there are multiple physical parameters or different interaction flavours, the graph edges have to be colour matched with the bonds of the lattice. If white graphs are used, we do not have to match any colours and this can be ignored; however, the white graph contributions have to be evaluated appropriately substituting the abstract expansion parameters with the correct physical parameter for each embedding. Regarding continuative reading on the embedding, we want to point out the standard literature for linked-cluster expansions [152,155,187], as well as Ref. [195]. Nonetheless, we show in the following how the embedding procedure works for models with nearest-neighbour interactions specifically on the example of the ground-state energy and one-quasiparticle processes yielding the 1qp dispersion of the Hamiltonian. Likewise, we derive the 1qp spectral weight. With these findings, we can turn to the embedding problem of long-range interacting models and eventually describe the Monte Carlo algorithm as a solution to this problem.

#### 4.7.1. Conventional Nearest-Neighbour Embedding

*Ground-state energy:* We start with the simplest case of calculating the ground-state energy on the infinite lattice L. We know that $H_{\mathrm{eff}}$ is cluster additive and that the ground-state energy is an extensive quantity, so we can directly calculate
(146)$$E_{0}=\sum_{G\subset L}W\left(G,L\right)E_{0}\left(G\right)\;.$$ In other words, this means we have to multiply the ground-state energy contributions on the graphs with the correct embedding factor and add up the resulting weighted contributions for every graph. For the definition of the embedding factor, we follow the formalism introduced by Ref. [195] and write
(147)$$W\left(G,L\right)=\frac{\left|\mathrm{Mono}(G,L)\right|}{\left|\mathrm{Auto}(G)\right|}=\frac{1}{s_{G}}\left|\mathrm{Mono}\left(G,L\right)\right|\;.$$ The embedding factor W is the number of subgraph isomorphisms (monomorphisms) of the graph G on the lattice L divided by the number of graph automorphisms, i.e., the symmetry number of the graph. It is necessary to divide by the symmetry number $s_{G}$ because the number of monomorphisms is in general not equal to the number of subgraphs because there may be multiple monomorphisms that map to the identical subgraph [195]. To properly account for this, we have to divide by the number of automorphisms.

In Figure 10, you see an example for the embedding problem. We can recognise the fact that there are multiple monomorphisms by the presence of arrows illustrating the ambiguity of mapping onto a specific subgraph embedding. Further, we always have to consider reduced graph contributions subtracting all subgraph contributions as
(148)$$E_{0}\left(G\right)=\langle 0|H_{\mathrm{eff}}|0\rangle -\sum_{G^{\prime }\subset G}W\left(G^{\prime },G\right)\;E_{0}\left(G^{\prime }\right)\;.$$ Here, $W(G^{\prime },G)$ is the embedding factor for subgraphs $G^{\prime }$ with respect to the considered graph G. Note again that, with an efficient white-graph implementation, reduced contributions are calculated on the fly without the need for explicit subtraction. We state the subtraction scheme for completeness. Going back to the embedding problem for the infinite lattice L, the embedding factor will be extensive as the ground-state energy is extensive as well. Usually, intensive quantities such as the energy per site $\epsilon_{0}=E_{0}/N$ are calculated, where *N* is the number of sites, and an arbitrary edge of the graph is fixed on the lattice. This then gives
(149)$$\epsilon_{0}=\sum_{G\subset L}w\left(G,L\right)E_{0}\left(G\right),$$
where the normalised embedding factor is
(150)$$w\left(G,L\right)=\frac{W(G,L)}{N}=\frac{q}{s_{G}}\left|\mathrm{Mono}\left(G_{c},L_{c}\right)\right|,$$
denoted with a lower case *w*, where *q* is the coordination number of the lattice (the number of neighbours of any site) [155,195]. One can think about fixing sites as colouring two adjacent vertices on the graph and two adjacent sites on the lattice and considering the monomorphism with respect to the additional colour attributes (for instance, $A_{V}=\left\{\left\{\mu ,\mathrm{yellow}\right\},\left\{\nu ,\mathrm{blue}\right\}\right\}$) of the coloured graph $G_{c}$ and lattice $L_{c}$. In Figure 10, we depict the number of embeddings for a given graph on the square lattice and show how the embedding factor is calculated for this example. Note that it would be equally valid to just fix a single site. Then, one would not have to account for the coordination number *q* and the number of monomorphism would be larger by a factor of *q*, making it computationally more expensive. In the end, we obtain the ground-state energy (per site) as a high-order perturbative series in an expansion parameter λ up to a computationally feasible maximal order $o_{\max}$ as in Equation (Equation 110).

*1qp dispersion:* We now turn to the one-quasiparticle channel and calculate the hopping amplitudes. For a quasiparticle hopping δ on the lattice with additional hopping within the unit cell from ξ to τ or a quasiparticle changing its flavour from ξ to τ, we need to calculate
(151)$$a_{\xi ,\tau }\left(\delta \right)=\langle 1;i+\delta ,\tau |H_{\mathrm{eff}}|1;i,\xi \rangle =\sum_{G\subset L}\sum_{(\mu ,\nu )\in P}w\left(G_{c},L_{c}\right)\;t_{\mu ;\nu }\left(G\right)\;,$$
where we fix the initial vertex μ and end vertex ν of the quasiparticle process on the graph to the initial site i and end site i+δ in real space. We can choose an arbitrary site i without loss of generality due to the translational symmetry of the models considered. (If we consider larger unit cells, we also need to fix the sites ξ, τ witin the unit cells i and i+δ.) The fixing of sites can be formally achieved by assigning colours to the initial and end sites on the graph and on the lattice. The second sum goes over all representative hopping processes P, as described above in Section 4.6.2. The embedding factor here is similar to Equation (Equation 150) and reads
(152)$$w\left(G_{c},L_{c}\right)=\frac{s_{G_{c}}}{s_{G}}\left|\mathrm{Mono}\left(G_{c},L_{c}\right)\right|\;,$$
where the colour attribute comes from fixing the sites to the hopping. We account for the reduced symmetry of coloured graphs stemming from the representative hopping by multiplying by the symmetry number of the coloured graph $s_{G_{c}}$.

The embedding factor is then calculated with respect to the coloured graph $G_{c}$ and lattice $L_{c}$. Again, we need to be careful as we have to consider reduced and additive contributions, i.e., we have to determine
(153)$$t_{\mu ;\nu }\left(G\right)=\langle 1;\nu |H_{\mathrm{eff}}|1;\mu \rangle -\delta_{\mu ,\nu }E_{0}\left(G\right)-\sum_{G^{\prime }\subset G}\sum_{(\mu ,\nu )\in P}W\left(G_{c}^{\prime },G_{c}\right)\;t_{\mu ;\nu }\left(G^{\prime }\right)\;,$$
for each contribution on a graph G. After having determined the ground-state energy per site $\epsilon_{0}$ and the hopping amplitudes $a_{\xi ,\tau }\left(\delta \right)$ on the lattice L, we can derive the effective one-quasiparticle Hamiltonian (Equation 113). As we have seen in Section 4.5, we can readily derive the one-quasiparticle gap as a series in the perturbation parameter λ as in Equation (Equation 117).

*1qp spectral weight:* Lastly, we can do the same for one-quasiparticle spectral weights. We calculate the process amplitudes:(154)$${\tilde{a}}_{\tau }\left(\delta \right)=\langle 1;\delta ,\tau |O_{\mathrm{eff},i}|0\rangle =\sum_{G\subset L}\sum_{(\mu ,\nu )\in P}w\left(G_{c},L_{c}\right)\;{\tilde{t}}_{\mu ;\nu }\left(G\right)\;.$$ Note that a process creating a quasiparticle at μ that subsequently hops to ν with the contribution ${\tilde{t}}_{\mu ;\nu }\left(G\right)$ is in general distinct to the inverse process, i.e., ${\tilde{t}}_{\mu ;\nu }\left(G\right)\neq {\tilde{t}}_{\nu ;\mu }\left(G\right)$. On the other hand, the 1qp hopping processes of the Hamiltonian fulfil $t_{\mu ;\nu }\left(G\right)=t_{\nu ;\mu }\left(G\right)$ as long as the hopping amplitudes are real. While for hopping processes of the Hamiltonian, we can use the same colour (for start and end vertex) as a topological attribute, here, we must use two different colours, leading to a smaller symmetry number of the coloured graphs. On the graph level, only subgraph contributions must be subtracted:(155)$${\tilde{t}}_{\mu ;\nu }\left(G\right)=\langle 1;\nu |O_{\mathrm{eff},\mu }|0\rangle -\sum_{G^{\prime }\subset G}\sum_{(\mu ,\nu )\in P}W\left(G_{c}^{\prime },G_{c}\right)\;t_{\mu ;\nu }\left(G^{\prime }\right)\;.$$ With the contributions ${\tilde{a}}_{\tau }\left(\delta \right)$, the spectral weight can be determined with Equation (Equation 131), and evaluating this quantity for example at the critical momentum $k_{c}$, we again obtain a series (Equation 132) in the perturbation parameter λ.

For the conventional embedding problem, we can also consider several generalisations, which we want to briefly mention. First, we could consider quasiparticle processes between different particle types. In a 1qp process, a particle flavour could change from ξ to τ and the graph contribution would be denoted by including the additional indices $t_{\mu ,\xi ;\nu ,\tau }$ and ${\tilde{t}}_{\mu ;\nu ,\tau }$. The rest of the formalism is identical. Second, we may want to consider different interaction types like in Equation (Equation 138) or more expansion parameter like in Equation (Equation 137). In such a case, when considering coloured graphs, we also have to consider their coloured edges, which must be matched with the appropriate bonds on the lattice. Then, the coloured graphs $G_{c}$ are given by the tuple $\left(V_{G},E_{G},A_{E},A_{V}\right)$, where $A_{E}$ are the edge colour attributes and $A_{V}$ the vertex colour attributes. Now, every embedding factor *w* needs to be determined using coloured graphs with respect to $A_{E}$ and $A_{V}$. When using white graphs, the additional edge colour can be ignored and the embedding is as before; however, we need to appropriately substitute the abstract expansion parameters with the actual physical expansion parameters.

By now, we have everything together to calculate the ground-state energy, the 1qp dispersion, and the 1qp spectral weight in the thermodynamic limit from a linked-cluster expansion set up as a full graph decomposition. Therefore, we need to calculate the embedding factor, multiply them with the associated graph contribution, and add up the resulting weighted contributions. The embedding factor can be determined using available graph libraries like the *Boost Graph Library* [203] as only automorphisms and monomorphisms (with colours) need to be determined. For long-range interactions, however, we cannot just simply calculate the embedding factor because, for every graph, there are infinitely many possible embeddings, even when accounting for translational invariance.

#### 4.7.2. Embedding for Models with Long-Range Interactions

In this section, we restrict ourselves to a single physical perturbation parameter. If we had more than one perturbation parameter, this would introduce an additional functional dependency, which we want to avoid as we perform the Monte Carlo summation. A convenient way around this is to sample the parameters and perform the Monte Carlo summation for each parameter ratio like it was performed in [31,35,206] for anisotropic XY interactions, distinct ladder interactions, and for XXZ interactions. In the following, we restrict ourselves to a single quasiparticle flavour due to simplicity and a trivial unit cell, as we have not generalised the algorithm to larger unit cells yet. The starting point to describe the embedding problem for long-range interactions is the embedding formulas given in the previous section in Equations (Equation 149), (Equation 151), and (Equation 154) for the ground-state energy per site, 1qp hopping processes, and the 1qp spectral weight, respectively, which we have to rewrite and adapt for long-range interactions.

*Ground-state energy:* Starting with the embedding of the ground-state energy per site (Equation 149), we can write
(156)$$\begin{array}{cc} \epsilon_{0} & =\sum_{G}w\left(G,L\right)E_{0}\left(G\right)\\ \; & =\sum_{o=2}^{o_{\max}}\sum_{G}w\left(G,L\right)E_{0}^{\left(o\right)}\left(G\right)\\ \; & =\sum_{o=2}^{o_{\max}}\sum_{n_{s}=2}^{o}\sum_{\begin{array}{c} G\\ |V_{G}|=n_{s} \end{array}}w\left(G,L\right)E_{0}^{\left(o\right)}\left(G\right)\;. \end{array}$$ We have replaced G⊂L with G in the sum to emphasise that the graphs are not restricted to the nearest-neighbour lattice geometry any longer due to the long-range interactions, making the lattice a fully connected graph. From the first to second line, we decomposed the ground-state energy contribution from graphs into contributions of individual orders $E_{0}\left(G\right)=\sum_{o=2}^{o_{\max}}E_{0}^{\left(o\right)}$. Note that the minimum order of the ground-state energy is o=2. From the second to third line, we introduced a second sum over vertices $n_{s}$ and restricted the sum over all graphs to a sum over graphs with a fixed number of vertices $n_{s}$. Next, we can reformulate the embedding factor in Equation (Equation 147) as
(157)$$w\left(G,L\right)=\sum_{c\in C}\frac{1}{s_{G}}\;,$$
where we replaced the expression for the number of monomorphisms (divided by *N*) with a sum over all possible configurations C. A configuration is nothing else than the current embedding of graphs. As we will see later, we can calculate the contributions of multiple graphs simultaneously as the Monte Carlo sum only depends on the number of sites (we use “site” also as a synonym for “vertex position”). When using the word configuration, we think about it as the current set of vertex positions on the lattice. The sum over all configurations comprises individual sums for each vertex over all lattice sites excluding configurations where vertices overlap, as shown in the next subsection in Equation (Equation 180). One may falsely conclude that a sum over all configurations should result in the un-normalised embedding factor W, but as we will see in the following, by substituting the abstract expansion parameters with the physical long-range interactions, only the relative distance between sites is relevant for the contribution in the summand, irrespective of the absolute position of the sites on the lattice. We can also see the reason behind splitting the graph set into sets with graphs of a fixed number of vertices because we can now group all graph contributions with a given number of sites into a single integrand because for long-range interactions there are no constraints on the embeddings (except for overlaps) and the integrand only depends on the (relative) position of the vertices. Further, the white graph contributions $E_{0}^{\left(o\right)}\left(G\right)$ still need to be replaced with the correct colour, i.e., the general expansion parameters need to be substituted with the algebraically decaying long-range interaction depending on the current configuration. In reality, the contribution $E_{0}^{\left(o\right)}\left(G,c\right)$ depends on the current configuration *c*. Thus, replacing the expression with an explicit sum is necessary as the contribution for each configuration is different and w(G,L) cannot just be a number. The substitution must look like
(158)$$E_{0}^{\left(o\right)}\left(G,c\right):=E_{0}^{\left(o\right)}\left(G;\{\lambda_{e}\mapsto |\delta {|}^{-(d+\sigma )}\}\right)=\sum_{m}v_{m}\left(G\right)\prod_{e\in E_{G}}\frac{1}{{\left|\delta \right|}^{-n_{e,m}\left(d+\sigma \right)}}\;,$$
where the index *m* of the sum runs over all monomials of the contribution $E_{0}^{\left(o\right)}\left(G,c\right)$ and the product is over all edges $e=\left\{\mu ,\nu \right\}\in E_{G}$ of the graph G with the adjacent vertices $\mu ,\nu \in V_{G}$. The power law in the product arises from substituting the expansion parameters $\lambda_{e}\mapsto {\left|\delta \right|}^{-\alpha }$ on the edges *e* with the appropriate algebraically decaying long-range interaction of the current embedding (cf. Equation (Equation 140)). The adjacent vertices μ,ν are embedded on the lattice sites $i_{\mu }$ and $i_{\nu }$ with the distance $\delta =i_{\nu }-i_{\mu }$, and the multiplicity $n_{e,m}\in N$ comes from the power $n_{e}$ of the associated expansion parameter $\lambda_{e}$. This way, we can reduce the many expansion parameters from the white graph contribution to a single physical perturbation parameter λ, and by reordering the expression (Equation 156) of the ground-state energy, we have
(159)$$\epsilon_{0}=\sum_{o=2}^{o_{\max}}\sum_{n_{s}=2}^{o}\sum_{c\in C}\sum_{\begin{array}{c} G\\ |V_{G}|=n_{s} \end{array}}\frac{1}{s_{G}}E_{0}^{\left(o\right)}\left(G,c\right)\lambda^{o}\;.$$ We can define
(160)$$\begin{array}{cc} f_{n_{s}}^{\left(o\right)}\left(c\right) & :=\sum_{\begin{array}{c} G\\ |V_{G}|=n_{s} \end{array}}\frac{1}{s_{G}}E_{0}^{\left(o\right)}\left(G,c\right)\;, \end{array}$$
(161)$$\begin{array}{cc} S[f_{n_{s}}^{\left(o\right)}] & :=\sum_{c\in C}f_{n_{s}}^{\left(o\right)}\left(c\right)\;, \end{array}$$
where $f_{n_{s}}^{\left(o\right)}\left(c\right)$ is the integrand function and S[·] denotes the associated sum over all possible embeddings on the lattice that will be evaluated using a classical Monte Carlo algorithm. Since Monte Carlo runs are usually performed for a batch of different seeds, we introduce an additional sum over seeds averaging the Monte Carlo runs, which we denote as
(162)$$\bar{S}\left[\cdot \right]=\frac{1}{N_{\mathrm{seeds}}}\sum_{s=1}^{N_{\mathrm{seeds}}}S\left[\cdot \right]\;.$$ Eventually, this yields the expression:(163)$$\epsilon_{0}=\sum_{o=2}^{o_{\max}}\sum_{n_{s}=2}^{o}\bar{S}\left[f_{n_{s}}^{\left(o\right)}\right]\lambda^{o}\;.$$ To express the ground-state energy $\epsilon_{0}$ as a perturbative series, we write
(164)$$\epsilon_{0}=p_{0}+\sum_{o=2}^{o_{\max}}p_{o}\lambda^{o}\;\mathrm{with}\;p_{i}=\sum_{n_{s}=2}^{o}\bar{S}\left[f_{n_{s}}^{\left(o\right)}\right]\;,$$
where we have to sum up the contributions of multiple Monte Carlo runs to obtain the series prefactors $p_{i}$ for i>0. The zeroth prefactor $p_{0}$ is simply given by the ground-state energy of $H_{0}$. As a result of the Monte Carlo summation, these prefactors $p_{i}$ carry an uncertainty that can be estimated from the standard deviation obtained by averaging over the results for all seeds.

*1qp dispersion:* We now turn to extracting the one-quasiparticle dispersion. In Equation (Equation 116), we have seen that the dispersion can be analytically determined in terms of the hopping amplitudes of Equation (Equation 151) up to some perturbative order. For nearest-neighbour interactions, it is an analytic function in k; however, for long-range interactions, there are infinitely many hoppings possible at any order, so we can neither explicitly determine the hopping amplitudes a(δ), nor is it possible to have the dispersion as an analytical function in k. This would introduce a functional dependence in the integrand that we cannot sample. Instead, we will have to evaluate the dispersion for certain values $k^{★}$ to obtain an explicit series in the perturbation parameter λ. Evaluating Equation (Equation 116) at $k=k^{★}$ and inserting Equation (Equation 151), we can write
(165)$$\begin{array}{cc} \omega (k=k^{★}) & =a\left(0\right)+2\sum_{\delta }a\left(\delta \right)\cos\left(k^{★}\delta \right)\\ \; & =\sum_{\delta }\Xi \left(k^{★},\delta \right)a\left(\delta \right)=\sum_{\delta }\Xi \left(k^{★},\delta \right)\sum_{G}\sum_{(\mu ,\nu )\in P}w\left(G_{c},L_{c}\right)t_{\mu ;\nu }\left(G\right)\;, \end{array}$$
where we rewrote the sum over δ by introducing the function:(166)$$\Xi \left(k^{★},\delta \right)=\left\{\begin{array}{cc} 1 & \delta =0,\\ 2\cos(k^{★}\delta ) & \delta \neq 0. \end{array}\right.$$ Again, we can split $t_{\mu ;\nu }$ into the contributions of individual orders $t_{\mu ;\nu }^{\left(o\right)}$ and split the graph set into subsets of a fixed number of sites $n_{s}=\left|V_{G}\right|$, yielding
(167)$$\omega \left(k^{★}\right)=\sum_{o=1}^{o_{\max}}\sum_{n_{s}=2}^{o+1}\sum_{\begin{array}{c} G\\ |V_{G}|=n_{s} \end{array}}\sum_{(\mu ,\nu )\in P}\sum_{\delta }\Xi \left(k^{★},\delta \right)w\left(G_{c},L_{c}\right)t_{\mu ;\nu }^{\left(o\right)}\left(G\right)\;.$$ Note here that the second sum goes until o+1, while for the ground-state energy, it runs only until o. The maximal number of sites at a given order is o+1 because graphs with o edges can maximally have o+1 sites. For the ground-state energy fluctuations, every quasiparticle that is created has to be annihilated again, so at order o, a process can only touch maximal o−1 edges, which restricts the sum to o sites.

Now, we argue that we can drop the sum over δ by thinking differently about this embedding problem for the dispersion. The information of the start and end vertex of the hopping process is encoded into vertex colours, and when finding the subgraph monomorphisms for the embedding on the infinite lattice L, the colours of the vertices must match the coloured sites on the lattice, i.e., the hopping vertices are fixed to the hopping sites on the lattice. Since the long-range interactions allow any hopping—i.e., of any distance—at any order, it is not useful to think in this picture. Instead, we should think about the embedding problem analogous to the one for the ground-state energy, where no such hopping constraint exists and the embedding factor W is simply proportional to a sum over all configurations. This is valid as we let the sum over all lattice sites and account for constraints on δ by multiplying with the symmetry number of the coloured graph $s_{G_{c}}$. The relevant hopping information of the vertices, which was previously fixed by coloured vertices, is anyway encoded into the cosine terms. Hence, we can make the substitution:(168)$$\sum_{\delta }\Xi \left(k^{★},\delta \right)w\left(G_{c},L_{c}\right)=\sum_{c\in C}\frac{s_{G_{c}}}{s_{G}}\cos\left(k^{★}\delta \right)\;,$$
where we account for the reduced symmetry of the graph due to the hopping by multiplying with the symmetry number of the coloured graph $s_{G_{c}}$. As before, we need to substitute the general white graph contribution with the actual algebraically decaying long-range interactions of the current embedding:(169)$$t_{\mu ;\nu }^{\left(o\right)}\left(G,c\right):=t_{\mu ;\nu }^{\left(o\right)}\left(G;\{\lambda_{e}\mapsto |\delta {|}^{-\alpha }\}\right)=\sum_{m}v_{m}\left(G\right)\prod_{e\in E_{G}}\frac{1}{{\left|\delta \right|}^{-n_{e,m}\alpha }}\;.$$ Inserting Equation (Equation 168) into Equation (Equation 167), we end up with the expression:(170)$$\omega \left(k^{★}\right)=\sum_{o=1}^{o_{\max}}\sum_{n_{s}=2}^{o+1}\sum_{c\in C}\sum_{\begin{array}{c} G\\ |V_{G}|=n_{s} \end{array}}\sum_{(\mu ,\nu )\in P}\frac{s_{G_{c}}}{s_{G}}t_{\mu ;\nu }^{\left(o\right)}\left(G,c\right)\cos\left(k^{★}\delta \right)\lambda^{o}\;.$$ For a lighter notation, we again define the integrand function and the Monte Carlo sum:(171)$$\begin{array}{cc} f_{n_{s},k^{★}}^{\left(o\right)}\left(c\right) & :=\sum_{\begin{array}{c} G\\ |V_{G}|=n_{s} \end{array}}\sum_{(\mu ,\nu )\in P}\frac{s_{G_{c}}}{s_{G}}t_{\mu ;\nu }^{\left(o\right)}\left(G,c\right)\cos\left(k^{★}\delta \right)\;, \end{array}$$(172)$$\begin{array}{cc} S[f_{n_{s},k^{★}}^{\left(o\right)}] & :=\sum_{c\in C}f_{n_{s},k^{★}}^{\left(o\right)}\left(c\right)\;. \end{array}$$ Introducing an average over a batch of seeds for the MC sum $\bar{S}\left[\cdot \right]$, we obtain
(173)$$\omega \left(k^{★}\right)=\sum_{o=1}^{o_{\max}}\sum_{n_{s}=2}^{o+1}\bar{S}\left[f_{n_{s},k^{★}}^{\left(o\right)}\right]\lambda^{o}\;.$$ The perturbative series of the dispersion evaluated at $k=k^{★}$ can then be expressed as
(174)$$\omega \left(k^{★}\right)=p_{0}+\sum_{o=1}^{o_{\max}}p_{o}\lambda^{o}\;\mathrm{with}\;p_{i}=\sum_{n_{s}=2}^{o+1}\bar{S}\left[f_{n_{s},k^{★}}^{\left(o\right)}\right]\;.$$ The sum prefactors $p_{i}$ for i>0 can be determined by summing up the individual contributions from the Monte Carlo runs and the prefactor $p_{0}$ is given by the energy gap of $H_{0}$.

*1qp spectral weight:* Lastly, the evaluation for the spectral weight observable is analogous to the 1qp dispersion. The integrand and Monte Carlo sum are defined as
(175)$$\begin{array}{cc} f_{n_{s},k^{★}}^{\left(o\right)}\left(c\right) & :=\sum_{\begin{array}{c} G\\ |V_{G}|=n_{s} \end{array}}\sum_{(\mu ,\nu )\in P}\frac{s_{G_{c}}}{s_{G}}{\tilde{t}}_{\mu ;\nu }^{\left(o\right)}\left(G,c\right)\cos\left(k^{★}\delta \right)\;, \end{array}$$
(176)$$\begin{array}{cc} S[f_{n_{s},k^{★}}^{\left(o\right)}] & :=\sum_{c\in C}f_{n_{s},k^{★}}^{\left(o\right)}\left(c\right)\;, \end{array}$$
which we use to calculate
(177)$$s\left(k^{★}\right)=p_{0}+\sum_{o=1}^{o_{\max}}p_{o}\lambda^{o}\;\mathrm{with}\;p_{i}=\sum_{n_{s}=2}^{o+1}\bar{S}\left[f_{n_{s},k^{★}}^{\left(o\right)}\right]\;.$$ We determine $s(k^{★})$ by again calculating the series prefactors $p_{i}$ for i>0 and then determine the 1qp spectral weight with $S^{1\mathrm{qp}}\left(k^{★}\right)={\left|s\left(k^{★}\right)\right|}^{2}$.

It should be noted that we have to perform
(178)$$n=\frac{o_{\max}\left(o_{\max}-1\right)}{2}\times N_{\mathrm{seeds}}$$ Monte Carlo runs for a series of order $o_{\max}$ with $N_{\mathrm{seeds}}$. This means the number of runs grows quadratically with the maximal order $o_{\max}$.

So far, we have derived the necessary formalism for how to express the embedding problem of models with long-range interactions, but we have not talked about how to evaluate the Monte Carlo sums S[·] for the integrand functions *f*. In the next section, we investigate how to evaluate such sums by introducing a suitable Monte Carlo algorithm.

#### 4.7.3. Monte Carlo Algorithm for the Long-Range Embedding Problem

We are left with evaluating the Monte Carlo sum S[·], which runs over all configurations C of graphs. The embeddings on the lattice depend only on the number of vertices of a graph G, and there is no constraint by the edge set as in the nearest-neighbour case because every site of the lattice interacts with any other site, making it a fully connected graph. The only restriction for models with long-range interactions is that the vertices of the graph are not allowed to overlap as they do not self-interact and an overlap would imply infinite interaction strength resulting from the term ${\left|\delta \right|}^{-(d+\sigma )}$. In conclusion, we can write the sum over all possible configurations as number-of-sites-many sums over all possible lattice positions: (179)S[fns(o)]=∑c∈Cfns(o)(c)=∑i1∑′⋯∑ins∑′fns(o)(i1,⋯,ins),(180)S[fns,k★(o)]=∑c∈Cfns,k★(o)(c)=∑i1∑′⋯∑ins∑′fns,k★(o)(i1,⋯,ins),
where the primed sum ∑i∑′ over vertex position i is the short notation for excluding overlaps with any other vertex position. For example, for the ground-state contribution with three sites on a one-dimensional chain, this sum would look like
(181)$$S\left[f_{n_{s}}^{\left(o\right)}\right]=\sum_{k=-\infty }^{\infty }\sum_{\begin{array}{c} j=-\infty \\ j\neq k \end{array}}^{\infty }\sum_{\begin{array}{c} i=-\infty \\ i\neq j\\ i\neq k \end{array}}^{\infty }f_{n_{s}}^{\left(o\right)}\left(i,j,k\right)\;.$$ Due to the overlap constraint, these are nested high-dimensional sums over the integrand functions $f_{n_{s}}^{\left(o\right)}$, which are in general hard to solve. The dimensionality of the MC problem is given by $d_{\mathrm{sum}}=n_{s}\cdot d$ because higher dimensions *d* of the system introduce additional sums for each component. If we wanted to evaluate the Monte Carlo sum in two dimensions for contributions with eight sites, which already occurs in eighth-order perturbation theory, we would have to determine the integral value of 16 nested sums over the integrand function $f_{8}^{\left(8\right)}$. This makes it clear that the evaluation of such sums becomes challenging very quickly. In the following, we use the short notation $f_{n_{s}}$, interchangeable for both the ground-state energy integrand $f_{n_{s}}^{\left(o\right)}$ and the 1qp process integrands $f_{n_{s},k^{★}}^{\left(o\right)}$.

The first approach to tackle the problem of evaluating these sums using conventional numerical integration techniques was pioneered by S. Fey and K.P. Schmidt already in 2016 [25]. They managed to successfully determine the phase diagram and critical exponents from the closing of the gap of the long-range transverse-field Ising model on a one-dimensional chain with ferro- and antiferromagnetic interactions. While they were successful with their approach over a large range of decay exponents in one dimension, the extraction of the critical properties for small decay exponents was challenging. The two-dimensional problem was out of reach with this approach as the number of nested sums doubles and the sums converge significantly more slowly. Here, Monte Carlo integration came into play as it is known to be a powerful integration technique for high-dimensional problems where conventional integration fails. The reason behind the slow convergence of such high-dimensional sums is often that the configuration space where the integrand mainly contributes to the integral is significantly smaller than the entire configuration space. In 2019, Fey et al. [29] introduced a Markov chain Monte Carlo algorithm to efficiently sample the relevant configuration space. They were able to determine the quantum-critical properties of the long-range TFIM on two-dimensional lattices to even higher precision than previously for the one-dimensional chain, extending the accessible range of decay exponents without having to forfeit perturbative orders in higher dimensions.

In the following, we describe the Markov chain Monte Carlo algorithm introduced by Ref. [29] to evaluate the high-dimensional nested sums. To sample the relevant configuration space efficiently, we use importance sampling with respect to some convenient probability weight π(c) with respect to a configuration *c* and the associated partition function $Z=\sum_{c}\pi \left(c\right)$. We can insert an identity into Equation (Equation 179) or Equation (Equation 180) and rewrite it as
(182)$$S\left[f_{n_{s}}\right]=\sum_{c\in C}\underset{=1}{\underset{︸}{\frac{\pi (c)}{Z}\frac{Z}{\pi (c)}}}f_{n_{s}}\left(c\right)=Z{\langle \frac{f_{n_{s}}\left(c\right)}{\pi (c)}\rangle }_{\pi }\;,$$
where π(c)/Z can be interpreted as the probability of being in configuration *c*. The integrand now reads as the contribution $f_{n_{s}}$ (we dropped the order o and momentum $k^{★}$ as indices to lighten the notation) from configuration *c* multiplied by its probability, which allows us to write the sum as the expectation value ${\langle \cdot \rangle }_{\pi }$ of $f_{n_{s}}\left(c\right)/\pi \left(c\right)$ with respect to the weight π. We later call this sum “target sum”. Since the partition function *Z* is not known a priori, we also introduce a “reference sum”:(183)$$S\left[f_{n_{s}}^{\mathrm{ref}}\right]=\sum_{c\in C}f_{n_{s}}^{\mathrm{ref}}\left(c\right)=Z{\langle \frac{f_{n_{s}}^{\mathrm{ref}}\left(c\right)}{\pi (c)}\rangle }_{\pi }\;,$$
over a reference function $f_{n_{s}}^{\mathrm{ref}}$. We require this sum to be analytically solvable to avoid introducing an additional source of error. We denote its analytical expression as $S_{n_{s}}^{\mathrm{ref}}$. The reference function $f_{n_{s}}^{\mathrm{ref}}$ should behave similarly to the integrand function of interest $f_{n_{s}}$. This means that the reference sum and target sum should have considerable contributions in the same area of the configuration space and their asymptotic behaviour should be similar as well. Although we could make, in principle, an arbitrary choice for the reference function, the latter properties guarantee to lead to good convergence. In one dimension, we choose the reference integrand as
(184)$$f_{n_{s}}^{\mathrm{ref}}\left(c\right)=\prod_{n=1}^{n_{s}-1}\frac{1}{|i_{n+1}-i_{n}{|}^{\rho }}=\prod_{n=1}^{n_{s}-1}\frac{1}{|\delta_{n}{|}^{\rho }}$$
with $\delta_{n}=i_{n+1}-i_{n}$ and the reference integrand exponent ρ, which is a free simulation parameter. We can solve the reference sum as follows:(185)Sfnsref=∑c∈C∏n=1ns−11|in+1−in|ρ=∑i1∑′⋯∑iN∑′1|i2−i1|ρ⋯1|ins−ins−1|ρ=ns!∑i1<i2⋯∑ins−1<ins1|i2−i1|ρ⋯1|ins−ins−1|ρ=ns!∑i1<i2⋯∑ins−1<ins1|δ1|ρ⋯1|δns−1|ρ=ns!∏n=1ns−1∑δ=1∞1δρ=ns!ζ(ρ)ns−1,
where ζ(ρ) is the Riemann ζ function and we accounted for $n_{s}!$ possibilities to randomly embed the vertices by ordering the indices of the sums. One major difference between the reference and the target sum is that, in the target sum, many different graph contributions contribute. In fact, the reference sum above is exactly the contribution of order $o=n_{s}-1$ of a chain graph with $n_{s}$ vertices and the contribution from the associated target sum is the same up to a linear factor. In higher dimensions, we cannot choose a contribution proportional to the one of a chain graph any longer since it cannot be solved analytically. Instead, we make simplifications to the reference sum and require that the reference sum is still good enough to capture the same properties as the target sum. We choose to decouple the dimensions in the reference integrand:(186)$$f_{n_{s}}^{\mathrm{ref}}\left(c\right)=\prod_{n=1}^{d}\prod_{\nu =1}^{n_{s}-1}\frac{1}{\left(1+\right|i_{\nu +1,n}-i_{\nu ,n}{\left|\right)}^{\rho }}=\prod_{n=1}^{d}\prod_{\nu =1}^{n_{s}-1}\frac{1}{\left(1+\right|\delta_{\nu ,n}{\left|\right)}^{\rho }}$$
and explicitly allow overlaps in the reference sum, such that it can be solved analytically as follows:(187)$$\begin{array}{cc} S\left[f_{n_{s}}^{\mathrm{ref}}\right] & =\sum_{i_{1,1}=-\infty }^{\infty }\cdots \sum_{i_{n_{s},d}=-\infty }^{\infty }\frac{1}{\left(1+\right|i_{2,1}-i_{1,1}{\left|\right)}^{\rho }}\cdots \frac{1}{\left(1+\right|i_{n_{s},d}-i_{n_{s}-1,d}{\left|\right)}^{\rho }}\\ \; & =\prod_{n=1}^{d(n_{s}-1)}\sum_{\delta =-\infty }^{\infty }\frac{1}{{\left(1+|\delta |\right)}^{\rho }}=\prod_{n=1}^{d(n_{s}-1)}\left(2\sum_{\delta =0}^{\infty }\frac{1}{{\left(1+\delta \right)}^{\rho }}-1\right)\\ \; & ={\left(2\zeta (\rho )-1\right)}^{d(n_{s}-1)}\;. \end{array}$$ Although the exponent ρ can be chosen freely, we want to achieve similar asymptotic behaviour as the target integrand; therefore, we choose ρ=1+σ in one dimension as the reference sum exactly behaves like the target integrand of a chain graph. In two dimensions, we made some simplifications to the reference sum, and we have to adopt the parameter as ρ=(d+σ)/2 for σ<5, ρ=3 for 5≤σ<7, and ρ=3.5 for σ≥7. This is by no means a rigorous choice, but it empirically proved to produce good convergence [29,207]. Solving Equation (Equation 183) for *Z* and inserting it into Equation (Equation 182), we obtain
(188)$$S\left[f_{n_{s}}\right]=\frac{{\langle \frac{f_{n_{s}}\left(c\right)}{\pi (c)}\rangle }_{\pi }}{{\langle \frac{f_{n_{s}}^{\mathrm{ref}}\left(c\right)}{\pi (c)}\rangle }_{\pi }}S_{n_{s}}^{\mathrm{ref}}\;.$$ We use the analytic expression of Equation (Equation 185) in 1D or Equation (Equation 187) in 2D for the reference sum $S_{n_{s}}^{\mathrm{ref}}$. We got rid of the partition function *Z* and now can use this expression in our Monte Carlo run to determine the sum S[·] using the analytic expression of the reference sum $S_{n_{s}}^{\mathrm{ref}}$ while tracking the running averages in the numerator and denominator expressions.

We are left with just one missing ingredient, which is the choice of the probability weight π(c). For our choice to be a probability weight, it must fulfil π(c)≥0, and we want the weight to be the largest if both the reference and the target integrand contribute the most. An obvious choice may be the quadratic mean:(189)$$\pi \left(c\right)=\sqrt{{\left(f_{n_{s}}^{\mathrm{ref}}\left(c\right)\right)}^{2}+{\left(f_{n_{s}}\left(c\right)\right)}^{2}}\;,$$
which is always ≥0 and rotationally invariant in $f_{n_{s}}^{\mathrm{ref}}$ and $f_{n_{s}}$. However, we also want both quantities to contribute equally to the probability weight on average over all configurations. As the contributions of the target and reference sum may differ significantly, we introduce a factor for rescaling:(190)$$R=\left|\frac{S\left[f_{n_{s}}^{\mathrm{ref}}\right]}{S\left[f_{n_{s}}\right]}\right|\;,$$
which can be estimated in an in-advance calibration run. We then use an adjusted probability weight:(191)$$\pi \left(c\right)=\sqrt{{\left(f_{n_{s}}^{\mathrm{ref}}\left(c\right)\right)}^{2}+R^{2}{\left(f_{n_{s}}\left(c\right)\right)}^{2}}$$
for the actual Monte Carlo run. The weight needs to be evaluated at every Monte Carlo step to track the running averages of the numerator and denominator in Equation (Equation 188).

Now, we have everything together to describe the actual Monte Carlo algorithm. We employ a Markov chain Monte Carlo algorithm, where we need to sample the configuration space C according to the probability weight π in Equation (Equation 191). Each configuration with a non-zero weight must be in principle accessible by the Markov chain. On the one hand, we propose a high acceptance rate of the Monte Carlo steps to sample the large configuration space efficiently, not staying in the same configuration for too long. On the other hand, we want to sample closely confined configurations with rather small distances between vertices more often than configurations that are farther apart (configurations with large distances between vertices) such that we capture the asymptotic behaviour of the model. The interaction strength decays algebraically with the distance between vertices, leading to smaller contributions for configurations in which sites are far apart. What we call a confined configuration, therefore, depends on the decay exponent of the long-range interaction σ. In the algorithm, we have the free exponent parameter ρ (for the reference sum) and γ (for probability distributions), which can be changed to tweak this behaviour, but are usually chosen similar or equal to d+σ. In two dimensions, we had to adapt the values of ρ to obtain a similar asymptotic behaviour for the reference and target sum due the approximations we made. An optimal choice of these parameters ensures fast convergence of the Monte Carlo sum. The current embedding configuration should be represented as an array container where the entries are the positions of the graph vertices. In one dimension, entries are simple integers, while in higher dimensions, the position needs to be represented as a mathematical vector. For small decay exponents σ, very large distances can occur between vertices from time to time, which need to be squared when calculating the absolute distance. This can lead to an integer overflow, and therefore, the use of a 128-bit integer may be considered. Further, we define functions in the programme for the target integrand $f_{n_{s}}$ and for the reference integrand $f_{n_{s}}^{\mathrm{ref}}$, where the current configuration is passed as a parameter and the function returns the contribution from the integrand evaluated for the current configuration.

Turning back to the sampling scheme, the idea of the Markov chain is to interpret the vertex positions on the lattices as random walkers. We randomly select a graph vertex and then draw a new position from a probability distribution such that the move fulfils the detailed balance condition. In each Monte Carlo step, we perform the following two moves:Shift move:This Monte Carlo move is implemented to introduce confined random fluctuations to the current configuration independent of the strength of the algebraically decaying long-range interactions. It is especially important for larger decay exponents σ when the configurations are much more likely to be confined. First, we randomly select a vertex $n_{\mathrm{sel}}\in \left\{1,\cdots ,n_{s}\right\}$ drawn from a discrete uniform distribution with $p^{\mathrm{sel}}=1/n_{s}$. Second, for the fluctuation, we draw a shift value $d^{\mathrm{prop}}\in \left\{-n_{s},\cdots ,n_{s}\right\}$ from a discrete uniform distribution $p^{\mathrm{shift}}=1/\left(2n_{s}+1\right)$. In one dimension, we have to draw a single time, and in higher dimensions, we draw repeatedly for each component. Subsequently, we add the shift to the position of the selected vertex and propose the position:
(192)$$i^{\mathrm{prop}}=i_{n_{\mathrm{sel}}}+d^{\mathrm{prop}}\;.$$We might have proposed a position that is already occupied by another vertex, so we have to check for overlaps. In one dimension, we reset the proposed position to the original one if there is an overlap, while in higher dimensions, we explicitly allow overlaps. As we remember from above, this distinction is also present in the reference sums in one dimension in Equation (Equation 185) compared to higher dimensions in Equation (Equation 187). If an overlap occurs in dimensions higher than one, then the target summand is explicitly set to zero such that these configurations cannot contribute (otherwise, the sum would become infinity). Then, we calculate the Metropolis acceptance probability:
(193)$$\begin{array}{cc} p_{\mathrm{acc}}^{\mathrm{shift}} & =\min\left(1,\frac{\pi (c^{\mathrm{prop}})}{\pi (c^{\mathrm{curr}})}\right)\\ \; & =\min\left(1,\frac{\sqrt{{\left(f_{n_{s}}^{\mathrm{ref}}\left(c^{\mathrm{prop}}\right)\right)}^{2}+R^{2}{\left(f_{n_{s}}\left(c^{\mathrm{prop}}\right)\right)}^{2}}}{\sqrt{{\left(f_{n_{s}}^{\mathrm{ref}}\left(c^{\mathrm{curr}}\right)\right)}^{2}+R^{2}{\left(f_{n_{s}}\left(c^{\mathrm{curr}}\right)\right)}^{2}}}\right)\;, \end{array}$$
by determining the probability weights π of the current and the proposed configuration. The result of the target and reference function calls should be saved into variables to prevent redundant and expensive function calls at each Monte Carlo step. Note that the transition weights $\tilde{T}\left(c^{\mathrm{curr}}\to c^{\mathrm{prop}}\right)=p^{\mathrm{sel}}\times p^{\mathrm{shift}}$ cancel out as we draw only from uniform distributions. Lastly, the minimum function is implemented by drawing a random number y∈0,1, and we accept the proposed move if $y<p_{\mathrm{acc}}^{\mathrm{shift}}$ and update the current configuration. An example of such a shift move is depicted in Figure 11a.Rift move:In contrast to the previous move, which should introduce fluctuations to the configuration independent of the current one and independent of the long-range interaction strength, “rift moves” are introduced to better capture the correct asymptotic behaviour induced by the algebraically decaying interactions. The moves are able to propose very large distances between vertices, but are also able to do the opposite, closing the “rift” between vertices when the configuration is split into essentially two clusters. At first, we select a site $n_{\mathrm{sel}}\in \left\{1,\cdots ,n_{s}-1\right\}$ from the vertex set with discrete uniform probability $p^{\mathrm{sel}}=1/\left(n_{s}-1\right)$, explicitly excluding the last site. In one dimension, we can order the vertex set such that the first vertex is the one with the smallest positional value and the last the one with the largest value, so we order by $i_{n}<i_{m}$, where n,m are vertex indices and $i_{n}$, $i_{m}$ the associated sites on the lattice. The same ordering was also performed when we solved the reference sum in Equation (Equation 185). In higher dimensions, a similar ordering comes at a much higher computational cost, so we stick to the vertex numbering given by the array indices, i.e., the order is n<m. Here, it is also important that the vertex labelling of the reference sum coincides with the labelling of the chain graph. To capture the physical asymptotics of the system, we draw random values from a ζ-function distribution. In one dimension, we draw from
(194)$$p^{\mathrm{rift}}\left(r^{\mathrm{prop}}\right)=\frac{{\left(r^{\mathrm{prop}}\right)}^{-\gamma }}{\zeta (\gamma )},$$
yielding a power-law distribution with $r^{\mathrm{prop}}>0$ with the free exponent parameter γ. We choose γ=d+σ for obvious reasons. The distance to the next vertex is given by $r^{\mathrm{curr}}=i_{n_{\mathrm{sel}}+1}-i_{n_{\mathrm{sel}}}$, and $r^{\mathrm{prop}}$ is the proposed distance drawn from the ζ distribution. Since we ordered by the position and only selected sites in $\left\{1,\cdots ,n_{s}-1\right\}$, it is sufficient to draw positive values only. We shift all indices $i_{n}>i_{n_{\mathrm{sel}}}$ according to
(195)$$i_{n}^{\mathrm{prop}}=i_{n}+\left(r^{\mathrm{prop}}-r^{\mathrm{curr}}\right)\;.$$In higher dimensions, we have no such ordering and, therefore, extend such a distribution to negative values (we refer to it as a “double-sided” ζ-function distribution) and draw random values from
(196)$$p^{\mathrm{rift}}\left(r^{\mathrm{prop}}\right)=\frac{\left(1+\right|r^{\mathrm{prop}}{\left|\right)}^{-\gamma }}{2\zeta (\gamma )-1}$$
for each component. Note that the additional one is introduced to prevent divergence when sites overlap. After drawing the new distance $r^{\mathrm{prop}}$, we shift all vertices componentwise with $n>n_{\mathrm{sel}}$ according to
(197)$$i_{n>n_{\mathrm{sel}}}^{\mathrm{prop}}=i_{n>n_{\mathrm{sel}}}+\left(r^{\mathrm{prop}}-r^{\mathrm{curr}}\right)\;.$$The underlying idea is that, if there is a large distance between two vertices $i_{n_{\mathrm{sel}}}$ and $i_{n_{\mathrm{sel}}+1}$, we can close the “rift” of the entire configuration instead of introducing a new one between $i_{n_{\mathrm{sel}}+1}$ and $i_{n_{\mathrm{sel}}+2}$. The transition weights for this move are given by
(198)$$\begin{array}{cc} \tilde{T}\left(c^{\mathrm{curr}}\to c^{\mathrm{prop}}\right) & =p^{\mathrm{sel}}\times p\left(r^{\mathrm{new}}\right)\;, \end{array}$$
(199)$$\begin{array}{cc} \tilde{T}\left(c^{\mathrm{prop}}\to c^{\mathrm{curr}}\right) & =p^{\mathrm{sel}}\times p\left(r^{\mathrm{curr}}\right)\;. \end{array}$$With these, we can calculate the Metropolis–Hastings acceptance probability in one dimension:
(200)$$\begin{array}{cc} p_{\mathrm{acc}}^{\mathrm{rift}} & =\min\left(1,\frac{\pi (c^{\mathrm{prop}})}{\pi (c^{\mathrm{curr}})}\frac{p(c^{\mathrm{curr}})}{p(c^{\mathrm{prop}})}\right)\\ \; & =\min\left(1,\frac{\sqrt{{\left(f_{n_{s}}^{\mathrm{ref}}\left(c^{\mathrm{prop}}\right)\right)}^{2}+R^{2}{\left(f_{n_{s}}\left(c^{\mathrm{prop}}\right)\right)}^{2}}}{\sqrt{{\left(f_{n_{s}}^{\mathrm{ref}}\left(c^{\mathrm{curr}}\right)\right)}^{2}+R^{2}{\left(f_{n_{s}}\left(c^{\mathrm{curr}}\right)\right)}^{2}}}\frac{{\left(r^{\mathrm{prop}}\right)}^{\gamma }}{{\left(r^{\mathrm{curr}}\right)}^{\gamma }}\right) \end{array}$$
and, likewise, in higher dimensions:
(201)$$\begin{array}{c} p_{\mathrm{acc}}^{\mathrm{rift}}=\min\left(1,\frac{\sqrt{{\left(f_{n_{s}}^{\mathrm{ref}}\left(c^{\mathrm{prop}}\right)\right)}^{2}+R^{2}{\left(f_{n_{s}}\left(c^{\mathrm{prop}}\right)\right)}^{2}}}{\sqrt{{\left(f_{n_{s}}^{\mathrm{ref}}\left(c^{\mathrm{curr}}\right)\right)}^{2}+R^{2}{\left(f_{n_{s}}\left(c^{\mathrm{curr}}\right)\right)}^{2}}}\frac{\prod_{n=1}^{d}\left(1+\right|r_{n}^{\mathrm{prop}}{\left|\right)}^{\gamma }}{\prod_{n=1}^{d}\left(1+\right|r_{n}^{\mathrm{curr}}{\left|\right)}^{\gamma }}\right)\;. \end{array}$$As above, we randomly draw y∈0,1, accept if $y<p_{\mathrm{acc}}^{\mathrm{rift}}$, and update the current configuration if the proposed configuration is accepted. In Figure 11b, you can find a typical rift move illustrated.
To implement the Monte Carlo algorithm, we just have to introduce a loop, where for each loop iteration we perform a Monte Carlo step consisting of those two moves.

To ensure that fluctuations to the current embedding introduce enough new configurations (shift moves) while making sure that the Monte Carlo algorithm does not spend too much time in physically unlikely configurations, the shift move probability was set to $p^{\mathrm{shift}}=0.7$ and the rift move probability to $p^{\mathrm{rift}}=0.3$ [29,207] (The “single-site rift move” was later completely replaced with the “multi-site” rift move presented in this review). This choice proved to produce good convergence behaviour over a large range of decay exponents and different models. After performing a move—accepted or not—we update the estimate for the target and reference sum in the numerator and denominator in Equation (Equation 188), respectively, and keep track of the statistics like the variance of target and reference sum. To determine the value $S[f_{n_{s}}]$ of the Monte Carlo run, the ratio of both quantities has to be multiplied by the analytical value of the reference sum as in Equation (Equation 188).

We can test the convergence of the MC algorithm by considering the 1qp gap Δ=ω(k=0) of the one-dimensional ferromagnetic long-range transverse-field Ising model (LRTFIM). We benchmark the second-order series coefficient $p_{2}$. According to Equation (Equation 174), the coefficient from the MC approach is $p_{2}^{\mathrm{MC}}=\bar{S}\left[f_{2,0}^{\left(2\right)}\right]+\bar{S}\left[f_{3,0}^{\left(2\right)}\right]$ and the exact value is given by $p_{2}^{\mathrm{exact}}=2\left(\zeta \left(2\left(1+\sigma \right)\right)-\zeta {\left(1+\sigma \right)}^{2}\right)$ (see Ref. [25]).

In Figure 12, we show the MC error:(202)$$\varepsilon_{\mathrm{MC}}=\left|\frac{p_{2}^{\mathrm{exact}}-p_{2}^{\mathrm{MC}}}{p_{2}^{\mathrm{exact}}}\right|$$
as a function of the number of steps $N_{\mathrm{steps}}$ for a hundred MC runs with a distinct seed for each. As can be seen from the figure, the MC error goes to zero with $N_{\mathrm{steps}}^{-1/2}$, as generally expected for MC simulations.

Further improvements to the algorithm can be made by recentring the configuration to the origin since the graphs on the lattice may drift away due to the random walk. Also, for integer powers occurring in the target integrand from powers of the expansion parameters, it may be better to use plain multiplication than using generic integer power functions. Most importantly, lookup tables should be used for the non-integer powers stemming from the algebraically decaying interaction strengths for distances $|i_{1}-i_{2}|<\delta_{\max}$ within a cutoff $\delta_{\max}$ that occur most commonly during the MC run. The integrand function also depends on k as cosine terms are present. It is useful to compile the code for a desired k value, so lookup tables can be defined at compile time due to the periodicity of the cosine. For instance, in the simplest case k=0, the cosine is always 1 and for k=π the cosine has only values of 1 and −1.

Let us emphasise that, for the pCUT+MC approach, we need to perform individual runs for every perturbative order, for every possible number of sites at a given order, for different values of momentum k if necessary (e.g., for a dispersion), and for different seeds so that an average value can be calculated. Typical runs are performed with $N_{\mathrm{seeds}}=5$–20 seeds for 6–24 h. For instance, for the long-range transverse-field Ising model, we were able to extract perturbative series up to order 10 for the 1qp spectral weight, order 11 for the 1qp gap, and order 13 for the ground-state energy. Much higher orders in perturbation are likely not feasible with the current implementation as the number of Monte Carlo runs scales quadratically with the order. In the future, to further improve the efficiency of the approach, one may come up with additional Monte Carlo moves for the Markov chain that can change the number of sites. As a result, the algorithm would only scale linearly with the perturbative order, but potentially at the cost of slower convergence. Maybe a compromise would be a favourable option, where the lower orders that converge faster are computed using an algorithm with the additional move and higher orders with the algorithm presented above.

Another important issue worth mentioning is the fact that, so far, the above algorithm can only be applied to lattices with a trivial single-site unit cell. To generalise the algorithm to arbitrary lattice geometries in the future, we would need to introduce another Monte Carlo move. We would keep the moves introduced above for moving the vertices along the underlying Bravais lattice. A new move then changes the position within the unit cell. However, new subtleties emerge from introducing a larger unit cell, as we would have to fix two vertices within the unit cells for hopping processes, while the remaining vertices can be moved freely. A larger unit cell also means that we have to calculate the entire matrix of the dispersion, as can be seen in Equation (Equation 115). Entries of the matrix are in general complex-valued, which needs to be accounted for as well. Lastly, the behaviour of the Monte Carlo algorithm for the system is altered due to the additional Monte Carlo move, which may impact the convergence. So, the choice of the reference sum should probably be adapted as well to achieve the desired convergence. See also the discussion in Ref. [207].

### 4.8. Series Extrapolation

We are interested in extracting the quantum-critical properties from the perturbative series obtained from the Monte Carlo embedding. DlogPadé extrapolations are an established and powerful method that allows us to extrapolate high-order series even beyond the radius of convergence and determine the quantum-critical point and associated critical exponents. A more elaborate description of DlogPadés and its application to critical phenomena can be found in Refs. [208,209].

We have given a high-order perturbative series of a physical quantity κ(λ) in the perturbation parameter λ. See Section 4.5 for typical quantities of interest. A Padé extrapolant is defined as
(203)$$P{\left[L,M\right]}_{\kappa }=\frac{P_{L}\left(\lambda \right)}{Q_{M}\left(\lambda \right)}=\frac{p_{0}+p_{1}\lambda +\cdots +p_{L}\lambda^{L}}{1+q_{1}\lambda +\cdots +q_{M}\lambda^{M}}\;,$$
where $p_{i},q_{i}\in R$ and the degrees *L*, *M* of the numerator polynomial $P_{L}\left(\lambda \right)$ and denominator polynomial $Q_{M}\left(\lambda \right)$ are restricted to $o_{\max}=L+M$, where $o_{\max}$ is the maximal perturbative order. The coefficients $p_{i}$ and $q_{i}$ are fixed by a set of linear equations by cross-multiplying Equation (Equation 203) with $Q_{M}\left(\lambda \right)$ and requiring $o_{\max}=L+M$, i.e., that all higher order terms must vanish on the left side of the equation [208]. We introduce
(204)$$P{\left[L,M\right]}_{D}=\frac{d}{d\lambda }\ln\left(\kappa \right)$$
as the Padé extrapolant of the logarithmic derivative of κ that must satisfy $o_{\max}-1=L+M$, as we lose one perturbative order due to differentiation. The DlogPadé extrapolant of κ can now be defined as
(205)$$dP{\left[L,M\right]}_{\kappa }=\exp\left(\int_{0}^{\lambda }P{\left[L,M\right]}_{D}\;d\lambda^{\prime }\right)\;.$$ Given that the quantity of interest κ shows a second-order phase transition with a dominant power law $\kappa ∼|\lambda -\lambda_{c}{|}^{-\theta }$ about the critical point $\lambda_{c}$, we can extract the critical point $\lambda_{c}$ and the associated critical exponent θ with DlogPadé extrapolants (Although we can extract the critical properties of a second-order quantum phase transition, we are blind to first-order phase transitions as the series expansion of a single physical quantity cannot capture level crossings of the analysed quantity at the critical point $\lambda_{c}$). We can determine estimates for the critical point $\lambda_{c}$ by analysing the poles of the extrapolant $P{\left[L,M\right]}_{D}$. We have to identify the physical pole whose position determines $\lambda_{c}$ and exclude spurious extrapolants that have non-physical poles in the complex plane close to the real line with $\lambda <\lambda_{c}$. If $\lambda_{c}$ is known, we can define biased DlogPadés by
(206)$$P{\left[L,M\right]}_{\theta^{★}}=\left(\lambda_{c}-\lambda \right)\frac{d}{d\lambda }\ln\left(\kappa \right)\;.$$ Here, defective extrapolants have to be removed as well by excluding all extrapolants that have poles in the vicinity of $\lambda <\lambda_{c}$. We can extract estimates for the critical exponent θ by calculating the residua. For an unbiased estimate, we calculate
(207)$$\theta =\mathrm{Res}\;P{\left[L,M\right]}_{D}{|}_{\lambda =\lambda_{c}}={\left.\frac{P_{L}\left(\lambda \right)}{\frac{d}{d\lambda }Q_{M}\left(\lambda \right)}\right|}_{\lambda =\lambda_{c}},$$
and for the biased estimate, we calculate
(208)$$\theta^{★}=\mathrm{Res}\;P{\left[L,M\right]}_{\theta^{★}}{|}_{\lambda =\lambda_{c}}={\left.\frac{P_{L}\left(\lambda \right)}{Q_{M}\left(\lambda \right)}\right|}_{\lambda =\lambda_{c}}\;.$$ Biased DlogPadé extrapolants were also used in the past trying to extract the exponent of multiplicative logarithmic corrections at the upper critical dimension [25,29,31,35]. At the upper critical dimensions, there are corrections to the dominant power-law behaviour of the form
(209)$$\kappa ∼{\left|\lambda -\lambda_{c}\right|}^{-\theta }{\left(\ln\left(\lambda -\lambda_{c}\right)\right)}^{p_{\theta }}\;.$$ We can bias the critical point $\lambda_{c}$ and the critical exponent θ to the known mean-field value and define the extrapolant:(210)$$P{\left[L,M\right]}_{p_{\theta }^{★}}=-\ln\left(1-\lambda /\lambda_{c}\right)\left[\left(\lambda_{c}-\lambda \right)D\left(\lambda \right)-\theta \right]\;,$$
so we can determine the estimate $p_{\theta }^{★}$ by calculating the residuum of the Padé extrapolant.

We take the series of mean coefficients $p_{i}$ for the quantities given by Equations (Equation 164), (Equation 174), and (Equation 177) and calculate a set of extrapolants to obtain reliable estimates for the critical properties of interest. We calculate estimates that satisfy $L+M=o\leq o_{\max}$. We exclude all defective extrapolants as briefly described above and arrange the remaining DlogPadés into families with L−M=const. Usually, we only allow |L−M|≤3 or |L−M|≤2 since more diagonal families are expected to converge faster to the real physical value as a function of the perturbative order o=L+M [208]. We further take the mean over the highest order extrapolants of each family that has at least two or three extrapolants to obtain an estimate for the critical point and exponents. The uncertainty obtained from averaging over the extrapolants is by no means a rigorous error, but is, rather, a “subjective” measure for the uncertainty obtained from systematically analysing extrapolants [209]. From experience, the estimated critical values are relatively stable to a small uncertainty in the series, and the uncertainty obtained from averaging over the highest order extrapolants is comparatively large. Therefore, the series of mean coefficients can simply be used for extrapolating the observables as performed in all previous publications [29,30,31,34,35,206]. A more rigorous approach can also be applied by repeatedly applying the extrapolation scheme to the series of individual seeds and averaging the critical values afterwards.

### 4.9. Workflow of Series Expansion Monte Carlo Embedding

In the previous subsections, we comprehensively described the pCUT+MC method for models with long-range interactions. The approach in its entirety is quite involved. Hence, to conclude this section, we want to give a short overview of the individual steps necessary and the workflow associated.

The workflow is sketched in Figure 13.

The approach starts with the generation of graphs with a software programme like “*nauty*” [201] and saving the graphs to bond files. The symmetry number $s_{G}$ of the graph can be obtained on the fly as it is usually a by-product of searching for further graph isomorphs. For the 1qp dispersion or spectral weight, we want to calculate quasiparticle processes on these graphs; however, we do not want to calculate every process on a graph since many are related by symmetries and give the exact same contribution. Therefore, we calculate additional symmetry numbers $s_{G_{c}}$ of coloured graphs. In a programme, we iterate over all possible vertex pairs and colours of the associated vertices (two colours for the 1qp spectral weight and one colour for 1qp processes of the Hamiltonian), choose a representative process, and save the associated vertices and the symmetry number of the coloured graph to a list. While, for the calculation of the ground-state energy, a list containing the graph names suffices, for particle processes, we want to have a list that contains the graph name, symmetry number, start and end vertex of the process, and a symmetry number associated with the process counting the number of equivalent processes. On the upside, it suffices to generate these lists and graphs once.

The second step is to iterate over the list entries, read the graph as a cluster, read the associated input states, and apply the pCUT method. Of course, such a programme must also read the pCUT coefficients C(m) or $\tilde{C}\left(m;i\right)$ for observables and the model information, i.e., the τ-operators. The programme iterates over the different operator sequences, through the operators from each sequence, systematically iterates over all edges of a graph, and applies the τ-operators. We additionally associate each edge of a graph with its own expansion parameter, such that we obtain the white graph contributions.

Having calculated the white graph contributions over every representative process for every graph, we need to convert the white graph contributions to an integrand function—also embeddable function—which is a callable function in our Monte Carlo algorithm. We have a programme that substitutes the expansion parameter with the algebraic decay expression ($\lambda_{e}\mapsto {\left|\delta \right|}^{-d+\sigma }$). This must be an expression in the programming language of choice, such that the Monte Carlo programme can call this function. Further, we multiply the white graph contribution with the symmetry factor $s_{G_{c}}/s_{G}$ for 1qp processes and with $1/s_{G}$ for the ground-state energy to properly account for the symmetries during the embedding. Also, all white graph contributions of a given order and number of vertices are grouped into one such function. The functions are saved in a file and included in the Monte Carlo algorithm code. If the white graph contributions are associated with models with different interaction flavours, we must sample for certain parameter values during this step.

The physical parameters are also fixed like the momentum k and the decay exponent d+σ, and of course, the code must be adapted for different lattice geometries. There are also simulation parameters like the number of seeds, the ratio between shift and rift moves, and the simulation exponents ρ of the reference sum and γ for the rift move ζ-function distribution. The Monte Carlo algorithm is then executed for each embeddable function with a fixed order, number of sites, and seed, yielding the target values of Equation (Equation 188).

We average the target values for multiple runs with different seeds, add them up according to Equations (Equation 164), (Equation 174), and (Equation 177), and obtain a perturbative series in λ. Afterwards, we employ DlogPadé extrapolation to extract critical quantities of interest like the critical point $\lambda_{c}$ and the associated critical exponent α for the ground-state energy, zν for the 1qp gap, or (2−z−η)ν for the 1qp spectral weight. We can also use the extrapolations to construct the dispersion close to the critical point.

## 5. Stochastic Series Expansion Quantum Monte Carlo

In this section, we discuss the method of stochastic series expansion (SSE) QMC. This class of QMC algorithms is closely related to path integral (PI) QMC and samples configurations according to the Boltzmann distribution of a quantum mechanical Hamiltonian. This sampling is achieved by extending the configuration space in the imaginary-time direction by operator sequences. The objective is to evaluate thermal expectation values for operators at a finite temperature on a finite system.

The canonical partition function of a system with a quantum mechanical Hamiltonian H can be expressed as
(211)$$\begin{array}{c} Z=\mathrm{Tr}\left\{\exp\left(-\beta H\right)\right\}=\sum_{|\alpha \rangle }\langle \alpha |\exp\left(-\beta H\right)|\alpha \rangle \end{array}$$
with β being the inverse temperature and the sum over an arbitrary orthonormal basis {|α〉}. The task is to bring Equation (Equation 211) into the form of
(212)$$\begin{array}{c} Z=\sum_{\omega \in C}\pi \left(\omega \right) \end{array}$$
with all weights π(ω) required to be non-negative.

Of course, for a Hamiltonian that is traced over in its eigenbasis (or a classical system), Equation (Equation 211) is already in the form of Equation (Equation 212), and the system can be directly sampled by a Metropolis–Hastings algorithm (see Equation (Equation 46)). For a general quantum mechanical problem, we do not have access to the eigenstates of a system and require a reformulation of Equation (Equation 211).

The SSE QMC idea resolves this issue in the following way: Given a Hamiltonian H, a computational orthonormal basis {|α〉} is chosen in which the trace is evaluated. Further, there should exist a decomposition of the Hamiltonian:(213)$$\begin{array}{c} H=-\sum_{i}H_{i} \end{array}$$
into operators $H_{i}$. $H_{i}$ and {|α〉} are chosen such that the following two conditions are met:No-branching rule:
(214)$$H_{i}|\beta \rangle \propto |\gamma \rangle \in \left\{|\alpha \rangle \right\}\;\forall \;H_{i}\;\forall |\beta \rangle \in \left\{|\alpha \rangle \right\},$$
ensuring that no superpositions of basis states are created by acting with $H_{i}$.Non-negative real matrix elements in the computational basis:
(215)$$\langle \beta |H_{i}|\gamma \rangle \geq 0\;\forall \;H_{i}\;\forall |\beta \rangle ,|\gamma \rangle \in \left\{|\alpha \rangle \right\}\;.$$
The second condition is not strictly necessary, but makes sure that no sign problem arises, which would lead to exponentially hard computational complexity. In general, it is not necessarily possible to find a computational basis in which this condition can be fulfilled for all Hamiltonians. However, if the negative matrix elements contribute in such a way that they always occur in pairs and the minus signs cancel, the condition can be relaxed without inducing a sign problem. We will encounter this case for the antiferromagnetic Heisenberg models in Section 5.2.

For operators $H_{i}$ that are diagonal in the computational basis, the conditions (Equation 214) and (Equation 215) never pose a problem as diagonal operators intrinsically obey the no-branching rule and can always be made non-negative by adding a suitable constant to the Hamiltonian. The main difficulty is to find a computational basis in which the off-diagonal matrix elements are non-negative, which is not necessarily possible, as mentioned above. In particular, for fermionic or frustrated systems, negative signs typically occur.

In order to reformulate Equation (Equation 211) in the form of Equation (Equation 212), a high-temperature expansion for the partition function is performed:(216)$$\begin{array}{cc} Z=\mathrm{Tr}\{\exp(-\beta H)\} & =\sum_{|\alpha \rangle }\langle \alpha |\exp\left(-\beta H\right)|\alpha \rangle \end{array}$$(217)$$\begin{array}{cc} \; & =\sum_{|\alpha \rangle }\sum_{n=0}^{\infty }\frac{\beta^{n}}{n!}\langle \alpha |{\left(-H\right)}^{n}|\alpha \rangle \end{array}$$(218)$$\begin{array}{cc} \; & =\sum_{|\alpha \rangle }\sum_{n=0}^{\infty }\frac{\beta^{n}}{n!}\langle \alpha |{\left(\sum_{i}H_{i}\right)}^{n}|\alpha \rangle \;. \end{array}$$ In general, the evaluation of $\langle \alpha |{\left(\sum_{i}H_{i}\right)}^{n}|\alpha \rangle$ is not feasible. The way the SSE tackles this expression is by expanding the product of sums as
(219)$$\begin{array}{c} {\left(\sum_{i}H_{i}\right)}^{n}=\sum_{S_{n}}\prod_{k=1}^{n}H_{i(k)} \end{array}$$
as a sum over all occurring operator sequences $S_{n}$ resulting from the exponentiation. The additional dimension created by the operator sequence is usually referred to as imaginary time in analogy to path-integral formulations. Inserting Equation (Equation 219) into Equation (Equation 218), one obtains
(220)$$\begin{array}{cc} Z & =\sum_{|\alpha \rangle }\sum_{n=0}^{\infty }\sum_{S_{n}}\frac{\beta^{n}}{n!}\langle \alpha |\prod_{k=1}^{n}H_{i(k)}|\alpha \rangle \;. \end{array}$$ Note that each of the summands in Equation (Equation 220) is non-negative by design due to the condition (Equation 215) and can be interpreted as the relative weight of a configuration. By comparing Equation (Equation 220) with Equation (Equation 212), we see that it is of a suitable form for a Markov chain Monte Carlo sampling. We identify the direct product of the set of all basis states {|α〉} with the set of all sequences as the configuration space:(221)$$\begin{array}{c} C=\left\{|\alpha \rangle \right\}\times \overset{\infty }{\underset{n=0}{⋃}}\left\{S_{n}\right\}\;. \end{array}$$ The weight of a configuration is given by
(222)$$\begin{array}{c} \pi \left(|\alpha \rangle ,S_{n};\beta \right)=\frac{\beta^{n}}{n!}\langle \alpha |\prod_{k=1}^{n}H_{i(k)}|\alpha \rangle \;. \end{array}$$ In the next step, we discuss the structure of each configuration consisting of a computational basis state |α〉 and an operator sequence $S_{n}$. Regarding Equation (Equation 220), we stress that the action of the product of operators in the sequence onto the basis state is crucial. Due to the no-branching rule (see Equation (Equation 214)), the action of $\prod_{k=1}^{n}H_{i(k)}$ onto the basis state can be interpreted as a discrete propagation of the state α in imaginary time according to the operators in the sequence. We define the state at propagation index p∈{0,⋯,n} to be
(223)$$\begin{array}{c} |\alpha (p)\rangle =\prod_{k=1}^{p}H_{i(k)}|\alpha \rangle \;. \end{array}$$ Due to the periodic boundary condition of the trace and the orthogonality of the basis states {|α〉}, only sequences for which |α(n)〉=|α(0)〉 have a non-zero weight (see Equation (Equation 222)).

At this point, we can demonstrate why it does not matter if some matrix elements are negative in some instances, e.g., for some antiferromagnetic spin models on bipartite lattices. If a matrix element of an operator is negative, but it is ensured by the Hamiltonian and the periodicity of the trace that there is always an even number of negative matrix elements in a sequence with non-vanishing weight, then the definition of weights as in Equation (Equation 222) is nevertheless possible.

In the discussion so far, we considered sequences of all possible lengths. In order to formulate algorithms sampling the configuration space efficiently, a scheme with a fixed sequence length L can be introduced, in which all sequences with n<L are padded with identity operators 1 to length L and all sequences n>L are discarded. The physical justification to discard all sequences above a certain fixed length L is that they are exponentially suppressed for sufficiently large L. In short, this is the case because the mean operator number 〈n〉 is proportional to the mean energy 〈H〉 and the variance is related to the specific heat in the following way:(224)$$\begin{array}{c} \langle H\rangle =-\frac{\langle n\rangle }{\beta }\;C=\langle n^{2}\rangle -{\langle n\rangle }^{2}-\langle n\rangle \;. \end{array}$$ A derivation and discussion of these statements can be found in Refs. [39,210,211,212,213]. From Equation (Equation 224), we can infer that the infinite sum over all sequence lengths can be truncated at a finite L∝βN. From rearranging Equation (Equation 224) and using the extensivity of the mean energy, the mean sequence length scales as 〈n〉∝βN proportional to the inverse temperature and system size *N*. As the mean sequence length has a finite value, the idea is to choose an L>〈n〉 great enough to be able to sample all but a negligible amount of operator sequences. Further, we will argue that the introduction of a large enough cutoff results in an exponentially small and negligible error. In the limit of β→∞, the specific heat has to vanish. Therefore, the variance of the mean sequence length is proportional to 〈n〉. From this, it is concluded that the weights of sequences vanish exponentially for large enough sequence lengths *n* [212].

We, therefore, introduce a large enough cutoff for the sequence length L and consider operator sequences with a fixed length L. The expression for the partition function can then be written as
(225)$$\begin{array}{c} Z\approx \sum_{S_{L}}\sum_{|\alpha \rangle }\frac{\beta^{n}\left(L-n\right)!}{L!}\langle \alpha |\prod_{k=1}^{L}H_{i(k)}|\alpha \rangle \;. \end{array}$$ The new configuration space includes all sequences of length L where the shorter sequences are padded by inserting unity operators. The random insertion of unity operators into a sequence of n<L non-trivial operators results in Ln=L!n!(L−n)! sequences of length L. The modified prefactor in Equation (Equation 225) accounts for this overcounting. Although Equation (Equation 225) is, strictly speaking, an approximation to the partition function, we want to note that, in practice, this does not cause a systematic error as L can be chosen large enough in a dynamical fashion such that, during the finite simulation time, no sequence of length n>L would occur. Therefore, we will not consider the fixed-length scheme as an approximation below.

To summarise: Following the argumentation discussed in this section, it is possible to bring any partition function into the form of Equation (Equation 225), whereby all the summands are non-negative if a suitable decomposition and computational basis can be found. The next step is to implement a Markov chain MC sampling on the configuration space. As the configuration space largely depends on the model, the sampling is performed in a model-dependent way. In Section 5.1, we will introduce an algorithm to sample the (LR)TFIM. In Section 5.2, we will further describe an algorithm to sample unfrustrated (long-range) Heisenberg models. In Section 5.3, the measurement of a variety of operators is discussed. Further, we will discuss how to sample systems at effectively zero temperature in Section 5.4 as this review considers the zero-temperature physics of quantum phase transitions. In Section 5.5, we give a short overview of other MC algorithms for long-range models based on path integrals.

### 5.1. Algorithm for Arbitrary Transverse-Field Ising Models

In this section, we describe the SSE algorithm to sample arbitrary transverse-field Ising models (TFIMs) of the form
(226)H=∑i≠jJi,jσizσjz−∑ihixσix
as introduced by A. Sandvik in Ref. [39]. The Pauli matrices $\sigma_{i}^{\kappa }$ with κ∈{x,z} describe *N* spins 1/2 located on the lattice sites i,j. The transverse-field strength at a lattice site *i* is $h_{i}>0$, and the Ising couplings between sites *i* and *j* have the strength $J_{i,j}\in R$. For $J_{i,j}>0$, the interaction is antiferromagnetic and it is energetically favourable for the spins to anti-align, while for $J_{i,j}<0$, the interaction is ferromagnetic and it is energetically favourable for the spins to align.

Choosing the $\sigma^{z}$-eigenbasis $\left\{|\alpha \rangle \}=\{|\sigma_{1}^{z},\cdots ,\sigma_{N}^{z}\rangle \right\}$ for the SSE formulation avoids the sign problem for arbitrary Ising interactions. The Hamiltonian is decomposed using the following operators: (227)$$\begin{array}{cc} H_{0,0} & =1 \end{array}$$(228)$$\begin{array}{cc} H_{i,0} & =h_{i}\sigma_{i}^{x}\;i>0 \end{array}$$(229)$$\begin{array}{cc} H_{i,i} & =h_{i}1 \end{array}$$(230)$$\begin{array}{cc} H_{i,j} & =|J_{i,j}|-J_{i,j}\sigma_{i}^{z}\sigma_{j}^{z}\;i,j>0,\;i\neq j\;. \end{array}$$ We call $H_{0,0}$ a trivial operator, $H_{i,0}$ a field operator, $H_{i,i}$ a constant operator, and $H_{i,j}$ an Ising operator. The operator $H_{i,i}$ is associated with the site *i*, even though it is proportional to 1. With these operators, we can rewrite Equation (Equation 226) up to an irrelevant constant as
(231)$$\begin{array}{c} H=-\sum_{i=1}^{N}\sum_{j=0}^{N}H_{i,j}\;. \end{array}$$ Note that Equation (Equation 231) is a sum over field, constant, and Ising operators. The constant operators are not part of the original Hamiltonian, but will be relevant for algorithmic purposes. The trivial operators (Equation (Equation 227)) are not relevant to express the Hamiltonian, but are necessary for the fixed-length sampling scheme. The proposed decomposition fulfils the no-branching property, and there are no negative matrix elements of the operators in the computational basis due to the positive constant $|J_{i,j}|$ that is added to the Ising operators. The possible matrix elements for Ising operators $H_{i,j}=\left|J_{i,j}\right|-J_{i,j}\sigma_{i}^{z}\sigma_{j}^{z}$ acting on a pair of spins i,j are given by
(232)$$\begin{array}{cc} \langle {↑↑}_{i,j}|H_{i,j}|{↑↑}_{i,j}\rangle & =\langle {↓↓}_{i,j}|H_{i,j}|{↓↓}_{i,j}\rangle =\left\{\begin{array}{cc} 2|J_{i,j}| & \mathrm{for}\;J_{i,j}<0\\ 0 & \mathrm{for}\;J_{i,j}>0 \end{array}\right. \end{array}$$
(233)$$\begin{array}{cc} \langle {↑↓}_{i,j}|H_{i,j}|{↑↓}_{i,j}\rangle & =\langle {↓↑}_{i,j}|H_{i,j}|{↓↑}_{i,j}\rangle =\left\{\begin{array}{cc} 0 & \mathrm{for}\;J_{i,j}<0\\ 2|J_{i,j}| & \mathrm{for}\;J_{i,j}>0\;. \end{array}\right. \end{array}$$ This implies that only sequences where (anti)ferromagnetic Ising bonds are placed on (anti)aligned spins have a non-vanishing weight.

The partition function (Equation (Equation 225)) in the fixed-length scheme reads
(234)$$\begin{array}{c} Z=\sum_{S_{L}}\sum_{|\alpha \rangle }\frac{\beta^{n}\left(L-n\right)!}{L!}\prod_{k=1}^{L}\langle \alpha (k)|H_{i(k),j(k)}|\alpha (k-1)\rangle \;. \end{array}$$ The propagated states |α(p)〉 only change in the imaginary-time propagation if a field operator, the only off-diagonal operator in the chosen basis, acts on it. A configuration only has a non-zero weight if all the Ising operators in the sequence are placed on spins that have the correct alignment. Constant operators are included in the Hamiltonian for algorithmic purposes, which will become important in the off-diagonal update described below.

Before going into the description of the Markov chain to sample the configuration space, it is illustrative to visualise a configuration with non-vanishing weight defined by a state |α〉 and an operator sequence $S_{L}$. In Figure 14, an exemplary configuration with non-vanishing weight is illustrated for the one-dimensional TFIM and visualisations of the different operators Equations (Equation 228)–(Equation 230) in such a configuration are shown.

Before going into the details of the algorithm, we introduce the main concept of the update scheme and the crucial obstacles that are encountered in setting up an efficient algorithm [39]. Each step of the Markov chain sampling of configurations is performed by performing a so-called diagonal update followed by an off-diagonal quantum cluster update [39]. In the diagonal update, one iterates over the sequence and exchanges trivial operators with constant or Ising operators and vice versa while propagating the states along the sequence. In the off-diagonal update, constant operators are exchanged with field operators while preserving the weight of the configuration. The main obstacle that is circumvented with this update procedure is the following: It is a non-trivial and non-local task to insert off-diagonal operators into the sequence without creating a configuration with vanishing weight. The first problem is that one cannot simply insert a single field operator into the sequence as it breaks the periodicity of the propagated states |α(L)〉≠|α(0)〉, leading to a vanishing weight due to the orthonormality of the computational basis. Therefore, field operators can only occur in an even number for each site to preserve the periodicity in imaginary time. The second issue involves Ising operators placed on a pair of sites *i* and *j*. If one places a field operator at one of the sites before and behind the Ising operator in the sequence, this preserves the periodicity in imaginary time, but the matrix element of the Ising operator becomes zero as the spins will be misaligned with respect to the sign of the Ising coupling (see Equation (Equation 232)). These issues can be tackled by a non-local off-diagonal update [39], which we will thoroughly discuss after the diagonal update.

In the diagonal update, the number of non-trivial operators *n* is altered by exchanging trivial operators with constant and Ising operators in the operator sequence and vice versa. This does not change the states |α(p)〉, including the state |α〉=|α(0)〉. Starting from propagation step p=0 and state |α(0)〉, one iterates over the sequence step-by-step and conducts the exchange of trivial operators with non-trivial diagonal operators. If a field operator is encountered at the current propagation step, the state is propagated by flipping the respective spin, and the iteration proceeds. If a diagonal operator is encountered, a local update following the Metropolis–Hastings algorithm as described in Section 3 is performed and the total transition probability is made of a proposition probability $\tilde{T}$ and an acceptance probability $p_{\mathrm{acc}}$.

If a trivial operator is encountered at propagation step *k*, a non-trivial diagonal operator gets proposed with the probability
(235)$$\begin{array}{c} \tilde{T}\left(H_{0,0}\to H_{i,j}\right)=\frac{M_{i,j}}{C} \end{array}$$
taking into account the weight $M_{i,j}$ of the proposed operator with the normalising constant *C*:(236)$$\begin{array}{cc} M_{i,j} & =\left\{\begin{array}{cc} 2|J_{ij}| & \;\mathrm{for}\;i\neq j\;,\\ h_{i} & \;\mathrm{for}\;i=j\;, \end{array}\right. \end{array}$$(237)$$\begin{array}{cc} C & =\sum_{i}h_{i}+2\sum_{i\neq j}\left|J_{ij}\right|\;. \end{array}$$ The weight is essentially given by the matrix elements of the respective operators $H_{i,j}$ with the special case that the Ising operators are a priori handled as if they were allowed. At a later stage, it is checked if the spins are correctly aligned at the considered propagation step *k* and state |α(k)〉, and the operator gets rejected if this is not the case. This has the benefit that one does not need to check every single bond for correct alignment for the insertion of a single operator, which scales like $O(N^{2})$ for long-range models. This sampling can be performed in a constant time complexity in the number of elements in the distribution by using the so-called walker method of aliases [214]. Instructions on how to set up a walker sampler from a distribution of discrete weights and how to draw from this distribution can be found in Appendix C or in Ref. [214]. Drawing from the discrete distribution of weights, an operator $H_{i,j}$ gets proposed to be inserted. On the other side, if a non-trivial diagonal operator is encountered, it is always proposed to be replaced by a trivial operator:(238)$$\begin{array}{c} \tilde{T}\left(H_{i,j}\to H_{0,0}\right)=1\;. \end{array}$$ The acceptance probabilities are then chosen according to the Metropolis–Hastings algorithm in Equation (Equation 46). For a constant field operator $H_{i,i}$, this gives the acceptance probability:
(239)$$\begin{array}{cc} p_{\mathrm{acc}}\left(H_{0,0}\to H_{i,i}\right) & =\min\left(1,\frac{\beta M_{i,i}}{L-n}\frac{C}{M_{i,i}}\right) \end{array}$$
(240)$$\begin{array}{cc} \; & =\min\left(1,\frac{\beta (\sum_{i}h_{i}+2\sum_{i\neq j}|J_{ij}|)}{L-n}\right)\;. \end{array}$$ Similarly, for an Ising operator $H_{i,j}$, one has
(241)$$\begin{array}{cc} p_{\mathrm{acc}}\left(H_{0,0}\to H_{i,j}\right) & =\min\left(1,\frac{\beta \langle \alpha (k)|H_{i,j}|\alpha (k-1)\rangle }{L-n}\frac{C}{M_{i,j}}\right) \end{array}$$
(242)$$\begin{array}{cc} \; & =\frac{\langle \alpha (k)|H_{i,j}|\alpha (k-1)\rangle }{M_{i,j}}\min\left(1,\frac{\beta (\sum_{i}h_{i}+2\sum_{i\neq j}|J_{ij}|)}{L-n}\right) \end{array}$$
up to a factor $\langle \alpha (k)|H_{i,j}|\alpha (k-1)\rangle /M_{i,j}$, which is either 1 or 0 depending on if the spins at the current propagation step *k* are correctly aligned or misaligned. An Ising bond with misaligned spins would lead to a vanishing weight of the configuration and is, therefore, not allowed.

Up to this factor, the acceptance probability is the same independent of the type of non-trivial diagonal operator that is proposed to be inserted as the operator weight $M_{i,j}$ arising in the weight for the newly proposed configuration cancels with the same factor in the proposition probability. This is because we already chose to propose an operator by considering its respective weight $M_{i,j}$. As the acceptance probabilities do not differ, one can, therefore, also first check if one accepts to insert any non-trivial diagonal operator and, only if this is accepted, draw the precise operator to be inserted. Of course, if the chosen operator is an Ising operator, it still has to be checked if the spins are correctly aligned to prevent a non-vanishing weight.

The cancellation of the operator weights $M_{i,j}$ also makes it easier to perform the reverse process and replace a non-trivial diagonal operator with a trivial one in the sense that the acceptance probability for inserting $H_{0,0}$:(243)$$\begin{array}{cc} p_{\mathrm{acc}}\left(H_{i,j}\to H_{0,0}\right) & =\min\left(1,\frac{L-(n-1)}{\beta (\sum_{i}h_{i}+2\sum_{i\neq j}|J_{ij}|)}\right)\; \end{array}$$
does not depend on the current non-trivial diagonal operator $H_{i,j}$.

If a proposition gets rejected, the iteration along the sequence continues and the procedure starts again for the next operator. After each diagonal update sweep (iteration over the whole sequence) during the equilibration, trivial operators are appended to the end of the sequence such that L>4n/3 [39]. This allows dynamically adjusting the sequence length L for the fixed-length scheme to ensure a sufficiently long sequence.

For an efficient implementation of the off-diagonal quantum cluster update, it is crucial to introduce the concept of operator legs. An operator at position *p* in the sequence has legs with the numbers 4p,⋯,4p+3. For an Ising operator, these legs are associated with two legs per site, one upper and one lower leg (see Figure 15a).

For a constant or field operator, only two of the four legs are real vertex legs since these operators act only on a single site (see Figure 15b). The remaining two legs are called ghost legs and are considered solely due to algorithmic reasons as it is numerically beneficial to let every operator have the same number of legs. This allows calculating the propagation index *p* of an operator from the leg numbers using integer division with p=⌊leg/4⌋. Trivial operators can be described with four ghost legs or can be ignored entirely in the sequence for the off-diagonal update.

For the chosen representation of the Ising model within the SSE, one can subdivide the configuration into disjoint clusters that extend in space, as well as in imaginary time with Ising operators acting as bridges in real space and with constant and field operators acting as delimiters in imaginary time. These clusters of spins can be flipped by replacing the delimiting constant operators with field operators and vice versa without changing the weight $\pi (|\alpha \rangle ,S_{L};\beta )$. If the cluster winds around the boundary in imaginary time, the respective spins in state |α〉 have to be updated as well. In order to flip half of the configuration on average and obtain a good mixing, all clusters are constructed and the probability to flip a specific cluster is chosen to be 1/2 for each cluster separately.

For the construction of the clusters, the propagated spin states along imaginary time are not needed. The entire problem can be dealt with in the language of vertices with legs. The relevant information for the cluster formation is which legs are connected. This information about the connection between legs of different operators is stored in a doubly linked list. The list is set up in the following way: At the index of a vertex leg *i*, we store the index *j* of the leg it is connected to, i.e., list[i]=j and vice versa list[list[i]]=list[j]=i. This segmentation is illustrated for an exemplary configuration in Figure 16 together with the doubly linked leg list for the depicted configuration. The doubly linked list can already be set up during the diagonal update when the sequence is traversed either way. An efficient algorithm to set up the data structure is described in Refs. [213,215].

In comparison to general off-diagonal loop updates [215,216,217], the formation of clusters in the presented off-diagonal cluster update for the TFIM is fully deterministic. The whole configuration is divided into disjoint clusters, all of which will be built and flipped with probability 1/2. It is beneficial to already decide whether a cluster is flipped or not before constructing the cluster so the constant and field operators and spin states can already be processed during the construction. A leg that is processed during the construction of a cluster is marked as visited in order to not process the same leg twice. This is also the reason why all the clusters have to be constructed even if they will not be flipped, as otherwise, the same cluster would get constructed starting from another leg later on. Further, we introduce a stack for the legs that were visited during the construction, but whose connections are yet to be processed.

The formation of each cluster starts by choosing a leg that has not yet been visited in the current off-diagonal update. At the beginning of the cluster formation, it is randomly determined if the cluster is flipped or not. If the leg corresponds to an Ising vertex, the cluster branches out to all four legs of the vertex. This means that all four legs of the operator are put on the stack. If the leg corresponds to a constant or field operator, the type of operator is exchanged if the cluster is flipped and only this primal leg is put on the stack. Next, the following logic is repeated until the stack is empty. We pop a leg *l* from the stack. If the leg has been visited already, we continue with a new leg from the stack. Else, we determine the new leg $l^{\prime }=\mathrm{list}\left[l\right]$ using the doubly linked list. We mark both legs *l* and $l^{\prime }$ as visited. If one passes by the periodic boundary in imaginary time while going from leg *l* to $l^{\prime }$ and the cluster will be flipped, the corresponding site belonging to the legs in the state |α〉 has to be flipped. If $l^{\prime }$ belongs to an Ising operator, we add all legs of the Ising operator that have not been visited yet to the stack. If $l^{\prime }$ belongs to a constant or field operator, we exchange the operator if the cluster is said to be flipped. This procedure is repeated until the stack is empty. After that, one proceeds with the next cluster starting from a leg that has not yet been visited in the current cluster update. If no such leg is left, the cluster update is finished.

In addition to this cluster update, spins that have no operators acting on them in the sequence can be flipped with a probability of 1/2 (thermal spin flip).

In summary, the off-diagonal update exchanges field $H_{i,0}$ and constant operators $H_{i,i}$ with each other and changes the state |α〉. Combining the diagonal update with the off-diagonal cluster update samples the entire configuration space.

### 5.2. Algorithm for Unfrustrated Heisenberg Models

In this section, we describe the algorithm to sample arbitrary unfrustrated spin-1/2 Heisenberg models with the SSE framework [213]. By unfrustrated Heisenberg models, we mean Hamiltonians:(244)$$\begin{array}{cc} H & =\sum_{b\in \mathrm{AF}}J_{b}{\vec{S}}_{i(b)}{\vec{S}}_{j(b)}+\sum_{b\in F}J_{b}{\vec{S}}_{i(b)}{\vec{S}}_{j(b)} \end{array}$$
written as a sum over three-component interactions between two sites i(b) and j(b) connected by bond *b*, where each bond is either ferromagnetic (F) with $J_{b}<0$ or antiferromagnetic (AF) with $J_{b}>0$, with the property that there is no loop of lattice sites connected by bonds that contains an odd number of antiferromagnetic bonds, as this would lead to frustration. The spin operators are a compact notation for a three-component vector of spin operators ${\vec{S}}_{i}={\left(S_{i}^{x},S_{i}^{y},S_{i}^{z}\right)}^{T}$. The coupling strengths $J_{b}$ can have a priori arbitrary amplitudes. It is crucial to look at unfrustrated Heisenberg models in order to define non-negative weights for configurations as the off-diagonal components of the antiferromagnetic operators have a negative matrix element. In an unfrustrated model, these matrix elements always occur an even number of times in the operator sequence of any valid configuration, which makes it possible to construct a non-negative SSE weight [213].

For the SSE algorithm, the $\sigma^{z}$-eigenbasis $\left\{|\alpha \rangle \}=\{|\sigma_{1}^{z},\cdots ,\sigma_{N}^{z}\rangle \right\}$ is chosen, but the $\sigma^{x}$- or $\sigma^{y}$-eigenbasis would work the same way. The Hamiltonian is decomposed into
(245)$$\begin{array}{cc} H & =-\sum_{b\in \mathrm{AF}}\left(H_{1,b}^{\mathrm{AF}}+H_{2,b}^{\mathrm{AF}}-\frac{|J_{b}|}{4}\right)-\sum_{b\in F}\left(H_{1,b}^{F}+H_{2,b}^{F}-\frac{|J_{b}|}{4}\right) \end{array}$$
using the following operators: (246)$$\begin{array}{cc} H_{1,b}^{F} & =\frac{|J_{b}|}{4}-J_{b}S_{i(b)}^{z}S_{j(b)}^{z} \end{array}$$(247)$$\begin{array}{cc} H_{2,b}^{F} & =-\frac{J_{b}}{2}\left(S_{i(b)}^{+}S_{j(b)}^{-}+S_{i(b)}^{-}S_{j(b)}^{+}\right) \end{array}$$(248)$$\begin{array}{cc} H_{1,b}^{\mathrm{AF}} & =\frac{|J_{b}|}{4}-J_{b}S_{i(b)}^{z}S_{j(b)}^{z} \end{array}$$(249)$$\begin{array}{cc} H_{2,b}^{\mathrm{AF}} & =-\frac{J_{b}}{2}\left(S_{i(b)}^{+}S_{j(b)}^{-}+S_{i(b)}^{-}S_{j(b)}^{+}\right)\;. \end{array}$$

The diagonal operators $H_{1,b}^{F}$ and $H_{1,b}^{\mathrm{AF}}$ in Equations (Equation 246) and (Equation 248) are defined in the same fashion as for the TFIM up to the factor of 1/4 due to the usage of spin operators $S^{\kappa }=\sigma^{\kappa }/2$ instead of Pauli matrices $\sigma^{\kappa }$ with κ∈{x,y,z}. The contribution to the weight of a sequence of these operators is either $|J_{b}|/2$ if the bond fulfils the (anti)ferromagnetic condition and zero otherwise. Although the expressions for the ferromagnetic and antiferromagnetic bonds look the same, we distinguish between these two bonds to highlight that these objects behave differently within the off-diagonal update.

As $J_{b}>0$ for antiferromagnetic bonds, the off-diagonal operators $H_{2,b}^{\mathrm{AF}}$ do not fulfil the non-negativity of matrix elements (see Equation (Equation 215)) in the computational basis. Therefore, they must always appear in an even number of times in the operator sequence to avoid the sign problem. For the ferromagnetic bonds, this restriction is not necessary.

Analogous to the TFIM, the operator sequence additionally contains trivial operators $H_{0,0}=1$, which are not part of the Hamiltonian, but are used to pad the sequence to a fixed length L. In contrast to the TFIM, there is no need for further artificial operators like the constant field operators in the TFIM used to limit the cluster in the cluster update. In the case of the Heisenberg model, the non-local off-diagonal update is constructed in the form of loops instead of a cluster with several branches that need to be limited in imaginary time. An exemplary SSE configuration for a Heisenberg chain using the decomposition from above can be seen in Figure 17.

Similar to the sampling of the TFIM in Section 5.1, each step of the Markov chain sampling of configurations is achieved by performing a diagonal update followed by a non-local off-diagonal update. The diagonal update exchanges trivial operators with diagonal operators, while the off-diagonal update exchanges diagonal bond operators with the respective off-diagonal bond operators.

Due to the structure of the diagonal operators (see Equations (Equation 246) and (Equation 248)), similar to the TFIM, the diagonal update is performed similarly to the diagonal update of the TFIM described in Section 5.1. Nonetheless, to make the description of the algorithm for unfrustrated Heisenberg models self-contained, we recapitulate the diagonal update and adapt it to the Heisenberg case.

The diagonal update exchanges trivial operators with diagonal operators in the sequence and vice versa. It will, therefore, not change the states |α(p)〉, including the state |α〉=|α(0)〉. Starting from |α〉 at k=0, the sequence is again iterated over by *k*. If the operator at the current position *k* in the sequence is an off-diagonal operator operator, the state |α(k−1)〉 is propagated by applying the off-diagonal operator and the iteration proceeds. If one encounters diagonal or trivial operators in the sequence, the Metropolis–Hastings algorithm (see Section 3) is performed and the transition probability to an altered configuration is again split into a proposition probability $\tilde{T}$ and an acceptance probability $p_{\mathrm{acc}}$.

Analogous to the algorithm for the TFIM, if a trivial operator is encountered, a non-trivial diagonal operator gets proposed with the probability:(250)$$\begin{array}{c} \tilde{T}\left(H_{0,0}\to H_{1,b}^{\mathrm{AF}/F}\right)=\frac{M_{b}}{C}, \end{array}$$
taking into account the weight $M_{b}$ of the proposed operator and the normalising constant *C*:(251)$$\begin{array}{c} M_{b}=\frac{|J_{b}|}{2}\;C=\frac{1}{2}\sum_{b}\left|J_{b}\right|\;. \end{array}$$ The diagonal update for the Heisenberg model only differs from the one of the TFIM by the detailed probabilities $M_{b}$ and the normalising constant *C*. The factor of 1/4 in the bond weights in comparison to the Ising model comes from using spin operators in contrast to Pauli matrices in the Hamiltonian. This sampling according to the weight $M_{b}$ can be performed in a constant-time complexity in the number of elements in the distribution by using the walker method of aliases [214]. An introduction on how to set up a walker sampler from a distribution of discrete weights and how to draw from this distribution can be found in Appendix C or in Ref. [214]. Drawing from the discrete distribution of weights, an operator $H_{1,b}^{\mathrm{AF}/F}$ gets proposed to be inserted. On the other side, if a non-trivial diagonal operator is encountered, it is always proposed to be replaced by a trivial operator:(252)$$\begin{array}{c} \tilde{T}\left(H_{1,b}^{\mathrm{AF}/F}\to H_{0,0}\right)=1, \end{array}$$
exactly as for the TFIM. The acceptance probabilities are then chosen according to the Metropolis–Hastings algorithm Equation (Equation 46). The respective Metropolis–Hastings probabilities with which the proposed replacements at position *k* in the operator sequence are accepted are given by the ratios of the configuration weights before and after the potential replacement and the ratio of the proposition probabilities:(253)$$\begin{array}{cc} p_{\mathrm{acc}}\left(H_{0,0}\to H_{1,b}^{\mathrm{AF}/F}\right) & =\min\left(1,\frac{\beta \langle \alpha (k)|H_{1,b}^{\mathrm{AF}/F}|\alpha (k-1)\rangle }{L-n}\frac{C}{M_{b}}\right) \end{array}$$(254)$$\begin{array}{cc} \; & =\frac{\langle \alpha (k)|H_{1,b}^{\mathrm{AF}/F}|\alpha (k-1)\rangle }{M_{b}}\min\left(1,\frac{\beta (\sum_{b}|J_{b}|)}{2(L-n)}\right)\;, \end{array}$$(255)$$\begin{array}{cc} p_{\mathrm{acc}}\left(H_{1,b}^{\mathrm{AF}/F}\to H_{0,0}\right) & =\min\left(1,\frac{2(L-(n-1))}{\beta (\sum_{b}|J_{b}|)}\right)\;. \end{array}$$

As in the case of the TFIM, the specific matrix element $\langle \alpha (k)|H_{1,b}^{\mathrm{AF}/F}|\alpha (k-1)\rangle$ for the propagated states |α(k−1)〉 and |α(k)〉=|α(k−1)〉 has to be considered, which is either $M_{b}$, and cancels with this factor in the acceptance probability, or is 0, and the acceptance probability vanishes. Intuitively, this factor simply checks if the spins are correctly aligned or misaligned with respect to the sign of the coupling $J_{b}$ and only allows the operator to be inserted if the spins i(b) and j(b) of the propagated state |α(k−1)〉 are correctly aligned. If the proposition is accepted, the replacement of the operator at index *k* is performed, and one continues with the next propagation index k+1 in imaginary time. If the proposition is denied, the current operator remains, and one continues with the next propagation index k+1 in imaginary time.

As for the TFIM, a fixed-length scheme is used and the sequence length L limiting the amount of non-trivial operators *n* has to be dynamically adjusted. During the thermalisation of the sampling procedure, trivial operators are appended to the end of the sequence after each diagonal update (iteration over the whole sequence) such that L>4n/3 [39].

In order to efficiently implement an off-diagonal loop update, it is crucial to introduce the concept of operator legs. An operator at position *p* in the sequence has four legs with the numbers 4p,⋯,4p+3, two legs (an upper and a lower leg) for each of the spin sites. Trivial operators can be described with four ghost legs or can be ignored entirely in the sequence for the off-diagonal update.

For the unfrustrated Heisenberg models, the off-diagonal update is performed using a loop update. During the update, loops are constructed in the space–time configuration (see Figure 17 for an example) and the spin states on the loop are flipped along the way, which changes the operator vertex types. Thereby, diagonal operators become off-diagonal operators and vice versa. It is noteworthy that the weight of a configuration is not modified by this since the weights of diagonal and off-diagonal operators are the same in the isotropic Heisenberg model. Therefore, flipping loops with a fixed probability satisfies detailed balance as long as the probability to construct the reverse loop is the same.

For the Heisenberg model, the creation of the loops is deterministic. Whenever a loop enters an operator during the loop construction, there is only one exit leg that creates a valid operator for the bond type (AF or F). Flipping the spins along the loop exchanges an diagonal operator $H_{1,b}^{\mathrm{AF}/F}$ with its off-diagonal counterpart $H_{2,b}^{\mathrm{AF}/F}$ and vice versa. The entrance leg and exit leg never belong to the same site. For ferromagnetic operators, the propagation direction of the loop in imaginary time remains the same (see Figure 18). For antiferromagnetic operators, the propagation direction of the loop in imaginary time changes (see Figure 18). When the loop crosses the periodic boundary in imaginary time at a site *i*, the computational basis state |α〉 has to be updated by flipping the spin at site *i*. As each leg only belongs to one loop, the configuration can be split into disjoint loops. Therefore, it is possible to construct all loops during the off-diagonal update and flip each loop independently with a probability of 1/2.

Combining the diagonal and off-diagonal update allows sampling arbitrary unfrustrated Heisenberg models. This includes antiferromagnetic Heisenberg ladders and bilayer systems with an unfrustrated long-range interaction.

### 5.3. Observables

In this section, we introduce some observables that can be easily measured within the SSE formalism introduced. Throughout this section, we do not use the fixed-length scheme, but keep the general form with operator sequences of fluctuating length to keep the notation short and simple. The sampling of observables can be performed analogously in the fixed-length scheme. When implementing the formulas for the observables in the fixed-length scheme, one just has to keep in mind that the sequences used in the formulas are the same ones, but without the padded trivial operators. This means, for instance, that a sum over all propagation steps *p* becomes a sum over all non-trivial propagation steps in the fixed-length scheme.

Up to this point in Section 5, we have focused on the partition function of a quantum-mechanical Hamiltonian:$$Z=\sum_{\omega \in C}\pi \left(\omega \right),$$
expressed as a sum over non-negative weights π(ω) of a configuration ω∈C. However, we are actually not interested in calculating the partition function itself, but rather, the expectation values of observables.

Analogous to the partition function, quantum mechanical expectation values can be expanded into a high-temperature series:(256)$$\langle A\rangle =\frac{1}{Z}\sum_{\left\{|\alpha \rangle \right\}}\sum_{n=0}\sum_{S_{n}}\frac{{\left(-\beta \right)}^{n}}{n!}\langle \alpha |A\prod_{k=1}^{n}H_{l(k)}|\alpha \rangle \;.$$ It is important to realise that configurations with a non-vanishing weight $\pi (\alpha ,S_{n};\beta )$ constituting the partition function do not necessarily contribute to the expectation value 〈A〉 with the same weight or a non-vanishing weight at all [213]. In general, only for operators *A* that are diagonal in the computational basis {|α〉}, the expectation value can be written as
(257)$$\langle A\rangle =\frac{1}{Z}\sum_{\left\{|\alpha \rangle \right\}}\sum_{n=0}\sum_{S_{n}}A\left(\alpha \right)\pi \left(\alpha ,S_{n};\beta \right)$$
with A(α)=Aα. For instance, the magnetisation in the *z*-direction or any *n*-th moment thereof is such a diagonal operator when using the $\sigma^{z}$-eigenbasis $\left\{|\alpha \rangle \}=\{|\sigma_{1}^{z},\cdots ,\sigma_{N}^{z}\rangle \right\}$ as a computational basis. The statistics of the MC estimates for such observables can be improved by realising that [218]
(258)$$\pi \left(\alpha ,S_{n};\beta \right)=\pi \left(\alpha \left(p\right),S_{n}\left(p\right);\beta \right)$$
with $S_{n}\left(p\right)$ being the sequence obtained from cyclically permuting $S_{n}$ for *p* times [218]. This means that the expectation value Equation (Equation 257) can be expressed as
(259)$$\langle A\rangle =\frac{1}{Z}\sum_{\left\{|\alpha \rangle \right\}}\sum_{n=0}\sum_{S_{n}}\frac{1}{n}\sum_{p=0}^{n-1}A\left(\alpha \left(p\right)\right)\pi \left(\alpha ,S_{n};\beta \right),$$
where A(α) is additionally averaged over imaginary time with A(α(p))=〈α(p)|A|α(p)〉.

For off-diagonal operators, one needs to find customised formulas for the expectation value. However, some of these expectation values are accessible in a quite general way. For instance, one can show that the mean energy has a rather simple formula [210,211,218]: (260)$$\langle H\rangle =-\frac{\langle n\rangle }{\beta }\;.$$ Moreover, for the operators $H_{i}$ (see Equation (Equation 213)), one finds [210,211,218]
(261)$$\langle H_{i}\rangle =-\frac{\langle n_{i}\rangle }{\beta }\;,$$
where $n_{i}$ is the amount of operators $H_{i}$ occurring in the operator sequence. Similarly, one can derive a formula for the heat capacity [210,211,218]: (262)$$\begin{array}{cc} C & =\langle n(n-1)\rangle -{\langle n\rangle }^{2}=\langle n^{2}\rangle -{\langle n\rangle }^{2}-\langle n\rangle \;. \end{array}$$ However, for small temperatures, the heat capacity is calculated as the small difference of large numbers.

#### Linear Response and Correlation Functions

An important class of observables is linear response functions like susceptibilities. The linear response of an observable *A* to a perturbation H→H−λB tuned by perturbation parameter λ is given by a Kubo integral [212,219]: (263)$$\begin{array}{cc} \chi_{A,B} & ={\left.\frac{\partial A}{\partial \lambda }\right|}_{\lambda =0} \end{array}$$(264)$$\begin{array}{cc} \; & =\int_{0}^{\beta }\langle A(\tau )B(0)\rangle d\tau -\beta \langle A\rangle \langle B\rangle \;. \end{array}$$ Once again, if *A* or *B* are off-diagonal operators, one has to consider case by case and find customised formulas that can be used to sample the specific observables within the SSE formulation. We will focus on the case where *A* and *B* are diagonal in the chosen computational basis. Important examples are the magnetic susceptibility or its local version of spin–spin correlation functions averaged over imaginary time, i.e., the correlation function at zero frequency when regarding Equation (Equation 264) as a Laplace transformation from imaginary time to frequency space. To make this type of observable accessible to a sampling in the SSE formulation, the imaginary-time correlation function 〈A(τ)B(0)〉 is expanded in temperature analogous to the partition function:(265)$$\begin{array}{cc} \langle A(\tau )B(0)\rangle & =\frac{1}{Z}\mathrm{Tr}\left(e^{-\beta H}e^{\tau H}Ae^{-\tau H}B\right) \end{array}$$(266)$$\begin{array}{cc} \; & =\frac{1}{Z}\sum_{\left\{|\alpha \rangle \right\}}\langle \alpha |\sum_{k=0}^{\infty }\left(\frac{{\left(\beta -\tau \right)}^{k}}{k!}{\left(-H\right)}^{k}\right)A\sum_{l=0}^{\infty }\left(\frac{\tau^{l}}{l!}{\left(-H\right)}^{l}\right)B|\alpha \rangle \end{array}$$(267)$$\begin{array}{cc} \; & =\frac{1}{Z}\sum_{\left\{|\alpha \rangle \right\}}\sum_{l,k=0}^{\infty }\frac{{\left(\beta -\tau \right)}^{k}}{k!}\frac{\tau^{l}}{l!}\langle \alpha |{\left(-H\right)}^{k}A{\left(-H\right)}^{l}B|\alpha \rangle \end{array}$$(268)$$\begin{array}{cc} \; & =\frac{1}{Z}\sum_{\left\{|\alpha \rangle \right\}}\sum_{l,k=0}^{\infty }\frac{{\left(\beta -\tau \right)}^{k}}{k!}\frac{\tau^{l}}{l!}A_{l}B_{0}\langle \alpha |{\left(-H\right)}^{l+k}|\alpha \rangle \;, \end{array}$$
where, in the last step, the operators *A* and *B* were replaced by their respective eigenvalues $A_{l}=\langle \alpha (l)|A|\alpha (l)\rangle$ and $B_{0}=\langle \alpha (0)|B|\alpha (0)\rangle$ of the state |α(p)〉 at the propagation steps p=l,0.

By replacing the sum over *k* by a sum over nl+k and inserting the decomposition of the Hamiltonian, this takes the form
(269)$$\begin{array}{cc} \langle A(\tau )B(0)\rangle & =\frac{1}{Z}\sum_{\left\{|\alpha \rangle \right\}}\sum_{n=0}^{\infty }\sum_{l=0}^{n}\sum_{S_{n}}\frac{{\left(\beta -\tau \right)}^{n-l}}{(n-l)!}\frac{\tau^{l}}{l!}A_{l}B_{0}\langle \alpha |\prod_{i=0}^{n-1}H_{a_{i},b_{i}}|\alpha \rangle \end{array}$$
(270)$$\begin{array}{cc} \; & =\frac{1}{Z}\sum_{\left\{|\alpha \rangle \right\}}\sum_{n=0}^{\infty }\sum_{l=0}^{n}\sum_{S_{n}}\frac{{\left(\beta -\tau \right)}^{n-l}}{(n-l)!}\frac{\tau^{l}}{l!}A_{l}B_{0}\frac{n!}{\beta^{n}}\pi \left(\alpha ,S_{n};\beta \right) \end{array}$$
(271)$$\begin{array}{cc} \; & =\frac{1}{Z}\sum_{\left\{|\alpha \rangle \right\}}\sum_{n=0}^{\infty }\sum_{l=0}^{n}\sum_{S_{n}}\frac{n!}{(n-l)!l!}{\left(1-\frac{\tau }{\beta }\right)}^{n-l}{\left(\frac{\tau }{\beta }\right)}^{l}A_{l}B_{0}\pi \left(\alpha ,S_{n};\beta \right) \end{array}$$
(272)=1Z∑{|α〉}∑n=0∞∑l=0n∑Snnl1−τβn−lτβlπ(α,Sn;β)1n∑p=0n−1Ap+lBp,
where, in the last step, the average over imaginary time was taken in order to improve the statistics. From Equation (Equation 269), the connection between the discrete propagation steps of the SSE and the continuous imaginary time τ becomes apparent [211,212]. An imaginary-time separation τ corresponds to a binomial distribution of separations *l* of SSE propagation steps:(273)B(l|τ,n)=nl1−τβn−lτβl,
which is peaked around l=nτ/β [211]. If one is interested in the spectral properties of the system, one can use this formula for sampling imaginary-time correlation functions 〈A(τ)B(0)〉. However, there are more efficient ways to calculate imaginary-time correlation functions by embedding the SSE configuration into continuous imaginary time. We refer to Refs. [212,220] for details, as this is out of the scope of this review.

To get to the linear response function $\chi_{A,B}$ in Equation (Equation 263), the imaginary time integral of in Equation (Equation 269) has to be calculated. This integral can be analytically calculated: (274)$$\begin{array}{cc} \int_{0}^{\beta }d\tau {\left(1-\frac{\tau }{\beta }\right)}^{n-l}{\left(\frac{\tau }{\beta }\right)}^{l} & =\beta \int_{0}^{1}du{\left(1-u\right)}^{n-l}u^{l} \end{array}$$(275)$$\begin{array}{cc} \; & =\beta \frac{(n-l)!}{n!}l!\int_{0}^{1}du{\left(1-u\right)}^{n} \end{array}$$(276)$$\begin{array}{cc} \; & =\beta \frac{(n-l)!}{n!}l!\frac{1}{n+1} \end{array}$$
by performing *l* partial integrations. Inserting Equation (Equation 269) into the formula (Equation 263) and performing the imaginary time integral Equation (Equation 274) yield
(277)$$\chi_{A,B}=\frac{1}{Z}\sum_{\left\{|\alpha \rangle \right\}}\sum_{n=0}^{\infty }\sum_{S_{n}}\pi \left(\alpha ,S_{n};\beta \right)\frac{\beta }{n(n+1)}\sum_{l=0}^{n}\sum_{p=0}^{n-1}A_{p+l}B_{p}-\beta \langle A\rangle \langle B\rangle \;.$$ One can further separate the l=n term while using the periodicity $A_{p+n}=A_{p}$ in imaginary time and rewrite the sums over *l* and *p* as a product of two sums. This eventually yields [210]
(278)$$\begin{array}{cc} \chi_{A,B}\; & =\langle {\frac{\beta }{n(n+1)}\left[\sum_{p=0}^{n-1}A_{p}B_{p}+\left\{\sum_{p=0}^{n-1}A_{p}\right\}\left\{\sum_{p=0}^{n-1}B_{p}\right\}\right]\rangle }_{\pi (\alpha ,S_{n};\beta )}-\beta \langle A\rangle \langle B\rangle \;. \end{array}$$ From an algorithmic perspective, the two sums in Equation (Equation 278) need to be calculated by traversing the operator sequence. The effort for measuring $\chi_{A,B}$, therefore, scales like the algorithm with complexity O(βN) when the observables $A_{p}$, $B_{p}$, and $A_{p}B_{p}$ are not calculated from scratch at every propagation step *p*, but gradually updated and averaged while propagating the state through the sequence.

For the special case of the susceptibility, one needs to set $A=m=\sum_{i}\sigma_{i}^{z}$ and B=M=N·m. The last term $N\beta m^{2}$ in Equation (Equation 278) can be dropped for simulations of finite systems due to m=0. Explicitly, this gives
(279)$$\begin{array}{cc} \chi & =N\langle {\frac{\beta }{n(n+1)}\left[\sum_{p=0}^{n-1}m_{p}m_{p}+\left\{\sum_{p=0}^{n-1}m_{p}\right\}\left\{\sum_{p=0}^{n-1}m_{p}\right\}\right]\rangle }_{\pi (\alpha ,S_{n};\beta )} \end{array}$$
for the susceptibility. Once again, we want to stress that this formula is not formulated in the fixed-length scheme and the average over the propagation index *p*, therefore, refers to the propagation steps of non-trivial operators. Averaging over all propagation steps in the fixed-length scheme is incorrect.

For the spin–spin correlation functions $G_{i,j}\left(\omega =0\right)$, the operators are set to $A=\sigma_{i}^{z}$ and $B=\sigma_{j}^{z}$, leading to
(280)$$\begin{array}{cc} G_{i,j}\left(\omega =0\right) & =\int_{0}^{\beta }\langle \sigma_{i}^{z}\left(\tau \right)\sigma_{j}^{z}\left(0\right)\rangle d\tau \end{array}$$
(281)$$\begin{array}{cc} \; & =\langle {\frac{\beta }{n(n+1)}\left[\sum_{p=0}^{n-1}\sigma_{i,p}^{z}\sigma_{j,p}^{z}+\left\{\sum_{p=0}^{n-1}\sigma_{i,p}^{z}\right\}\left\{\sum_{p=0}^{n-1}\sigma_{j,p}^{z}\right\}\right]\rangle }_{\pi (\alpha ,S_{n};\beta )}\;, \end{array}$$
where it was already used that $\langle \sigma_{i}^{z}\rangle =0$ for any finite system. Similar to the zero-frequency spin–spin correlation function, one can also calculate an equal-time spin–spin correlation function: (282)$$\begin{array}{cc} G_{i,j}\left(\tau =0\right) & =\langle \sigma_{i}^{z}\sigma_{j}^{z}\rangle ={\langle \frac{1}{n}\sum_{p=0}^{n-1}\sigma_{i,p}^{z}\sigma_{j,p}^{z}\rangle }_{\pi (\alpha ,S_{n};\beta )}\;. \end{array}$$ Sampling the correlations among all sites *i* and *j* at all propagation steps *p* leads to a complexity of $O(\beta N^{3})$, which is worse than the computational complexity O(βN) of the SSE QMC updates. Even though one could eliminate the average over imaginary time in the case of Equation (Equation 282) to reduce the complexity to $O(N^{2})$ at the expense of statistical accuracy, this is not possible for Equation (Equation 280) as the sum over imaginary time is intrinsic in the definition of the zero-frequency correlation function. However, by realising that the correlations $\sigma_{i}^{z}\sigma_{j}^{z}$ between two sites *i* and *j* are not altered at the order O(βN)-times in imaginary time, but only O(β), one can measure the correlation functions Equations (Equation 280) and (Equation 282) in $O(\beta N^{2})$ time. This is achieved by traversing imaginary time while memorising the propagation step last[i] of the last preceding spin flip operator for every site separately. When a spin flip occurs at a site *j* at propagation step *p*, one needs to update the correlation function:(283)$$G\left[i,j\right]\to G\left[i,j\right]+\sigma_{i,p}^{z}\sigma_{j,p}^{z}\left(p-\max\left(\mathrm{last}\left[i\right],\mathrm{last}\left[j\right]\right)\right)$$
for every site *i* before the local magnetisation is propagated with $\sigma_{j,p+1}^{z}=-\sigma_{j,p}^{z}$. One further needs to update last[j]=p. In order to avoid that the measurement dominates the simulation for large systems, we only measure the correlations between a fixed site i=1 and j∈{1,⋯,N}. This finally reduces the complexity of the measurement to O(βN).

Correlation functions are observables that contain much information about a system’s state and, in particular, its order. Moreover, its decay at the critical point with the distance between two spins is connected to the critical exponent η (anomalous dimension). Away from the critical point, the correlation function usually decays exponentially with long-range models being one exception to this. The length scale of this exponential decay is given by the correlation length ξ, which itself is an interesting quantity at a continuous phase transition as its divergence is the reason for systems becoming scale-free at the critical point. As the correlation function does not decay exponentially for long-range system, which we focus on in this review, we will use the more general term “characteristic length scale” to refer to the length scale at which the correlations switch to their long-distance behaviour. In particular, with respect to the Q-FSS for systems above the upper critical dimension, the characteristic length scale is a crucial quantity as its scaling was predicted to be different from the standard FSS with $\xi_{L}∼L^{ϙ}$ instead of $\xi_{L}∼L$ (see Section 2.3 or Ref. [34] for details on scaling above the upper critical dimension in quantum systems). This quantity, therefore, played a crucial role in confirming the Q-FSS hypothesis. However, the characteristic length scale is a subtle quantity, which is hard to extract or even define on a finite lattice. For a finite system, the definition of a characteristic length scale is not unique, and there are several definitions for $\xi_{L}$, which will converge to $\xi_{\infty }$ for L→∞ [221]. For long-range systems, finding a suitable definition for the characteristic length scale is even more difficult, as the correlation function decays algebraically even away from the critical point [222].

Common definitions that are tailored for correlation lengths, which specify the exponential decay of a correlation function at long distances, such as the second moment: (284)$$\xi_{\infty }=\sqrt{\frac{1}{2D}\frac{\int |\vec{r}{|}^{2}G\left(\vec{r}\right)dr}{\int G(\vec{r})dr}}$$
therefore, might yield $\xi_{\infty }=\infty$ in an infinite system not only at the critical point, but also away from the critical point [223].

One possible definition for long-range quantum systems is [224]
(285)$$\begin{array}{cc} \xi_{L}^{\left(\mathrm{LR}\omega \right)} & =\frac{1}{q_{\min}}{\left[\frac{{\tilde{G}}_{L}\left(0,\omega =0\right)-{\tilde{G}}_{L}\left(q_{\min},\omega =0\right)}{{\tilde{G}}_{L}\left(q_{\min},\omega =0\right)}\right]}^{1/\sigma } \end{array}$$
with G(q,ω) being the Fourier transform of $G_{i,j}\left(\omega \right)$ from real space to momentum space and $q_{\min}=2\pi /L$ the smallest wavevector fitting on the finite lattice. By inserting the Gaussian propagator $\tilde{G}\left(q,\omega =0\right)∼{\left(b_{\sigma }q^{\sigma }+m^{2}\right)}^{-1}$ for long-range interactions [20,21] into Equation (Equation 285), one obtains
(286)$$\begin{array}{cc} \xi_{L}^{\left(\mathrm{LR}\omega \right)} & =\frac{1}{q_{\min}}{\left[\frac{b_{\sigma }q_{\min}^{\sigma }+m_{L}^{2}}{m_{L}^{2}}-1\right]}^{1/\sigma }\\ \; & =b_{\sigma }^{1/\sigma }m_{L}^{-2/\sigma }, \end{array}$$
so that the momentum dependency cancels. Another definition for the same quantity, but using the equal-time Gaussian propagator $G_{L}\left(0,\tau =0\right)∼{\left(2\sqrt{\tilde{g}}\sqrt{b_{\sigma }q^{\sigma }+m^{2}}\right)}^{-1}$ [212], is
(287)$$\begin{array}{cc} \xi_{L}^{\left(\mathrm{LR}\tau \right)} & =\frac{1}{q_{\min}}{\left[\frac{{\tilde{G}}_{L}^{2}\left(0,\tau =0\right)-{\tilde{G}}_{L}^{2}\left(q_{\min},\tau =0\right)}{{\tilde{G}}_{L}^{2}\left(q_{\min},\tau =0\right)}\right]}^{1/\sigma }\\ \; & =b_{\sigma }^{1/\sigma }m_{L}^{-2/\sigma }\;. \end{array}$$ By inserting the scaling of the gap $m_{\infty }∼{\left|r\right|}^{z\nu }={\left|r\right|}^{\sigma \nu /2}$ for the *n*-component quantum rotor model in the long-range mean-field regime [20,21], which is relevant for most applications discussed below, one obtains the scaling of $\xi^{\left(LR\right)}$ in the thermodynamic limit to be
(288)$$\xi_{\infty }^{\left(LR\right)}∼{\left|r\right|}^{-\nu }\;,$$
which is the singularity one expects from the characteristic length scale.

### 5.4. Sampling at Effectively Zero Temperature

The SSE QMC approach can be used to stochastically calculate the thermal average of operators for systems of a finite size at a finite temperature. We discussed in Section 2 that QPTs are a concept defined in the thermodynamic limit at zero temperature. Further, we presented in Section 2.2 and Section 2.3 how the critical point, as well as critical exponents can be extracted from certain observables calculated on finite systems at zero temperature. The goal of many works that use SSE QMC sampling is to calculate the expectation value of these observables on finite systems at an effective zero temperature [32,34]. We define the notion of effective zero temperature as a temperature at which, in practice, only the ground state contributes to the thermal average, as all excited states are exponentially suppressed [32,34]. This definition can be applied only to gapped systems as gapless systems at infinitely small, but finite temperature lead to a thermal average over the infinitesimally low-lying excitations. As the SSE QMC approach is sampling finite systems, there cannot be a non-analytic point in the ground-state energy associated with a second-order QPT [113]. Closely related with this statement, the ground state of a finite system cannot spontaneously break the symmetry of the Hamiltonian, as it is the case at a second-order QPT [113]. An intuition to this statement can be built from a perturbative point of view. On a finite system, there is always a finite-order perturbative process coupling the hypothetical ground states of a symmetry-broken phase. Due to this finite-order coupling, there will always be a level repulsion between the states and no true degeneracy occurs. This means that, on a finite system, there is always a finite-size energy gap between the ground state and the excitations. As discussed in Section 2.2 on a finite system, there is a pseudo-critical point close to the parameter values of the QPT in the thermodynamic limit. For a system with gapped phases on both sides of the QPT, we expect the relevant finite-size gap to be the smallest in the vicinity of the pseudo-critical point. The finite-size gap at the pseudo-critical point is expected scales as $L^{-zϙ}$ with the linear system size *L* to the power of the dynamical correlation length exponent *z* [34,113,129,212] and ϙ=1 for a QPT below the upper critical dimension (see Section 2.3). This statement can be directly derived from the finite-size scaling form of the energy gap: (289)$$\begin{array}{c} \Delta_{L}\left(r\right)=L^{-zϙ}\Psi_{\Delta }\left(L^{ϙ/\nu }r\right)\;. \end{array}$$ Note that the scaling dependence for systems above the upper critical dimension has not been confirmed yet by numerical studies. To perform simulations at effective zero temperature, there are two main approaches in the contemporary literature. Firstly, one just scales the simulation temperature with *L* for different linear system sizes. This is a valid approach for many systems as z=1 is a common value for the critical exponent, indicating a space–time symmetry in the correlations. For long-range interacting systems such as the LRTFIM or unfrustrated long-range Heisenberg models, one has z≤1, which makes the scaling sufficient, but overly ambitious [21]. An improved version of this naive approach is to scale the simulation temperature with $L^{zϙ}$ for different linear system sizes. This follows the expected scaling of the finite-size gap. The correct prefactor of the scaling is not known and accounted for in the scaling, and corrections to scaling for small system sizes can lead to errors. A more sophisticated approach to determine a suitable effective zero temperature for a finite system is discussed in Refs. [32,34,225]. This technique was introduced in Ref. [225] as the beta-doubling method. The idea is to study the convergence of observables $\langle O_{L}\left(r,\beta \right)\rangle$ for a fixed system size and parameter value *r* at successively doubled inverse temperatures $\beta =2^{n}$ of the simulation. The fraction of $\langle O_{L}\left(r,\beta \right)\rangle /\langle O_{L}\left(r,\beta_{\max}\right)\rangle$ is considered with $\beta_{\max}$ being the largest considered β value. This quantity is used to probe the convergence of the observable $\langle O_{L}\left(r,\beta \right)\rangle$ towards zero temperature. The temperature convergence can be gauged using a plot, as demonstrated in Figure 19. Note that the value of $\langle O_{L}\left(r,\beta_{\max}\right)\rangle /\langle O_{L}\left(r,\beta_{\max}\right)\rangle$ is always at one. Therefore, as a rule of thumb, one can validate simulation temperatures for which at least the two points prior to the last point also have a value close to one.

Although the beta-doubling scheme has more overhead than a naive scaling with $L^{zϙ}$, it provides multiple advantages. First of all, one does not need to know the dynamical critical exponents *z* beforehand. Secondly, in most cases, it is easy to implement: if one uses either way a simulated annealing scheme by successively cooling the simulation temperature during the thermalisation of the algorithm, the beta-doubling scheme can be easily incorporated in this procedure. Thirdly, the beta doubling provides a direct demonstration that a simulation is sampling zero-temperature properties. It captures intrinsically the relevant scaling $L^{zϙ}$, as well as potentially large prefactors and corrections to this scaling. Fourthly, it does not need to be applied to all parameter values *r*, as it suffices to perform the beta-doubling procedure only for the parameter values where the relevant finite-size gap is expected to be the smallest and use this temperature for the whole parameter range.

In general, the beta-doubling scheme can help to reduce computational cost by performing simulations at a tailored effective zero temperature. If computational time is not a critical factor, in many practical applications, one can use the temperature that is sufficient for the largest system and sample all smaller systems at the same temperature.

### 5.5. Overview: Path Integral Quantum Monte Carlo Algorithms for Long-Range Models

The QMC methods discussed so far are both based on the SSE formulation [39,213]. In parallel with the development of the SSE-based QMC methods for long-range systems, path integral (PI) QMC methods operating in continuous imaginary time have been introduced for extended Bose–Hubbard models with long-range density–density interactions [215,226,227,228] (see (Equation 290)). The main application of these methods in the context of long-range interactions is the determination and classification of ground states [64,229,230]. Since we focus on reviewing QPTs in long-range interacting systems, which are usually simulated using the SSE, we only skim over the basic concepts relevant for the PI QMC methods. One motivation for the development of these methods was the study of the Mott insulator to superfluid transition in the presence of disorder [231,232,233]. Further, long-range density–density interactions were added to the algorithms [226]. A great application case for these algorithms is the study of extended Bose–Hubbard models [64,229,230], which are an effective description of ultracold atomic or molecular gases trapped in optical lattice potentials [56,58,61,62,64,68,229,234,235,236,237,238]. For a comprehensive explanation on how to derive effective Bose–Hubbard models for these systems, see Refs. [56,236,237,238].

A basic Hamiltonian studied with the world-line QMC methods in this context [226,227,228] reads
(290)H=−μ∑ini+∑ijVi,jninj−t∑〈i,j〉ai†aj†+aj†ai†
with (hard-core) bosonic degrees of freedom on the lattice site *i* described by creation $a_{i}^{†}$, annihilation ai†, and particle number $n_{i}$ operators. The coupling μ describes a chemical potential and $V_{i,j}\leq 0$ the density–density interaction between particles at sites *i* and *j*, and the hopping amplitude t>0.

PI QMC methods formulated in imaginary time are based on the PI formulation of the partition function:(291)$$\begin{array}{c} Z=\mathrm{Tr}\left\{e^{-\beta H}\right\}=\lim_{K\to \infty }\mathrm{Tr}\left\{{\left(e^{-\Delta \tau H}\right)}^{K}\right\}\;\mathrm{with}\;\Delta \tau =\beta /K \end{array}$$
using the Suzuki–Trotter decomposition [115]. The PI formulation is used to extend the configuration space to all imaginary-time trajectories |α(τ)〉 of computational basis states [39,226], which are periodically closed in β(|α(0)〉=|α(β)〉) and which are connected by an imaginary-time evolution $\left(\langle \alpha (\tau_{2})|e^{-H(\tau_{2}-\tau_{1})}|\alpha (\tau_{1})\rangle \right)$. These methods can be implemented in discrete [239] or continuous [226,240] imaginary time.

In the SSE approach, the configuration space is extended using the operator sequences arising from a high-temperature expansion of the exponential in the partition function. These configurations can be regarded as a series of discretely propagated states |α(p)〉 with an artificial propagation index *p* labelling its position in the sequence. This introduction of an additional dimension to obtain a configuration space that can be sampled using MC in the PI and SSE QMC approach hints that both approaches are closely related. Studies of the close relationships between the SSE and PI representations can be found in Refs. [211,241]. Concepts developed in one of the two pictures can be equivalently formulated in the other one. For example, the concept of loops [215,216,242] and directed loops [215,217,242] can be applied in both formulations. The sampling of the Heisenberg model as described in Section 5.2 can be transferred to the PI QMC picture, as it is a direct application of the directed loop idea [217,225].

Since we will focus in this review on quantum-critical properties of magnetic systems, we only briefly summarise the major applications of PI QMC methods for extended (hard-core) Bose–Hubbard models with dipolar density–density interactions [64,229,230]. The PI QMC is used in this field to determine ground-state phase diagrams studying the emergence of solid, supersolid, and superfluid phases [64,229,230]. It has been demonstrated in Ref. [64] that long-range interactions stabilise more solid phases than the chequerboard solid with a filling of 1/2 present in systems with nearest-neighbour interactions. It has also been shown that long-range interactions lead to the emergence of supersolid ground states [64]. The great benefit of the QMC simulations in comparison to other methods (e.g., mean-field calculations [111]) is the quantitative nature of these methods. Therefore, the numerical study of these models is highly relevant, since experimental progress in cooling and trapping atoms and molecules with dipolar electric or magnetic moments [67,69,72,73,243] enables experiments to realise extended Hubbard models with long-range interactions [68]. We expect the implementation of SSE QMC techniques for this kind of model to be straightforward using the directed loop approach [217,225].

## 6. Long-Range Transverse-Field Ising Models

In this section, we review the ground-state quantum phase diagrams of the long-range transverse-field Ising model (LRTFIM) with algebraically decaying long-range interactions. We emphasise how the Monte Carlo-based techniques introduced in this work are a reliable way to obtain critical exponents of quantum phase transitions (QPTs) in this model. The Hamiltonian of the LRTFIM is given by
(292)$$\begin{array}{c} H=\frac{J}{2}\sum_{i\neq j}\frac{1}{|{\vec{r}}_{i}-{\vec{r}}_{j}{|}^{d+\sigma }}\sigma_{i}^{z}\sigma_{j}^{z}-h\sum_{i}\sigma_{i}^{x} \end{array}$$
with Pauli matrices $\sigma_{i}^{\kappa }$ and κ∈{x,z} describing spins 1/2 located on the lattice sites ${\vec{r}}_{i}$. The Ising coupling is tuned by the parameter *J*. For J>0, the Ising interaction is antiferromagnetic and, for J<0, ferromagnetic. The amplitude of the transverse field is denoted by *h*. The positive parameter (d+σ) governs the algebraic decay of the Ising interaction. Here, *d* denotes the spatial dimension of the system and σ is a tunable real-valued parameter. The limiting cases of the algebraically decaying long-range interaction are the nearest-neighbour interaction for σ=∞ and an all-to-all coupling for (d+σ)=0. As a side note, there is literature where α≡d+σ is used as a parameter to tune the decay of the long-range interaction. For this review, we stay with d+σ in order to treat systems with different spatial dimensions *d* on the same footing.

The Hamiltonian (Equation 292) can be treated using the pCUT+MC method as described in Section 4. A generic, perturbative starting point for the LRTFIM is the high-field limit. The transverse field term is regarded as the unperturbed Hamiltonian and the Ising interaction as the perturbation. Using the Matsubara–Matsuda transformation [25,29,89]:(293)σix=1−2bi†bi†σiz=bi†+bi†,
the Hamiltonian (Equation 292) can be brought into a hard-core bosonic quasi-particle picture:(294)H=ϵ0N+∑ibi†bi†+∑i≠jλi,jbi†bj†+bi†bj†+bi†bj†+bi†bj†. Here, $\epsilon_{0}$ is a constant, *N* is the number of sites, and $b_{i}^{\left(†\right)}$ is a hard-core bosonic quasiparticle annihilation (creation) operator. The perturbation parameters are given by
(295)$$\begin{array}{c} \lambda_{i,j}=\frac{J}{4h}\frac{1}{|{\vec{r}}_{i}-{\vec{r}}_{j}{|}^{\left(d+\sigma \right)}}\;. \end{array}$$ In the pCUT language, the perturbation associated with the perturbation parameter λ can be written as $V=T_{-2}+T_{0}+T_{2}$, where the hopping processes bi†bj† are contained in $T_{0}$ and the pair creation (annihilation) processes are in $T_{2}$ ($T_{-2}$).

Further, the Hamiltonian (Equation 292) is of the form to straightforwardly apply the SSE QMC algorithm reviewed in Section 5.1 (see Equation (Equation 226)). We recall that this QMC algorithm is sign-problem-free for arbitrary, potentially frustrated, Ising interactions.

The studied effects of algebraically decaying long-range Ising interactions on the ground-state phase diagram can be categorised into three scenarios:

First, for ferromagnetic interactions, there is a QPT with $Z_{2}$ symmetry breaking between a ferromagnetic low-field phase and a symmetric *x*-polarised high-field phase. The intriguing effect of long-range interaction is the emergence of three regimes with distinct types of universality of the QPT. For large values of the parameter $\sigma >2-\eta_{\mathrm{SR}}$, the universality class is the same as in the nearest-neighbour model. For intermediate σ values of $2d/3<\sigma <2-\eta_{\mathrm{SR}}$, the critical exponents change as a function of σ and the criticality can be described by a non-trivial long-range theory. For σ<2d/3, the long-range interaction lowers the upper critical dimension below the physical dimension of the model and the transition becomes a long-range mean-field transition. We review the recent studies investigating this model in Section 6.1 with a particular focus on the $\phi^{4}$-theory in Section 6.1.1 and on numerical studies investigating the critical properties in Section 6.1.2. The aspects regarding FSS above the upper critical dimension are summarised in Section 6.1.3.

The second relevant category is antiferromagnetic Ising interactions where the nearest-neighbour couplings form a bipartite lattice. In this case, there is a QPT with $Z_{2}$ symmetry breaking between an antiferromagnetic low-field phase and a symmetric *x*-polarised high-field phase. The current state of literature indicates no change in the universality class of the phase transition in dependence of the long-range interaction. We summarise and review the recent progress in studying this case in Section 6.2.

The last aspect is scenarios where the nearest-neighbour couplings do not form a bipartite lattice. In this case, the long-range interactions may substantially alter the quantum phase diagram of the nearest-neighbour model. We summarise and review the recent progress regarding this case in Section 6.3.

### 6.1. Ferromagnetic Long-Range Transverse-Field Ising Models

The ferromagnetic TFIM with nearest-neighbour interactions in d≥1 dimensions is a paradigmatic model to display a QPT in a (d+1)D-Ising universality class. This transition describes the non-analytic change in the ground-state between a $Z_{2}$ symmetry-broken low-field phase and a symmetric *x*-polarised high-field phase. By adding algebraically decaying long-range interactions to this model, one can study how such interactions can alter non-universal, as well as universal properties of a QPT and which new features can emerge.

In the case of ferromagnetic interactions, the algebraically decaying long-range interaction stabilises the symmetry-broken phase and the QPT shifts to larger *h* values [25,29,32,34]. An intuitive way to understand this behaviour is that, due to the long-range interaction, more connections are introduced that align the spins in the *z*-direction, and the energy cost for a single spin flip increases. In the limit σ→0, the critical value of the transverse field diverges as the cost for a single spin flip becomes extensive.

In this review, we focus exclusively on the regime of so-called weak long-range interactions with an algebraic decay exponent σ>0 [21]. The research field of strong long-range interactions considers σ≤0 [5,42,43,44,45]. For weak long-range interactions, the Hamiltonian (Equation 292) is well defined in the thermodynamic limit, and common properties such as the ground-state additivity and thermodynamic quantities are well defined [5]. For strong long-range interactions, the energy of the ferromagnetically aligned state is superextensive, and in order to study the model in the thermodynamic limit, an appropriate rescaling of the Ising coupling with the system size is required [5].

The significance of the ferromagnetic LRTFIM comes from its paradigmatic nature of displaying changes in the universal behaviour due to the long-range interaction [20,21,25,29,32]. It has been known since the advent of the theory of classical phase transitions that algebraically decaying long-range interactions are a potential knob to alter the fixed-point structure of the renormalisation flow and, therefore, change the universal properties (e.g., critical exponents) [1,2,140]. Coming from the limit σ=∞ of nearest-neighbour interactions, the QPT of a *d*-dimensional model remains in the (d+1)D-Ising universality class until $\sigma =2-\eta_{\mathrm{SR}}$ ($\eta_{\mathrm{SR}}$ is the anomalous dimension critical exponent of the nearest-neighbour universality class). This means that the critical exponents are constant as a function of σ and the fixed point associated with the QPT remains the one of the nearest-neighbour model. For $\sigma =2-\eta_{\mathrm{SR}}$, the RG fixed point associated with the QPT changes to the one of a non-trivial long-range interacting theory, and the critical exponents become σ-dependent [21]. The upper critical dimension of this theory with long-range interaction is lowered with respect to the short-range model as $d_{\mathrm{uc}}=3\sigma /2$. If the (fixed) spatial system dimension *d* is larger than or equal to the σ-dependent upper critical dimension, the universality class describing the QPT enters a long-range mean-field regime. The three different regimes, as well as their boundaries in dependence of the dimensionality of the system are visualised in Figure 20.

In the following, we recapitulate the basic field-theoretical arguments leading to the distinction between three universality regimes (see Section 6.1.1). We will further emphasise the most relevant aspects for the two long-range regimes. In the non-trivial, intermediate regime, the precise values of the critical exponents depend on σ and need to be determined numerically. Therefore, we review the recent numerical studies computing the critical exponents in Section 6.1.2. To conclude the discussion, we outline how to utilise ferromagnetic long-range Ising models to study QPTs above the upper critical dimension (see Section 6.1.3).

#### 6.1.1. $\phi^{4}$-Theory for Quantum Rotor Models with Long-Range Interactions

Starting with the short-range interacting action (see Equation (Equation 334)) of the *n*-component quantum rotor model, Dutta et al. [20] introduced an action for long-range interacting rotor models, by adding the algebraic decay between interacting fields as an additional term to the action in real space. Note the the action for the Ising model is equivalent to the one-component (scalar) rotor model [113]. By performing a Fourier transformation of the resulting action, the authors obtained the action:(296)$$\begin{array}{cc} S_{\tilde{\phi }}= & \int \frac{d^{d}q}{{\left(2\pi \right)}^{d}}\int \frac{d\omega }{2\pi }\left[\tilde{g}\omega^{2}+r+aq^{\sigma }+bq^{2}\right]\tilde{\phi }\left(q,i\omega \right)\tilde{\phi }\left(-q,-i\omega \right)\\ & +u\int \frac{d\omega_{1}}{2\pi }\cdots \frac{d\omega_{4}}{2\pi }\int \frac{d^{d}q_{1}}{{\left(2\pi \right)}^{d}}\cdots \frac{d^{d}q_{4}}{{\left(2\pi \right)}^{d}}\delta^{d}\left(q_{1}+\cdots +q_{4}\right)\delta \left(\omega_{1}+\cdots +\omega_{4}\right)\times \\ & \times \left[\tilde{\phi }\left(q_{1},i\omega_{1}\right)\tilde{\phi }\left(q_{2},i\omega_{2}\right)\right]\left[\tilde{\phi }\left(q_{3},i\omega_{3}\right)\tilde{\phi }\left(q_{4},i\omega_{4}\right)\right] \end{array}$$
with a,b>0, σ from the decay exponent d+σ of the microscopic model, and r,u coupling constants as in the nearest-neighbour case. The $q^{\sigma }$ term, which is new compared to the nearest-neighbour theory, results from the Fourier transformation of the added long-range interacting term:(297)$$\int d^{d}x\;\int d^{d}y\;\int d\tau \;\frac{\phi (x,\tau )\phi (y,\tau )}{{\left|x-y\right|}^{d+\sigma }}$$
of the order-parameter field ϕ [40]. As the relevant modes for the QPT are the ones with long wavelengths, it is clear that, for σ≥2, the $q^{2}$ term gives the leading contribution in *q*; the $q^{\sigma }$ term can be neglected and, therefore, the system is, in this case, described by the short-range interacting theory. In contrast to this, a different critical behaviour distinct from the short-range case is possible for σ<2. With the same argument as above, the $q^{2}$ term becomes negligible for σ<2. In the following, we will focus on this regime, where the behaviour differs from the nearest-neighbour case. In order to gain insight into the QPT affected by long-range interactions, an investigation of the Gaussian theory is a good starting point to enter the framework of perturbative renormalisation group calculations [20,21,113] like the ϵ-expansion in Ref. [20]. From Equation (Equation 296), one can directly read off the propagator of the Gaussian theory as
(298)$$G_{0}{\left(q,\omega \right)}^{-1}=q^{\sigma }+\tilde{g}\omega^{2}+r\;.$$ From field-theoretical power-counting arguments, Dutta et al. [20] derived several properties of the Gaussian theory. Using the divergence of the mass renormalisation term, the lower critical dimension below which no phase transition is occurring can be derived as $d_{\mathrm{lc}}=\sigma /2$ [20]. By regarding the Gaussian propagator, one finds directly that η=2−σ as η is defined by the deviation from 2 of the leading power by which the momentum enters the propagator (see Equation (Equation 368)). Note that, in the long-range case, the Gaussian theory has an η≠0. From further power-counting analysis, it is possible to derive more critical exponents from the Gaussian theory [20]:(299)$$\begin{array}{c} \gamma =1\;\nu =\frac{1}{\sigma }\;z=\frac{\sigma }{2}\;\eta =2-\sigma . \end{array}$$ In the limit σ→2, the short-range mean-field exponents are recovered [20]. With the argument that the hyperscaling relation still holds directly at the upper critical dimension, Dutta et al. [20] derived the upper critical dimension by inserting the long-range Gaussian exponents (see Equation (Equation 299) and α=0).
(300)$$2-\alpha =\nu \left(d+z\right)⟶2=\frac{1}{\sigma }\left(d_{\mathrm{uc}}+\frac{\sigma }{2}\right)\Leftrightarrow d_{\mathrm{uc}}=\frac{3\sigma }{2}\;.$$ In addition to the investigation of the Gaussian part of the action in Equation (Equation 296), one can derive non-trivial exponents below the upper critical dimension by deriving perturbative corrections to the Gaussian exponents in an ϵ-expansion around the upper critical dimension [113]. Dutta et al. [20] established with a one-loop renormalisation group expansion of flow equations that the Gaussian fixed point is stable for $d\geq d_{\mathrm{uc}}$. Therefore, the long-range Gaussian exponents (see Equation (Equation 299)) are valid for σ≤2D/3. For $d<d_{\mathrm{uc}}$, the correction indicates a flow to a non-trivial fixed point. The first-order corrections of the ϵ-expansion for ν and γ are provided [20], and the first-order corrections to η and *z* are argued to be zero [20]. It should be noted that the dynamic correlation exponent z≠1 for σ<2 and, therefore, correlations are not isotropic in the space and imaginary-time direction [20,21]. “For any value of σ<2, η sticks to its mean-field value 2−σ, because the renormalization does not generate new $q^{\sigma }$ terms” [20]. This also fits well with the claim that $\sigma =2-\eta_{\mathrm{SR}}$, because, then, there is no discontinuity at the boundary [1,2,3,21,244].

A recent study by Defenu et al. [21] with a functional renormalisation group approach investigated the flow of couplings in an effective action with anomalous dimension effects. This analysis showed that the boundary between the short-range and the non-trivial long-range regime does not occur at σ=2, but at $\sigma =2-\eta_{\mathrm{SR}}$. So, it is shifted towards lower σ by the anomalous dimension of the short-range criticality. This anomalous dimension consideration in [21] is mainly driven by the following argument: From the functional renormalisation group ansatz, the authors derived two flow equations for the coupling of the long-range coupling $Z_{k}$ (connected to *a* in Equation (Equation 296)) and the short-range coupling $Z_{2,k}$ (connected to *b* in Equation (Equation 296)):(301)$$\begin{array}{cc} \partial_{t}Z_{k} & =\left(2-\sigma -\eta \right)Z_{k} \end{array}$$(302)$$\begin{array}{cc} \partial_{t}\eta & =\frac{\partial_{z}Z_{2,k}}{Z_{2,k}}\;. \end{array}$$ The first flow Equation (Equation 301) has fixed points for η=2−σ or $\lim_{k\to 0}Z_{k}=0$. The first fixed point results in long-range exponents, whereas the second fixed point means that the long-range coupling becomes irrelevant.

In addition to this anomalous dimension effect, the authors in [21] also calculated critical exponents in the non-trivial intermediate phase with their functional renormalisation group technique. They also compared their results with Monte Carlo simulations of the dissipative non-Ohmic spin chains, which can be mapped to the same behaviour as the long-range interacting quantum Ising model [245], showing reasonable agreement.

#### 6.1.2. Critical Exponents and Critical Points for One- and Two-Dimensional Systems

The criticality of the ferromagnetic LRTFIM has been studied by a wide variety of methods with different strengths and weaknesses. Overall, all methods qualitatively agree with each other and confirm the three expected regimes from the field-theoretical analysis [20,21], namely a long-range mean field, a short-range regime, and an intermediate non-trivial long-range regime, which connects the two limiting regimes with monotonously changing exponents. In the limit σ→0, the value of the critical field $h_{c}$ diverges as the energy cost for a single spin flip in the low-field phase becomes extensive and the quantum fluctuations introduced by the transverse field fail to compete with the Ising interaction. In the limit σ→∞, the value of $h_{c}$ converges towards the exact value for the nearest-neighbour transverse-field Ising model [25,29,32,246].

In the case of the LRTFIM on the one-dimensional linear chain, the model has been studied by perturbative continuous unitary transformation with Monte Carlo embedding (pCUT + MC) [29,31,34,247], the functional renormalisation group (FRG) [21], the density matrix renormalisation group (DMRG) [28,248], as well as stochastic series expansion (SSE) [32,34,247,249], path integral (PI) [33] QMC, and QMC with stochastic parameter optimisation (QMC+SPO) [250].

The critical value of the transverse field was calculated with the pCUT method [29,31,34] by estimating the gap closing using DlogPadé extrapolation of the perturbative gap series coming from the high-field phase, by the DMRG using finite-size scaling of the fidelity susceptibility [28], and by SSE QMC calculations using finite-size scaling of the magnetisation [32,249]. The finite-temperature transition points were studied by PI QMC calculations [33] for σ=0.5 in the long-range mean field regime and σ=−0.95 in the strong long-range regime.

Depending on the method, different critical exponents were extracted. Only in the case of pCUT + MC and SSE QMC calculations [34,247], all three regimes were investigated. By extracting three independent critical exponents, the authors were able to provide the full set of critical exponents for each method individually. In the case of the pCUT + MC approach [34,247], the exponents were computed by (biased) DlogPadé extrapolants of high-order series of the gap (zν), the one-quasiparticle static spectral weight ((2−z−η)ν), and the control-parameter susceptibility (α). In SSE QMC studies [34,247], the exponents were calculated using data collapses of the magnetisation (β/ν and ν) and the order-parameter susceptibility (γ/ν). The DMRG study [28] applied the method of data collapse to the fidelity susceptibility to extract the single exponent ν in the non-trivial long-range regime. Another DMRG study [248] extracted critical exponents by using finite-size scaling for the energy gap (*z*) and the squared order parameter (2β/ν and ν). Using the FRG [21], the exponents ν and *z* were calculated, from which one can additionally calculate the exponent α. In the PI QMC study [33], the finite-temperature criticality of the LRTFIM was investigated for σ=0.5 in the long-range mean field regime and σ=−0.95 in the strong long-range regime. The study includes the extraction of an exponent $\theta_{t}$, which can be related to ν, from the Binder cumulant and classical order-parameter susceptibility via finite-size scaling, as well as the exponent $\gamma \theta_{t}$ from finite-size scaling of the classical order-parameter susceptibility. In Figure 21, the critical field $h_{c}$ and the canonical critical exponents are plotted for the discussed studies.

The two-dimensional case of the square lattice is less studied. The critical values of the transverse field were only calculated by the pCUT + MC [29] and SSE QMC method [32,249]. There is so far no study that has extracted the full set of critical exponents altogether. With pCUT [29], the gap exponents zν were also extracted by DlogPadé extrapolants of the gap series. In the FRG study [21], the critical exponents *z* and ν were calculated, and in the SSE QMC study [32], the critical exponents β and ν were extracted from the data collapses of the magnetisation. These results are summarised in Figure 22. Note that we present the results from pCUT + MC calculations from Ref. [29] with one additional order to the maximal order in the perturbation parameter.

Overall, the different methods all perform well even for relatively small σ. For the pCUT and SSE QMC studies, where all three regimes were simulated, the limiting cases of long-range mean-field criticality and nearest-neighbour criticality were correctly reproduced, which underlines the reliability of the presented results in the intermediate regime. The main challenge that the methods are facing is to sharply resolve the regime boundaries due to rounding effects, which are probably due to the limited length scales in the simulation. Additionally, the boundary to the long-range mean field regime is spoiled by logarithmic corrections to scaling at the upper critical dimension. Even in the case of pCUT + MC, which is a method operating in the thermodynamic limit, there is an intrinsic limit of the length scale due to the finite order of the series. In particular, the rounding makes it hard to verify (or falsify) the claim that the boundary between the short-range and non-trivial long-range regime is shifted $2\to 2-\eta_{\mathrm{SR}}$. For 1D, the boundary is shifted to σ=1.75, which could be resolvable in extensive simulation. However, in 2D, the boundary is only marginally shifted to σ≈1.964, which is probably not resolvable by the reviewed methods. The best boundary resolution is probably given by the SSE QMC study [34]. If one is interested in studying the shift of the boundary, one could, therefore, push the SSE QMC simulations of the one-dimensional chain to larger systems for σ⪅2. Another difficulty for the methods operating on finite systems is the breakdown of the common hyperscaling relation and standard FSS above the upper critical dimension, which affects the processing of data in the long-range mean-field regime for σ<2d/3. The scaling above the upper critical dimension was already outlined in Section 2.3. However, we will recapitulate the most important points in the following Section 6.1.3, not only because this modified scaling was used in several of the above-mentioned methods to extract critical exponents [32,33,34], but also because the LRTFIM constitutes a testing ground for quantum Q-FSS scaling and the modified scaling of the correlation length was confirmed in a study of the LRTFIM [34].

#### 6.1.3. Scaling above the Upper Critical Dimension

As discussed in Section 6.1.1, for $\sigma <2-\eta_{\mathrm{SR}}$, the action describing the QPT and its universality changes from short range to long range. In this long-range regime, the upper critical dimension $d_{\mathrm{uc}}=3\sigma /2$ was found to depend on the value of σ [20,21]. This means that the connectivity of the system increases for decreasing σ and a smaller spatial dimension d<3 is already sufficient to reach or even exceed the upper critical dimension. The sigma below which the model is above its upper critical dimension is $\sigma_{\mathrm{uc}}=2d/3$. In this regime, it is not possible to correctly extract all critical exponents from the standard FSS, and the common hyperscaling relation:(303)$$2-\alpha =\left(d+z\right)\nu$$
becomes invalid as the critical exponents become independent of the dimension *d*. The origin of this discrepancy comes from the effect of dangerous irrelevant variables that one has to take into account in the derivation of scaling above the upper critical dimension. When doing so, one obtains a modified hyperscaling relation: (304)$$2-\alpha =\left(\frac{d}{ϙ}+z\right)\nu$$
with the exponent $ϙ=\max(1,d/d_{\mathrm{uc}})$. This relation can be used for the conversion of different critical exponents [34,35].

Additionally, the standard FSS scaling form of an observable O with power-law singularity $O\left(r,L^{-1}=0\right)∼{\left|r\right|}^{\omega }$ is modified to [34]
(305)$$\begin{array}{cc} O(r,L^{-1}) & =L^{-\omega ϙ/\nu }\Psi \left(L^{ϙ/\nu }r\right)\; \end{array}$$
following the formulation of the Q-FSS for classical [15,16,17,18,19,41] and quantum [34] systems. Finite-size methods that rely on the method of data collapse or other FSS-based techniques have to use this adapted formula to extract all the critical exponents successfully [34]. For a more elaborate description of quantum Q-FSS, please refer to Section 2.3 or to Ref. [34].

On the other side, the LRTFIM can also be used as a testing ground for quantum Q-FSS. One key difference that distinguishes the Q-FSS from standard FSS is that the correlation sector is also affected by the DIV in the case of the Q-FSS. This leads to a modified scaling of the characteristic length scale $\xi_{L}\left(r\right)=L^{ϙ}\Xi \left(L^{ϙ/\nu }r\right)$ with the system size instead of $\xi_{L}\left(r\right)∼L\Xi \left(L^{1/\nu }r\right)$, as it is for standard FSS. The characteristic length scale was measured in the SSE QMC study [34] in the long-range mean-field regime, and the exponent ϙ was extracted by a data collapse with fixed mean field value $\nu =\sigma^{-1}$. The extracted values coming from Ref. [34] are presented in Figure 23 and are clearly in line with the predictions made by the Q-FSS.

### 6.2. Antiferromagnetic Long-Range Transverse-Field Ising Models on Bipartite Lattices

In this section, we review results for the antiferromagnetic LRTFIM on lattices where the coupling structure of the nearest-neighbour couplings forms a bipartite lattice. A lattice—or, more generally, a graph—is called bipartite if the sites can be split into two disjoint sets *A* and *B* so that there are no edges between sites within each of the two sets *A* and *B*. In the case of a lattice, *A* and *B* are called bipartite sublattices. Note that, only in the nearest-neighbour limit on bipartite lattices, there is an exact duality between the ferromagnetic and antiferromagnetic TFIM, and the quantum-critical properties coincide. This duality comes from a sublattice rotation of π along the *x*-axis for the spins in one of the two sublattices. In the nearest-neighbour limit, there is an *x*-polarised symmetric high-field phase and a $Z_{2}$ symmetry-broken low-field phase. The low-field phase is adiabatically connected to the zero-field limit. In the zero-field limit, there are two ground states associated with the states where the spins on the sites in one set of the bipartite sublattices are pointing in one direction while the others point in the other direction. Note that these states can be directly mapped onto the two ferromagnetic ground states by the above-mentioned sublattice rotation. As for the ferromagnetic nearest-neighbour TFIM, the QPT between these two phases is of (d+1)D-Ising universality.

The antiferromagnetic long-range interaction beyond the nearest-neighbours induces a hierarchy of competing interactions. Due to the long-range interactions, there is no longer a bipartite coupling structure as the spins cannot be aligned such that the energy of all interactions is minimised. Nevertheless, there is evidence that the two ground states in the low-field phase of the nearest-neighbour limit are adiabatically connected to the ground states in the low-field phase for all (d+σ)>0 [23,24,25,26,29,31,32,111].

For the all-to-all connected case (d+σ)=0, every state with zero *z*-magnetisation is a ground state of the system [30,40] at h=0. The Hamiltonian (Equation 292) can then be written in terms of total spin $\sigma_{\mathrm{tot}}^{\kappa }=\sum_{i}\sigma_{i}^{\kappa }$ with κ∈{x,z} as
(306)$$\begin{array}{c} H=-h\sigma_{\mathrm{tot}}^{x}+\frac{J}{2}{\left(\sigma_{\mathrm{tot}}^{z}\right)}^{2}-\frac{J}{2}N \end{array}$$
with the total number of spins N→∞ [30,40]. For all finite transverse fields h>0, the ground state of Equation (Equation 306) is directly located in the *x*-polarised phase [40,254].

The natural questions arising for the QPT in the antiferromagnetic LRTFIM on the bipartite lattices are similar to the ferromagnetic LRTFIM: How does the algebraically decaying long-range interaction influence the critical field value of the QPT? Is there a change in the universality class of the QPT depending on the precise values of the decay exponent (d+σ)?

Regarding the critical field values in one-dimensional chains, there are MC-based studies using the SSE QMC (see Section 5) [32] and pCUT+MC (see Section 4) [25,31], as well as the DMRG [24,26,248] and matrix product states [23]. We summarise the results regarding critical field values and selected critical exponents of the studies mentioned above in Figure 24. In two-dimensional systems, there are MC-based studies for the square lattice using SSE QMC (see Section 5) [32] and pCUT+MC (see Section 4) [29] (for selected results, see Figure 25). Variational methods in two dimensions such as projected entangled pair states [255,256] or DMRG calculations [257] have not been applied for antiferromagnetic systems on bipartite lattices. Entanglement-based methods become more challenging from the increased entanglement due to the area law in two dimensions [258]. Further, there are reports about the violation of the area law for one-dimensional systems with long-range interactions [23,24]. On general grounds, one cannot efficiently represent algebraically decaying long-range interactions in these techniques [259,260]. However, there exist DMRG calculations for antiferromagnetic LRTFIMs on non-bipartite lattices [27,54,55,109].

In general, it has been numerically demonstrated that the critical field values decrease monotonically as a function of σ from the nearest-neighbour value at σ=∞ to zero at (d+σ)=0 [23,24,25,26,29,31,32,248]. Regarding the QPT on the antiferromagnetic chain, the established picture is that it is of the (1+1)D-Ising type for all σ≥1.25 [23,24,25,26,31,32,248]. In the regime of ultra-long-range couplings σ≤1.25, recent finite-size DMRG findings by G. Sun [26] and R. Puebla et al. [248] suggest that the (1+1)D-Ising universality even holds for any σ>−1. The only study indicating a change in criticality below the boundary of σ≤1.25 uses matrix product states generalising the time-dependent variational principle (TDVP) [23]. Note that there is no further theoretical motivation for this boundary besides this single numerical study, which does not provide convincing evidence for its existence. The more recent results from Refs. [26,248] challenge these findings and establish the (1+1)D-Ising universality for all σ>−1. The studies using MC-based techniques [25,31,32] cannot extract reliable critical exponents in this regime. In the SSE QMC study [32], the increasing competition of the long-range interactions is the major problem making the numerical simulations in this regime impractical. If the long-range interaction becomes more prominent, this leads to similar algorithmic challenges as in frustrated systems [32,40,261,262]. The autocorrelation times of the SSE QMC algorithm increase as more and more bond operators are present in the operator sequence, and the field operators diffuse only slowly [32]. Nevertheless, following the trend of the results in Ref. [32], the scenario of a single universality class across the entire σ range is plausible. Regarding the pCUT+MC studies [25,31], the main obstacle to extract information about the quantum criticality in the regime σ<1 is the fact that the critical field value is decreasing. Refs. [25,31] use a perturbative expansion around the high-field limit; therefore, if the critical field value increases, then the extrapolation of the gap series becomes increasingly challenging.

**Figure 24 entropy-26-00401-f024:**
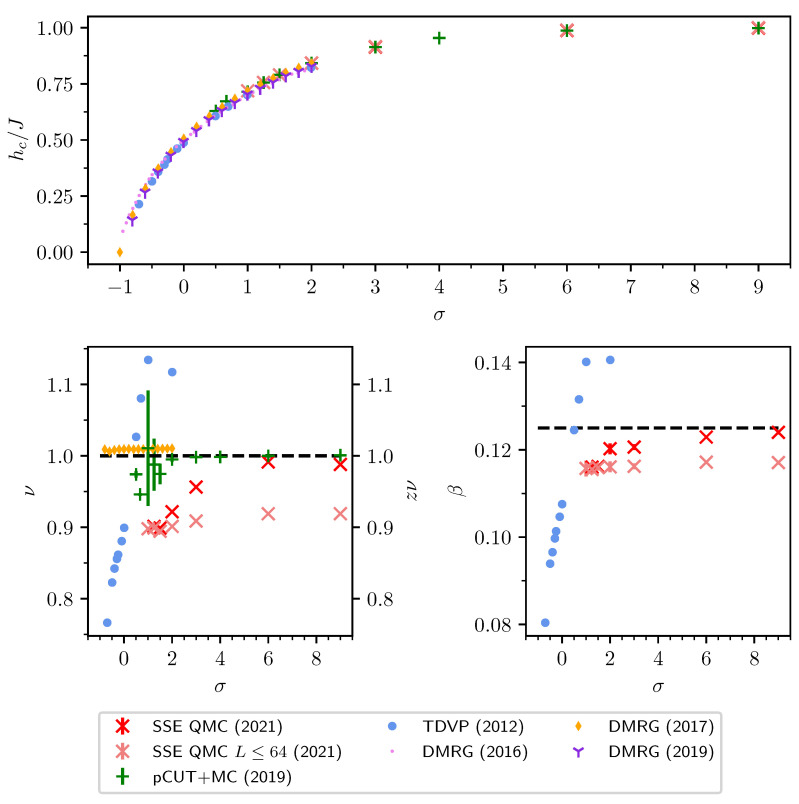
Critical field values and exponents from numerical studies of the antiferromagnetic LRTFIM on the chain. The upper panel displays critical field values $h_{c}/J$; the lower left panel displays values for the critical exponents ν and zν and, the lower right panel, values for the critical exponents β. The data points “SSE QMC (2021)” and “SSE QMC L≤64 (2021)” for $h_{c}/J$, ν, and β are from Refs. [32,249]. The data points “pCUT+MC (2019)” for $h_{c}/J$ and zν are from Ref. [29]. The data points “TDVP (2012)” for $h_{c}/J$, ν, and β originate from Ref. [23]. The exponents ν and β are calculated from the scaling dimensions in Ref. [23] under the assumption of z=1, which is reasonable according to [248]. The data points “DMRG (2016)” for $h_{c}/J$ are from Ref. [24]. The data points “DMRG (2017)” for $h_{c}/J$ and ν originate from Ref. [26]. The data points “DMRG (2019)” for $h_{c}/J$ are from Ref. [248]. The values for the critical exponents of the transition in the nearest-neighbour model ν=1, β=1/8, and z=1 [246,251,252,263] are given by the black dashed lines.

To summarise, there are studies indicating a change from the (1+1)D-Ising criticality on the chain for small σ values [23,24]. These studies also report a possible breakdown on the area law of the entanglement entropy for the ground state in this regime [23,24], which is a vital prerequisite for the methods they implemented. Two more recent studies [26,248] demonstrate no change in the universality class, which is backed by [25,30,31]. For two-dimensional systems, the universality class is demonstrated to be (2+1)D-Ising for σ>0.5 [29,32], and no trend is reported that the universality class should change for smaller values of σ.

**Figure 25 entropy-26-00401-f025:**
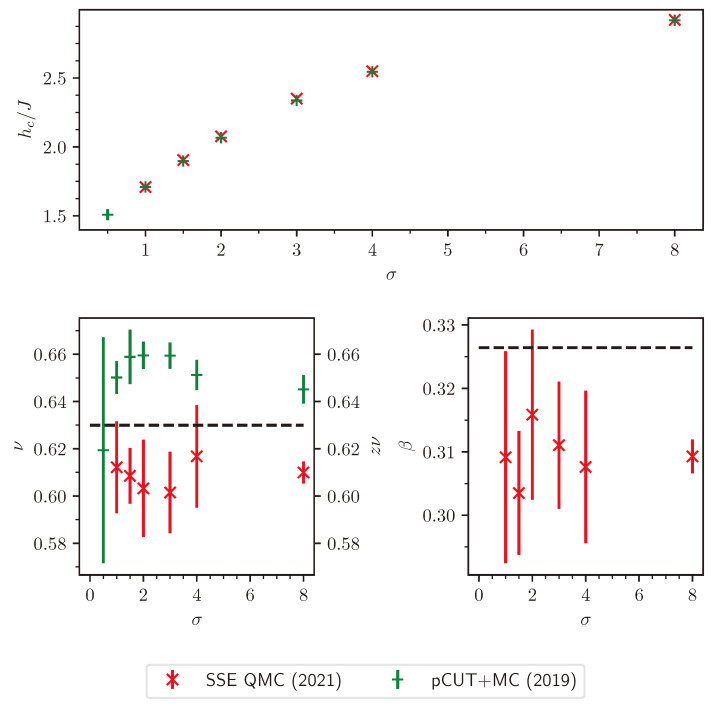
Critical field values and exponents from numerical studies of the antiferromagnetic LRTFIM on the square lattice. The upper panel displays critical field values $h_{c}/J$; the lower left panel displays values for the critical exponents ν and zν and, the lower right panel, values for the critical exponents β. The data points “SSE QMC (2021)” for $h_{c}/J$, ν, and β are from Refs. [32,249]. The data points “pCUT+MC (2019)” for $h_{c}/J$ and zν originate from Ref. [29]. The values for the critical exponents of the transition in the nearest-neighbour model ν=0.629971(4), β=0.326419(3), and z=1 [253] are given by the dashed lines.

In general, a great algorithmic challenge that remains for the MC-based methods with regard to the antiferromagnetic LRTFIM on bipartite lattices is a reliable study of the regime of small values (d+σ). Here, as discussed above, all of the commonly applied methods have a handicap in some way.

### 6.3. Antiferromagnetic Long-Range Transverse-Field Ising Models on Non-Bipartite Lattices

We start with an antiferromagnetic nearest-neighbour interacting TFIM on a non-bipartite lattice (e.g., the sawtooth chain, triangular lattice, Kagome lattice, pyrochlore lattice) at zero field. In the case of non-bipartite lattices, one can always find an odd number of Ising bonds (edges) that form a closed loop. The presence of a loop of odd length means it is impossible to satisfy all antiferromagnetic couplings simultaneously. The phenomenon of not being able to minimise all interactions due to geometrical lattice constraints is called geometrical frustration [264]. A notion for the strength of geometric frustration can be defined using the resulting ground-state degeneracy [265]. A theoretical tool to access the ground-state degeneracy is through Maxwell counting arguments [265,266,267]. For systems with a large degree of frustration, e.g., the antiferromagnetic nearest-neighbour TFIM on the triangular or Kagome lattice, there is a finite residual entropy per site in the thermodynamic limit [168,268,269,270,271].

In general, extensively degenerate ground-state spaces due to frustration pose a great resource for emergent exotic quantum phenomena. Quantum fluctuations introduced, e.g., by a transverse field result in a breakdown of the extensive ground-state degeneracy and the potential emergence of non-trivial ground states [270,271]. In general, it is useful to think about the low-field physics as a degenerate perturbation theory problem on the zero-field ground-state space [30,32,270,271]. We will later review that this is a reasonable framework in order to treat the breakdown of the degenerate subspace due to fluctuations and algebraically decaying long-range interactions on the same footing [30,32,112].

Perturbing extensively degenerate ground-state spaces with fluctuations may result in several distinct scenarios. First, a distinct symmetry-broken order can emerge for infinitesimal perturbations (order-by-disorder) [168,270,271,272,273]. Further, a direct realisation of a symmetry-unbroken phase may occur (disorder-by-disorder). This phase can either be trivial [168,274] or exotic, e.g., quantum spin liquids [99,100,101,275,276].

For the first part of this section, we review the antiferromagnetic LRTFIM on the triangular lattice [27,29,30,32,40,110,277]. In the nearest-neighbour limit of that model, the zero-field case has an extensively degenerate ground state [268,270,271,273]. When adding an infinitesimal transverse field, this ground-state degeneracy breaks down and a $Z_{2}\times Z_{3}$-symmetry-broken clock order emerges from an order-by-disorder mechanism [270,271,273]. The effective Hamiltonian describing the breakdown of the degeneracy in leading order can be expressed as a quantum dimer model [270,271,273]. By investigating the degenerate ground-state space, it was observed that there are local spin configurations in which spins can be flipped without leaving the ground-state space. These configurations, called flippable plaquettes, consist of a spin surrounded by six spins with alternating orientation. It can be easily seen that flipping the spin in the middle of the flippable plaquette results in turning the plaquette by π/3 (see Figure 26).

The effective low-field quantum dimer model [270,271,273], consisting of the plaquette rotation term, is, therefore, the hexagonal-lattice version of the Rokhsar–Kivelson quantum dimer model with t=h and v=0 [270,271,278]: (307)
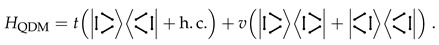
 From this mapping, the nature of the state at h>0 was predicted, as well as the breakdown of the order with a 3D-XY QPT [270,271]. Further numerical studies extended the insights into this system [168,262,273]. The critical field value at which the phase transition between the symmetric *x*-polarised paramagnetic high-field phase and the symmetry-broken gapped clock-ordered low-field phase occurs is at h/J=1.65±0.05 [168,273].

To infer the nature of the full phase diagram including the transverse field and the long-range interaction, the next step is to discuss the zero-field limit of the Ising model with long-range interactions. As the antiferromagnetic long-range interactions introduce a hierarchy of constraints due to the interaction between sites further apart, the long-range interaction breaks the ground-state degeneracy of the nearest-neighbour zero-field as well [30,110,277,279]. It is important to emphasise that the long-range interaction lifts the degeneracy in a different way and stabilises a gapped stripe phase breaking the translational symmetry in a different way than the clock order promoted by the transverse field [30,110,277,279,280]. This plain stripe state is sixfold degenerate by rotations around π/3, as well as spin flips [30,110,277,279]. The spins are aligned in straight lines with alternating *z*-orientation [30,110,277,279]. An important aspect is that these stripe states are gapped and, therefore, stable against finite transverse fields [30,110,277].

Regarding the limit of large transverse fields, pCUT+MC was used to investigate the QPT between the *x*-polarised high-field phase and the clock-ordered phase [29,30]. Similar to the antiferromagnetic models on the bipartite lattices, the critical field value is reduced by the long-range interaction towards smaller field strengths [29,30] (see Figure 27). On the triangular lattice, the universality class of the transition from the high-field to a clock-ordered phase is of the (2+1)D-XY type [270,271,273]. It is reported to remain in this category for all σ values investigated (σ≥1) [29]. For small σ values, the extrapolation of high-field series expansion is inconclusive [29,30]. Infinite DMRG (iDMRG) investigations on triangular lattice cylinders suggest that the clock order vanishes for small values of σ and a direct transition to a low-field stripe phase occurs [27]. A first-order phase transition between the *x*-polarised high-field phase and a low-field stripe phase would be consistent with the incapability of the high-field gap to track the transition [27,29,30,277]. Note that the stripe phase determined from the iDMRG study [27] does not agree with subsequent studies focusing on the low-field ground states of the model [30,32,277,279,280].

Complementary to the high-field analysis and numerical iDMRG studies, approaches in the low-field limit have also been conducted [30,110,277] (see Figure 27). As already mentioned, it was discussed in the literature for some time [30,279,280] and recently demonstrated using a unit cell-based optimisation technique [110] that gapped plain stripes are the zero-field ground state as soon as long-range interactions are present. Therefore, in a phase diagram with a σ and a transverse-field axis, the clock order phase is wedged between the high-field phase from above and the plain stripe phase from below [27,29,30,277] (see Figure 27). The extent of both the stripe phase and the high-field phase increases with stronger long-range interactions [27,29,30,277]. This behaviour was also demonstrated perturbatively from the nearest-neighbour zero-field limit by treating the long-range interaction, as well as the transverse field as perturbations on the degenerate subspace [30,277].

We sketch a qualitative picture of the quantum phase diagram in Figure 28. The precise ground state at small values of σ is still an open research question. We believe that there is some evidence that, after a certain value of σ, only a transition between the high-field phase and the low-field stripes remains, which is expected to be of first order. However, there is, by now, no numerical technique to make reliable statements in this regime. In this context, the high-field series expansions cannot detect a first-order phase transition from the elementary excitation gap. There is an SSE QMC study of the system by S. Humeniuk [40] applying the algorithm discussed in Section 5.1. The direct QMC simulation was used to determine the critical point between the high-field phase and the clock-ordered phase for α=3 [40]. For smaller transverse fields, the author identified a “region dominated by classical ground states [40]”, which are adiabatically connected to the zero-field states. Therefore, there is a qualitatively sketch of a phase diagram in Ref. [40] similar to Figure 28. Nevertheless, the naive application of the SSE QMC approach as discussed in Section 5.1 is not as efficient as for models with interactions that are not competing [40,261,262]. However, there are ideas on how to implement efficient quantum cluster algorithms for frustrated Ising models in a transverse field [261,262], which need to be adjusted for long-range interacting Ising models.

The antiferromagnetic LRTFIM on the triangular lattice is a well-studied example of the interplay between long-range interactions and order-by-disorder. Recently, several more examples arose from the field of Rydberg atom quantum simulators [54,55,109,110,112]. In Rydberg atom simulators, atoms are positioned in a desired configuration using optical tweezers [52] and are laser-driven to a Rydberg state [52]. The Rydberg blockade mechanism leads to an algebraically decaying interaction between Rydberg excitations, which decays with (d+σ)=6 [52]. Using the Matsubara–Matsuda transformation [89], the excitations/non-excitations of a Rydberg atoms are associated with spin degrees of freedom [52]. The density–density interaction transforms into an Ising interaction [110,112]. Rydberg atom quantum simulators are capable of simulating the antiferromagnetic LRTFIM with (d+σ)=6 and a longitudinal field [52,54,55,109]. Order-by-disorder scenarios are described for the Kagome lattice at a filling of f=2/9 Rydberg excitations [54], the Ruby lattice at a filling of f=1/4 Rydberg excitations [55,109], and at vanishing longitudinal field [112]. Regarding these examples, the order-by-disorder was studied for a long-range interaction truncated after the third-nearest neighbours [54,55,109,112]. Considering the remaining long-range interaction as a perturbation to the degenerate ground-state space for most of these mechanisms, a similar scenario as for the triangular lattice is expected [54,55,109,110]. Using the same unit cell-based optimisation method as for the triangular lattice, Koziol et al. [110] have calculated ground states of the zero-field model using the full untruncated long-range interaction. For the Kagome lattice at a filling of f=2/9 Rydberg excitations [54] and the Ruby lattice at a filling of f=1/4 Rydberg excitations [54,55,109], gapped ground states were determined [110], which do not coincide with the order arising from the order-by-disorder mechanism. Interestingly, in all these examples, the zero-field ground state determined by the long-range interaction possess none of the motifs relevant for the leading-order order-by-disorder mechanism [110]. For example, on the triangular lattice, the plain stripe states contain zero flippable plaquettes, which are the relevant motif to lower the energy at a finite transverse field in leading order. Recently, an order-by-disorder mechanism for the antiferromagnetic $J_{1}$-$J_{2}$-$J_{3}$ TFIM on the Ruby lattice was introduced by A. Duft et al. [112]. The remarkable aspect of the mechanism is that adding long-range interactions does stabilise the same order, which is also selected by the quantum fluctuations [112]. A general theory, under which conditions the long-range interactions stabilise the same order as quantum fluctuations or stabilise a completely disjoint order, is yet to be found.

Conclusively, the breakdown of extensively degenerate ground-state spaces due to an interplay of long-range interactions and quantum fluctuations is a highly vibrant and relevant field. Long-range interactions are relevant for a wide range of quantum–optical quantum–simulation platforms, including cold atoms [52,70,72] and ions [74,75,76,77,79,80,81,82,84]. These platforms can be used to realise exotic phases of matter, e.g., quantum spin liquids [54,55,109], glassy behaviour [283], or clock-ordered states [29,30,112,277]. To understand the emergence of these exotic states of matter, simplified models are oftentimes considered [54,55,109,112,283], which truncate the long-range interaction. Therefore, it is imminent to understand the effects of the full long-range interactions on the mechanisms driving the emergence of these exotic phases. At the moment, the commonly used tools to investigate the phase diagrams of these frustrated systems consist of the following:The derivation of effective models [30,54,55,109,112]. These can be used in order to predict exotic emergent phases and the nature of QPTs [30,54,55,109,112].DMRG calculations [27,54,55,109], which are used to obtain a numerical insight into the ground-state phase diagrams.QMC calculations, in particular the SSE, as discussed in Section 5.1, can be used for an unbiased sampling of ground-state properties [40]. In order to omit the slowdown of the algorithm due to the geometric frustration, algorithmic improvements are required [261,262]. It is still an open research question how to set up an algorithm that samples long-range interaction and frustration efficiently.High-order high-field series expansions using a graph decomposition. This is also a very capable tool to track the first QPT coming from the high-field limit. With this method, the critical field value, as well as critical exponents can be determined [29,30,112]. It is also possible to infer information about the phase on the other side of the phase transition by studying the momentum at which the elementary excitation gap is located [29,30,112]. An MC embedding of white graphs can be performed to study the entire algebraically decaying long-range interaction [29,30], while an ordinary graph decomposition can be used for systems with truncated interactions [112].

## 7. Long-Range Transverse-Field XY Chain

In this section, we review results on the XY chain with long-range interactions in a transverse field from Ref. [31]. The Hamiltonian of the long-range transverse-field anisotropic XY model (LRTFAXYM) on a one-dimensional chain is given by
(308)$$H=h\sum_{i}\sigma_{i}^{z}-\frac{J}{4}\sum_{i\neq j}\frac{1}{{\left|i-j\right|}^{1+\sigma }}\left[\left(1+\beta \right)\sigma_{i}^{x}\sigma_{j}^{x}+\left(1-\beta \right)\sigma_{i}^{y}\sigma_{j}^{y}\right]\;,$$
with Pauli matrices $\sigma_{i}^{\kappa }$ and κ∈{x,y,z} describing spin-1/2 degrees of freedom on the *i*-th lattice site of the chain. The transverse-field strength is given by h>0, and the strength of the coupling between sites *i* and *j* is given by $∼{J/|i-j|}^{1+\sigma }$. The coupling is (anti)ferromagnetic for J>0 (J<0). We include a continuous interpolation parameter β∈[0,1] to tune the system from the XY chain with isotropic interactions (β=0) to Ising interactions (β=1). The exponent σ determines the decay of the long-range interaction. The limit of short-range interactions is recovered for σ=∞ and the limit of all-to-all couplings for σ=−1.

Similar to the discussion of the transverse-field Ising model, it is useful to express Equation (Equation 308) in terms of hard-core bosonic operators bi†, $b_{i}^{†}$ using the Matsubara-Matsuda transformation [89]
(309)σiz=1−2bi†bi†σix=bi†+bi†σiy=i(bi†−bi†).

We obtain
(310)H=ϵ0N+∑ibi†bi†−∑i≠jλ41|i−j|1+σ[(1+β)(bi†bj†+bi†bj†)+(1−β)(bi†bj†−bi†bj†)+h.c.]
in units of 2h with $\epsilon_{0}=-1/2$ being the unperturbed ground-state energy per site, *N* the number of sites, and λ=J/(2h) the perturbation parameter associated with the perturbation V (second sum) taking the form $V=T_{-2}+T_{0}+T_{2}$ in the pCUT language, so that a high-order series expansion about the high-field limit can be performed.

In the limit J=0, the ground state is a nondegenerate *z*-polarised state that serves as an unperturbed reference state. Elementary excitations are called local spin flips at an arbitrary site *i*, which can be annihilated (created) by the hard-core bosonic operator $b_{i}^{\left(†\right)}$. Upon increasing λ, these quasiparticles (qps) become spin flips dressed by quantum fluctuations induced by the perturbation V. In the limit of h=0, the system exhibits ferromagnetic or antiferromagnetic magnetic order depending on the sign of *J* in the *x*-direction (*y*-direction) for β>0 (β<0). When tuning the interpolation parameter β from pure Ising to isotropic XY interactions, ordering in the *x*-direction starts to compete with ordering in the *y*-direction.

In order to better appreciate the results for σ<∞, we briefly discuss the quantum phase diagram in Figure 29 of the model (Equation 308) with nearest-neighbour interactions [284,285,286,287,288,289,290,291,292,293]. For any β≠0, the system with nearest-neighbour interactions undergoes a (1+1)D-Ising QPT at J=±h from the paramagnetic symmetric high-field phase to the (anti)ferromagneti- cally ordered low-field phase, where the ground state spontaneously breaks the $Z_{2}$ symmetry of the Hamiltonian [284]. The transition along the β=0 line between the two distinct $Z_{2}$ symmetry-broken phases is of the (Ising)^2^ type, which is conformally and U(1) invariant [284]. These critical lines meet at β=0 and J=±h in multicritical points $M_{1/2}$ with critical exponents z=2 and ν=1/2 [284].

For β=0, the Hamiltonian (Equation 310) becomes particle-conserving as the terms annihilating (creating) two quasiparticles $b_{i}^{\left(†\right)}b_{j}^{\left(†\right)}$ cancel each other, making the isotropic XY Hamiltonian U(1) symmetric. Due to this particle-conserving nature of Equation (Equation 310), there are no quantum fluctuations dressing the symmetric λ=0 ground state for λ>0 and the perturbative treatment of the high-field dispersion becomes exact in first order of the perturbation theory [31]. This distinguished symmetry property of the β=0 case motivates that we review the properties of the isotropic transverse-field XY chain in the ferromagnetic and antiferromagnetic case first in Section 7.1. Subsequently, we discuss the anisotropic XY chain for ferromagnetic interactions in Section 7.2 and for antiferromagnetic interactions in Section 7.3. Here, we show improved results of the ones shown in Ref. [31] since we noticed right after their publication that the Monte Carlo runs were performed with a suboptimal choice of the simulation parameters.

### 7.1. Isotropic Long-Range XY Chain in a Transverse Field

To consider the isotropic XY chain, β=0 is set in the Hamiltonians in Equations (Equation 308) and (Equation 310). The quantum criticality of this model was studied in Ref. [31] in an analytical fashion, by evaluating the ground-state energy and the dispersion of the elementary excitations analytically. Setting β=0, Equation (Equation 310) reads
(311)H=ϵ0N+∑ibi†bi†−∑i≠jλ21|i−j|1+σ[bi†bj†+h.c.]
since the pair creation and annihilation terms $b_{i}^{\left(†\right)}b_{j}^{\left(†\right)}$ cancel, as stated above. The perturbation V acting on the unperturbed *z*-polarised ground state does not introduce any quantum fluctuations. Therefore, the *z*-polarised state becomes an exact eigenstate for arbitrary λ and stays in the ground state until a QPT occurs [31]. Analogously, the one-quasiparticle dispersion is exact in first-order perturbation theory. The one-particle dispersion in the symmetric high-field phase reads [31]
(312)$$\omega \left(k\right)=1-2\lambda \sum_{\delta =1}^{\infty }\frac{\cos(k\delta )}{\delta^{1+\sigma }}\;.$$ The critical value $\lambda_{c}$ can be determined from the dispersion (Equation 312) via the closing of the quasiparticle gap Δ(λ). The critical exponent zν associated with the gap closing can be extracted from the dominant power-law behaviour of the gap near $\lambda_{c}$:(313)$$\begin{array}{c} \Delta \left(\lambda \right)\propto |\lambda -\lambda_{c}{|}^{z\nu }\;. \end{array}$$ Further, it is possible to separate the dynamic *z* and static ν exponent by evaluating the 1qp dispersion ω(k) at the quantum-critical point such that
(314)$$\begin{array}{c} {\omega \left(k\right)|}_{\lambda =\lambda_{c}}\propto {\left|k-k_{c}\right|}^{z}\;, \end{array}$$
which allows the extraction of the critical dynamic exponent *z*. For ferromagnetic XY interactions, the minimum of the dispersion (Equation 312) is located at the momentum k=0. Therefore, the gap series is given by Δ(λ)=1−2λζ(1+σ) and $\lambda_{c}\left(\sigma \right)={\left(2\zeta \left(1+\sigma \right)\right)}^{-1}$. The Riemann-zeta function $\zeta \left(1+\sigma \right):=\sum_{n=1}^{\infty }n^{-(1+\sigma )}$ is convergent for all σ>0 and diverges for σ→0. This mathematical observation coincides with the qualitative behaviour found in ferromagnetic Ising systems discussed in Section 6.1 in the sense that $\lim_{\sigma \to 0}\lambda_{c}\left(\sigma \right)=0$. Since the expression for the gap Δ(λ) is linear in λ, the critical exponent is zν=1, independent of the σ value. Using the expression for the dispersion (Equation 312), the knowledge about $\lambda_{c}$, and the definition of the dynamic critical exponent (Equation 314), Adelhardt et al. [31] evaluated *z* and, therefore, also ν as a function of σ. The results of their calculations are presented in Figure 30. The exponents resulting from the consideration of the dispersion can be categorised into two distinct regimes: For σ>2, the critical exponents of the nearest-neighbour transition are found (z=2 and ν=1/2). For σ<2, the exponent *z* decreases linearly from 2 to 0 and ν increases like 1/z from 1/2 to *∞*. Note that this linear decrease of *z* and the divergence of ν appear similar to the behaviour of the ferromagnetic LRTFIM at small σ values (see Section 6.1) [20,21]. To explain this behaviour, a quantum field theory was proposed in Ref. [31]. The idea was to add a $k^{\sigma }$ term (analogous to [20,21]) to the well-studied bosonic action used for the study of the isotropic short-range transition [113,231,284,294]. The suggested action reads
(315)$$\begin{array}{c} S=\frac{1}{2}\int_{k,\omega }\left(ak^{2}+bk^{\sigma }+ig\omega +r\right)|\psi_{k,\omega }{|}^{2}+u\int_{k,\omega }{\left|\psi_{k,\omega }\right|}^{4} \end{array}$$
with ψ being a complex *c*-number order-parameter field of the transition, a,b>0, and the real constants u,g and *r* [31]. The notation of Equation (Equation 315) is taken from Refs. [31,231]. From power counting, one obtains the critical exponents: (316)$$\begin{array}{cc} z & =\left\{\begin{array}{c} 2\;\mathrm{for}\;\sigma \geq 2\\ \sigma \;\mathrm{for}\;\sigma <2 \end{array}\right. \end{array}$$(317)$$\begin{array}{cc} \nu & =\left\{\begin{array}{c} 1/2\;\mathrm{for}\;\sigma \geq 2\\ 1/\sigma \;\mathrm{for}\;\sigma <2 \end{array}\right. \end{array}$$
from the Gaussian part of the action in Equation (Equation 315) [31]. Following the arguments of Ref. [231], these exponents hold “presumably” [231] for $1\leq d\leq d_{\mathrm{uc}}$ below the upper critical dimension $d_{\mathrm{uc}}=2$. Refs. [231,294] provide arguments that the self-energy vanishes for every order in *u* and the renormalisation of *u* can be performed in all orders via ladder diagrams [31]. The vanishing self-energy is explained by the fact that, in each diagram, every pole in ω lies in the complex upper-half plane. The frequency integral can be deformed into the lower half-plane to give zero [294]. The fact that the free propagator of the field theory is not changed by a self-energy is a manifestation of the absence of fluctuations that do not preserve the particle number [31]. The predictions of the field theory (Equation 315) for *z* in Equation (Equation 316) and ν in Equation (Equation 317) are in perfect agreement with the results of the high-field excitation gap (see Figure 30) [31].

Note that, in the nearest-neighbour case, the low-field ground state does not break the U(1) symmetry of the Hamiltonian due to the Hohenberg–Mermin–Wagner (HMW) theorem, ruling out continuous symmetry breaking [113,130,131,284,295,296]. Long-range interactions are a known mechanism to circumvent the HMW theorem [22,35,297,298,299,300,301,302,303,304,305,306]. With the high-field approach discussed above, it is not possible to determine the nature of the low-field ground state and if a continuous symmetry breaking occurs for sufficiently small decay exponents. We are not aware of studies addressing this topic.

For the remainder of this section, we discuss the antiferromagnetic isotropic XY chain in a transverse field along the same lines as for the ferromagnetic case. The minimum of the dispersion is at momentum k=π, and the gap can be expressed as Δ(λ)=1−2λη(1+σ) with η(1+σ) being the Dirichlet eta function (also known as the alternating ζ function) [31]. The Dirichlet eta function η(1+σ) is convergent for all σ>−1; therefore, the definition of an excitation gap is also possible in the regime of strong long-range interactions (σ≤0). Analogous to the discussion of the Ising interaction in Section 6.2, the extent of the low-field phase is diminishing for decreasing σ values. For σ→−1, the energy gap closes at diverging $\lambda_{c}\to \infty$ [31]. With a similar analysis to the ferromagnetic case, the same critical exponents z=2 and ν=1/2 are found along the entire σ line [31]. This is also analogous to the antiferromagnetic LRTFIM discussed in Section 6.2, where the nearest-neighbour criticality is believed to be the correct universality class for all values of σ.

### 7.2. Ferromagnetic Anisotropic Long-Range XY Chain in a Transverse Field

The fundamental difference between the isotropic β=0 and anisotropic β>0 XY chain in a transverse field is that, for β≠0, the Hamiltonian is no longer U(1)-invariant (particle-conserving); see Equation (Equation 310). Therefore, a perturbative treatment of the anisotropic model with long-range interaction from the high-field limit requires the entire procedure described in Section 4 to perform pCUT+MC calculations. In Ref. [31], this procedure was applied to obtain the critical values $\lambda_{c}$ and critical exponents zν from the gap. We review these results in this section, followed by the results for the antiferromagnetic case in Section 7.3.

Critical values $\lambda_{c}$ and critical exponents zν for several β values as a function of σ are depicted in Figure 31. The overall σ dependence of $\lambda_{c}$ for β>0 is similar to the β=0 case (see the inset of Figure 31). However, the low-field ferromagnetic phase becomes more stable for increasing β values [31]. Note that the perturbative parameter is λ=J/2h; therefore, a QPT at smaller λ values means a larger extent of the phase in h/J. In Ref. [307], the QPT of the anisotropic LRTFAXYM was studied with exact diagonalisation, and the critical point was determined for β=1/2, which coincides perfectly with the value from pCUT+MC (see Figure 31). For all β>0, Adelhardt et al. [31] identified the same three regimes for the universality of the QPT as for the ferromagnetic LRTFIM. The gap exponent behaves analogously to the LRTFIM (β=1), as depicted in Figure 31. Solely the β=0 exponent behaves in a special manner, being zν=1 for all σ values, in full agreement with the existence of a multicritical point in the nearest-neighbour model. For β>0, the first domain identified in Ref. [31] is the long-range mean-field regime for σ<2/3 with zν=1/2. For large values of σ, the gap exponent is zν=1 and the transition belongs to the 2D-Ising universality class [31]. Third, in the intermediate regime, the critical exponents vary continuously between the two values in a non-trivial fashion [31].

The criticality regimes for β>0 can be understood using the same field-theoretical argument (see Section 6.1.1) as for the ferromagnetic LRTFIM, since the underlying symmetry that is spontaneously broken is, in both cases, the same $Z_{2}$ symmetry of the Hamiltonian. Therefore, at σ=2/3, the upper critical dimension as a function of σ becomes smaller than one and the transition enters the regime above the upper critical dimension. The boundary between nearest-neighbour criticality and the intermediate long-range non-trivial regime is expected to be at $\sigma =2-\eta_{\mathrm{SR}}$ [21]. The data from Ref. [31], presented in Figure 31, are in good agreement with these three regimes. Nevertheless, the changes between the regimes cannot be determined accurately by the pCUT+MC approach. In Figure 31, we can observe that the interfaces between the nearest-neighbour and intermediate long-range regime become more pronounced for increasing β values, while at the interface between the long-range mean-field and intermediate regime, the deviation is the same for all values of β>0. The reason for the distinct deviations at the nearest-neighbour interface may be due to the finite nature of the perturbative series or may also be indicative of different corrections to scaling. At σ=2/3, where the system dimension equals the upper critical dimension of the transition, there are multiplicative logarithmic corrections to the critical exponents [125,135,308,309,310]. From the biased DlogPadé extrapolations at σ=2/3, Adelhardt et al. [31] determined these corrections to scaling at the upper critical dimension with a qualitatively good agreement for different β values.

To summarise the findings, we provide the following: Similar to the ferromagnetic LRTFIM, the ferromagnetic LRTFAXYM is a paradigmatic toy model extending the LRTFIM. With the pCUT+MC method described in Section 4, the critical exponents as a function of the decay exponent σ can be determined as demonstrated in Ref. [31]. The LRTFAXYM displays three distinct regimes of quantum criticality including a mean-field regime above the upper critical dimension for σ<2/3 for all β>0. The sensitivity of the series expansion method suffices to even study corrections to scaling at the upper critical dimension. The limiting case β=0 is especially interesting, as in the high-field limit, the ground state contains no quantum fluctuations, the dispersion is exactly solvable in first-order perturbation theory, the underlying QFT is different from the case β>0, and two instead of three critical regimes were found. We would expect that the setup of an SSE QMC algorithm following the directed loop idea [215,217,311] is conceptually possible for the model.

### 7.3. Antiferromagnetic Anisotropic Long-Range XY Chain in a Transverse Field

For frustrated antiferromagnetic interactions, we begin our discussion with the Ising case at β=1. Here, we have already discussed in Section 6.2 that, for the antiferromagnetic LRTFIM, there is strong numerical evidence that it remains in the 2D-Ising universality class for all σ>−1. It was also demonstrated that it is possible to determine critical gap exponents up to σ≈0.5 with the pCUT+MC method [29,31,35].

Following the same procedure as for the ferromagnetic model, the critical values $\lambda_{c}$ and the critical gap exponents zν were studied in Ref. [31], and we summarise these results in Figure 32. The critical values $\lambda_{c}$ shift towards larger values when decreasing σ, eventually diverging for σ→−1. It is possible to reliably determine the critical exponents for the anisotropic XY chain until σ≈0.5, but it becomes increasingly challenging to extract the exponents for smaller σ due to the shifting of $\lambda_{c}$ [31]. For β≠0, the critical values $\lambda_{c}$ behave qualitatively in a similar way to β=0. However, the larger β, the more the high-field phase is stabilised (see the inset of Figure 32). As a consequence, we observe more reliable exponent estimates in the regime σ<0.5 for β values closer to zero. In general, the observations in Ref. [31] are in line with a constant gap exponent zν=1 within the limitations of the method.

This promotes the scenario that the nearest-neighbour 2D-Ising universality class remains the correct universality class for all considered σ values and β>0. For β=0, the critical exponents zν=1 and critical values $\lambda_{c}={\left(2\eta \left(1+\sigma \right)\right)}^{-1}$ were determined analytically in Section 7.1, and it was shown that the universality class of the nearest-neighbour isotropic chain remains for all σ>−1 [31]. A recent study [312] investigated the multicritical point using exact diagonalisation for comparably small system sizes. The critical points for β=1/2 and β=1/5 agree well with the exact limit and the pCUT+MC values. The study further confirms 2D-Ising universality up to σ>−0.4; however, in contrast to Ref. [31], signatures of a different crossover criticality at σ=−0.4 are claimed.

## 8. Long-Range Heisenberg Models

We review the results obtained from SSE and pCUT+MC calculations for Heisenberg models with long-range interactions. The starting point is the nearest-neighbour Heisenberg model, which can be written as
(318)$$H=\sum_{\langle i,j\rangle }{\vec{S}}_{i}{\vec{S}}_{j}=\sum_{\langle i,j\rangle }\left(S_{i}^{x}S_{j}^{x}+S_{i}^{y}S_{j}^{y}+S_{i}^{z}S_{j}^{z}\right)$$
with the spin-1/2 operators $S^{\kappa }=\sigma^{\kappa }/2$ and κ∈{x,y,z}. Instead of a transverse field that induces quantum fluctuations in the (anti)ferromagnetic phase of the (LR)TFIM or (LR)TFAXYM, a magnetic field would immediately break the SU(2)-symmetry of the Heisenberg Hamiltonian. Here, we introduce a dimerisation limit. We consider two layers of lattices with Heisenberg spins on each site stacked on top of each other, where each spin couples to its nearest-neighbour. The nearest-neighbour coupling between the layers gives rise to interlayer dimers, also referred to as rungs. In one dimension, the layers are just two chains, resulting in a ladder system, while in two dimensions, we consider two square lattice layers, resulting in a square lattice bilayer model. Generalising this model for long-range interactions gives rise to intralayer, as well interlayer long-range couplings. The generic antiferromagnetic Hamiltonian is then given by
(319)$$H=J_{⊥}\sum_{i}{\vec{S}}_{i,1}{\vec{S}}_{i,2}-\frac{1}{2}\sum_{i\neq j}\left[J_{\left|\right|}\left(i-j\right)\underset{\mathrm{intralayer}}{\underset{︸}{\left({\vec{S}}_{i,1}{\vec{S}}_{j,1}+{\vec{S}}_{i,2}{\vec{S}}_{j,2}\right)}}+J_{\times }\left(i-j\right)\underset{\mathrm{interlayer}}{\underset{︸}{\left({\vec{S}}_{i,1}{\vec{S}}_{j,2}+{\vec{S}}_{i,2}{\vec{S}}_{j,1}\right)}}\right],$$
where the first term couples the nearest-neighbour spins between the layers with coupling strength $J_{⊥}>0$, the second term describes the long-range intralayer coupling with $J_{\left|\right|}$, and the third term the long-range interlayer coupling of Heisenberg spins with $J_{\times }$. The long-range interaction is given by
(320)$$J_{\mu ,\nu }\left(i-j\right)=J\frac{{\left(-1\right)}^{{\left||{\vec{r}}_{i,\mu }-{\vec{r}}_{j,\nu }|\right|}_{1}}}{|{\vec{r}}_{i,\mu }-{\vec{r}}_{j,\nu }{|}^{d+\sigma }},$$
where *J* is the coupling constant, *i* and *j* are the rung dimer indices, and μ and ν are the indices of the respective legs (layers) of the ladder (bilayer) model. In Equation (Equation 319), we use the notation $J_{\left|\right|}\left(i-j\right)=J_{1,1}\left(i-j\right)=J_{2,2}\left(i-j\right)$ and $J_{\times }\left(i-j\right)=J_{1,2}\left(i-j\right)=J_{2,1}\left(i-j\right)$. The long-range interaction introduced is similar to the one introduced in the sections before, however with the difference that the alternating sign in the numerator gives rise to staggered, non-frustrating antiferromagnetic long-range interactions, as we chose J>0. The exponent of the numerator is given by the one-norm ${\left||\cdot |\right|}_{1}$, while the usual geometric distance in the denominator given by |·| can be identified as the two-norm $\left|\cdot \right|={\left||\cdot |\right|}_{2}$. When increasing the overall interaction strength *J* of intralayer and interlayer coupling and, likewise, making the long-range decay exponent σ smaller, the Heisenberg spins will anti-align at odd distances and align at even distances, inducing a Néel-ordered antiferromagnetic phase. For σ≤0, the system becomes superextensive (analogous to the ferromagnetic LRTFIM in Section 6.1), and for σ→∞, we recover the antiferromagnetic short-range ladder or bilayer model.

In order to employ the pCUT+MC method, we need an exactly soluble limit about which we can perform the series expansion. We introduce the perturbation parameter $\lambda =J/J_{⊥}$ rescaling the Hamiltonian by $J_{⊥}$ and identify the first term of Equation (Equation 319) as the unperturbed part $H_{0}$, which corresponds to a sum over Heisenberg dimers that can be easily diagonalised. For a single dimer, the lowest lying state with total spin S=0 is the antisymmetric combination of S=1/2 spins:(321)$$|s\rangle =\;\frac{1}{\sqrt{2}}\left(|↑↓\rangle -|↓↑\rangle \right),$$
also called a rung singlet in our case with the associated energy $\epsilon_{0}=-3/4$. The total spin S=1 states:(322)$$\begin{array}{c} \begin{array}{cccc} |t_{-}\rangle & =|↓↓\rangle ,\;|t_{0}\rangle & =\frac{1}{\sqrt{2}}\left(|↑↓\rangle +|↓↑\rangle \right),\;|t_{+}\rangle & =|↑↑\rangle \end{array} \end{array}$$
are three-fold degenerate with the energy 1/4 and are called triplets. Alternatively, we can use an SU(2)-symmetric basis, where the singlet and triplet states read
(323)$$\begin{array}{c} |s\rangle =\;\frac{1}{\sqrt{2}}\left(|↑↓\rangle -|↓↑\rangle \right),\\ \begin{array}{cccc} |t_{x}\rangle & =-\frac{1}{\sqrt{2}}\left(|↑↑\rangle -|↓↓\rangle \right),\;|t_{y}\rangle & =\frac{i}{\sqrt{2}}\left(|↑↑\rangle +|↓↓\rangle \right),\;|t_{z}\rangle & =\frac{1}{\sqrt{2}}\left(|↑↓\rangle +|↓↑\rangle \right). \end{array} \end{array}$$ There is also a convenient mapping from the spin operators to bosonic, SU(2)-symmetric operators creating and annihilating these states introduced in Ref. [313], which can be readily adapted to hard-core bosonic operators for triplet excitations [314]. We use the mapping
(324)S→i,1α=12ti,α†+ti,α†−iϵα,β,γti,β†ti,γ†,S→i,2α=12ti,α†+ti,α†+iϵα,β,γti,β†ti,γ†,
where $\epsilon_{\alpha ,\beta ,\gamma }$ is the Levi-Civita symbol, and insert these expressions into Equation (Equation 319). After some straightforward manipulations, we obtain
(325)H=ϵ0NR+∑i,αti,α†ti,α†+14∑i≠jλ||(i−j)ti,α†tj,α†+ti,α†tj,α†−ti,α†tj,α†ti,β†tj,β†+ti,α†tj,β†ti,β†tj,α†+h.c.+14∑i≠jλ×(i−j)−ti,α†tj,α†−ti,α†tj,α†−ti,α†tj,α†ti,β†tj,β†+ti,α†tj,β†ti,β†tj,α†+h.c.,
where $N_{R}$ is the number of rungs, $\lambda_{\left|\right|}\left(i-j\right)=J_{\left|\right|}\left(i-j\right)/J_{⊥}$, and $\lambda_{\times }\left(i-j\right)=J_{\times }\left(i-j\right)/J_{⊥}$. The second and third line constitute the perturbation $V=T_{-2}+T_{0}+T_{2}$ associated with the physical perturbation parameter λ. The $T_{1}$ and $T_{-1}$ operators are absent, as terms contributing to these operators cancel out due to the reflection symmetry about the centre of the rung dimers.

For λ=0, the ground state is given by a trivial product state of rung singlets, and for small, but finite λ, the ground state is still adiabatically connected to this product state. We refer to this adiabatically connected ground state as the rung-singlet ground state. Elementary excitations in this phase are called triplons [315], corresponding to dressed triplet excitations. For large λ≫0, we expect the non-frustrating antiferromagnetic long-range interaction to induce an antiferromagnetic phase giving rise to a quantum phase transition (QPT) between these two phases. In contrast to the transverse-field Ising model and the anisotropic XY model, where the ground state in the ordered phase spontaneously breaks the discrete $Z_{2}$ symmetry of the Hamiltonian, the Néel-ordered antiferromagnetic ground state for strong long-range couplings must break the continuous SU(2) symmetry of the Heisenberg Hamiltonian. Related to this observation, we present in the following two prominent theorems of great importance that apply to Hamiltonians with continuous symmetries:

First, there is the Hohenberg–Mermin–Wagner (HMW) theorem [130,131,295], which rules out the spontaneous breaking of continuous symmetries in one- and two dimensional systems for T>0. From the quantum–classical correspondence (see Ref. [113]), we can infer that, for one-dimensional systems, a QPT at T=0 breaking such symmetry should be ruled out as well. Indeed, it was shown rigorously by Pitaevskii and Stringari in 1991 [296] that it is prohibited in one dimension for T=0 as well. However, there is the restriction that the interaction must be sufficiently short-ranged, i.e., the condition σ>2 for T>0 must hold for the long-range decay exponent. A stronger condition was given by Ref. [316] with σ≥d, where *d* is the spatial dimension of the system. For the quantum case, it was shown in Ref. [317] for a staggered antiferromagnetic long-range Heisenberg chain (d=1) that long-range order is absent for all σ>2.

Second, Goldstone’s theorem [318,319,320] applies. The theorem states that, when spontaneous breaking of a continuous symmetry occurs, it gives rise to massless excitations, also known as Nambu–Goldstone modes [318,319,320]. For instance, magnons that are quantised spin–wave excitations are a manifestation of such gapless Nambu–Goldstone modes inside the ordered phase of Heisenberg systems. The recent findings in Ref. [321] actually go beyond the conventional scenario of Goldstone modes due to long-range interactions. The authors identified three regimes depending on the decay exponent σ: for ferromagnetic (antiferromagnetic) interactions, they found standard Goldstone modes for σ≥2 (σ≥0) and an anomalous Goldstone regime with $\omega ∼{\left|k\right|}^{s}$ and s<2 (s<1) for σ<2. Interestingly, a third regime for strong long-range interactions σ≤0 (σ≤−2) was found where the Goldstone modes become gapped via a generalised Higgs mechanism [321]. A recent large-scale QMC study [38] investigating the dynamic properties of the non-frustrating staggered antiferromagnetic square lattice Heisenberg model confirmed these three scenarios and found evidence that the Higgs regime already occurs in the regime σ≤0.2 when the Hamiltonian is still extensive [38]. Let us mention at this point that previous results from linear spin–wave theory [22,322] already indicated the existence of a sublinear dispersion in the staggered antiferromagnetic long-range Heisenberg chain a decade earlier. Remarkably, the existence of gapped Goldstone modes in the Heisenberg model with long-range interactions was already pointed out in the 1960s by Refs. [323,324].

The HMW theorem, as well as Nambu–Goldstone modes are a distinguishing feature of quantum systems with continuous symmetries like Heisenberg antiferromagnets. The quantum field theory describing dimerised Heisenberg antiferromagnets [113] is given by the action
(326)$$S_{\phi }=\int d^{d}x\int_{0}^{\beta }d\tau \;\left[\left\{g{\left(\partial_{\tau }\phi \left(\tau ,x\right)\right)}^{2}+{\left(\nabla_{x}\phi \left(\tau ,x\right)\right)}^{2}+r\phi {\left(\tau ,x\right)}^{2}\right\}+u\phi {\left(\tau ,x\right)}^{4}\right]\;,$$
which is the same *n*-component $\phi^{4}$-theory as the one describing the transverse-field Ising model. Here, the order-parameter field ϕ(τ,x) is now a 3-component field instead of a 1-component one. We can readily include long-range interactions by adding the term
(327)$$\int d^{d}x\;\int d^{d}y\;\int d\tau \;a\frac{\phi (\tau ,x)\phi (\tau ,y)}{{\left|x-y\right|}^{d+\sigma }}$$
to the action (Equation 326). For a more detailed discussion of the short-range $\phi^{4}$ action, we refer to Appendix A and to Section 6.1.1, where the implications of long-range interactions were already discussed for n=1. The classical equivalent of the n=3 action can describe the finite temperature phase transition of the ferromagnetic Heisenberg model [1,36,113,325]. Interestingly, the action of zero-temperature Heisenberg ferromagnets is not given by the above action [113]. In fact, the quantum field theory describes a class of dimerised Heisenberg antiferromagnets, which can be appropriately described by Equation (Equation 326) since their low-energy physics can be mapped onto the n=3 quantum rotor model because pairwise antiferromagnetically coupled Heisenberg spins are an effective low-energy representation of quantum rotors [113]. We can see this correspondence by looking at the Hamiltonian of the O(3) quantum rotor model:(328)$$H=H_{\mathrm{kin}}+V=\frac{K}{2}\sum_{i}L_{i}^{2}-J\sum_{\langle i,j\rangle }n_{i}n_{j}\;,$$
where the first part $H_{\mathrm{kin}}=K/2\sum_{i}L_{i}^{2}$ with K>0 and L the angular momentum operator gives the kinetic energy of the rotors. The second part $V=-J\sum_{\langle i,j\rangle }n_{i}n_{i}$ with J>0 is the interaction between the three-component quantum rotors inducing parallel ordering of the rotors [113]. The physical picture is that the kinetic energy is minimised when the orientation of the rotors is maximally uncertain and the energy of the interaction term is minimised when the rotors are aligned [113]. The eigenvalues of the kinetic term of a single quantum rotor are given by $E_{\mathrm{kin}}\left(l\right)=K/2\;l\left(l+1\right)$ with l∈N corresponding to a (2l+1)-fold degenerate state. Now, given two spins *S* interacting antiferromagnetically with coupling strength $J_{⊥}$ forming a dimer, the total spin is $0\leq S_{\mathrm{tot}}\leq 2S$ and the eigenenergies are given by $E_{\mathrm{dimer}}\left(S_{\mathrm{tot}}\right)=J_{⊥}/2\;\left(S_{\mathrm{tot}}\left(S_{\mathrm{tot}}+1\right)-2S\left(S+1\right)\right)$ with a $\left(2S_{\mathrm{tot}}+1\right)$-fold degeneracy. This gives a one-to-one correspondence to the kinetic energy of a rotor, but with an upper bound for the energy. Thus, this mapping is only valid when considering the low-energy properties of the models [113] and holds certainly for large K/J. We just introduced a powerful mapping from the low-energy properties of the antiferromagnetic Heisenberg systems like ladder and bilayer models forming Heisenberg dimers on the rungs (see Equation (Equation 319)) to an O(3) quantum rotor model, which is described by the action (Equation 326). It turns out that this mapping is not only valid for large *K*/*J*, but also in the ordered phase and at the quantum critical point [113]. To include algebraically decaying long-range interactions, the term (Equation 327) must be added to the action. The implications of long-range interactions for an *n*-component order-parameter field in the $\phi^{4}$ field theory was already extensively discussed in the 1970s [1,2,3] for the classical action, but it took until 2001 for the quantum analogue to be studied by Ref. [20]. More studies followed by N. Defenu et al. employing functional RG approaches [21,244,326].

The predictions for long-range Heisenberg models are in large part the same as for the LRTFIM. We expect long-range mean-field behaviour with a Gaussian fixed point for $\sigma \leq \sigma_{\mathrm{uc}}$, a regime of nearest-neighbour criticality for $\sigma \geq \sigma_{*}$, and a non-trivial regime with continuously varying critical exponents in between [20,21,244,326]. For d=1, however, there is the major difference that a QPT associated with the spontaneous breaking of continuous symmetries is ruled out by the HMW theorem [130,131,295,296], at least for σ>2 [21,244,317,326]. The boundary is then called lower critical exponent $\sigma_{\mathrm{lc}}$. For the Heisenberg model, the previously discussed action (Equation 326) with long-range interactions (Equation 327) holds, which together with the HMW theorem rules out the breaking of the order-parameter field for sufficiently short-range interactions and, therefore, a QPT in the regime $\sigma >\sigma_{\mathrm{lc}}$. For $\sigma \leq \sigma_{\mathrm{lc}}$, a QPT breaking the continuous symmetry is predicted [20,21,244,326]. Indeed, it has been long confirmed by several numerical studies [22,297,298,299,300,301,302,303,304,305,306,327] that the HMW theorem can be circumvented if the long-range interaction is sufficiently strong. Notably, in recent experiments with trapped ions [328] and Rydberg atoms [329] continuous symmetry breaking was realized in one and two dimensions which would be prohibited in the abscence of long-range interactions.

So far, most of the studies considered only Heisenberg Hamiltonians with long-range interactions in one-dimensional systems [22,35,297,298,299,300,301,302,303,304,305,306,327], however often considering various modifications to the Hamiltonian. Some studies considered a one-dimensional Heisenberg chain with non-frustrating (staggered) antiferromagnetic long-range interactions [22,300,305], while others included an anisotropy along the *z*-components resulting in the XXZ-model [302,303,327]. An interesting example with frustrating long-range interactions is the exactly solvable Haldane–Shastry model [330,331] with $∼r^{-2}$, which is known to show quasi-long-range order (QLRO) just like the conventional Heisenberg chain, but with vanishing logarithmic corrections [332]. For the dimerisation transition between the QLRO and a valence bond solid (VBS), the frustrated long-range interaction seems to play only a minor role compared to the $J_{1}$-$J_{2}$ model [213]. Based on this, in Refs. [213,297,298,299,333], a model combining the frustrated $J_{1}$-$J_{2}$ model with additional non-frustrating staggered long-range interaction was investigated. The initial intention was to realise a one-dimensional analogue of deconfined criticality [334,335,336,337] between a Néel-order phase and a VBS. As the transition was found to be of first order, the authors in Ref. [338] later turned to the one-dimensional long-range *J*-*Q* model, indeed finding evidence for a continuous deconfined QPT. Interestingly, a very recent paper [327] also predicts a deconfined QPT in the antiferromagnetic long-range XXZ chain. There are also studies considering larger spin S=1. In Ref. [304], a Heisenberg chain with single-ion anisotropy was studied, while in Ref. [301], the XXZ chain was under scrutiny. In both cases, a rich phase diagram was found with intriguing quantum critical properties. For instance, continuous symmetry breaking due to long-range interactions gives rise to a new, possibly exotic, tricritical point in the XXZ chain with no analogue in short-range one-dimensional spin systems [301].

In Section 6 and Section 7, we reviewed the results of various numerical studies investigating the quantum-critical properties of the LRTFM [23,24,25,26,28,29,30,32,33,34] and the LRTFAXYM [31] with the focus on extracting the critical exponent as a function of the long-range decay σ. In comparison to the LRTFIM, for long-range Heisenberg models, just recently, efforts have been made to extract the critical exponents associated with the QPT as a function of the long-range decay exponent [22,35,37,206]. Reference [22], studying a one-dimensional long-range Heisenberg chain, is one notable exception in the sense that it was ahead of its time, performing large-scale SSE QMC simulations for long-range Heisenberg systems already in 2005, way before the rapid progress in implementing quantum simulators in quantum optics reignited the interest in long-range interacting systems.

In the following, we will first review the results from Refs. [37,206,339] for the long-range square-lattice Heisenberg bilayer model in Section 8.1. Then, we proceed with the discussion of the results from Ref. [35] for long-range Heisenberg ladders in Section 8.2. Finally, we conclude this section with the discussion of the long-range Heisenberg chain from Ref. [22] in Section 8.3.

### 8.1. Staggered Antiferromagnetic Long-Range Heisenberg Square Lattice Bilayer Model

In this section, we consider two square lattices stacked directly on top of each other, where the Heisenberg spins on each lattice site interact with their nearest neighbours. When the two spins on top of each other form a rung dimer interacting with coupling strength $J_{⊥}$, the Hamiltonian describing the bilayer system is given by Equation (Equation 319). Here, we neglect the long-range interlayer interactions, which results in the Hamiltonian
(329)$$H=J_{⊥}\sum_{i}{\vec{S}}_{i,1}{\vec{S}}_{i,2}-\frac{1}{2}\sum_{i\neq j}\left[J_{\left|\right|}\left(i-j\right)\left({\vec{S}}_{i,1}{\vec{S}}_{j,1}+{\vec{S}}_{i,2}{\vec{S}}_{j,2}\right)\right].$$ We consider staggered non-frustrating long-range interactions along the layers $J_{\left|\right|}\left(i-j\right)$ of the form (Equation 320). The short-range model (σ→∞) was the subject of several studies from the 1990s onward [162,183,184,340,341,342,343,344,345,346] investigating its quantum-critical properties. For a critical coupling ratio $\lambda =J_{/}J_{⊥}$, the system undergoes a QPT breaking the continuous SU(2) symmetry of the Hamiltonian from a ground state adiabatically connected to the product singlet state towards an antiferromagnetic Néel ground state with gapless magnon excitations (Nambu–Goldstone modes). For long-range interactions, we expect that these Goldstone modes are altered [38,321] and that the criticality of the system changes as a function of the decay exponent σ. We expect short-range criticality until $\sigma \geq \sigma_{*}=2-\eta_{\mathrm{SR}}$, a non-trivial regime of continuously varying criticality in $\sigma_{\mathrm{uc}}\leq \sigma <\sigma_{*}$ with $\sigma_{\mathrm{uc}}=4/3$, and a long-range mean-field regime for $\sigma <\sigma_{\mathrm{uc}}$ [20,21,244,326].

Until recently, to the best of our knowledge, there was a complete absence of numerical confirmation of this scenario. The first data for the critical point and exponents as a function of σ were published by Song et al. [37] using the SSE QMC algorithm and is now supplemented by pCUT+MC data from Adelhardt and Schmidt [206,339].

In Figure 33, the critical values from Refs. [37,206,339] are plotted together with the results from functional renormalisation group (FRG) calculations from Ref. [21] for the O(3) quantum rotor model. The critical points $\lambda_{c}$ for both methods are in excellent agreement over the entire σ-range. However, the critical exponents cannot be directly compared since the FSS of the magnetisation curves from SSE QMC simulations give β and ν, while pCUT+MC calculations for the gap and spectral weight give zν and (2−z−η)z. While the QMC data show relatively large error bars in the long-range mean-field regime, the pCUT approach overestimates the critical exponents in the short-range regime. For the two-dimensional long-range Heisenberg bilayer, the deviation is larger than for the LRTFIM in two dimensions. This is no surprise, as a previous series expansion for the nearest-neighbour square-lattice Heisenberg bilayer model showed similar deviations [347]. Also, the critical exponents from QMC are better at capturing the boundaries of the long-range mean-field and the short-range regime. In general, both approaches show good agreement with the FRG results from Ref. [21] for the non-trivial intermediate regime (which apparently has its own shortcomings at the boundary to the short-range regime).

In conclusion, we summarised the results from SSE QMC and pCUT+MC studies for the square-lattice Heisenberg bilayer model with staggered antiferromagnetic interactions. Both approaches are in good agreement and confirm the three critical regimes [1,2,3,20,21,244,326] from the two-dimensional O(3) quantum rotor model within their limitations.

### 8.2. Staggered Antiferromagnetic Long-Range Heisenberg Ladder Models

After the discussion of the square-lattice bilayer model, we can imagine the Heisenberg ladder models as effectively reducing the dimension of the square-lattice bilayer model by one now considering two linear chains coupled by rung interactions. The long-range ladder Hamiltonian then reads
(330)$$H=J_{⊥}\sum_{i}{\vec{S}}_{i,1}{\vec{S}}_{i,2}-\frac{1}{2}\sum_{i\neq j}\left[J_{\left|\right|}\left(i-j\right)\left({\vec{S}}_{i,1}{\vec{S}}_{j,1}+{\vec{S}}_{i,2}{\vec{S}}_{j,2}\right)+J_{\times }\left(i-j\right)\left({\vec{S}}_{i,1}{\vec{S}}_{j,2}+{\vec{S}}_{i,2}{\vec{S}}_{j,1}\right)\right],$$
with long-range coupling $J_{\left|\right|}\left(i-j\right)$ along the legs of the ladder and $J_{\times }\left(i-j\right)$ between spins of different legs. Recalling the long-range interactions of Equation (Equation 320) and the definitions of $J_{\left|\right|}\left(i-j\right)$ and $J_{\times }\left(i-j\right)$, thereafter, we define two distinct Hamiltonians $H_{\left|\right|}{=H|}_{J_{\times =0}}$ and $H_{⋈}=H$, where the first one includes long-range interactions only along the legs, while the second one is the original Hamiltonian including long-range interactions both along and in between the legs.

A sketch of these ladders is provided in Figure 34. As for the bilayer model, for small $J/J_{⊥}$ (cf. Equation (Equation 320)), the ground state is adiabatically connected to the product state of rung singlets (rung-singlet ground state), while for strong coupling ratios, the long-range interactions want to induce an antiferromagnetic ground state. However, in contrast to the bilayer model in Section 8.1, there is no QPT for the nearest-neighbour Heisenberg ladder [350,351,352], due to the HMW theorem ruling out continuous symmetry breaking for one-dimensional quantum models [296]. Note that the HMW theorem only rules out a QPT with continuous symmetry breaking and not a QPT in general. For instance, the nearest-neighbour isotropic XY model in a transverse field exhibits a QPT without breaking the U(1) symmetry [284,285,286,287,288,289,290,291,292,293], while in the nearest-neighbour Heisenberg ladder, there is no QPT at all [350,351,352]. In fact, a QPT was ruled out until σ≥2 in another one-dimensional model, namely the staggered antiferromagnetic long-range Heisenberg chain in Ref. [317]. As there is a one-to-one correspondence between antiferromagnetic Heisenberg ladders and the low-energy properties of the one-dimensional O(3) quantum rotor model [113], we can expect a QPT predicted by the quantum field theory given by the action (Equation 326) with Equation (Equation 327) for $\sigma \leq \sigma_{\mathrm{lc}}=2-\eta_{\mathrm{sr}}$ with $\eta_{\mathrm{SR}}=0$ [20,21,244]. The existence of a lower critical decay exponent $\sigma_{\mathrm{lc}}$ has the consequence that there are only two quantum-critical regimes. One is the long-range mean-field regime for σ≤2/3, and the other one is a regime of a continuously varying critical exponent with a non-trivial fixed point. The third regime in these models is non-critical.

The only study investigating the full parameter space of this model is Ref. [35] using pCUT+MC and complementary linear spin–wave calculations. A previous study [306] using QMC and DMRG investigated the λ=1 parameter line for $H_{⋈}$. There are also known limiting cases of decoupled staggered long-range Heisenberg chains at λ=∞, where a QPT from a QLRO phase towards the same Néel-ordered antiferromagnetic phase occurs. We can compare the critical points determined by pCUT+MC for $H_{\left|\right|}$ and the linear spin–wave results to the SSE QMC data from Ref. [22] and also to the linear spin–wave calculations in Refs. [22,322] of the long-range Heisenberg chain. In fact, the linear spin–wave calculations for $H_{\left|\right|}$ can be seen as a generalisation of the ones for the Heisenberg chain and, therefore, exactly including them as a limiting case.

We can find a plot in Figure 35 showing a ground-state phase diagram for both $H_{\left|\right|}$ and $H_{⋈}$ from Refs. [35,353]. The figure also includes other known values from Refs. [22,306], which fit into the overall picture of the pCUT+MC results. We also show the critical exponents zν, (2−z−η)ν, and α determined by the pCUT+MC approach. It is easy to identify the two critical regimes of long-range mean-field behaviour and continuously varying critical exponents as predicted by quantum field theory [20,21,244]. In the non-trivial regime, the exponents shown here diverge when approaching the lower critical dimension $\sigma \to \sigma_{\mathrm{lc}}$, which is in reasonable agreement with FRG results in Refs. [21,244]. The largest deviation can be seen for α, which is the hardest exponent to extract from DlogPadé extrapolations. Using the (hyper)scaling relations (Equation 3)–(Equation 9), the remaining critical exponents can be determined. See Figure A2 in Appendix D showing all critical exponents for the Heisenberg ladders. It should be noted that there are three main difficulties discussed in Ref. [35]. First, as just stated, the α exponent is difficult to determine. Second, it becomes increasingly hard to extrapolate the perturbative series in the regions σ≳1.1 (σ≳1.2) for $H_{\left|\right|}$ ($H_{⋈}$). Third, the presence of logarithmic corrections to the dominant power-law behaviour close to the critical point spoils the exponents around the upper critical dimension at σ=2/3. In the end, all factors play a role when determining all critical exponents due to error propagation. We can observe in Figure A2 that several exponents from the pCUT+MC approach deviate from the FRG exponents significantly in the non-trivial regime. For instance, the ν exponent seems to approach a constant value ν≈1 for $\sigma \to \sigma_{\mathrm{lc}}$ for pCUT+MC, while the FRG predicts a diverging exponent. Another important finding is that the lower critical decay exponent $\sigma_{\mathrm{lc}}$ is apparently not universal in these two models and considerably smaller than the predicted value $\sigma_{\mathrm{lc}}=2$ from quantum field theory [21,244]. This claim is made in Ref. [35] due to the known limiting case of the long-range Heisenberg chain [22] from SSE QMC simulations and due to linear spin–wave calculations [35].

Beyond these interesting discrepancies, there was speculation about a possible deconfined quantum critical point along the λ=1 parameter line for $H_{⋈}$ in Ref. [306]. The reason for this was the fact that the staggered long-range Heisenberg ladder undergoes a QPT from a disordered phase with a non-local string order parameter towards a Néel-ordered phase with conventional order. Also, they found a sharp peak and a gap in the dynamic structure factor at the ordering momentum $k_{c}$ in the ordered phase, which could be indicative of deconfined excitations in terms of spinons [306,354]. Usually, when there is a QPT between two competing ordered phases, the system undergoes a first-order phase transition or there must be a coexistence phase. There is also a much more exotic scenario beyond the Landau–Ginzburg–Wilson theory of phase transitions [334]. A deconfined quantum critical point [334,335,336,337] is a second-order QPT that is not described by a “confining” order parameter, but by an emergent U(1)-symmetric gauge field with “deconfined” degrees of freedom accompanied by a fractionalisation of the order parameters [334]. A paradigmatic example is the deconfined QPT between a Néel-ordered antiferromagnetic ground state and a VBS state on a two-dimensional lattice [334,335,337,355,356,357,358,359,360]. While deconfined criticality was originally proposed in two dimensions, similarities in terms of a conventional Luttinger Liquid theory description have been drawn in one dimension [361,362,363,364,365,366,367,368]. Interestingly, a very close analogy to a two-dimensional deconfined critical point was found in Ref. [338] using a toy model with six-spin Heisenberg interactions and long-range two-spin interactions inducing a continuous phase transition between a VBS phase and an antiferromagnetic phase. A scenario between a conventional order and non-local string order as proposed by Ref. [306] would go even beyond the one-dimensional scenarios found so far. However, it was argued in Ref. [35] that there should be no such deconfined critical point in the above long-range Heisenberg ladders due to the critical exponents found and the fact that the rung-singlet phase is adiabatically connected to the trivial product state of rung singlets not falling into the category of symmetry-protected topological phases despite the presence of a non-local string order parameter [35,369]. Another possible interpretation of the finding in Ref. [306] is probably along the lines of Refs. [38,321]. The observed gap in the dynamic structure factor is in agreement with strong finite-size artefacts arising from the altered dispersion $\omega ∼{\left|k\right|}^{s}$ of sublinear behaviour s<1 in the anomalous Goldstone regime σ≤2.

In this subsection, we have seen that the antiferromagnetic Heisenberg ladders with staggered long-range interactions show two quantum-critical regimes: one regime with long-range mean-field behaviour and a second non-trivial regime with continuously varying critical exponents. There is also a third regime that does not show any QPT because continuous symmetry breaking is ruled out by the HMW theorem. Despite the numerical confirmation of the two critical regimes predicted from quantum field theory [20,21,244,326], a discrepancy between the upper bounds of the Néel-ordered phase and the predicted lower critical exponent $\sigma_{\mathrm{lc}}$ was identified by Ref. [35]. On the other hand, the data in Refs. [35,353] are in agreement with other literature for the limiting case of long-range Heisenberg spin chains [22,322]. We also discussed briefly the possibility of a deconfined QPT and the presence of anomalous Goldstone modes in one dimension for the above ladder models. To put it briefly, the Hamiltonian (Equation 330) hosts some intriguing physics with several aspects that need further clarification.

### 8.3. Staggered Antiferromagnetic Long-Range Heisenberg Chain

Lastly, we consider another one-dimensional system, the Heisenberg chain with staggered non-frustrating antiferromagnetic long-range interactions:(331)$$H=\sum_{i}\left[{\vec{S}}_{i}{\vec{S}}_{i+1}-\sum_{j=2}^{\infty }\lambda \left(i-j\right){\vec{S}}_{i}{\vec{S}}_{i+j}\right].$$ In the previous two models (bilayer and ladders), the unperturbed part at λ=0 consisted of uncoupled dimers with a trivial product singlet state as its ground state and local triplet excitations above. Here, the unperturbed part is the nearest-neighbour Heisenberg chain with a ground state exhibiting quasi-long-range order (QLRO) and fractionalised elementary excitations. These excitations are referred to as spinons and can be seen as propagating domain walls carrying S=1/2 degrees of freedom. The perturbation consists of long-range interactions that couple sites beyond the nearest-neighbours. This interaction is of the same algebraic form as Equation (Equation 320). Because of the non-frustrating nature of the antiferromagnetic long-range interactions, it induces a Néel-ordered antiferromagnetic phase upon increasing its coupling strength. Again, a QPT breaking the SU(2)-symmetry of the Hamiltonian is only allowed when the long-range decay exponent satisfies σ<2 due to the HMW theorem [22,317], and thus, such a transition can be ruled out for larger decay exponents. In this model, a QPT from a QLRO towards an ordered phase is expected to occur, and therefore, the $\phi^{4}$ theory of Equation (Equation 326) does not apply. The k=1 Wess–Zumino–Witten non-linear σ model [370,371] is known to describe the low-energy physics of Heisenberg chains and includes topological coupling to account for the presence of QLRO in the lattice model [22,213]. As for the $\phi^{4}$ theory, a long-range coupling analogous to Equation (Equation 327) can be added to describe the Hamiltonian (Equation 331) [22].

Reference [22] is a comprehensive study of the Heisenberg chain (Equation 331) using large-scale SSE QMC simulations to extract the critical properties of the QPT. The results for the critical point, as well as the critical exponents η and *z* can be found in Figure 36. The overall behaviour of the critical point as a function of the decay exponent σ is very similar to the one found in the previous subsection for the Heisenberg ladder. Here, the critical point diverges at about σ≈1.8 when approaching the lower critical exponent $\sigma \to \sigma_{\mathrm{lc}}$. The hard boundary for a QPT is again given by the HMW theorem. Yet, there is a significant difference. While, for the Heisenberg ladders, the disordered rung-singlet phase exists for any σ>0, for the Heisenberg chain, the QLRO phase only exists for σ>1. Thus, for the Heisenberg chain, the critical line terminates in a marginal point at σ=1 and λ=0. For any σ<1, the perturbation parameter λ becomes irrelevant, and the system is always in the antiferromagnetic phase [22]. The long-range Heisenberg chain (Equation 331) was also studied in Refs. [213,297,333] in the context of a Heisenberg chain with frustrated next-nearest-neighbour and non-frustrating long-range interactions, i.e., the $J_{1}-J_{2}$ chain with staggered long-range interactions. In both models, the QLRO-Néel QPT can be identified by a level crossing between a triplet S=0 and quintuplet S=2 excitations (The transition was initially misidentified as a level crossing between two S=0 states in Refs. [213,297] until it was later clarified to be a level crossing between the S=0 and S=2 states [333]). The critical point from finite-size scaling of the level crossing from the ED [297] and DMRG [333] is in very good agreement with the QMC results at λ=1 (see Figure 36). Proceeding with the critical exponents, we can see that the critical exponent η matches well with the field-theoretical expectations η=2−σ (linear in at least leading order, but also a simple scaling argument predicts the linear behaviour [22]) in the range 1≤σ≤1.3. For σ>1.3, the η from SSE QMC simulations starts to deviate from the linear behaviour. One interesting observation was pointed out in Ref. [35]. In both the Heisenberg ladders and the Heisenberg chain, the linear behaviour η=2−σ is expected from the underlying quantum field theory, yet the data from pCUT+MC and SSE QMC studies indicate a deviation from this with η≤2−σ for the Heisenberg ladders and η≥2−σ for the Heisenberg chain. It should be noted, however, that the exponent from the pCUT+MC approach is determined using scaling relations and, therefore, can suffer from unfavourable error propagation, especially when the α exponent is involved. The dynamical exponent *z* of the Heisenberg chain was extracted from SSE QMC simulations as well. The exponent *z* is one at the marginal point σ=1 and then quickly drops to z≈0.75, where it seems to be constant within the error bars up to σ=1.7. This finding is also in contrast to the RG prediction, where z=1 in leading order, and the exponent is expected to be constant even in higher orders [22]. In Ref. [213], for λ=1, the dynamic exponent *z* was determined in excellent agreement with the QMC results (see Figure 36). Also, the results for the $J_{1}-J_{2}$ model with non-frustrating long-range interactions, where the same QLRO-Néel transition is realised, the exponent is in agreement with z≈0.75 [213,297]. Further, the RG analysis in Ref. [22] gave also a prediction for ν. However, this could not be compared with QMC as it was not possible to obtain accurate estimates of ν [22]. Note, also, that neither the long-range transverse-field XY chain nor the Heisenberg chain show long-range mean-field behaviour.

The Heisenberg chain with staggered long-range interactions is another prime example of long-range models hosting intriguing critical behaviour. The remaining discrepancy between the numerical SSE QMC results and the underlying field-theoretical description shows that the critical properties are not yet fully settled and further exploration of the model is necessary.

## 9. Summary and Outlook

In this review, we gave an overview of recent advances in the investigation of the quantum-critical properties of quantum magnets with long-range interactions focusing on two techniques, both based on Monte Carlo integration, but complementary in spirit. On the one hand, we described pCUT+MC, where classical Monte Carlo integration is decisive in the embedding scheme of white graphs. This allows extracting series expansions of relevant physical quantities directly in the thermodynamic limit. On the other hand, SSE QMC enables calculations on large finite systems where finite-size scaling can be used to determine the physical properties of the infinite system. Both quantitative and unbiased approaches take the full long-range interaction into account and can be used a priori in any spatial dimension for any geometry.

In recent years, both techniques, alongside other methods, have been applied successfully to one- and two-dimensional quantum magnets involving long-range Ising, XY, and Heisenberg interactions on various bipartite and non-bipartite lattices. In this work, we have summarised the obtained quantum-critical properties including quantum phase diagrams and the (full sets of) critical exponents for all these systems coherently. Further, we reviewed how long-range interactions are used to study quantum phase transitions above the upper critical dimension and how the scaling techniques are extended to extract these quantum critical properties from the numerical calculations. This is indeed generically the case for all unfrustrated systems in this review, with the exception of the one-dimensional isotropic XY and Heisenberg chain. For frustrated systems, one can apply both MC techniques successfully for Ising interactions, while in general, only the pCUT+MC method is applicable (if an appropriate perturbative limit exists) due to the sign problem of SSE QMC. Nevertheless, in all frustrated cases, the small-σ regime of long-range interactions is challenging for both approaches, and further technical developments are desirable.

In the future, several extensions and research directions are interesting. As mentioned in Section 6.3 of this review, the interplay between long-range interactions and geometric frustration is a vibrant research field at the moment. It has been demonstrated numerically and experimentally that this interplay provides a great resource to engineer exotic phases of matter [54,55,109,283,372] with the most spectacular example being the $Z_{2}$ quantum spin liquid on the Ruby lattice [55,109,372]. We expect further rapid development in the field since many promising theoretical proposals can be realised in analogue quantum simulation platforms (e.g., programmable Rydberg atom quantum simulators [52,55]).

In terms of methods, pCUT+MC is yet to be extended to arbitrary unit cells, larger spin values, and multi-spin interactions. The access to larger unit cells will enable the investigation of the interplay between long-range interactions and frustration on even more relevant lattice structures, e.g., the Kagome or Ruby lattice. There are systems with multi-spin interactions hosting deconfined quantum criticality [338,355,356,357,358,359], and an introduction of long-range interactions to this type of systems seems to be an interesting research topic [338]. We hope to spark further interest in the development and application of pCUT+MC by other users.

The SSE QMC approach is a widely used numerical tool for the calculation of unbiased thermal averages of observables. A large variety of distinct QPTs is potentially accessible by these QMC simulations. We envision that, for all systems that do not suffer from a sign problem, SSE QMC, in combination with appropriate zero-temperature protocols and finite-size scaling, can be used to study how long-range interactions affect QPTs beyond the standard O(n)-symmetry. The SSE QMC method has also been extended to tackle frustrated systems in a more efficient way [261,262,373]. However, an efficient treatment of both the long-range interaction and frustration has not been introduced yet [32,40]. Reference [374] developed an SSE QMC approach to access the toric code quantum spin liquid regime [55,109,372]. An application of the SSE QMC approach to extended long-range interacting Bose–Hubbard models (see Section 5.5) along the lines of directed loop updates [215,216,217] would also be a natural development. This would enable numerically calculating observables with SSE QMC for ultracold gas experiments with optical lattices. A possible application could be the study of complex crystalline phases and their breakdown in frustrated Bose–Hubbard systems with long-range interactions [64,73,111,230].

Finally, in the context of the long-range mean-field regime above the upper critical dimension, much research has been conducted regarding finite-size scaling in classical systems [12,13,14,15,16,17,18,19,41,125,128,142,145], including the study of multiplicative logarithmic corrections for the characteristic length scale at the upper critical dimension [17] and the investigation of the role of Fourier modes and boundary conditions [19]. On the contrary, its quantum counterpart has only been treated successfully in recent years [32,34]. In addition to the transfer of established concepts from classical to quantum Q-FSS, one interesting open question is a detailed understanding of the crossover regime between classical and quantum Q-FSS for small temperatures. Even though we focused on the quantum version of the Q-FSS of Ref. [34] due to the quantum nature of the models analysed in this review, the ground work has been conducted by the inventors of classical Q-FSS (Refs. [15,16,17,18,19,41,125]) and, in general, many other researchers who provided valuable insight into the scaling above the upper critical dimension, e.g., Refs. [10,12,13,14,127,128,142]. Overall, it is exciting that the abstract concept of dangerous irrelevant variables and the physics above the upper critical dimension are accessible in quantum–optical platforms realising long-range interactions.

## Figures and Tables

**Figure 1 entropy-26-00401-f001:**
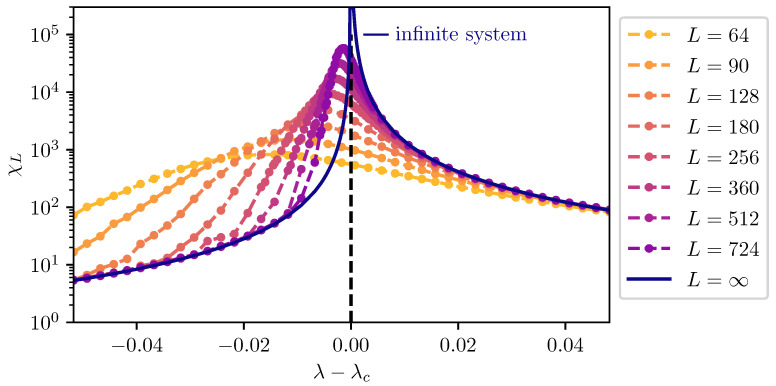
Susceptibility $\chi_{L}$ of the long-range transverse-field Ising chain for different linear system sizes from L=64 to L=724. The smaller the system, the farther away from the critical point the susceptibility starts to deviate from the thermodynamic limit and the farther the peak position shifts away from the critical point marked by the black dotted line.

**Figure 2 entropy-26-00401-f002:**
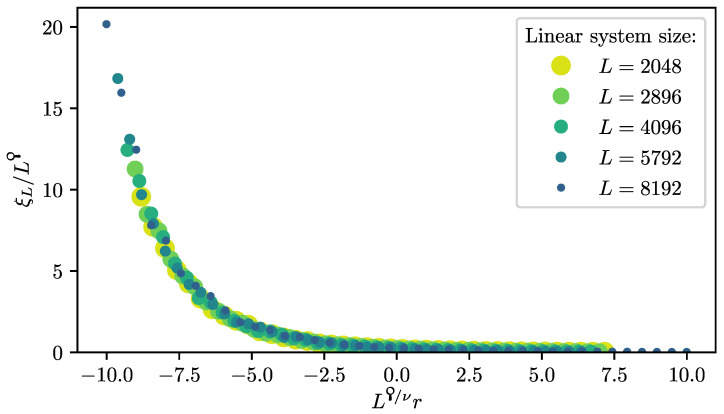
Rescaled correlation length according to Equation (Equation 37) for a model above the upper critical dimension with $ϙ=20/9\approx 2.\bar{2}$ (to be specific, the long-range transverse-field Ising chain with σ=0.3). The control parameter $r∼h-h_{c}$ is proportional to the transverse field. The collapse of the data around the critical point r=0 verifies the scaling Equation (Equation 37) and, therefore, demonstrates that $\xi_{L}$ is indeed—in contrast to the prior belief—not bound by the linear system size, but $\xi_{L}∼L^{ϙ}$.

**Figure 3 entropy-26-00401-f003:**
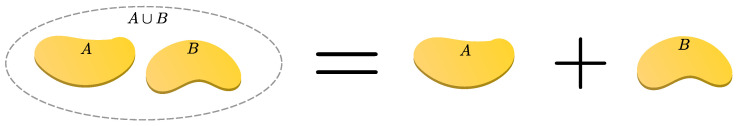
Schematic illustration of cluster additivity. The contribution of a disconnected cluster A∪B (clusters within dashed circle) made up of individual connected clusters *A* and *B* (yellow areas) is the sum of its individual parts.

**Figure 4 entropy-26-00401-f004:**
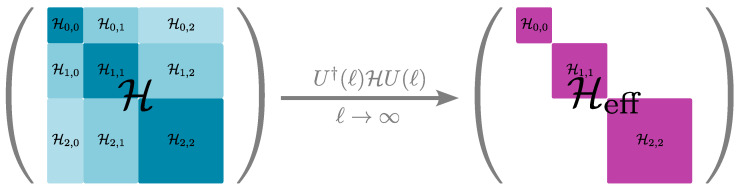
Illustration of the method of perturbative continuous unitary transformations (pCUTs) transforming the original Hamiltonian on the left to a block-diagonal quasiparticle-conserving effective Hamiltonian on the right. The desired effective Hamiltonian is given in the limit ℓ→∞ of the flow parameter *ℓ* of the continuous unitary transformation $H\left(\ell \right)=U^{†}\left(\ell \right)H\left(0\right)U\left(\ell \right)$. While the different quasiparticle sectors interact with each other by the off-diagonal blocks in the original Hamiltonian, the off-diagonal blocks are zero in the effective Hamiltonian as they are “rotated away” during the flow.

**Figure 5 entropy-26-00401-f005:**
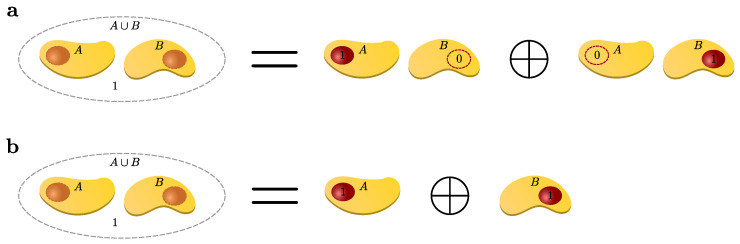
Schematic illustration of cluster additivity (**a**) and particle additivity in the one-particle basis (**b**) on a disconnected cluster A∪B (grey area) consisting of individual connected clusters *A* and *B* (yellow areas). (**a**) Cluster additivity in the one-particle basis translates to one particle being on cluster *A* and zero on *B* and vice versa. To calculate the contribution on the disconnected cluster A∪B, both contributions need to be considered including the cases of zero occupancy. (**b**) Particle additivity is fulfilled when the one-particle contribution on a disconnected cluster A∪B is simply the sum of one-particle contributions on the connected clusters *A* and *B*.

**Figure 6 entropy-26-00401-f006:**
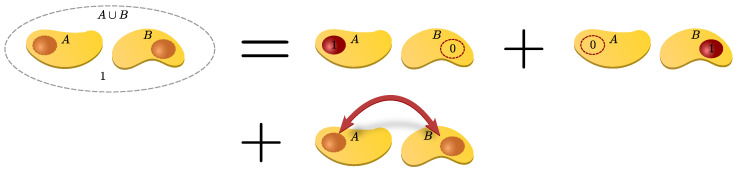
For operators that are not cluster additive, the contribution on the disconnected cluster A∪B originates not only from the sum of the contributions where a single particle is on *A* or *B*, but also from contributions where the particle can hop between the two connected clusters.

**Figure 7 entropy-26-00401-f007:**
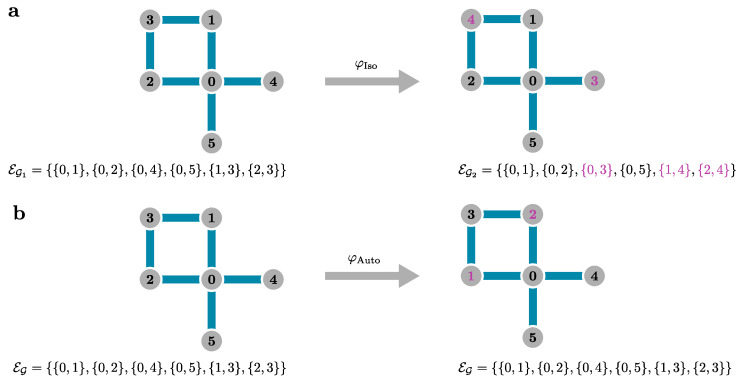
Illustration of a graph isomorphism $\varphi_{\mathrm{Iso}}$ and automorphism $\varphi_{\mathrm{Auto}}$ map for an example graph. (**a**) The mapping $\varphi_{\mathrm{Iso}}\left(3\right)=4$, $\varphi_{\mathrm{Iso}}\left(4\right)=3$ and identity for the remaining vertices is a graph isomorphism preserving the adjacency of vertices. If such an isomorphism exists between two graphs, they are topologically equivalent. (**b**) Under a graph automorphism, the edge set $E_{G}$ remains invariant, i.e., the graph is mapped onto itself. Here, it is exemplified for the mapping $\varphi_{\mathrm{Auto}}\left(1\right)=2$, $\varphi_{\mathrm{Auto}}\left(2\right)=1$ and identity for the remaining vertices. A graph automorphism is, therefore, a special case of a graph isomorphism, which leaves the edge set invariant.

**Figure 8 entropy-26-00401-f008:**
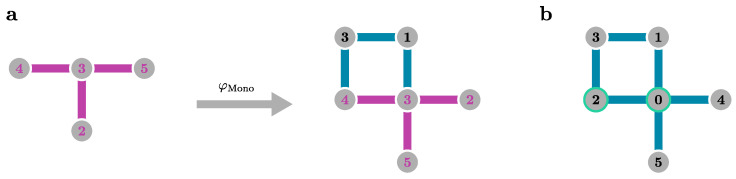
Depiction of a graph monomorphism $\varphi_{\mathrm{Mono}}$ and a graph with colour attributes. (**a**) An example showing a graph monomorphism. The smaller graph on the left is mapped onto the bigger graph on the right, which is the same graph as in Figure 7. Explicitly, the mapping $\varphi_{\mathrm{Mono}}$ is given by $\varphi_{\mathrm{Mono}}\left(2\right)=4$, $\varphi_{\mathrm{Mono}}\left(3\right)=0$, $\varphi_{\mathrm{Mono}}\left(4\right)=2$, and $\varphi_{\mathrm{Mono}}\left(5\right)=5$. (**b**) A coloured graph with the “green” colour attribute assigned to vertices 0 and 2 ($A_{V}=\left\{\left\{0,\mathrm{green}\right\},\left\{2,\mathrm{green}\right\}\right\}$). If mappings are applied to coloured graphs, the colour set must be left invariant, i.e., vertices with a colour must be mapped onto each other and vertices with no colour as well.

**Figure 9 entropy-26-00401-f009:**
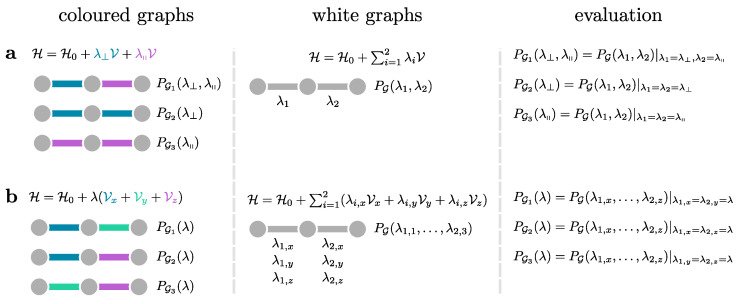
In contrast to the conventional approach using coloured graphs (left), where different expansion parameters or different interaction types are associated with an edge colour, for white graphs (centre) the edge colour is ignored in the topological classification of graphs. Instead additional information is tracked, e.g., by associating each link with abstract expansion parameters and only substituting these abstract contribution during the embedding procedure, reintroducing the correct colour information (right), hence the name white graphs. (**a**) For the problem of Equation (Equation 137) on linear graphs with two edges, there are three distinct graphs as the expansion parameters $\lambda_{⊥}$ and $\lambda_{\left|\right|}$ are associated with individual edge colours (left), but there is only one white graph, as we associated one abstract expansion parameter for each edge (centre). When substituting the abstract expansion parameters with the physical one, reintroducing the correct colour, we can recover the polynomial contributions of the conventional approach (right) (cf. Ref. [152]). (**b**) For the problem of Equation (Equation 138) also on linear graphs with two edges, there are also three topologically distinct graphs (left), but for the white graph contribution, we have to introduce multiple abstract expansion parameters for each edge, due to the three flavours f∈{x,y,z} (centre). The substitution works analogously to recover the polynomial contribution form the conventional approach (right). Parameters that are not explicitly set are set to zero (cf. Ref. [204]).

**Figure 10 entropy-26-00401-f010:**
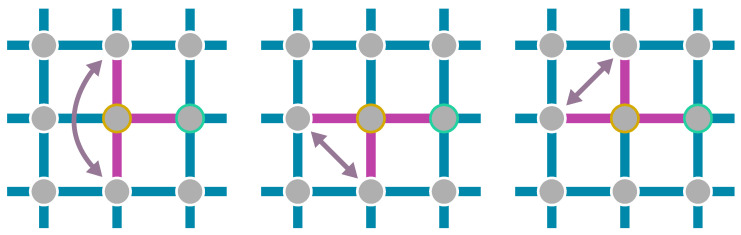
The graph with three edges given in Figure 8 is embedded on the infinite lattice. We consider a coloured graph with additional colour attributes (yellow and green) for two specific vertices. The reason behind colouring the vertices might by to simply fix the graph due to translational and rotational symmetry of the lattice as it is done when calculating the ground-state energy or due to the presence of a one-quasiparticle process from one coloured site to the other. For the ground-state energy contribution, the embedding factor would be $w\left(G_{c},L_{c}\right)=\frac{q}{s_{G}}\left|\mathrm{Mono}\left(G_{c},L_{c}\right)\right|=\frac{4}{6}\times 6=4$. There are six possible embeddings (monomorphisms) on the infinite lattice, when the coloured vertices on the graph are correctly mapped to the coloured sites on the lattice. There are only three geometrically distinct embeddings, but the number of monomorphisms is two-times bigger due to an ambiguity of mapping the subgraph on the graph as illustrated by the arrows.

**Figure 11 entropy-26-00401-f011:**
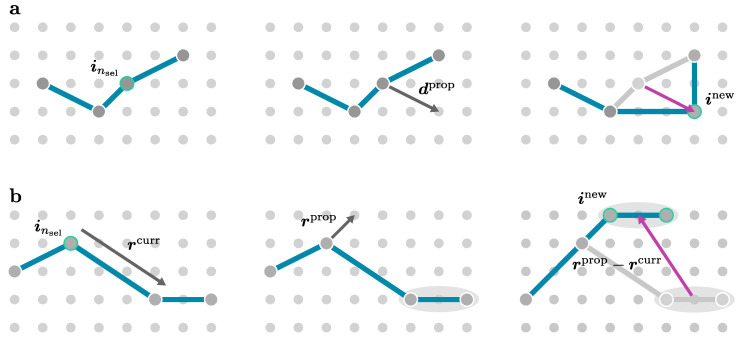
Exemplary Monte Carlo moves for a linear graph on a two-dimensional square lattice. (**a**) During a shift move, a vertex $n_{\mathrm{sel}}\in \left\{1,\cdots ,n_{s}\right\}$ is selected randomly from a uniform distribution. Then, a shift vector $d^{\mathrm{prop}}$ is drawn (uniformly for each component), which moves the selected site to $i^{\mathrm{prop}}=i_{n_{\mathrm{sel}}}+d^{\mathrm{prop}}$ if accepted. (**b**) For rift moves, a vertex is selected from $n_{\mathrm{sel}}\in \left\{1,\cdots ,n_{s}-1\right\}$. Instead of drawing from uniform distributions, rift moves account for the correct asymptotic behaviour of the system by drawing a new distance to the next vertex from a ζ-function distribution (from a normal ζ function in one dimension and from a double-sided ζ function in higher dimension for each component). If accepted, the vertices $n>n_{\mathrm{sel}}$ are shifted to the new position $i_{n>n_{\mathrm{sel}}}^{\mathrm{prop}}=i_{n>n_{\mathrm{sel}}}+\left(r^{\mathrm{prop}}-r^{\mathrm{curr}}\right)$.

**Figure 12 entropy-26-00401-f012:**
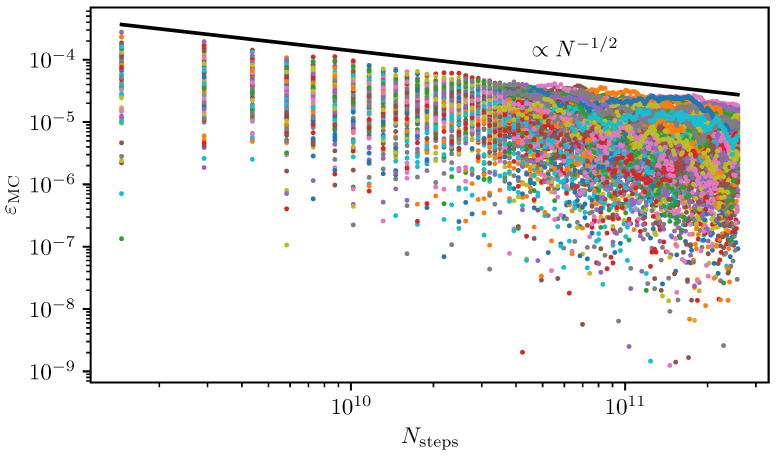
Monte Carlo error $\varepsilon_{\mathrm{MC}}=|p_{2}^{\mathrm{exact}}-p_{2}^{\mathrm{MC}}\left|/\right|p_{2}^{\mathrm{exact}}|$ (coloured data points) on a log–log scale as a function of the number of steps $N_{\mathrm{steps}}$. The plot shows data from a hundred 12-hour-long MC runs with a distinct seed each. The MC error $\varepsilon_{\mathrm{MC}}$ goes to zero following a $N_{\mathrm{steps}}^{-1/2}$ convergence behaviour (indicated by the black line), as generally expected from an MC algorithm.

**Figure 13 entropy-26-00401-f013:**
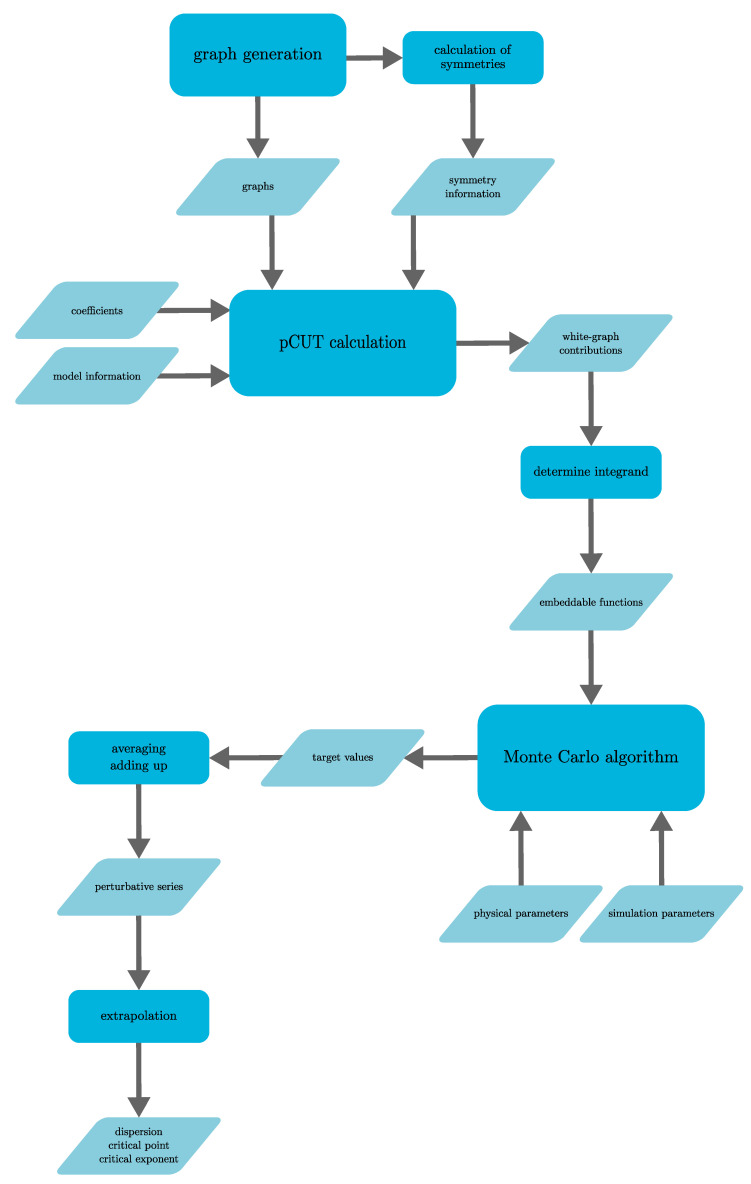
The workflow of the presented pCUT+MC method consists of several steps. There are three major steps. First, there is the graph generation, which only has to be performed once. Second is the calculation of the graph contributions with the pCUT method, and third is the Monte Carlo algorithm to embed the contributions on the lattice to determine the perturbative series of the quantity of interest in the thermodynamic limit.

**Figure 14 entropy-26-00401-f014:**
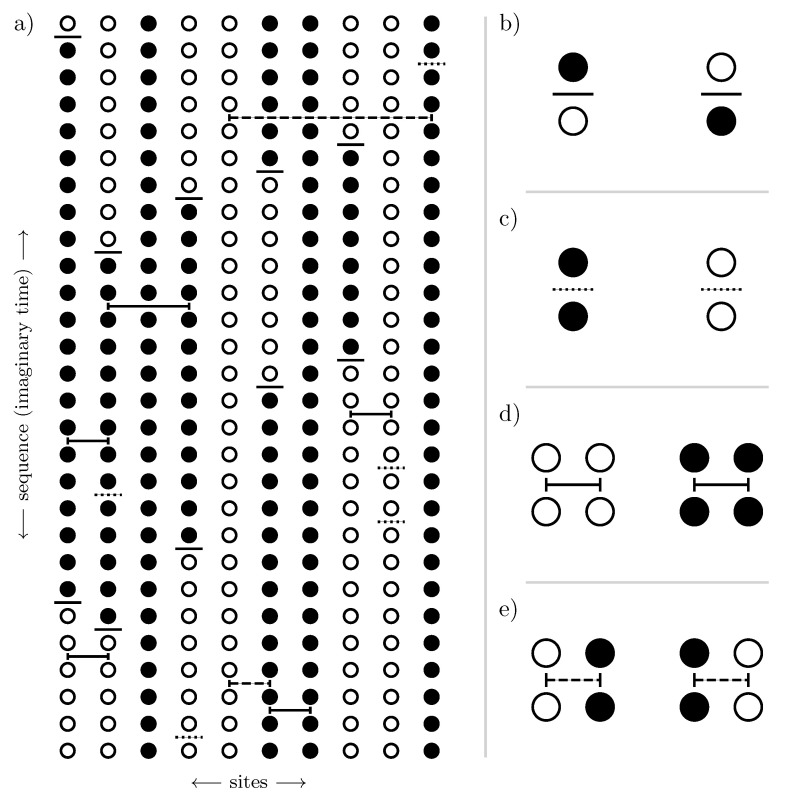
(**a**) An SSE configuration of a transverse-field Ising chain with N=10 sites and a sequence length L=27. The spatial spin direction goes from left to right, and the imaginary-time dimension goes from bottom to top. The number of trivial operators in the operator sequence $S_{L}$ is 8. Filled (empty) circles represent spins aligned in the $\sigma_{i}^{z}=+1\;\left(-1\right)$ direction. The propagated states |α(p)〉 correspond to the *p*-th row from below with the lowest row being state |α〉. (**b**) A depiction of the possible vertices for field operators. (**c**) A depiction of the possible vertices for constant operators. (**d**) A depiction of the allowed vertices for ferromagnetic Ising operators. Note that a ferromagnetic Ising vertex can only connect sites that are connected by a ferromagnetic bond in the Hamiltonian. (**e**) A depiction of the allowed vertices for antiferromagnetic Ising operators. Note that an antiferromagnetic Ising vertex can only connect sites that are connected by an antiferromagnetic bond in the Hamiltonian.

**Figure 15 entropy-26-00401-f015:**
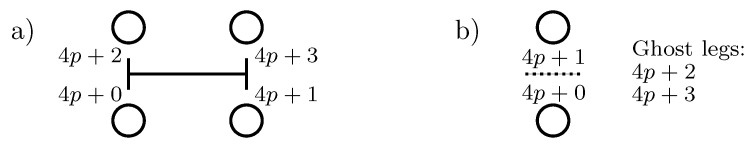
Assignment of leg numbers to operator legs for an operator at propagation step *p*. (**a**) Illustration of a two-site operator with four real vertex legs. (**b**) Illustration of a single-site operator with only two real vertex legs and two ghost legs.

**Figure 16 entropy-26-00401-f016:**
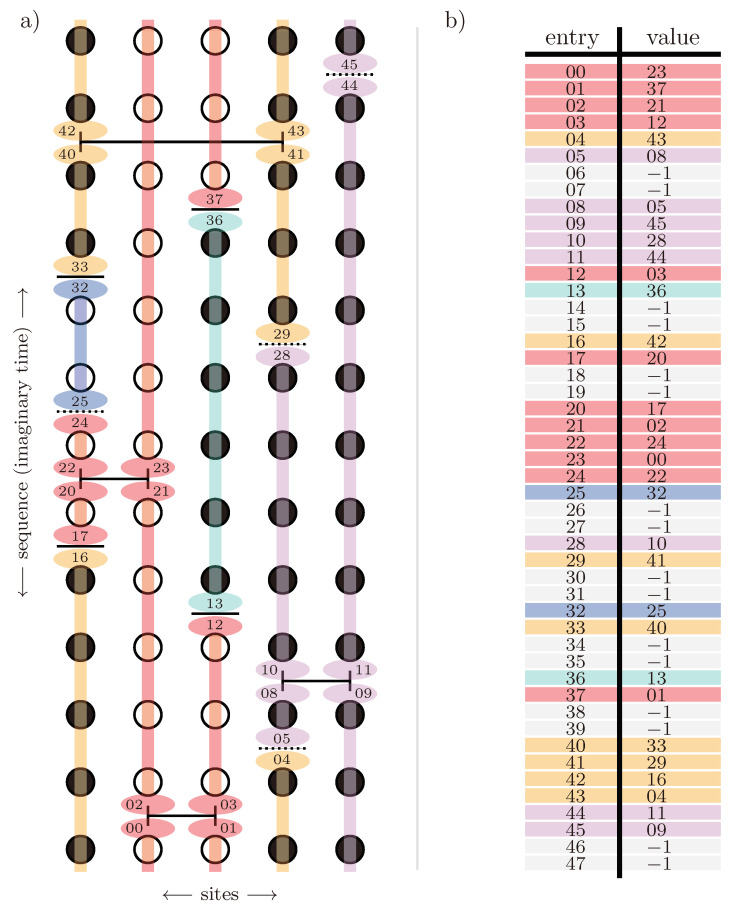
Illustration for the segmentation and construction of the doubly linked list for an exemplary configuration of the ferromagnetic transverse-field Ising model. (**a**) Segmentation of a configuration into disjoint clusters including the numbers of the operator legs. Ghost legs are not depicted. The coloured lines with ellipses at each end depict the operator legs that are linked. Each colour represents one cluster in the off-diagonal update. (**b**) Depiction of a doubly linked list for the configuration shown in (**a**). The left column represents the entry numbers in the list and the right column the corresponding legs to which the entry is connected. The colour represents the clusters in (**a**) to which the connection belongs. Ghost legs are linked to the value −1 and are shaded in grey.

**Figure 17 entropy-26-00401-f017:**
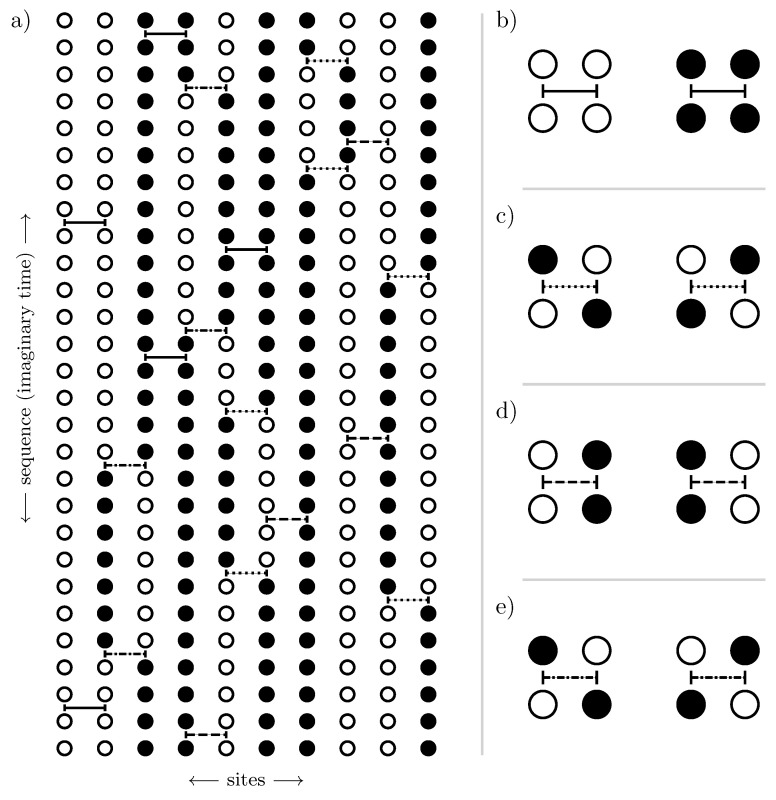
(**a**) An SSE configuration of a Heisenberg chain with ten sites and a sequence length of 27. The number of trivial operators is 5. Filled (empty) circles represent spins in the $\sigma_{i}^{z}=+1\left(-1\right)$ direction. The propagated states |α(p)〉 correspond to the *p*-th row in the configuration. (**b**) A depiction of the allowed vertices for ferromagnetic diagonal operators. (**c**) A depiction of the allowed vertices for ferromagnetic off-diagonal operators. (**d**) A depiction of the allowed vertices for antiferromagnetic diagonal operators. (**e**) A depiction of the allowed vertices for antiferromagnetic off-diagonal operators. Note that (anti)ferromagnetic vertices can only connect sites that are connected by (anti)ferromagnetic bonds in the Hamiltonian.

**Figure 18 entropy-26-00401-f018:**
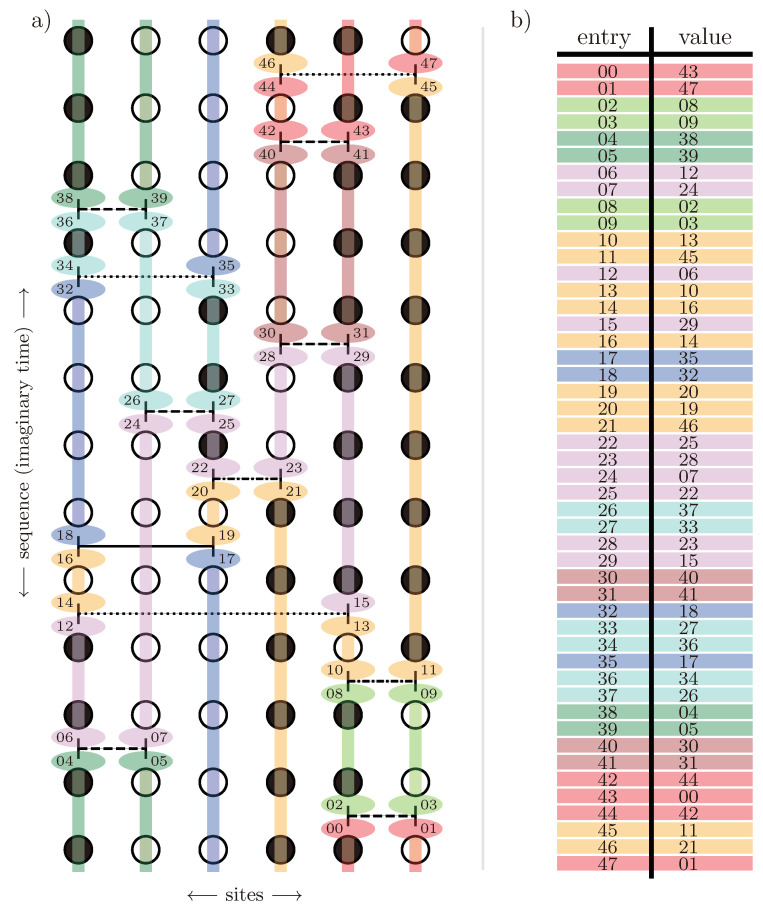
Illustration showing the construction of the doubly linked vertex list and the off-diagonal deterministic loop update for unfrustrated Heisenberg models. As an example, a Heisenberg chain with periodic boundary conditions and nearest-neighbour antiferromagnetic and next-nearest-neighbour ferromagnetic interactions is considered. (**a**) Illustration of a configuration including the numbers of the operator legs belonging to the respective operators. The coloured lines with ellipses at each end depict the operator legs that are linked. Each colour represents one loop in the off-diagonal update. (**b**) Depiction of a doubly linked list for the configuration shown in (**a**). The left column represents the entry numbers in the list and the right column the corresponding legs to which the entry is connected. The colour represents the loops in (**a**) to which the connection belongs.

**Figure 19 entropy-26-00401-f019:**
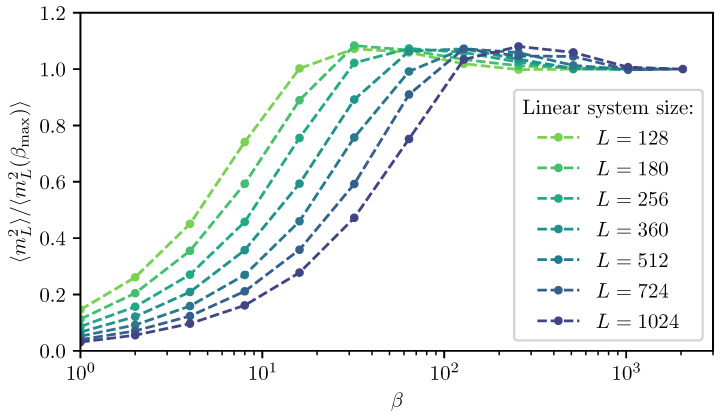
Illustration of the beta-doubling method for the one-dimensional LRTFIM in the short-range regime with decay exponent σ=2.5 at a transverse field of h=1.25. The simulation starts at β=1 (leftmost points) for every system size. The inverse temperature is then doubled in every beta-doubling step until the maximum $\beta_{\max}=2048$ is reached. All of the shown magnetisation curves seem to be converging to zero temperature. Larger systems with L>1024 were discarded as they do not appear to be fully converged yet.

**Figure 20 entropy-26-00401-f020:**
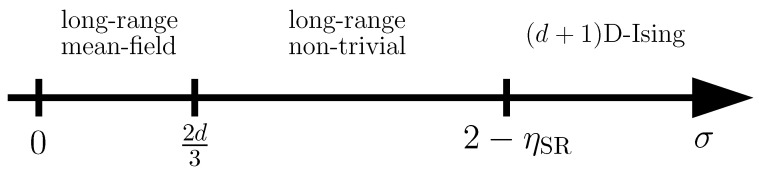
Sketch of the three distinct universality regimes of the QPT in the ferromagnetic LRTFIM. For one- and two-dimensional systems, all three regimes exist, and the boundaries can be obtained using the expressions in the figure. For d≥3, there is only the long-range mean-field and the nearest-neighbour mean-field regime with a boundary at σ=2.

**Figure 21 entropy-26-00401-f021:**
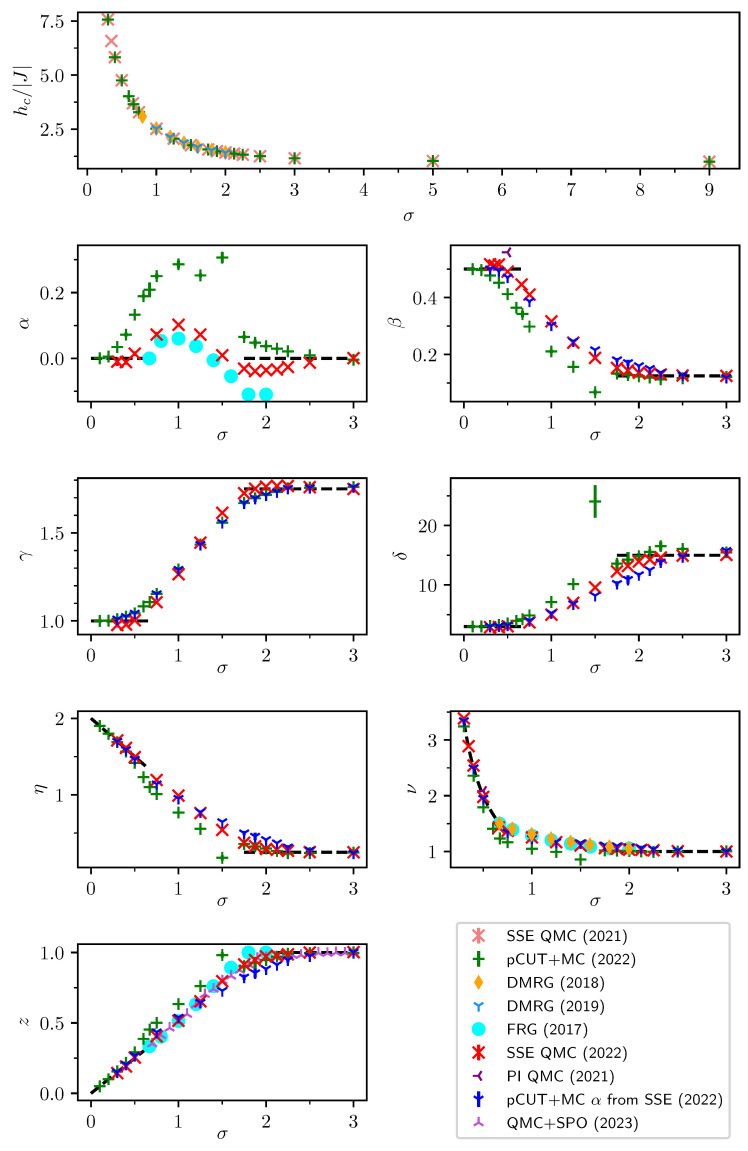
Critical field values and exponents from numerical studies of the ferromagnetic LRTFIM on the linear chain. The panels display $h_{c}/\left|J\right|$ (top), α (second row left), β (second row right), γ (third row left), δ (third row right), η (fourth row left), ν (fourth row right), and *z* (bottom). The labels refer to the references in the following way: “SSE QMC (2021) [32,249]”, “SSE QMC (2022) [34,247]”, “pCUT+MC (2022) [34,247]”, “pCUT+MC α from SSE (2022) [34,247]”, “DMRG (2018) [28]”, “DMRG (2019) [248]”, “FRG (2017) [21]”, “PI QMC (2021) [33]”, and “QMC+SPO (2023) [250]”. The values for the critical exponents of the transition in the nearest-neighbour model α=0, β=1/8, γ=7/4, δ=15, η=1/4, ν=1, and z=1 [246,251,252] and in the long-range mean-field regime α=0, β=1/2, γ=1, δ=3, η=2−σ, ν=1/σ, and z=σ/2 [20,21] are given by the dashed lines.

**Figure 22 entropy-26-00401-f022:**
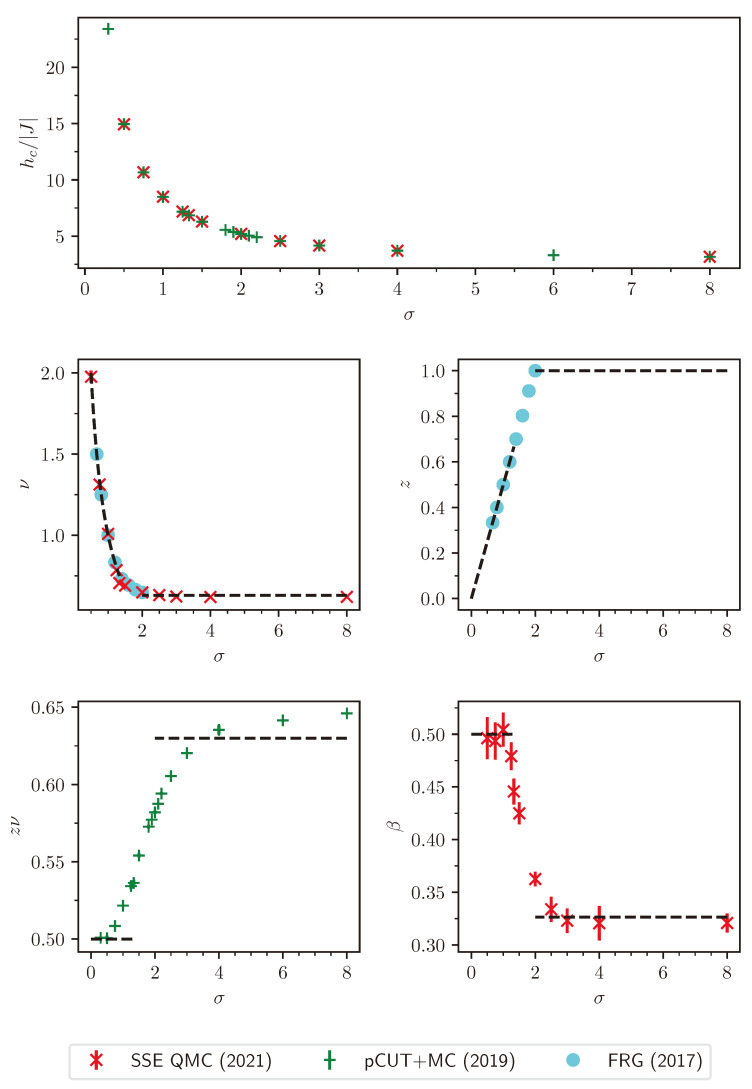
Critical field values and exponents from numerical studies of the ferromagnetic LRTFIM on the two-dimensional square lattice. The panels display $h_{c}/\left|J\right|$ (top), ν (middle left), *z* (middle right), zν (bottom left), and β (bottom right). The data points “SSE QMC (2021)” for $h_{c}/J$, ν, and β are from Refs. [32,249]. The data points “pCUT+MC (2019)” for $h_{c}/J$ and zν are from Ref. [29]. The data points “FRG (2017)” for ν and *z* originate from the functional RG study in Ref. [21]. The values for the critical exponents of the transition in the nearest-neighbour model ν=0.629971(4), β=0.326419(3), and z=1 [253] and in the long-range mean-field regime ν=1/σ, z=σ/2, and β=0.5 [20,21] are given by the dashed lines.

**Figure 23 entropy-26-00401-f023:**
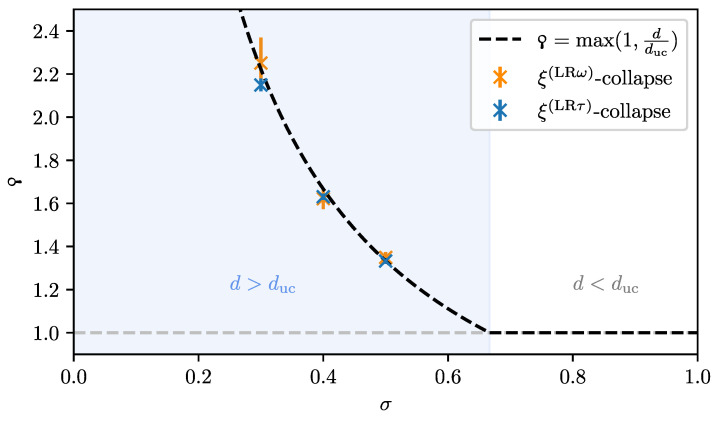
Exponent ϙ extracted by the data collapse of $\xi^{\left(LR\omega \right)}$ and $\xi^{\left(LR\tau \right)}$ for different decay exponents of the LRTFIM on the linear chain [34]. The black dashed line depicts the prediction by the Q-FSS $ϙ=\max(1,d/d_{\mathrm{uc}})$, while the grey dashed line shows the prediction from standard FSS ϙ=1. In the regime above the upper critical dimension for σ<2/3, the predictions start to deviate and the extracted values for ϙ are clearly in line with the Q-FSS scenario. Figure adapted from Ref. [34].

**Figure 26 entropy-26-00401-f026:**
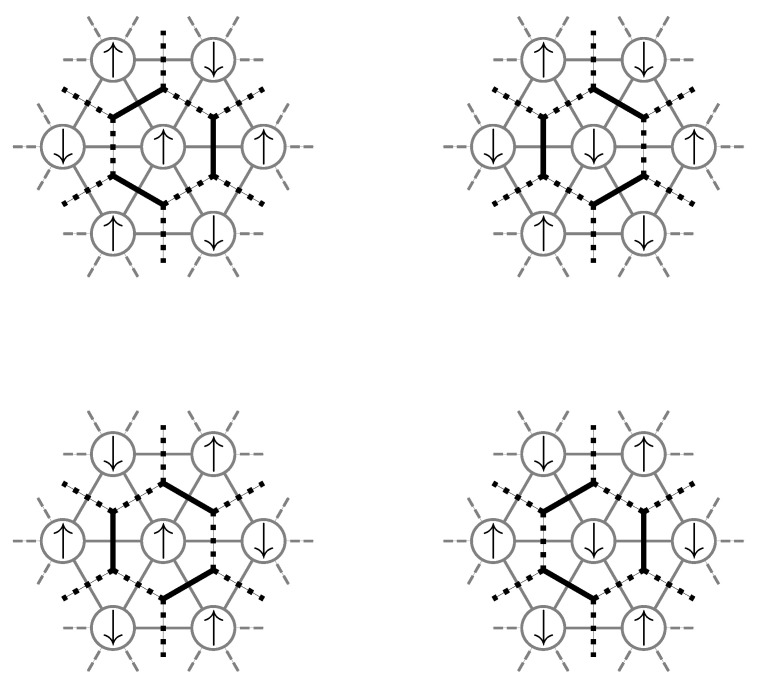
Illustration of the four local spin configurations called flippable plaquettes. The arrows denote the local spin orientation in the *z*-direction. The grey lines in the back visualise the triangular lattice. Solid (dotted) black lines are depicted on (anti)ferromagnetically aligned bonds. Note that flipping the spin in the centre of each configuration maps the left to the right configuration in each row and vice versa.

**Figure 27 entropy-26-00401-f027:**
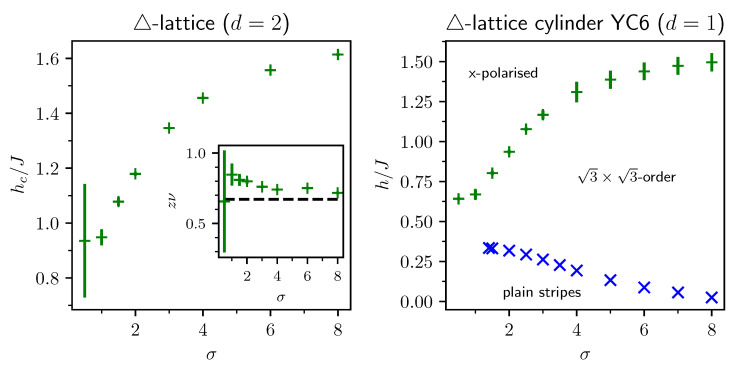
Phase diagrams for the long-range transverse-field Ising model on the triangular lattice (left panel) and triangular lattice cylinders with infinite extent in the *x*-direction and a circumference of 6 sites in the *y*-direction (YC6) (right panel). Right panel: The critical field values $h_{c}/J$ for the triangular lattice for the QPT between the high-field polarised phase originate from pCUT+MC calculations [29]. The inset of the right panel represents the critical exponents zν determined from the series expansion [29]. The black dashed line represents the critical value zν=0.67175(10) of the 3D-XY universality class [281,282]. The critical exponent zν confirms the (2+1)D-XY universality class within the limitations of the series expansion. Left panel: The transitions values between the high-field *x*-polarised phase and the $\sqrt{3}\times \sqrt{3}$-clock-ordered phase are from the gap closing of the high-field series obtained from pCUT+MC [30]. The transition between the plain stripe low-field phase and the $\sqrt{3}\times \sqrt{3}$-clock-ordered phase is terminated via a level crossing of both ground-state energies, which were calculated perturbatively [30]. The phase diagrams for small σ values are not yet conclusively determined [27,29,30].

**Figure 28 entropy-26-00401-f028:**
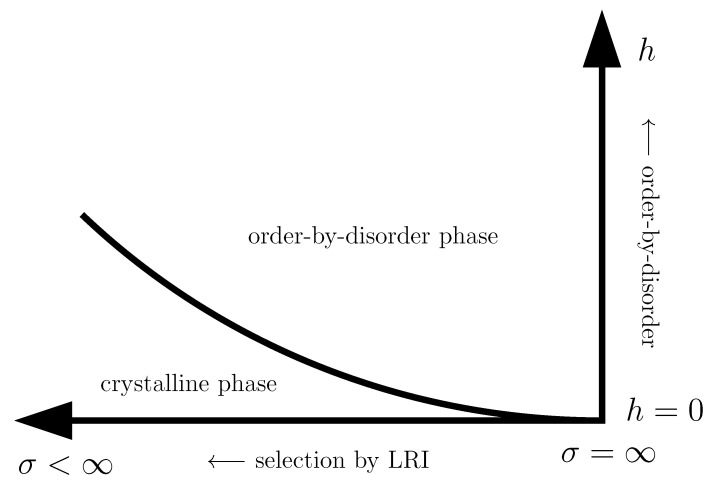
Sketch of a generic phase diagram of a long-range transverse-field Ising model (LRTFIM) for which a degenerate subspace at σ=∞ and h=0 breaks down in an order-by-disorder scenario for h>0 and into a different crystalline state due to the long-range interactions (LRIs) σ<∞. An example is the LRTFIM on the triangular lattice, where the order-by-disorder phase is the clock-ordered phase and the crystalline phase is the plain stripe phase [27,29,30,110,277,279,280]. The transition between the crystalline phase and the order-by-disorder phase is believed to be a first-order lever-crossing transition [29,30,110,277].

**Figure 29 entropy-26-00401-f029:**
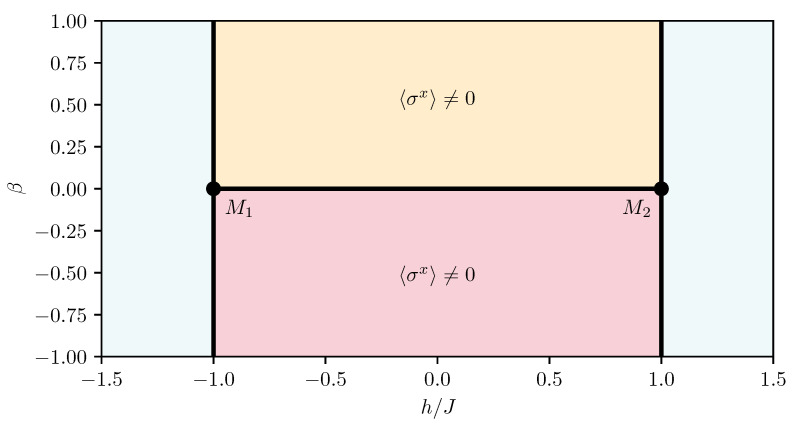
Quantum phase diagram of the ferromagnetic nearest-neighbour XY chain in a transverse field (see Ref. [284]). $M_{1}$ and $M_{2}$ denote multicritical points. The phase transition between the symmetric high-field polarised phases and magnetically ordered low-field phases are of (1+1)D-Ising universality for β≠0. For β=0, the transition at the multicritical points $M_{1}$ and $M_{2}$ has critical exponents z=2 and ν=1/2. The transition between $\langle \sigma_{i}^{x}\rangle \neq 0$ and $\langle \sigma_{i}^{y}\rangle \neq 0$ is of the (Ising)^2^ type.

**Figure 30 entropy-26-00401-f030:**
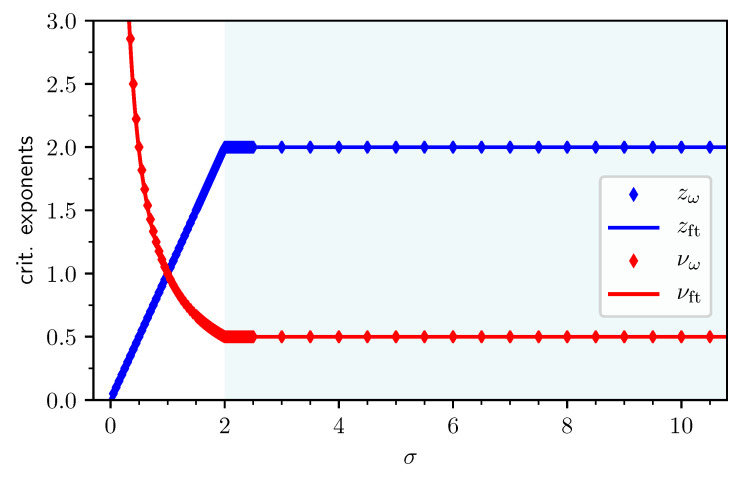
Critical exponents *z* and ν for the ferromagnetic long-range transverse-field XY model as a function of σ. The data points $z_{\omega }$ and $\nu_{\omega }$ were determined in Ref. [31] by studying the dispersion of the elementary excitations at the critical point that can be determined exactly in first-order perturbation theory from the high-field limit. The lines $z_{\mathrm{ft}}$ and $\nu_{\mathrm{ft}}$ are theoretical predictions from the QFT investigated in Ref. [31]. The region with the blue-shaded background denotes the regime in which the nearest-neighbour XY universality occurs [284], while for σ<2, the QPT is in a long-range regime with continuously varying exponents.

**Figure 31 entropy-26-00401-f031:**
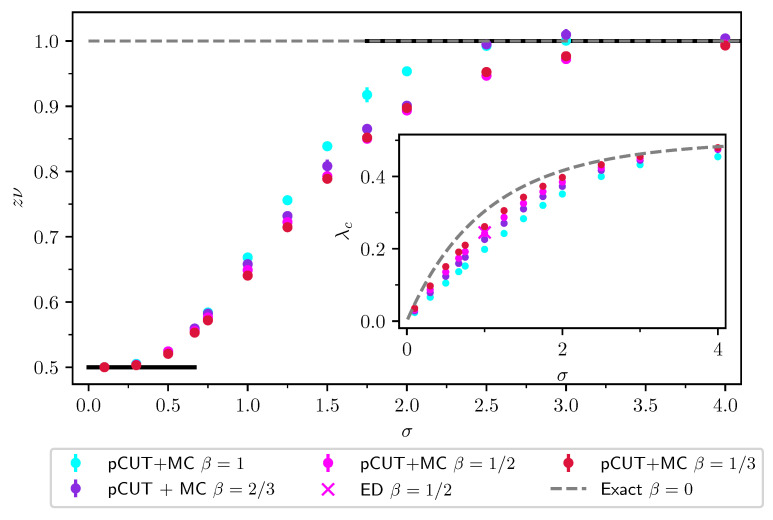
Critical gap exponent zν and critical values of λ (see inset) as a function of σ for the anisotropic ferromagnetic transverse-field XY model. The data points “pCUT+MC” are improved results from Ref. [31], and the data point “ED” is from Ref. [307]. The anisotropy parameter β is tuned from β=1 (Ising) to β=0 (isotropic XY). As discussed in Section 7.1, the isotropic case is analytically solvable with zν=1 and $\lambda_{c}={\left(2\zeta \left(\sigma +1\right)\right)}^{-1}$. The black lines denote the values of zν if the ferromagnetic LRTFIM is in the nearest-neighbour and the long-range mean-field regime.

**Figure 32 entropy-26-00401-f032:**
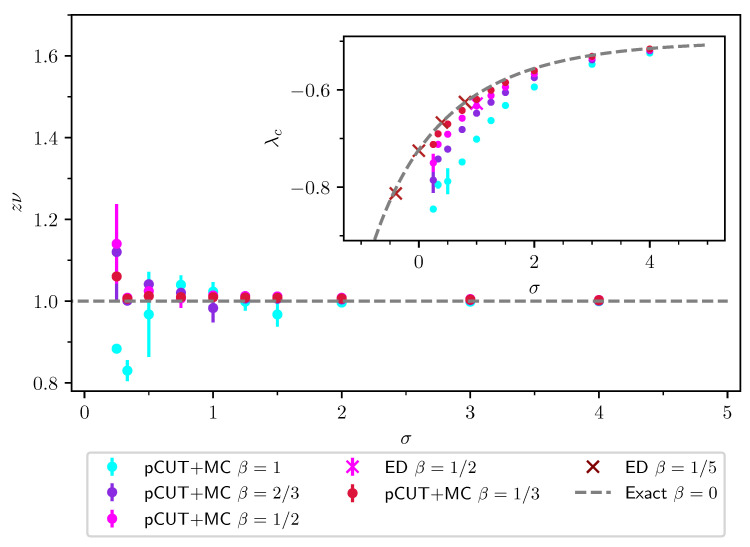
Critical gap exponent zν and critical values of λ (see inset) as a function of σ for the anisotropic antiferromagnetic transverse-field XY model [31]. The anisotropy parameter β is tuned from β=1 (Ising) to β=0 (isotropic XY). As discussed in Section 7.1, the isotropic case is analytically solvable with zν=1 and $\lambda_{c}={\left(2\eta \left(\sigma +1\right)\right)}^{-1}$.

**Figure 33 entropy-26-00401-f033:**
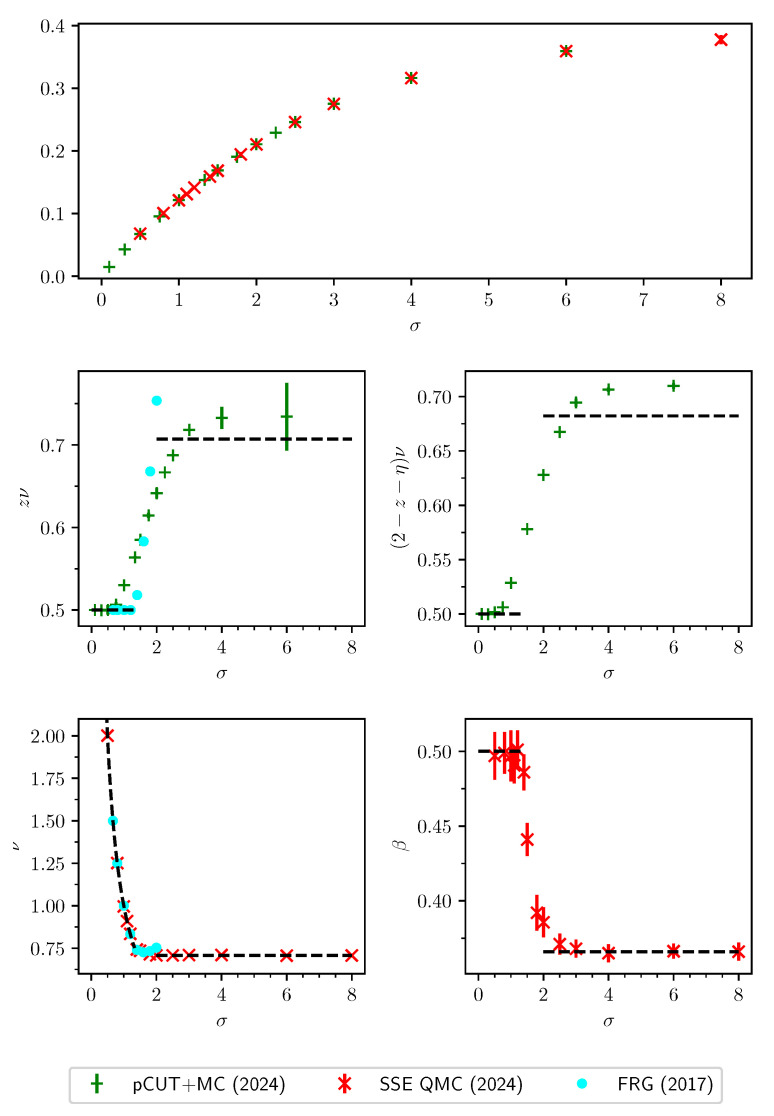
Critical values and exponents from numerical studies of the Néel-ordering transition in the unfrustrated antiferromagnetic long-range Heisenberg square-lattice bilayer model. The upper panel shows the critical values. The middle left panel displays critical exponent values zν, the middle right panel the exponent (2−z−η)ν, the lower right panel the exponent ν, and the lower right panel the one-particle spectral weight exponent β. The data points “QMC (2024)” for ν and β originate from Ref. [37]. The data points “pCUT+MC (2024)” for zν and (2−z−η)ν originate from Refs. [206,339]. The data points “FRG (2017)” for ν and zν originate from Ref. [21]. The black dashed lines denote the critical exponents in the regime of short-range O(3) criticality (ν=0.7116(10), β=0.36932(16), zν=0.7116(10), and (2−z−η)ν=0.6847(10) [348,349]) and long-range mean-field criticality (ν=1/σ, β=1/2, zν=1/2, and (2−z−η)ν=1/2 [20,21]).

**Figure 34 entropy-26-00401-f034:**
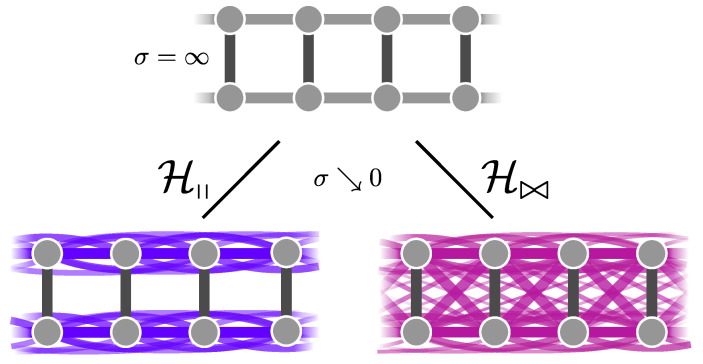
Illustration of the quantum spin ladders with long-range interactions. For nearest-neighbour interactions (σ=∞), both long-range ladders $H_{\left|\right|}$ (left) and $H_{⋈}$ (right) reduce to the same Heisenberg ladder. The coupling on the rungs $∼J_{⊥}$ is illustrated with black lines, and the long-range coupling along the legs $∼J_{\left|\right|}\left(i-j\right)$ and in between the legs $∼J_{\times }\left(i-j\right)$ is depicted in blue for $H_{\left|\right|}$ and in purple for $H_{⋈}$. The figure is adapted from Refs. [35,353].

**Figure 35 entropy-26-00401-f035:**
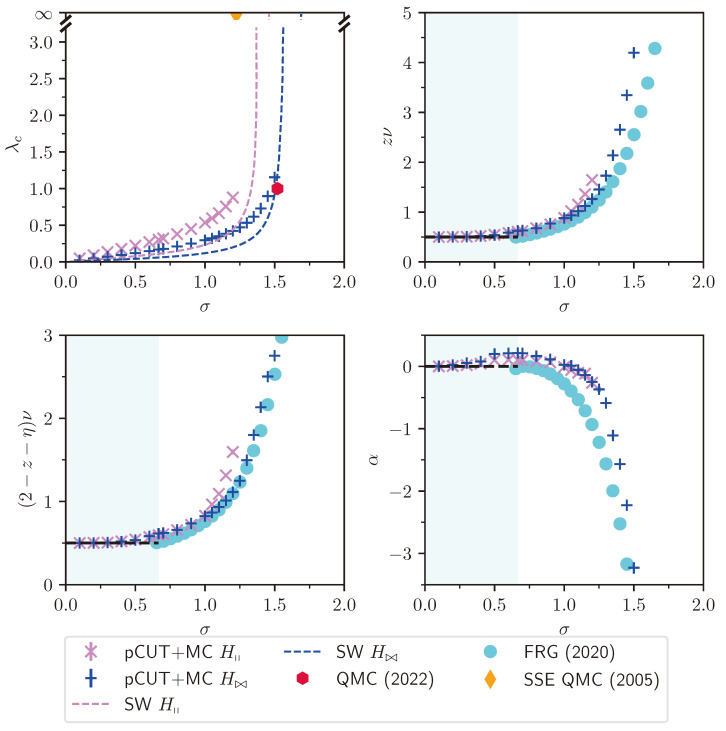
Critical values and exponents from numerical studies of the Néel-ordering transition in the unfrustrated antiferromagnetic long-range Heisenberg ladders. The panels show the critical values $\lambda_{c}$ (upper left), zν (upper right), (2−z−η)ν (lower left), and α (lower right). The data points “pCUT+MC $H_{\left|\right|}$”, “SW $H_{\left|\right|}$”, “pCUT+MC $H_{⋈}$”, and “SW $H_{⋈}$” originate from Refs. [35,353] and refer to parallel (||) and parallel + diagonal (⋈) interactions. The “SSE QMC (2005)” data point from Ref. [22] shows a $\lambda_{c}=\infty$ value on the long-range Heisenberg chain, which corresponds to the limiting case of decoupled legs. The “QMC (2022)” data [306] shows a $\lambda_{c}=1$ value for $H_{⋈}$. The data points “FRG 2020” are from Ref. [244] and show the critical exponents for the one-dimensional O(3) quantum rotor model. The blue-shaded region denotes the σ regime in which long-range mean-field criticality is expected. The black-dashed lines denote long-range mean field critical exponents.

**Figure 36 entropy-26-00401-f036:**
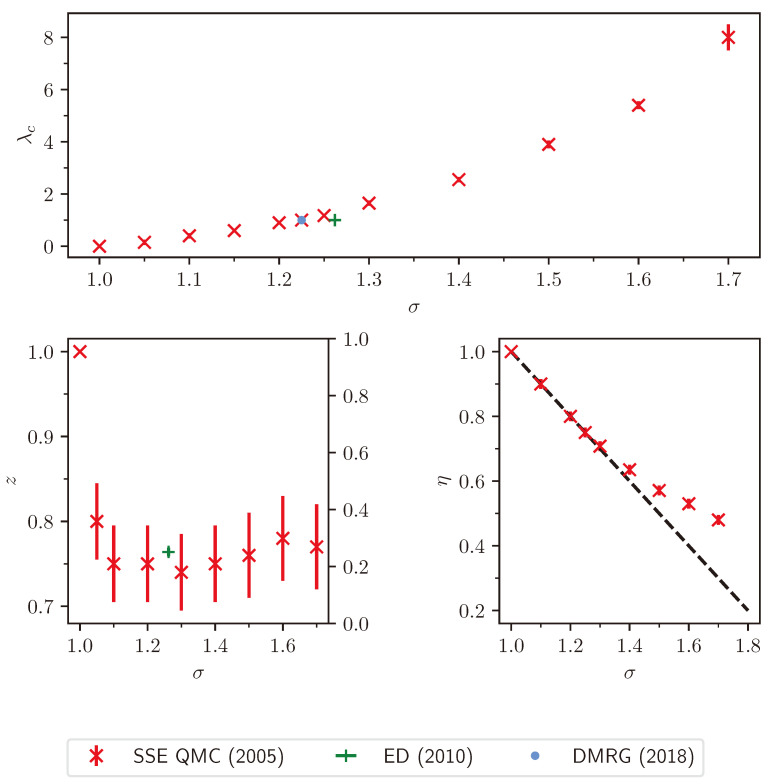
Critical values and exponents from numerical studies of the QLRO-Néel transition in the unfrustrated antiferromagnetic long-range Heisenberg chain. The upper panel shows critical values $\lambda_{c}$; the lower left shows the critical exponent *z*; the lower right shows η. The data points “SSE QMC 2005” are from Ref. [22]; “ED 2010” are from Refs. [213,297]; the single data point “DMRG 2018” is from Ref. [333]. The dashed line is for η and is the prediction from the first-order RG and scaling arguments provided in Ref. [22].

**Table 1 entropy-26-00401-t001:** Definitions of critical exponents by means of the singularities of thermodynamic quantities for a magnetic phase transition. The free energy density is denoted by *f*. Note that the control parameter susceptibility associated with the critical exponent α coincides with the heat capacity only for thermal phase transitions, where $r=\left(T-T_{c}\right)/T_{c}$, while for QPTs, the meaning depends on the control parameter triggering the phase transition [129].

Observable	Definition	Crit. Exp.	Singularity
Characteristic length ξ	via $G(\vec{r})$	ν	$\xi \left(r\to 0,H=0\right)∼{\left|r\right|}^{-\nu }$
Energy gap Δ Charact. time scale $\xi_{\tau }$	via $G(\vec{r},\omega )$ $\xi_{\tau }∼\Delta^{-1}$	zν	$\Delta \left(r\to 0\right)∼{\left|r\right|}^{z\nu }$ $\xi_{\tau }\left(r\to 0\right)∼\xi^{z}∼{\left|r\right|}^{-z\nu }$
Order parameter *m*	$m=\frac{\partial f}{\partial H}{|}_{H=0}$	β	$m\left(r\to 0^{-},H=0\right)∼{\left|r\right|}^{\beta }$
		δ	$m\left(r=0,H\to 0\right)∼{\left|H\right|}^{1/\delta }$
Order-parameter susceptibility χ	$\chi =\frac{\partial m}{\partial H}{|}_{H=0}$	γ	$\chi \left(r\to 0,H=0\right)∼{\left|r\right|}^{-\gamma }$
Control-parameter susceptibility $\chi_{r}$	$\chi_{r}=\frac{\partial^{2}f}{\partial r^{2}}$	α	$\chi_{r}\left(r\to 0,H=0\right)∼{\left|r\right|}^{-\alpha }$
Correlation function $G(\vec{r})$	${\left.\frac{\partial \langle m(\vec{r})\rangle }{\partial H_{\vec{r}=0}}\right|}_{H_{0}=0}$	η	$G(\vec{r}\to \infty ,r=0,H=0)$ $∼\frac{1}{|\vec{r}{|}^{d-2+\eta }}$

## Data Availability

The data presented in this study are available on request from the corresponding author.

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
