# Peer review of "Monte Carlo Based Techniques for Quantum Magnets with Long-Range Interactions"

_entropy, 2024, doi:10.3390/e26050401_

Round 1
Reviewer 1 Report
Comments and Suggestions for Authors
This manuscript provides an excellent review on theoretical backgrounds, numerical methods and current status of the research on the quantum phase transitions in the long-range spin models. It includes comprehensive tutorials on the pCUT+MC and SSE-QMC methods, which are very important tools to study quantum phase transitions. Also, I appreciate very much the overview of the past research results given in this manuscript, which I think is a very good summary that can greatly help researchers in this field.
On this ground, I recommend this paper for publication in Entropy. But, I'd like to suggest some minor points that the authors need to consider before publication.
1. There is another important DMRG calculation on the LRTFIM done by Puebla et al. published in Phys. Rev. A 100, 032115 (2019). Please see Fig. 1 of this paper. This reference is currently missing in the manuscript, and I think that Figures 20 and 23 would provide richer information by adding these data, or at least, the results in Fig. 1 of their paper should be mentioned at some places in Section 6.
2. I have a few questions for Figure 23.
(1) I can see there are data points from TDVP (2012) [23] in the panel for the exponent "beta", but in Ref. 23, I can't find any measurements of "beta".
(2) It is also about the data point from TDVP results. Reference 23 actually provides their measurement of scaling dimensions "Delta_x" and "Delta_z". I believe that the data points in the panel for nu came from "Delta_z". Is that correct? But, there is the dimension involved in the conversion between the two, and in the quantum version, isn't there the dynamical exponent z also involved in the conversion? I think, at least, the source of these data points needs little more clarifications.
(3) In the second paragraph (L2879-) on page 89, it says "the established picture is that it is of (1+1)D Ising type for all \sigma > 1.25", and the value "1.25" is mentioned in the several places. But, as far as I understood from the references provided, it seems that this particular value 1.25 does not have a solid theoretical ground. The TDVP (2012) paper mentioned this value for the first time, but they didn't provide any convincing evidence, in my opinion. I think the theoretical origin of this particular value 1.25 needs more clarifications in this section.
Author Response
----------------------------------------------------------------------
Response to the first referee
----------------------------------------------------------------------
We thank the referee for the examination of our work and pointing out points of improvement and for the recommendation for publication in Entropy.
Below we address the points raised by the referee:
1.) We thank the referee for the provided reference. We have added the critical field values from the reference in the respective figures as well as discussed the criticality in the corresponding passages in the main text.
We approached the authors of the reference to obtain the data points for the critical exponents since it is impossible to accurately extract them from the figures in the reference.
As soon as there is a reply from the authors we will add the data points to our figures.
2.1) We have used the following conversions to obtain the canonical critical exponents form the operator scaling powers: $\beta=\Delta_x/(d+z-\Delta_z)$ and $\nu=1/(d+z-\Delta_z)$. We have assumed that $z=1$ which is backed by the findings in Phys. Rev. A 100, 032115 (2019). We have added in the text that we assumed $z=1$ for the conversion of the quantities.
2.2) We thank the referee for this point. We have indeed assumed that $z=1$ for the conversion of the scaling powers to the critical exponents. We have clarified this in the caption of Fig.~24.
2.3) We thank the referee for this point. We totally agree with the referee that the value of $\sigma = 1.25$ introduced by the TDVP paper is not motivated and the numerical evidence is not convincing. We have strengthened our statements with regards to this reference and clarified that the "state of the art" belief in literature is a $(1+1)$D Ising universality up to $\sigma>-1$.
Reviewer 2 Report
Comments and Suggestions for Authors
This review article is devoted to the description of quantum phase transitions in lattices. The first half of the article is dedicated to a detailed explanation of various Monte Carlo techniques which can be efficiently applied to address this problem. The second half of the article is basically a review of the properties of different models. Overall, I find the article to be well written. The results seem to be credible. I recommend the publications once my comments below are addressed.
1. Introduction. To me this looks like a good place to explain in general how evaluation of the static properties can be expressed in a form convenient for Monte Carlo calculations, and why Monte Carlo methods are advantageous in evaluation of multidimensional integrals.
2. Page 4, Section “2 Quantum Phase Transitions”. It looks appropriate add Ehrenfest classification of phase transitions here
3. P. 13, Fig 2. I suggest inverting the order in which the data points are shown, so that smaller L are on the top.
4. P 25, “TFIM” looks to be used without being introduced
5. P 40 “Usually, we calculate…” it looks like some reasoning is left behind, for example “Usually, intensive quantities such as the energy per site are calculated …”
6. P 40 “Due to translational symmetry we can fix site i”, rephrase
7. P 41 “Note that the process f(G)…” sounds like a jargon, one can sum amplitudes in an equation, but summing processes in an equation does not sound right.
8. P 41 “several generalization” -> “several generalizations”
9. P 41 “In this section we only consider a single physical perturbation parameter because there cannot be an additional functional dependency on a parameter when we later perform numerical Monte Carlo summation.” Please rephrase
10. P 43 It looks like a proper way to introduce how statistical errors in the observables are calculated
11. P 46 finally it is commented that “Monte Carlo integration came into play as it is known to be a powerful integration technique for high-dimensional problems where conventional integration fails”, to me it sounds to be more appropriate for the Introduction.
12. P 51 “From trying out different probabilities we set the probability to perform a shift move to pshift = 0.7 and for the rift move we set it to pshift = 0.3”. Some details are missing, which quantity is optimized by this specific choice and also why the two probabilities are that different.
13. P 52 simulation time is provided without specifying what is the goal accuracy. It would be preferable to have a figure of the accuracy versus simulation time.
14. P 52 and also in other parts of the text, “Monte Carlo dynamics” the word “dynamics” might be confusing as it typically refers to the time evolution. I suggest avoiding this word.
15. P 52 above Eq. (202) We have given a high-order perturbative series of a physical quantity κ(λ) in the perturbation parameter λ.” It would be useful to provide some examples of which quantities can be considered, as it is not clear from the context on which quantities can be calculated in this way, i.e. local, nonlocal, etc.
16. P 53 below (204) “Given that the quantity of interest κ shows a second-order phase transition”, again it is not clear which quantity is considered here.
17. P 56 “measurable systematic error” does not sound right, if there is any error, it can be measured
18. P 56 “to bring any Hamiltonian”, “any” is a dangerous word, are there any restrictions or not?
19. P 59, the first phrase contains two abbreviations not yet defined in this section, I would suggest adding the expanded form at least of the second one.
20. Fig 13 15 16 17 and similar ones. I suggest adding “sites” and “imaginary time” labels written horizontally and vertically along the axes in (a)
21. P 57, Eq (218)“.. by expanding…” It is not clear if this is an approximate expansion or is an exact expression.
22. P 74 “The SSE QMC approach is used” -> “can be used”
23. P 79, Fig 19 looks like a phase diagram, if this is the case, this should be explicitly stated
24. P 80 it is not clear why sigma=2 corresponds to the MF case
25. P 82 Fig 20 and discussion, as the figure shows the cases with different values of sigma it will be interesting to have discussion how the properties change in the weak and strong long range cases
26. P 87 “Monte Carlo dynamics” might sound misleading
27. P 89 Fig 24 It is not clear what is the dimensionality (1D 2D 3D) of the system
28. P 90 to me “formidable resource” sounds too strong for a scientific article
29. P 92, I suggest starting both horizontal and vertical axes from 0
30. P 93 Fig 27, I suggest to add the meaning of the used abbreviation (LRFTIM)
31. P 94, “glassy phase” is mentioned, it can be interesting to see what are the critical exponents in the glass phase.
32. P 102 “should be sufficiently long range” -> “long ranged”
33. P 103 “is now a 3-component instead of one-component field”-> “is now a 3-component field instead of one-component one”
34. P 109 “about the critical point”, may “in the vicinity” would sound better
35. P 112 “advancements” -> “advances”?
36. General: for a review, I would prefer to use bibliography style such that the reference is formed by the first author and publication year, if this is allowed by the journal.
37. General: this review provides a detailed description of the algorithms presented in a quite rigorous and abstract way. At the same time there are explicit references to programming languages such as C++ and python, as well as specific libraries. This feels like a large jump from an abstract explanation to practical implementation. It might be better to present the use of libraries (p. 34, 41) by making at least references which algorithms are used there.
38. General: use of \frac{x}{y} vs x/y notation is not consistent. I suggest using \frac{x}{y} for numbered equations and x/y inside of the text and captions.
39. General: figures: - it is a good practice to present figures in a self-explanatory way so that it can be understood by reading the caption. For the caption, it is good to have the first sentence providing essentially the title of the figure, then explanation of which quantity is shown as a function of which quantity. Also it is good to use both the mathematical and physical terms, that is instead of just saying “z” or “dynamical exponent”, saying “dynamical exponent z” for an easier comprehension. Avoid using lengthy abbreviations in the captions, or at least the first time it is used, explain its meaning. It is convenient to add labels (a,b,c,…) to all panels and insets, so that instead of referring to “inset of the right panel of Fig. N” the reference is simply “Fig. Na”
40. General: it is a good practice to write Fig.~(), Ref.~[], etc, in LaTeX so that the number is never separated to the next line. Same with “scaling relations~(3)-(9)
41. General, in some journals the first word of a sentence should not be abbreviated, i.e. “Refs [229,293]” would be expanded into “References []”, same with Fig.~, Sec.~, etc.
42. General, for me a phrase like “We use Simpson to calculate a multidimensional integral” sounds like jargon. Instead “We use the Simpson method/algorithm/technique ..” sounds better. In a similar way “We use Monte Carlo/QMC/SSE QMC/etc to calculate …” sounds like jargon as the authors do not refer to a city in Monaco but rather to a method.
The English grammar is good, I have found only a few typos or grammatical suggestions.
Author Response
----------------------------------------------------------------------
Response to the second referee
----------------------------------------------------------------------
We thank the referee for the thorough examination of our work and addressing many points for improvement.
Below we address the minor points raised by the referee:
1.) We added two sentences about the advantages of Monte Carlo integration in the introduction on p. 4.
2.) We added an introduction of first- and second-order phase transitions in the Ehrenfest classification scheme.
3.) We think that the large Ls are more important as they are closer to the thermodynamic limit and should be on top. However, we adjusted the size of data points such that the smaller system sizes are more visible and not hidden behind the larger system sizes anymore.
4.) We added the expanded version of the abbreviation.
5.) We included the suggested improved sentence.
6.) We rephrased the sentence.
7.) We rephrased the sentence.
8.) The typo is corrected.
9.) We split it into two sentences and rephrased it
10.) We addressed the calculation of errors in 4.7.2. and in 4.8 in more detail.
11.) We now mention the advantages of MC integration in the motivation, but we believe that the reader can benefit from restating the advantage at this point.
12.) We agree with the point raised by the referee and explained the choice in more detail.
13.) We agree with the referee and added a figure showing the Monte Carlo error as a function of the number of Monte Carlo steps (which is proportional to the simulation time). We commented on the convergence in the main body of the text.
14.) We adjusted the wording: dynamics -> behaviour; changed -> altered.
15.) We now refer to a previous section, where we introduced the quantities we consider.
16.) See point 15.)
17.) We removed the word "measurable" from the sentence.
18.) We replaced "Hamiltonian" with the more appropriate word "partition function". There are no restrictions to the partition function.
19.) We included the expanded version of the abbreviation "TFIM". The abbreviation "SSE" was already defined in section 5..
20.) We added the labels in the respective figures as suggested.
21.) The expression is exact. We substituted "expanding it" with "expanding the product of sums" to make this more clear.
22.) We accepted the suggested improved verb.
23.) Fig 20. is not a phase diagram, but a diagram visualizing which kind of universality classes are expected in dependence of the decay exponent. We substituted "distinct critical regimes" with "distinct universality regimes" as this might be a source of confusion.
24.) We are not entirely sure which statement needs clarification. If the comment by the referee is about the sentence "In the limit σ → 2, the short-range mean-field exponents are recovered" after Eq. (298), we would like to point out that a Gaussian theory is considered here and even before setting sigma=2, the exponents are mean-field exponents. The main point of the sentence is that for sigma=2, the exponents are the short-range version of mean-field exponents because for sigma = 2 the action coincides with the short-range action.
25.) In Fig. 21 and the following discussion we present only the regime of weak long-range interactions ($\sigma > d$). Here the three regimes of criticality (LR-mean-field, LR-non-trivial, SR) occur. For discussing the regime of strong long-range interactions ($\sigma \leq d$) Kac-rescaled coupling constants for the Ising interactions are required. As pointed out on several occasions in the article we choose not to discuss models where rescaling is required.
26.) We deleted the word "dynamics" as what we meant with the word "dynamics" is already included in "autocorrelation times" and additionally added the word "algorithm" in that place to make it a proper sentence.
27.) We included "two-dimensional" in front of "square lattice" to improve clarity.
28.) We substituted "formidable resource" with "great resource"
29.) We were not able to associate the comment with a figure where it is, in our opinion, beneficial to adjust the $x$ and $y$-ranges of the plot.
30.) We included the expanded version of the abbreviation and at the same time corrected the typo "LRFTIM" -> "LRTFIM".
31.) We thank the referee for this point. We mention the "glassy behaviour" only in a listing of recent interesting theoretical predictions of states that can be realized in Rydberg atom quantum simulators. Since we provide the corresponding primary reference we choose not to dive further into this topic.
32.) We corrected the wording.
33.) We accepted the improved phrasing.
34.) We substituted "about the critical point" with "close to the critical point"
35.) We accepted the improved wording.
36.) We follow the citation guidelines of the journal: "In the text, reference numbers should be placed in square brackets [ ], and placed before the punctuation; for example [1], [1–3] or [1,3].".
37.) We removed all mentions of programming languages (occurrence in Sec. 4.4.). The only mention of specific libraries is in the context of graphs. We believe that mentioning specific graph libraries is valuable to the reader because this is the only part of the approach where libraries are readily available. For the graph generation we cite McKay's algorithm for the interested reader. However implementing these algorithms is a non-trivial task that is also why we believe mentioning libraries is important.
38.) We thank the referee for this general point. We have removed all $\frac{x}{y}$ from everything besides numbered formulas and replaced it by $x/y$.
39.) We thank the referee for this general point. We implemented the suggestions by the referee where required, reasonable and feasible.
40.) We thank the referee for this general point. We agree with the referee and use the LaTex features as much as possible to keep formulas in one line. We will wait for the production version of the manuscript for final adjustments regarding the placement of formulas.
41.) We thank the referee for this general point. We have removed abbreviations like Ref., Sec., Fig., ... from the beginning of sentences on all occasions that we noticed.
42.) We thank the referee for this general point. We have tried to adjust our wording on several occasions across the entire manuscript.